# The $U$-plane of rank-one 4d $\mathcal{N} = 2$ KK theories

Cyril Closset[1] and Horia Magureanu[2]

**1** School of Mathematics, University of Birmingham, Watson Building,
Edgbaston, Birmingham B15 2TT, UK
**2** Mathematical Institute, University of Oxford, Andrew-Wiles Building,
Woodstock Road, Oxford, OX2 6GG, UK

## Abstract

The simplest non-trivial 5d superconformal field theories (SCFT) are the famous rank-one theories with $E_n$ flavour symmetry. We study their $U$-plane, which is the one-dimensional Coulomb branch of the theory on $\mathbb{R}^4 \times S^1$. The total space of the Seiberg-Witten (SW) geometry – the $E_n$ SW curve fibered over the $U$-plane – is described as a rational elliptic surface with a singular fiber of type $I_{9-n}$ at infinity. A classification of all possible Coulomb branch configurations, for the $E_n$ theories and their 4d descendants, is given by Persson's classification of rational elliptic surfaces. We show that the global form of the flavour symmetry group is encoded in the Mordell-Weil group of the SW elliptic fibration. We study in detail many special points in parameters space, such as points where the flavour symmetry enhances, and/or where Argyres-Douglas and Minahan-Nemeschansky theories appear. In a number of important instances, including in the massless limit, the $U$-plane is a modular curve, and we use modularity to investigate aspects of the low-energy physics, such as the spectrum of light particles at strong coupling and the associated BPS quivers. We also study the gravitational couplings on the $U$-plane, matching the infrared expectation for the couplings $A(U)$ and $B(U)$ to the UV computation using the Nekrasov partition function.


# 1   Introduction

Supersymmetric quantum field theories (SQFT) with a least eight supercharges are particularly amenable to exact, non-perturbative methods. The infrared (IR) physics on the Coulomb branch (CB) of 4d $\mathcal{N} = 2$ supersymmetric field theories, in particular, is famously encoded in a Seiberg-Witten (SW) geometry [1,2]. For 4d $\mathcal{N} = 2$ field theories of rank one – that is, when the Coulomb branch is one dimensional – the SW geometry consists of an elliptic fibration over the Coulomb branch, and the effective gauge coupling for the low-energy $U(1)$ vector multiplet is identified with the modular parameter of the elliptic fiber.

The original SW geometries for $SU(2)$ gauge theories were derived based on the semi-classical analysis and on deep physical intuition. Once the SW geometry of a 4d $\mathcal{N} = 2$ theory is given to us, on the other hand, we can gain further insights into strong-coupling phenomena by analysing the SW geometry throughout its full parameter space. The Argyres-Douglas (AD) superconformal field theories (SCFTs) were discovered in that way on the Coulomb branch of 4d gauge theories [3,4]. Postulating the existence of other SW geometries with more complicated singularities also led to the discovery of 4d SCFTs with $E_n$ global symmetry, the Minahan-Nemeschansky (MN) theories [5,6].

All these 'classic' rank-one 4d $\mathcal{N} = 2$ field theories can be understood within the framework of geometric engineering[1] in Type-IIA string theory on local Calabi-Yau threefold singularities [9,10], with an interesting plot twist: from that particular string-theory point of view, the 'most natural' rank-one theories are not these 'ordinary' 4d $\mathcal{N} = 2$ SQFTs. Instead, they are *five-dimensional* field theories compactified on a circle. This fact is true for IIA geometric engineering of $\mathcal{N} = 2$ SQFTs of any rank, since it is a simple consequence of the Type-IIA/M-

---

[1]Of course, there are also methods for 'engineering' these theories on D-branes or M5-branes – see *e.g.* [7,8]. We will focus on the 'purely geometric' string-theory engineering.

theory duality [11, 12]. At rank one, one can obtain in this way a small family of 5d SCFTs with $E_n$ symmetry [13,14], which we will often call 'the $E_n$ theories'. Their circle compactification gives us 4d $\mathcal{N} = 2$ supersymmetric field theories 'of Kaluza-Klein (KK) type' – intuitively speaking, with an infinite number of 'fields' organised in KK towers –, which we denote by $D_{S^1}E_n$. The 5d $E_n$ theories are all related by mass deformations, starting from the $E_8$ theory and mass-deforming to smaller $E_n$ subgroups. The 5d $E_8$ theory itself can be obtained as a deformation of the so-called $E$-string theory [15], which is a 6d $\mathcal{N} = (1,0)$ SCFT with $E_8$ symmetry, compactified on a circle. Similarly, the 4d MN theories naturally arise as subsectors of the $D_{S^1}E_n$ theories for $n = 6, 7, 8$. By taking the so-called geometric-engineering limit [10] on the Type-IIA complexified Kähler parameters, one obtains the ordinary 4d $\mathcal{N} = 2$ $SU(2)$ gauge theories with $N_f \leq 4$ flavours as a limit of the $D_{S^1}E_n$ theories with $n = N_f + 1$.

In this work, we revisit the Seiberg-Witten geometries of all these 'classic' rank-one theories, with our main focus being on $D_{S^1}E_n$, the circle compactifications of the 5d $E_n$ SCFTs. Their SW curves are best understood in terms of local mirror symmetry in string theory [11, 12, 16–24]. We provide a detailed analysis of their Coulomb branches by a variety of methods, including: by a direct analysis of the electromagnetic periods $a$, $a_D$ using Picard-Fuchs equations; by a global analysis using the mathematical language of rational elliptic surfaces; by taking advantage of the beautiful modular properties that exist in many special cases. The main upshot of our analysis is threefold:

- We reach a systematic understanding of the possible CB configurations in all cases, including a classification of the Argyres-Douglas points that can arise. Part of the original motivation for this endeavour was to understand some RG flows between the 5d $E_n$ theories and Argyres-Douglas theories, whose existence was implied by the recent work of Bonelli, Del Monte and Tanzini [25] relating some 5d BPS quivers [26] to the gauge/Painlevé correspondence [27–29]. The most striking of such RG flows may be:

$$D_{S^1}E_3 \rightarrow H_2 , \tag{1}$$

  where $H_2$ is the AD theory with flavour symmetry algebra $\mathfrak{su}(3)$. The reason why this flow may look surprising is that $H_2$ appears on the CB of 4d $SU(2)$ with $N_f = 3$ flavours [4] while the $E_3$ theory is related to 5d $SU(2)$ with $N_f = 2$. In hindsight, this is not too disturbing, since the AD points arise in the strong-coupling region of the Coulomb branch. In this example, the $\mathfrak{su}(3)$ flavour symmetry of $H_2$ in inherited from the symmetry $E_3 = \mathfrak{su}(3) \oplus \mathfrak{su}(2)$ of the larger theory, which arises due to 'infinite-coupling effects' from the 5d gauge theory point of view and is related to the condensation of instanton particles – see $e.g.$ [30]. As we will see explicitly, the AD points are ubiquitous on the extended Coulomb branch of the $D_{S^1}E_n$ theories as we tune the mass parameters. For instance, one can show that there are exactly 35 distinct CB configurations of the $D_{S^1}E_8$ theory with at least one $H_2$ AD point (33 configurations have one such point, 2 configurations have 2 distinct $H_2$ points).

- We explain how to determine the global symmetry of the rank-one SQFTs directly from the SW geometry itself. The SW geometry is an elliptic fibration with a distinguished section (the zero section), and it can have a non-trivial Mordell-Weil (MW) group. Firstly, we show that the free generators of the MW group correspond to abelian factors of the flavour symmetry group $G_F$. Secondly, we argue that the precise form of the flavour symmetry *group* (as opposed to its Lie algebra) is determined by the torsion part of the MW group. Thirdly and relatedly, we conjecture that the one-form symmetry of the theory [31] is also encoded in the MW torsion in a specific way.[2] For the 5d and 4d $E_n$

---

[2]Our precise findings differ from the discussion of one-form symmetries in appendix A of [32].

theories, our derivation of the flavour symmetry group from the 4d infrared confirms the recent results in [33] and [34], respectively. Interestingly, our analysis also implies that the flavour symmetry group for the AD theories $H_1$ and $H_2$ is $SO(3) = SU(2)/\mathbb{Z}_2$ and $PSU(3) = SU(3)/\mathbb{Z}_3$, respectively – for $H_1$, this was recently argued in [35], while for $H_2$ this seems to be a new observation.

- Throughout our analysis, we emphasise the key role played by modularity, especially in the case of the massless $E_n$ theories on a circle. For all the massless $D_{S^1}E_n$ theories with $n < 8$ and a semi-simple flavour symmetry algebra, the Coulomb branch is actually a modular curve $\mathbb{H}/\Gamma$ for a finite-index subgroup $\Gamma \subset \mathrm{PSL}(2, \mathbb{Z})$ – see table 2 below. This is similar to the role played by the congruence subgroup $\Gamma^0(4)$ for the pure $SU(2)$ gauge theory [2], for instance. Somewhat relatedly, we also discuss the gravitational couplings $A(U)$ and $B(U)$ on the Coulomb branch of the $D_{S^1}E_n$ theories (in the so-called toric case), which gives a 5d generalisation of the same computation for the 4d $SU(2)$ gauge theories in [36]. These considerations are very important for the computation of supersymmetric partition functions as in *e.g.* [37–39], as we will discuss elsewhere [40].

In the rest of this introduction, we explain in more detail our general picture and our new results. We also discuss how our approach relates to the vast previous literature, and we emphasise unresolved issues and challenges for future work.

## The $U$-plane, CB configurations and rational elliptic surfaces

The one-dimensional Coulomb branch of the $D_{S^1}E_n$ theory is parameterised by a single complex parameter:

$$U = \langle W \rangle, \tag{2}$$

which is the expectation value of a five-dimensional $\mathcal{N} = 1$ supersymmetric 'Wilson loop' operator wrapping the circle. (More precisely, when the 5d $E_n$ theory is mass-deformed to a 5d $SU(2)$ gauge theory with $N_f = n-1$ flavours, the operator $W$ flows to the fundamental Wilson loop.) Our main object, in this paper, is to understand the $U$-plane physics as thoroughly as we can – we would like to explore the Coulomb branch 'as pedestrians' [41], as it were.

When studying 4d $\mathcal{N} = 2$ and 5d $\mathcal{N} = 1$ supersymmetric quantum field theories, one can gain invaluable insight by embedding these field theories within string theory. In fact, in the five-dimensional case, a definition of the 5d field theory via string theory is so-far unavoidable. In string-theory approaches to SQFT, many features of the field theory become 'geometrised'. In the case at hand, the Seiberg-Witten geometry of 4d $\mathcal{N} = 2$ field theories can be understood in terms of local mirror symmetry [21]:

$$\text{Type IIA on } \mathbb{R}^4 \times \widetilde{\mathbf{X}} \qquad \longleftrightarrow \qquad \text{Type IIB on } \mathbb{R}^4 \times \widehat{\mathbf{Y}}. \tag{3}$$

Here, $\widetilde{\mathbf{X}}$ will be the local $dP_n$ (or $\mathbb{F}_0$) geometry, *i.e.* the total space of the canonical line bundle over a del Pezzo surface of degree $9 - n$ – see *e.g.* [12, 14, 20, 42–44]. This geometrically engineers the $D_{S^1}E_n$ theories for $n \leq 8$. (For $n = 9$, this engineers the $E$-string theory on $T^2$.) The local mirror threefold in Type IIB, $\widehat{\mathbf{Y}}$, can be viewed as the suspension of an elliptic curve, which is precisely the SW curve of the $D_{S^1}E_n$ theory.

Any rank-one SW geometry is given by a family of elliptic curves over the $U$-plane. (For strictly 4d theories, we should call it 'the $u$-plane' instead, following standard terminology.) This can always be written in Weierstrass normal form:

$$y^2 = 4x^3 - g_2(U; M_F)x - g_3(U; M_F), \tag{4}$$

Table 1: Correspondence between the singular fiber at infinity and the 5d and 4d theories. The field theories are denoted by their flavour symmetry algebra, except for the $SU(2)$ gauge theory with $N_f = 1$ and $N_f = 0$ flavours (denoted by $N_f = 1$ and $N_f = 0$, respectively). Thus, $D_{N_f}$ denotes the $SU(2)$ theory with $N_f$ flavours. The bottom line corresponds to the 3 AD theories $H_{N_f - 1}$, which can be found on the CB of the $SU(2)$, $N_f > 0$ gauge theory for $N_f = 1, 2, 3$. We also give the number of distinct CB configurations, $\#\mathcal{S}$, in each case.

| $F_\infty$ | $I_1$ | $I_2$ | $I_3$ | $I_4$ | $I_5$ | $I_6$ | $I_7$ | $I_8$ | $I_9$ |
|---|---|---|---|---|---|---|---|---|---|
| $D_{S^1}\mathcal{T}_{5d}$ | $E_8$ | $E_7$ | $E_6$ | $E_5$ | $E_4$ | $E_3$ | $E_2$ | $E_1$ or $\widetilde{E}_1$ | $E_0$ |
| $\#\mathcal{S}$ | 227 | 140 | 77 | 51 | 26 | 16 | 6 | $2 + 2$ | 1 |
| $F_\infty$ | $II$ | $III$ | $IV$ | $I_0^*$ | $I_1^*$ | $I_2^*$ | $I_3^*$ | $I_4^*$ | |
| $\mathcal{T}_{4d}$ | $E_8$ | $E_7$ | $E_6$ | $D_4$ | $D_3$ | $D_2$ | $N_f = 1$ | $N_f = 0$ | |
| $\#\mathcal{S}$ | 137 | 93 | 49 | 19 | 13 | 6 | 2 | 1 | |
| $F_\infty$ | | | | | $IV^*$ | $III^*$ | $II^*$ | | |
| $\mathcal{T}_{4d}$ | | | | | $A_2$ $(H_2)$ | $A_1$ $(H_1)$ | $-$ $(H_0)$ | | |
| $\#\mathcal{S}$ | | | | | 8 | 4 | 2 | | |

where $M_F$ denotes all the mass parameters. At fixed $M_F$, the total space of the elliptic fibration over the $U$-plane can be viewed as a *rational elliptic surface*, $\mathcal{S}$, by compactifying the point at infinity:

$$E \to \mathcal{S} \to \mathbb{P}^1 \cong \{U\}. \tag{5}$$

For all the 5d and 4d theories, the SW fibration is singular at $U = \infty$, while the point at infinity is regular for the $E$-string compactified on $T^2$.

Rational elliptic surfaces (RES) have been very thoroughly studied by mathematicians – see [45] for a beautiful and detailed introduction. In particular, they were fully classified by Persson and Miranda over 30 years ago [46–48]. A RES $\mathcal{S}$ is partially characterised by its set of singular fibers, $\{F_v\}$.[3] The possible singular fibers are given by the Kodaira classification, which is closely related to the ADE classification of simply-laced Lie algebras. In our physical setup, the singularity type of the fiber at infinity, $F_\infty$, essentially determines the field theory. All the theories studied in this work appear in table 1. In particular, the $D_{S^1}E_n$ theories are characterised by $F_\infty = I_{9-n}$, which is equivalent to having a prescribed monodromy at infinity $\mathbb{M}_\infty = T^{9-n}$. This follows from general considerations similar to the weak-coupling analysis for 4d $SU(2)$ gauge theories [1, 2]; here, the monodromy is determined by the 5d gauge theory one-loop $\beta$-function coefficient [13, 19], as we will show.

Let us then denote by $\mathcal{T}_{F_\infty}$ the corresponding field theory. Any *CB configuration* of $\mathcal{T}_{F_\infty}$ corresponds to a RES with a fixed set of singular fibers:

$$U\text{-plane of } \mathcal{T}_{F_\infty} \text{ at fixed } M_F \qquad \longleftrightarrow \qquad \mathcal{S} \text{ with } \{F_\infty, F_1, \cdots, F_k\}. \tag{6}$$

The Kodaira singularities *in the interior* of the $U$ plane have a well-understood low-energy physics interpretation – see *e.g.* [11]. For instance, the AD fixed points $H_0$, $H_1$ and $H_2$ arise from Kodaira singularities of type $II$, $III$ and $IV$, respectively. We can then use the classification of

---

[3]It is fully characterised by $\{F_v\}$ and $\Phi$, its Mordell-Weil group, to be discussed below.

rational elliptic surfaces to map out all possible CB configurations of a given rank-one 4d $\mathcal{N} = 2$ field theory, simply by fixing $F_\infty$.[4] The number of distinct configuration in each case is given in table 1 for all the 4d and 5d theories. The remaining possibility is for the fiber at infinity to be smooth, $F_\infty = I_0$, which corresponds to a CB configuration of $D_{T^2}E_8^{(6d)}$, the $E$-string theory on a torus; since the total number of distinct rational elliptic surfaces is 289 [47, 48], this is also the number of distinct CB configuration for $D_{T^2}E_8^{(6d)}$. We shall focus on the 4d and 5d theories (the 'rational' and 'trigonometric' cases) in this work, which are special limits of the $E$-string theory (the 'elliptic' case).

The relation between rank-one SW geometries and rational elliptic surfaces has appeared repeatedly in the string-theory literature – see in particular [24, 49–57]. More recently, Caorsi and Cecotti used the RES formalism to explore the physics of strictly four-dimensional $\mathcal{N} = 2$ field theories [58], in relation to the Argyres-Lotito-Lü-Martone classification of rank-one 4d $\mathcal{N} = 2$ SCFT [59–62]. See also [63] for a closely related approach. The present work builds on the approach of [58] by noting that *all* rational elliptic surfaces with a fixed $F_\infty$ have a nice CB interpretation, once we consider the 5d $E_n$ theories and the $E$-string theory. In addition, we provide an improved understanding of the flavour symmetry group of the 4d $\mathcal{N} = 2$ theories in that language, as we will discuss next. On the other hand, we focus exclusively on the 5d $E_n$ theories and on their 4d descendants. The latter are the 4d $SU(2)$ gauge theories with $0 \leq N_f \leq 4$ flavours and the 6 'classic' 4d SCFTs without marginal couplings – the 3 AD theories [4] and the 3 Minahan-Nemeschansky theories [6]. We believe that our analysis can be (and should be) generalised to include the remaining rank-one 4d SCFTs [59, 60, 64] by building on the insights from [58, 63] and on the S-fold approach [65–67]. This is left for future work.

## Mordell-Weil group and global symmetries

The flavour symmetries of the rank-one theories can be deduced directly from the structure of the associated SW fibration. This can be understood by using the particular structure of the Type IIB mirror geometry, which can be equivalently described in terms of a single D3-brane probing a collection of 7-branes in an F-theory construction [23, 49, 50, 68]. In the F-theory language, the Kodaira singularities on the $U$-plane are non-compact 7-branes, which therefore give rise to flavour symmetry algebras of ADE type. We are particularly interested in determining the actual flavour symmetry *group* which, by definition, acts faithfully on gauge-invariant states.

In that respect, an important role is played by the Mordell-Weil (MW) group of rational sections of the SW geometry $\mathcal{S}$, which is an abelian group of the general form:

$$\Phi = \mathrm{MW}(\mathcal{S}) \cong \mathbb{Z}^{\mathrm{rk}(\Phi)} \oplus \Phi_{\mathrm{tor}}, \qquad \Phi_{\mathrm{tor}} \cong \mathbb{Z}_{k_1} \oplus \cdots \oplus \mathbb{Z}_{k_t}, \tag{7}$$

where $\mathrm{rk}(\Phi) \in \mathbb{Z}_{\geq 0}$ is the rank of the MW group. The $\mathrm{rk}(\Phi)$ generators of the free part of $\Phi$ correspond to abelian symmetries. On the other hand, the torsion sections constrain the non-abelian part of the flavour symmetry group, similarly to the way the gauge group is determined in F-theory compactifications [52, 69–73]. In our setup, a key role is played by the Kodaira singularity at infinity, $F_\infty$, which does not contribute to the IR flavour symmetry. It then turns out to be important to compute precisely how the sections $P \in \Phi$ intersect the reducible fibers in the interior and at infinity. We will define $\mathcal{Z}^{[1]} \subset \Phi_{\mathrm{tor}}$ to be the maximal subgroup of torsion sections that intersect 'trivially' (*i.e.* like the zero section) all the fibers in the interior of the $U$-plane, and we will define the abelian group $\mathscr{F}$ to be the cokernel of the inclusion map:

$$0 \to \mathcal{Z}^{[1]} \to \Phi_{\mathrm{tor}} \to \mathscr{F} \to 0. \tag{8}$$

---

[4]From table 1, one can see that the $F_\infty = I_8$ is either the $E_1$ or the $\widetilde{E}_1$ theory. The distinction between the two will be explained momentarily.

Table 2: Modular groups for the massless $D_{S^1}E_n$ theories, and their index in $\mathrm{PSL}(2,\mathbb{Z})$. They are all congruence subgroups. The massless CB for $E_8$, $E_2$ and $\widetilde{E}_1$ are not modular.

| $D_{S^1}\mathcal{T}_{5d}$ | $E_7$ | $E_6$ | $E_5$ | $E_4$ | $E_3$ | $E_1$ | $E_0$ |
|---|---|---|---|---|---|---|---|
| $\Gamma$ | $\Gamma^0(2)$ | $\Gamma^0(3)$ | $\Gamma^0(4)$ | $\Gamma^1(5)$ | $\Gamma^0(6)$ | $\Gamma^0(8)$ | $\Gamma^0(9)$ |
| $(\Gamma : \Gamma(1))$ | 3 | 4 | 6 | 12 | 12 | 12 | 12 |

Then, we claim that:

- $\mathcal{Z}^{[1]}$ gives the one-form symmetry of the 4d field theory. In particular, it does not change as we vary the mass parameters. Incidentally, this distinguishes between the $E_1$ and $\widetilde{E}_1$ configurations for $F_\infty = I_8$, in which case $\mathcal{Z}^{[1]} \cong \mathbb{Z}_2$ for $E_1$ while it is trivial for $\widetilde{E}_1$.

- The IR flavour symmetry group $G_F$, for any given CB configuration $\mathcal{S}$ of $\mathcal{T}_{F_\infty}$, takes the schematic form:

$$G_F \cong \left( U(1)^{\mathrm{rk}(\Phi)} \times \prod_{v \neq \infty} \widetilde{G}_v \right)\Big/ \mathscr{F}, \tag{9}$$

where $\widetilde{G}_v$ is the simply-connected group with Lie algebra $\mathfrak{g}_v$, associated to each non-reducible Kodaira fiber $F_v$ in the interior of the $U$-plane. It follows from the general mathematical theory of RES that $\mathscr{F}$ injects into the center of $\prod_v \widetilde{G}_v$, so that the quotient in (9) is well-defined. The expression (9) is slightly imprecise when $\mathrm{rk}(\Phi) > 0$, because we should specify the normalisations of the abelian charges, which are determined by the way the non-torsion sections intersect the reducible fibers.

We will explain all these statements in section 3, after reviewing the necessary mathematical background. Our identification of the one-form symmetry agrees with known results for all the theories under consideration. A complete 'physics proof' of the identification of $\mathcal{Z}^{[1]}$ with the one-form symmetry, which would relate our approach to [74–79], is left for future work. On the other hand, we provide a detailed explanation, and much direct evidence, for the determination of the flavour symmetry group using the MW torsion. This confirms the recent results in [33,34], which were derived by different (albeit related) methods. It is also tempting to identify $\Phi_{\mathrm{tor}}$ itself as a 2-group [80–83], which can be non-trivial for 5d SCFTs [33]; that is another interesting point that we leave for future work.

## Modularity on the $U$-plane

It is well-known that the Coulomb branch of the pure $SU(2)$ gauge theory is a modular curve for the congruence subgroup $\Gamma^0(4)$ of $\mathrm{PSL}(2,\mathbb{Z})$ [2]. Similarly, the massless $u$-plane of 4d $SU(2)$ with $N_f = 2$ and $N_f = 3$ is modular with respect to $\Gamma(2)$ and $\Gamma_0(4)$, respectively [37,84].[5] By definition, a CB configuration is modular if the $U$-plane is a modular curve of genus zero:

$$\{U\} \cong \mathbb{H}/\Gamma, \qquad \tau \mapsto U(\tau), \ \forall \tau \in \mathbb{H}, \tag{10}$$

for some modular group $\Gamma \subset \mathrm{PSL}(2,\mathbb{Z})$. In that case, the $U$-plane is isomorphic to a fundamental domain for $\Gamma$ on the upper half-plane. The function $U(\tau)$ is called the Hauptmodul, or

---

[5]Note that the $\Gamma(2)$ curve also appeared in the original SW paper to describe the pure $SU(2)$ CB [1]. Our perspective here is that the 'correct' pure $SU(2)$ CB is the $\Gamma^0(4)$ curve – see *e.g.* the discussion in [41] – which is the one that arises naturally from local mirror symmetry.

principal modular function, for $\Gamma$. Modularity often simplifies the analysis of the low-energy physics, as we will see in many examples. For instance, it allows us to easily identify the dyons that become massless at cusps, once we have chosen a fundamental domain. The latter can often be used to visualize how other configurations can be obtained by merging singularities, such as in the traditional case of the AD theories arising on the $u$-planes of the $SU(2)$ gauge theories [4]. This also allows us to easily derive BPS quivers [85–87] directly from the $U$-plane in many cases. This last point certainly deserves further study. Other recent discussions of BPS quivers in the present context include *e.g.* [88–94].

The modular groups for the massless 'semi-simple' $E_n$ theories on a circle are shown in table 2. Most of these modular groups appeared in one guise or another in the string-theory literature when discussing local mirror symmetry – see *e.g.* [95–102]. On the gauge theory side, the recent work by Aspman, Furrer, and Manschot on the $SU(3)$ Coulomb branch [103] was a source of inspiration to us.[6] Incidentally, it would be important, but probably challenging, to generalise our systematic approach to rank-two (and higher-rank) theories, in order to make contact with the recent renewed efforts in mapping out rank-two 4d SCFTs [105–107].

### Gravitational couplings on the $U$-plane

In the last section of this paper, section 9, which can be read in combination with section 2 and independently of the rest of the paper, we study the effective gravitational couplings $A(U)$ and $B(U)$ on the $U$-plane, focussing on the simpler 'toric' cases. These are the additional effective couplings that arise in addition to the prepotential in the low-energy effective field theory on a non-trivial four-manifold, and are necessary for S-duality [108]. Our objective, there, is to verify explicitly the infrared prediction for these couplings, which is given in terms of the SW curve [37, 38, 108], by matching it to the microscopic prediction that follows from the asymptotic expansion of the Nekrasov partition function [109, 110] – see also [111–113]. This amounts to a 5d generalisation of some of the computations in a recent work by Manschot, Moore and Zhang [36].

This paper is organised as follows. In section 2, we review the geometric engineering of the $E_n$ 5d SCFTs in M-theory and their circle reduction in Type IIA, and we discuss the SW curves for the $D_{S^1}E_n$ theories using local mirror symmetry. In section 3, we discuss the rank-one SW geometries in terms of rational elliptic surfaces, and we explain how to find the flavour symmetry group in that context. In section 4, we illustrate our general approach to the $U$-plane by revisiting the 'classic' four-dimensional $\mathcal{N} = 2$ theories. In sections 5, 6, 7 and 8, we study the Coulomb branch of the $D_{S^1}E_n$ theories in full detail. In section 9, we discuss the gravitational couplings on the $U$-plane. Some useful additional material is relegated to various appendices.

## 2 $E_n$ SCFT from M-theory, local mirror and Seiberg-Witten geometry

In this section, we review the geometric engineering of the $E_n$ 5d SCFTs from M-theory on del Pezzo ($dP$) singularities. These are of course the simplest 5d SCFTs we could consider – the geometric engineering of general 5d SCFTs has attracted a lot of interest in recent years, see *e.g.* [76, 77, 114–145]. The circle compactification of the $E_n$ theory is described by Type

---

[6]One day after the first version of this paper appeared on the arXiv, another paper by Aspman, Furrer and Manschot appeared [104] which studies 4d $SU(2)$ gauge theories. Their results partially overlap with section 4 of this paper.

Table 3: Correspondence between $E_n$ SCFTs and del Pezzo surfaces. Here, $\mathrm{Bl}_k(\mathcal{B}_4)$ denotes the blow-up of the complex surface $\mathcal{B}_4$ at $k$ generic points. Note that $dP_1$ can also be viewed as the Hirzebruch surface $\mathbb{F}_1$.

| $E_n$ | $E_0$ | $E_1$ | $\widetilde{E}_1$ | $E_n \, (n = 2, \cdots, 8)$ |
|---|---|---|---|---|
| $\mathcal{B}_4$ | $\mathbb{P}^2$ | $\mathbb{F}_0 \cong \mathbb{P}^1 \times \mathbb{P}^1$ | $\mathbb{F}_1 \cong dP_1 = \mathrm{Bl}_1(\mathbb{P}^2)$ | $dP_n = \mathrm{Bl}_n(\mathbb{P}^2) \cong \mathrm{Bl}_{n-1}(\mathbb{F}_0)$ |

IIA string theory on the same $dP_n$ singularity, and the local mirror description in Type IIB gives us the Seiberg-Witten geometry we are interested in. After reviewing some standard facts about families of elliptic curves and Seiberg-Witten geometry, we discuss the $E_n$ Seiberg-Witten curves and derive the Picard-Fuchs (PF) equations satisfied by their periods. We also review the relation between the 5d $E_n$ theories and the 4d MN theories.

## 2.1 Geometric engineering at del Pezzo singularities

We are interested in the small family of 10 distinct rank-one 5d SCFTs with flavour symmetry algebra $E_n$ [13,14], namely:

$$
\begin{aligned}
&E_0 = \varnothing, &&E_2 = \mathfrak{su}(2) \oplus \mathfrak{u}(1), &&E_5 = \mathfrak{so}(10), \\
&\widetilde{E}_1 = \mathfrak{u}(1), &&E_3 = \mathfrak{su}(3) \oplus \mathfrak{su}(2), &&E_n = \mathfrak{e}_n \ \ (n = 6,7,8). \\
&E_1 = \mathfrak{su}(2), &&E_4 = \mathfrak{su}(5),
\end{aligned}
\tag{11}
$$

These 5d fixed points are all related to each other by five-dimensional RG flows, starting from the $E_8$ model and breaking down the flavour symmetry to $E_{n<8}$ by appropriate real-mass deformations [13,14,146]. In this section, we will only discuss the flavour symmetry *algebra* of the $E_n$ theories. The global form of the flavour symmetry was recently derived in [33]. Those results also follow from the Coulomb-branch analysis of the KK theory, as we will explain in section 3.

   These rank-1 5d SCFTs can be 'geometrically engineered' as the low-energy limit of M-theory on $\mathbb{R}^5 \times \mathbf{X}_{E_n}$, where the $\mathbf{X}_{E_n}$ is a canonical singularity that admits a crepant resolution with a single exceptional divisor [14,43]. Let $\mathcal{B}_4$ denote a Fano surface – that is, either a del Pezzo surface or the Hirzebruch surface $\mathbb{F}_0 \cong \mathbb{P}^1 \times \mathbb{P}^1$. We consider the local Calabi-Yau threefold obtained as the total space of the canonical line bundle over $\mathcal{B}_4$:

$$
\widetilde{\mathbf{X}}_{E_n} \cong \mathrm{Tot}\left(\mathcal{K} \to \mathcal{B}_4\right).
\tag{12}
$$

By blowing down the zero section, one obtains the canonical singularity $\mathbf{X}_{E_n}$. The correspondence between del Pezzo surfaces and $E_n$ theories is summarized in table 3.

   The smooth threefold (12) provides a crepant resolution of $\mathbf{X}_{E_n}$, which corresponds physically to going onto the extended Coulomb branch (ECB) of the 5d SCFT, by turning on the real Coulomb branch VEV, $\langle \varphi \rangle \neq 0$, as well as $n$ real mass parameters $m_i$ $(i = 1, \cdots, n)$. The $n$ real masses should be understood as VEVs for real scalars in vector multiplets valued in the Cartan subalgebra $\oplus_{i=1}^n \mathfrak{u}(1)$ of $E_n$. In the M-theory geometric point of view, the full ECB is identified with the extended Kähler cone of $\widetilde{\mathbf{X}}_{E_n}$ [43]. The $E_n$ symmetry at the fixed point arises because of M2-branes wrapping vanishing curves. Indeed, it is a beautiful mathematical fact that the second homology lattice of $dP_n$ can be decomposed as:

$$
H_2(\mathcal{B}_4, \mathbb{Z}) \cong \Lambda_{-\mathcal{K}} \oplus E_n^-,
\tag{13}
$$

with $\Lambda_{\mathcal{K}} \cong \mathbb{Z}$ generated by a choice of anticanonical divisor, $-\mathcal{K}$, of $\mathcal{B}_4$. Here, $E_n^-$ denotes 'minus' the $E_n$ root lattice,[7] which is generated by the curves orthogonal to $-\mathcal{K}$. As reviewed

---

[7]That is, with an intersection pairing that is minus the Cartan matrix of $E_n$, in some appropriate basis.

in appendix A, one can pick a basis of curves, denoted by $\mathcal{C}_{\alpha_i}$, which are in one-to-one correspondence with the simple roots $\alpha_i$ of the flavour algebra $E_n$ and intersect according to its Dynkin diagram.

The $E_n$ fixed point is also the UV completion of a non-normalizable 5d gauge theory with $\mathcal{N} = 1$ supersymmetry, consisting of an $SU(2)$ vector multiplet coupled to $N_f = n - 1$ hyper-multiplets[8] with inverse gauge coupling $m_0 = 8\pi^2 g_{5d}^{-2}$ [13]. This gauge theory description is obtained by a mass deformation of the SCFT that breaks $E_n$ down to $\mathfrak{so}(2n-2) \oplus \mathfrak{u}(1)$:

$$
\begin{array}{ccc}
\text{[Dynkin diagram with } \alpha_3 \text{ branch, nodes } \alpha_1, \alpha_2, \alpha_4, \alpha_5, \cdots, \alpha_n] & \longrightarrow & \text{[Dynkin diagram with filled node } \mathfrak{u}(1), \alpha_3 \text{ branch, nodes } \alpha_2, \alpha_4, \alpha_5, \cdots, \alpha_n]
\end{array}
\tag{14}
$$

In the geometry, this corresponds to a partial resolution of the singularity. To describe the $SU(2)$, $N_f = n - 1$ gauge theory geometrically, one should pick a ruling of the exceptional divisor $\mathcal{B}_4$. For our purposes, this consists of a choice of 'fiber' and 'base' rational curves, $\mathcal{C}_f \cong \mathbb{P}^1$ and $\mathcal{C}_b \cong \mathbb{P}^1$ respectively. For $n = 1$, we have the Hirzebruch surfaces:

$$
\mathcal{C}_f \to \mathbb{F}_p \to \mathcal{C}_b, \qquad p = 0, 1.
\tag{15}
$$

The trivial ($p = 0$) or non-trivial ($p = 1$) fibration of $\mathcal{C}_f$ over $\mathcal{C}_b$ gives us the $SU(2)_0$ or $SU(2)_\pi$ gauge theory in the limit where the fiber curve collapses to a point; the M2-brane wrapping $\mathcal{C}_f$ gives the $SU(2)$ $W$-boson, and the M2-brane wrapping $\mathcal{C}_b$ gives the 5d instanton particle. For $n > 1$, we view $\mathcal{B}_4 = dP_n$ as the blow-up of $\mathbb{F}_0$ at $N_f = n - 1$ generic points. By a slight abuse of notation, we then denote by $\mathcal{C}_f$, $\mathcal{C}_b$ the same curves pulled back through the blow-down map $dP_n \to \mathbb{F}_0$. The $N_f$ exceptional curves are denoted by $E_i$, $i = 1, \cdots, n - 1$, and the corresponding wrapped M2-branes give us the hypermultiplets.

In this work, we are interested in the 5d SCFT compactified on a finite-size circle with radius $\beta$. This gives us a 4d $\mathcal{N} = 2$ supersymmetric theory 'of Kaluza-Klein (KK) type', which we denote by $D_{S^1} E_n$. By the M-theory/Type IIA duality, we can engineer the theory as the low-energy limit of Type IIA string theory on $\mathbb{R}^4 \times \mathbf{X}_{E_n}$:

$$
\begin{aligned}
D_{S^1} E_n \equiv E_n \text{ 5d SCFT on } \mathbb{R}^4 \times S^1 \qquad &\longleftrightarrow \quad \text{M-theory on } \mathbb{R}^4 \times S^1 \times \mathbf{X}_{E_n} \\
&\longleftrightarrow \quad \text{IIA on } \mathbb{R}^4 \times \mathbf{X}_{E_n}.
\end{aligned}
\tag{16}
$$

The Coulomb-branch physics of $D_{S^1} E_n$ is rather more subtle and interesting. This is due to quantum corrections, which kick in as soon as we compactify on a circle. In the geometric-engineering picture, we have worldsheet instanton corrections in Type IIA. Equivalently, in M-theory, we have to account for M2-branes wrapping $\mathcal{C} \times S^1$, with $\mathcal{C}$ some curve inside $\widetilde{\mathbf{X}}_{E_n}$.

Note that the 4d $\mathcal{N} = 2$ theory $D_{S^1} E_n$ is a massive theory, since we introduced the KK-scale $m_{\text{KK}} = 1/\beta$. For generic values of the parameters, this is an 'abstract' strongly coupled quantum field theory defined by the IIA geometry. In some particular limit on the Kähler parameters, called the geometric engineering limit [10, 21], we recover the 4d $\mathcal{N} = 2$ $SU(2)$ theory with $N_f$ flavours, at least when $N_f \leq 4$, and the Coulomb branch physics is then governed by the celebrated Seiberg-Witten solution [1, 2]. (We will review what happens for $N_f > 4$ at the end of this section.) More generally, the 5d gauge theory description remains useful for $m_0 \gg m_{\text{KK}}$ [12, 19].

---

[8]5d $\mathcal{N} = 1$ $SU(2)$ with $N_f = n - 1$, for short. For $n = 1$, we have $SU(2)$ with $\theta$ angle 0 or $\pi$, corresponding to $E_1$ or $\widetilde{E}_1$, respectively. The $E_0$ fixed point does not have a gauge theory interpretation but can be obtained as a deformation of the $\widetilde{E}_1$ theory [14].

## 2.2 Introducing the $U$-plane: a gauge theory perspective

One can gain some useful intuition about the Coulomb-branch physics of $D_{S^1}E_n$ from the 5d gauge-theory description [19]. Firstly and most importantly, the Coulomb branch is a one-complex dimensional variety because the 5d real scalar $\sigma$ in the abelian vector multiplet for $U(1) \subset SU(2)$ is paired with the $U(1)$ holonomy along the circle. Let us then define the dimensionless scalar:

$$a = i\left(\beta\sigma + iA_5\right), \qquad A_5 \equiv \frac{1}{2\pi}\int_{S^1} A_M dx^M. \qquad (17)$$

The classical $SU(2)$ Coulomb branch is then of the form $(\mathbb{R} \times S^1)/\mathbb{Z}_2$, which is spanned by $a \in \mathbb{C}$ modulo $a \to -a$ (the $SU(2)$ Weyl group action) and $a \to a + 1$ (the five-dimensional large gauge tranformations along $S^1$). It will be useful to parameterize the Coulomb branch in a gauge invariant way, as:

$$U = e^{2\pi i a} + e^{-2\pi i a}. \qquad (18)$$

This corresponds to the classical expectation value of a five-dimensional supersymmetric Wilson line in the fundamental representation of $SU(2)$, wrapping the circle:

$$U \equiv \langle W \rangle, \qquad W \equiv \mathrm{Tr}\, \mathrm{P}\exp\left(i\int_{S^1}(A - i\sigma d\psi)\right). \qquad (19)$$

For each $U(1)_i \subset E_n$ symmetry on the ECB, we similarly introduce the complexified flavour parameters:

$$\nu_i = i\left(\beta m_i^{(F)} + iA_{i,5}^{(F)}\right), \qquad M_{Fi} \equiv e^{2\pi i \nu_i}, \qquad (20)$$

which include flavour Wilson lines along the circle.

Secondly, the classical relation (18) will be modified by quantum corrections. Let us consider (19) as the intrinsic definition of $U$, valid in the full quantum theory. Recall that the 4d $\mathcal{N} = 2$ low-energy description on the CB is fully determined, in flat space, by some holomorphic prepotential $\mathcal{F}(a)$. In particular, we have the effective gauge coupling:

$$\tau = \frac{\partial^2 \mathcal{F}(a)}{\partial a^2}, \qquad (21)$$

at any given point on the Coulomb branch. The challenge is then to write down the low-energy parameter $a$ in terms of the VEV $U$ in (19). More generally, we will have:

$$a = a(U, M_F), \qquad (22)$$

including a dependence on the flavour parameters $M_{Fi}$. Then, (21) gives us the effective gauge coupling on the CB as a function of $U$ and $M_{Fi}$.

At large distance on the CB, namely for $U \to \infty$, one can compute the prepotential at the one-loop order similarly to the 4d gauge-theory case, by resumming the KK towers [19]. For $SU(2)$ with $N_f$ flavours, one finds:

$$\begin{aligned}
\mathcal{F} &= \mathcal{F}_0 + \frac{2}{(2\pi i)^3}\mathrm{Li}_3\left(e^{4\pi i a}\right) - \frac{1}{(2\pi i)^3}\sum_{i=1}^{n-1}\sum_{\pm}\mathrm{Li}_3\left(e^{2\pi i(\pm a + \mu_i)}\right) \\
&\approx \mathcal{F}_0 + \frac{2}{(2\pi i)^3}\mathrm{Li}_3\left(\frac{1}{U^2}\right) - \frac{1}{(2\pi i)^3}\sum_{i=1}^{n-1}\mathrm{Li}_3\left(\frac{1}{U}\right),
\end{aligned} \qquad (23)$$

with $\mathcal{F}_0 = \frac{1}{2}\mu_0 a^2$ a classical contribution, and the trilogarithms arise at one-loop. Here, the $\mu_i$'s are the complexified masses of the $n-1$ fundamental hypermultiplets, and we assumed $|a| \gg |\mu_i|$ on the second line. The mass parameters $\mu_0$, $\mu_i$ are related to the parameters $\nu_i$ in (20) in a specific way, as explained below and in appendix A.

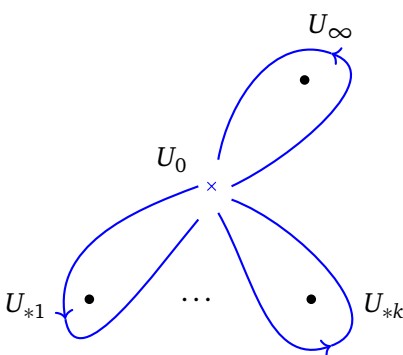

Figure 1: Paths $\gamma_v$ generating the fundamental group of the $U$-plane. The path around infinity is equal to minus the sum of all the other paths, $\gamma_\infty = -(\gamma_1 + \cdots + \gamma_k)$.

### 2.3   Monodromies, periods and Seiberg-Witten geometry

The low-energy scalar field $a$ is not a single-valued function of the parameter $U$. This is already true, in a somewhat trivial way, in the large distance approximation, where we have:

$$a = \frac{1}{2\pi i} \log\left(\frac{1}{U}\right) + \mathcal{O}\left(\frac{1}{U}\right). \tag{24}$$

The presence of a logarithmic branch cut is equivalent to the statement that $a$ and $a+1$ are gauge equivalent. More importantly, the effective gauge coupling itself is not single-valued. As we go around the point at infinity, $U^{-1} = 0$, we have:

$$U^{-1} \to e^{2\pi i} U^{-1} \qquad : \qquad \tau \to \tau + 9 - n , \tag{25}$$

which follows from (23). This gives us a shift of the effective 4d $\theta$-angle by $2\pi b_0$, with $b_0$ the $\beta$-function coefficient of the 5d gauge theory [13]:

$$b_0 = 8 - N_f = 9 - n . \tag{26}$$

In the interior of the $U$-plane, one should then have more singularities, around which the effective gauge coupling $\tau$ transforms by some non-trivial elements of $\mathrm{PSL}(2,\mathbb{Z})$, exactly like in the case of purely four-dimensional $SU(2)$ theories [1,2]. Such singularities are physically allowed because of the electric-magnetic duality of the 4d $\mathcal{N} = 2$ abelian vector multiplet. Let $a_D$ denote the scalar field magnetic dual to $a$. It can be obtained in terms of the effective prepotential by:

$$a_D = \frac{\partial \mathcal{F}}{\partial a} . \tag{27}$$

Semi-classically, at large distance on the $U$-plane, the field $a_D$ describes a BPS monopole. The low-energy effective theory is fully determined by the data of a section $(a_D, a)$ of a rank-two holomorphic affine bundle[9] over the $U$-plane, with structure group $\mathbb{C}^2 \rtimes \mathrm{SL}(2,\mathbb{Z})$, such that the effective gauge coupling is given by:

$$\tau = \frac{\partial a_D}{\partial a} . \tag{28}$$

---

[9]This is an affine bundle instead of a vector bundle because of the presence of masses, as we will discuss momentarily. For now, let us focus on the $\mathrm{SL}(2,\mathbb{Z})$ part of the structure group.

The low-energy scalars $a$ and $a_D$ are called the electric and magnetic 'periods', respectively. As we go around any singularity $U = U_*$ on the $U$-plane (including the point at infinity) in a clockwise fashion, the periods transform as:

$$\begin{pmatrix} a_D \\ a \end{pmatrix} \rightarrow \mathbb{M}_* \begin{pmatrix} a_D \\ a \end{pmatrix}, \qquad \mathbb{M}_* \in \mathrm{SL}(2, \mathbb{Z}). \tag{29}$$

The $\mathrm{SL}(2, \mathbb{Z})$ matrix $\mathbb{M}_*$ is the so-called monodromy around that point. Let us denote the $k + 1$ singularites on the $U$ plane (including the point at infinity, $U_\infty = \infty$) by $\widehat{\Delta} = \{U_{*1}, \cdots, U_{*k}, U_\infty\}$, and let:

$$\mathcal{M}_C = \{U\} - \widehat{\Delta} \tag{30}$$

be the Coulomb branch with its singular points removed. Given one of our rank-one theories with fixed mass parameters $M_{Fi}$, the quantum Coulomb branch data is an affine bundle $E$ with $\mathbb{C}^2$ fibers:

$$\mathbb{C}^2 \lhook\joinrel\longrightarrow E \xrightarrow{\pi} \mathcal{M}_C. \tag{31}$$

By definition, the monodromy group at some base point $U_0$ is a representation of the fundamental group $\pi_1(\mathcal{M}_C, U_0)$ on the fiber $\mathbb{C}^2 \cong \pi^{-1}(U_0)$. It is generated by the matrices $\mathbb{M}_{*l}$, for some convenient choice of base point and of paths $\gamma_\nu$, where each 'vanishing path' goes once along a single singularity as shown in figure 1. We then have the obvious constraint:

$$\mathbb{M}_\infty \prod_{l=1}^{k} \mathbb{M}_{*l} = \mathbf{1}. \tag{32}$$

A good part of paper is dedicated to a thorough study of this elementary structure for the $D_{S^1} E_n$ theories. In particular, we would like to give a detailed account of the Coulomb branch singularities, and of the corresponding low-energy physics.

The modular group $\mathrm{SL}(2, \mathbb{Z})$ is generated by the two matrices:

$$S = \begin{pmatrix} 0 & -1 \\ 1 & 0 \end{pmatrix}, \qquad T = \begin{pmatrix} 1 & 1 \\ 0 & 1 \end{pmatrix}. \tag{33}$$

Let us also denote by $P = S^2 = -\mathbf{1}$ the generator of the $\mathbb{Z}_2$ center of $\mathrm{SL}(2, \mathbb{Z})$. The monodromy at $U = \infty$ can be computed from (23) and (24), which gives:

$$a_D \rightarrow a_D + (9 - n)a + \mu_0 - \sum_{i=1}^{n-1} \mu_i, \qquad a \rightarrow a + 1. \tag{34}$$

We then have the following $\mathrm{SL}(2, \mathbb{Z})$ monodromy at infinity:

$$\mathbb{M}_\infty = T^{9-n} = \begin{pmatrix} 1 & 9-n \\ 0 & 1 \end{pmatrix}. \tag{35}$$

Note that this is tied to the Witten effect [147]: a shift of the 4d $\theta$-angle as in (25) induces an electric charge for the monopole, turning it into a dyon.

### 2.3.1 Central charge, massless BPS particles and $T^k$ monodromies

Half-BPS massive particle excitations on the Coulomb branch of $D_{S^1} E_n$ have a mass determined by their electromagnetic and flavour charges:

$$\gamma \in \Gamma \cong \mathbb{Z}^{n+3}, \qquad \gamma \cong (m, q, q_F^i, n_{\mathrm{KK}}), \tag{36}$$

according to the central-charge formula, $m_\gamma = |Z_\gamma|$. The integer-quantized charges consist of the magnetic and electric charges, $(m, q)$, the $E_n$ flavour charges $q_F^i$, and the KK momentum $n_{KK}$ [26]. Using the KK scale as the unit of mass, let us define the dimensionless central charge $\mathcal{Z} \equiv \beta Z$. At any given point on the extended Coulomb branch, the central charge is a map $\mathcal{Z} : \Gamma \to \mathbb{C}$, which is given explicitly by:

$$\mathcal{Z}_\gamma(U, M) = \mathbf{q}\,\Pi = ma_D + qa + q_F^i \mu_i + n_{KK}, \tag{37}$$

in terms of the electromagnetic periods. The parameters $\mu_i$ and $\mu_{KK}=1$ are 'exact periods', as we will review below. Around any singularity on the $U$-plane, we have an enlarged monodromy of the form:

$$\Pi \to \mathbf{M}_* \Pi, \qquad \Pi \equiv \begin{pmatrix} a_D \\ a \\ \mu_i \\ 1 \end{pmatrix}, \qquad \mathbf{M}_* = \begin{pmatrix} m_{11} & m_{12} & n_1^i & n_1^0 \\ m_{21} & m_{22} & n_2^i & n_2^0 \\ 0 & 0 & 1 & 0 \\ 0 & 0 & 0 & 1 \end{pmatrix}, \tag{38}$$

with $m_{11}m_{22} - m_{12}m_{21} = 1$, the upper-left corner of $\mathbf{M}_*$ being the electromagnetic monodromy (29). Note that the monodromy can be equivalently understood as acting on the electromagnetic and flavour charges as:

$$\mathbf{q} \to \mathbf{q}\mathbf{M}_*, \qquad \mathbf{q} \equiv (m, q, q_F^i, n_{KK}). \tag{39}$$

In general, we keep to the 'electric' duality frame dictated by the non-abelian gauge theory limit (or, more precisely, the large volume limit, as we will see below). The local infrared physics is invariant under electric-magnetic duality, however. When analysing the physics at a given point on the $U$-plane, it can be convenient to change the duality frame. This change of basis leaves the central charge invariant and therefore acts on the charges and periods as:

$$\mathbf{q} \to \mathbf{q}\mathbf{B}^{-1}, \qquad \Pi \to \mathbf{B}\Pi, \tag{40}$$

with $\mathbf{B}$ the basis-change matrix.

The simplest type of singularity that can occur in the interior of the $U$-plane is when a single charged particle becomes massless. In the appropriate duality frame, the low-energy physics at that point is then governed by SQED, namely a $U(1)$ gauge field coupled to a single massless hypermultiplet of charge 1, denoted by $\tilde{a}$. Let us assume that a dyon of electromagnetic charge $(m, q)$ becomes massless at $U_*$, with $m$ and $q$ mutually prime, so that $\tilde{a} = ma_D + qa$. Due to the $\beta$-function of SQED, the local monodromy is given by $T$, in the variables $(\tilde{a}_D, \tilde{a})^T = B(a_D, a)^T$, and it thus follows that a massless dyon at $U_*$ induces a monodromy:

$$\mathbb{M}_*^{(m,q)} = B^{-1}TB = \begin{pmatrix} 1 + mq & q^2 \\ -m^2 & 1 - mq \end{pmatrix}. \tag{41}$$

Any such singularity with a monodromy conjugate to $T$ is called an $I_1$ singularity. Similarly, we could have SQED with $k$ electrons (or some hypermultiplets of charges $q_j$ such that $\sum_j q_j^2 = k$), with a monodromy conjugate to $T^k$ [1, 2]. That is called an $I_k$ singularity. Other types of singularities are possible, as we will review momentarily.

For the $D_{S^1}E_n$ theory at generic values of the mass parameters $M_F$, there are $k = n + 3$ singularities of type $I_1$ in the interior of the $U$-plane, at each of which a single BPS particle becomes massless. This number of Seiberg-Witten points can be understood in various ways. From the perspective of local mirror symmetry, which we take below, $n + 3$ is the number of generators of the third homology of the Type IIB mirror threefold, which equals the total number of generators of the even homology of $\widetilde{\mathbf{X}}$. From the point of view of the 5d gauge theory, if we admit that the pure 5d $SU(2)$ gauge theory on a circle has 4 CB singularities [19], then adding $N_f = n - 1$ massive flavours adds $N_f$ singularities, which are semi-classical in some particular large-mass regime.

### 2.3.2 Seiberg-Witten geometry, Kodaira singularities and low-energy physics

For any rank-one 4d $\mathcal{N} = 2$ field theory, the physical problem is to find exact expressions for the electromagnetic periods $(a_D, a)$ such that the CB metric is positive definite, namely:

$$\text{Im}\,\tau = \text{Im}\,\frac{\partial a_D}{da} > 0\,, \qquad ds^2(\mathcal{M}_C) = \text{Im}\,\tau\, da d\bar{a}\,, \tag{42}$$

and which otherwise match the known asymptotics. The original Seiberg-Witten solution for 4d $\mathcal{N} = 2$ $SU(2)$ gauge theories was obtained by realising that, given some physical ansatz for the singularities and monodromies on the Coulomb branch, a positive-definite metric can be elegantly obtained by viewing the low-energy fields $(a_D, a)$ as the periods of a meromorphic one-form, $\lambda_{\text{SW}}$, on a family of elliptic curves [1,2].

Let us review some useful facts about families of elliptic curves, and their relationship to 4d physics. At fixed mass parameters, we wish to consider a one parameter-family of elliptic curves, which we generally call 'the Seiberg-Witten geometry':

$$\Sigma \to \mathcal{S}_{\text{CB}} \to \overline{\mathcal{M}}_C \cong \{U\}\,. \tag{43}$$

Here, $\mathcal{S}_{\text{CB}}$ denotes a one-parameter family of elliptic curve over the $U$-plane, including the singularities. At each smooth point $U \in \mathcal{M}_C$ on the Coulomb branch, we have an elliptic curve $\Sigma_U \cong E$. One then identifies $\tau(U)$ with the modular parameter of that curve. The latter is computed as $\tau = \frac{\omega_D}{\omega_a}$, where $\omega_D$ and $\omega_a$ are the periods of the holomorphic one-form $\boldsymbol{\omega}$ along the $A$- and $B$-cycles in $\Sigma_U$:

$$\omega_D = \int_{\gamma_B} \boldsymbol{\omega}\,, \qquad \omega_a = \int_{\gamma_A} \boldsymbol{\omega}\,. \tag{44}$$

We call these periods the 'geometric periods'. The holomorphic one-form of an elliptic curve is unique up to rescaling. The Seiberg-Witten differential $\lambda_{\text{SW}}$ is a meromorphic one-form such that:

$$\frac{d\lambda_{\text{SW}}}{dU} = \boldsymbol{\omega}\,, \tag{45}$$

modulo an exact 1-form. The 'physical periods' are then defined as:

$$a_D = \int_{\gamma_B} \lambda_{\text{SW}}\,, \qquad a = \int_{\gamma_A} \lambda_{\text{SW}}\,. \tag{46}$$

We then indeed have:

$$\omega_D = \frac{da_D}{dU}\,, \qquad \omega_a = \frac{da}{dU}\,, \qquad \tau = \frac{\omega_D}{\omega_a} = \frac{\partial a_D}{\partial a}\,. \tag{47}$$

The SW curve of $D_{S^1}E_n$, similarly to the case of the massive 4d $SU(2)$ gauge theories [2], can be viewed, perhaps more precisely, as a genus-one Riemann surface with (generically) $n + 1$ punctures, where the SW differential has simple poles with residues given by the masses (or 'flavour periods') $\mu_i$ and $\mu_{\text{KK}}$. For our purpose in this work, however, we can mainly bypass an explicit determination of the SW differential. It will often be enough to determine the geometric periods before using (47) to determine the electromagnetic periods up to integration constants. The latter will be fixed by matchings to known asymptotics.

Table 4: Kodaira classification of singular fibers and associated 4d low-energy physics. The $I_k$ fibers are also called 'multiplicative' or 'semi-stable' fibers. ($I_0$ is the 'stable' generic smooth fiber.) All the other types of fibers are called 'additive' or 'unstable'.

| fiber | $\tau$ | ord($g_2$) | ord($g_3$) | ord($\Delta$) | $\mathbb{M}_*$ | 4d physics | $\mathfrak{g}$ flavour |
|---|---|---|---|---|---|---|---|
| $I_k$ | $i\infty$ | 0 | 0 | $k$ | $T^k$ | SQED | $\mathfrak{su}(k)$ |
| $I_k^*$ | $i\infty$ | 2 | 3 | $k+6$ | $PT^k$ | $SU(2), N_f = 4+k > 4$ | $\mathfrak{so}(2k+8)$ |
| $I_0^*$ | $\tau_0$ | $\geq 2$ | $\geq 3$ | 6 | $P$ | $SU(2), N_f = 4$ | $\mathfrak{so}(8)$ |
| $II$ | $e^{\frac{2\pi i}{3}}$ | $\geq 1$ | 1 | 2 | $(ST)^{-1}$ | $AD[A_1, A_2] = H_0$ | - |
| $II^*$ | $e^{\frac{2\pi i}{3}}$ | $\geq 4$ | 5 | 10 | $ST$ | MN $E_8$ | $\mathfrak{e}_8$ |
| $III$ | $i$ | 1 | $\geq 2$ | 3 | $S^{-1}$ | $AD[A_1, A_3] = H_1$ | $\mathfrak{su}(2)$ |
| $III^*$ | $i$ | 3 | $\geq 5$ | 9 | $S$ | MN $E_7$ | $\mathfrak{e}_7$ |
| $IV$ | $e^{\frac{2\pi i}{3}}$ | $\geq 2$ | 2 | 4 | $(ST)^{-2}$ | $AD[A_1, D_4] = H_2$ | $\mathfrak{su}(3)$ |
| $IV^*$ | $e^{\frac{2\pi i}{3}}$ | $\geq 3$ | 4 | 8 | $(ST)^2$ | MN $E_6$ | $\mathfrak{e}_6$ |

**Weierstrass model and $J$-invariant.** All the SW curves considered in this work can be brought to the standard Weierstrass form:

$$y^2 = 4x^3 - g_2(u)x - g_3(u),\qquad(48)$$

by a change of coordinates.[10] Here, $u = (U, M, \cdots)$ denotes local coordinates on the base of the elliptic fibration. The discriminant of the curve is the function:

$$\Delta(u) \equiv g_2(u)^3 - 27g_3(u)^2.\qquad(49)$$

At fixed mass parameters, the roots of the polynomial $\Delta(U)$ give us the locations of the $U$-plane singularities. It is also very useful to consider the $J$-invariant:

$$J(u) = \frac{1}{1728}j(u) = \frac{g_2(u)^3}{\Delta(u)},\qquad(50)$$

which is an absolute invariant of the curve (while $g_2$ and $\Delta$ depend on the choice of variables). Importantly, $J$ is a modular function when written in terms of the modular parameter $\tau$. It takes the following universal form:

$$J(\tau) = \frac{E_4(\tau)^3}{E_4(\tau)^3 - E_6(\tau)^2},\qquad(51)$$

in terms of Eisenstein series. We refer to appendix B for a review of useful facts about modular groups and modular forms. Let us note that the zeroes of the Eisenstein series on the canonical fundamental domain of the upper half-plane are at $\tau = \zeta_3$ and $\tau = i$ for $E_4$ and $E_6$, respectively, with $\zeta_3 = e^{\frac{2\pi i}{3}}$. These are elliptic points for the SL$(2, \mathbb{Z})$ group. We have $J(\zeta_3) = 0$ and $J(i) = 1$.

---

[10]When viewing the SW curve as a compact curve, the Weierstrass equation can be read as the cubic $Y^2 = 4X^3 - g_2 X Z^4 - g_3 Z^6$ with $[X, Y, Z] = \mathbb{P}^2_{[2,3,1]}$. Here we are working on the patch $Z = 1$. In fact, even though we call $\Sigma_U$ 'an elliptic curve', we will remain somewhat agnostic about the precise mathematical definition. In some string-theory geometric engineering scenarios, it appears more natural to view the SW curve as an affine curve in $(\mathbb{C}^*)^2$ instead of a curve in projective space. These subtle differences of perspective will not affect our physical discussion. We used SAGE [148] to find the explicit Weierstrass form of various curves (for instance for the 'toric' mirror curves reviewed below or for various 4d curves from the literature).

**Kodaira classification and infrared physics.** The possible singularities of our Seiberg-Witten geometry are captured by the Kodaira classification of singular fibers. The singularity type can be read off from the Weierstrass form of the curve by looking at the order of vanishing at $U = U_*$ of $g_2$, $g_3$ and of the discriminant:

$$g_2 \sim (U - U_*)^{\text{ord}(g_2)}, \qquad g_3 \sim (U - U_*)^{\text{ord}(g_3)}, \qquad \Delta \sim (U - U_*)^{\text{ord}(\Delta)}. \tag{52}$$

The different types of fibers, in Kodaira's notation, are listed in table 4. This gives us a crucial tool to identify the types of singularities in the low-energy physical description, given the Seiberg-Witten geometry [11].[11]

We already discussed the $I_k$ fibers. In the context of the 5d $E_n$ SW curves, we have an $I_{9-n}$ singularity at $U = \infty$, as shown in (35). We can also have an $I_k$ singularity in the interior of the $U$-plane, which is interpreted as $k$ BPS particles of the same charge becoming massless. The local physics at the singularity is then that of massless SQED with $k$ electrons, which is an IR-free theory. This is consistent with the fact that the effective inverse gauge coupling is $\tau = i\infty$ at that point. This theory has a Higgs branch which is isomorphic to the moduli space of one $SU(k)$ instanton.[12] Therefore, there is a 'quantum Higgs branch' emanating from such a point on the $U$-plane and, in particular, there is an $\mathfrak{su}(k)$ flavour symmetry associated to this type of singularity.

The second and third type of singularity in table 4, called $I_k^*$, has a monodromy conjugate to $PT^k$. The low-energy physics is that of a 4d $\mathcal{N} = 2$ $SU(2)$ gauge theory with $N_f = 4 + k$ flavours, which is IR-free for $k > 0$, and conformal for $k = 0$. Its Higgs branch is the moduli space of one $SO(8 + 2k)$ instanton, and the flavour symmetry algebra is $\mathfrak{so}(8 + 2k)$.

The Kodaira singularities of type $II$, $III$ and $IV$ realise the three 'classic' rank-one Argyres-Douglas theories [3,4]. These are non-trivial 4d $\mathcal{N} = 2$ SCFTs with fractional scaling dimensions for the Coulomb branch operator ($\frac{6}{5}$, $\frac{4}{3}$ and $\frac{3}{2}$, respectively). The flavour symmetry of the $H_0$ theory (Kodaira fiber $II$) is trivial, while the flavour symmetry of the $H_1$ and $H_2$ theory is $\mathfrak{su}(2)$ and $\mathfrak{su}(3)$, respectively. The latter two have a Higgs branch which is the moduli space of one $SU(2)$ or $SU(3)$ instanton, respectively.

Finally, the Kodaira singularities of type $II^*$, $III^*$ and $IV^*$ correspond to the $E_n$ Minahan-Nemeschansky theories for $n = 6, 7, 8$ [5,6], as indicated in the table. These 4d SCFTs have a Higgs branch isomorphic to the moduli space of one $E_n$ instanton.

**Picard-Fuchs equations.** Consider a one-parameter families of curves, $\Sigma_U$, by setting the various mass parameters to definite values. We consider the Weierstrass form (48), with $g_2(U)$ and $g_3(U)$ some polynomials in $U$, and we would like to determine the geometric periods:

$$\omega = \int_\gamma \boldsymbol{\omega}, \qquad \boldsymbol{\omega} = \frac{dx}{y}, \tag{53}$$

with $\gamma$ any one-cycle $\gamma \in H_1(\Sigma_U)$. These periods satisfy a second-order linear differential equation, the Picard-Fuchs equation, which reads:

$$\Delta(U) \frac{d^2\omega}{dU^2} + P(U) \frac{d\omega}{dU} + Q(U) \omega = 0, \tag{54}$$

---

[11]In this work, we will only consider fibers that are 'split' in the sense of [9] – see also [149].

[12]This follows, for instance, from compactification to 3d together with 3d $\mathcal{N} = 4$ mirror symmetry [150].

with $P(U)$ and $Q(U)$ determined in terms of $g_2$ and $g_3$:

$$
\begin{aligned}
P(U) &= \frac{27g_3 J^2}{g_2^4 J'}\Big(-7g_2{}^3 g_2' g_3' + 9g_2{}^2 g_2'^2 g_3 + 108 g_2 g_3 g_3'^2 - 135 g_2' g_3{}^2 g_3' \\
&\qquad + 2g_2{}^4 g_3'' - 3g_2{}^3 g_2'' g_3 - 54 g_2 g_3{}^2 g_3'' + 81 g_2'' g_3{}^3\Big), \\
Q(U) &= \frac{27 g_3 J^2}{16 g_2^4 J'}\Big(-8g_2{}^3 g_2'' g_3' + 8g_2{}^3 g_2' g_3'' + 216 g_2'' g_3{}^2 g_3' - 216 g_2' g_3{}^2 g_2'' \\
&\qquad - 18 g_2{}^2 g_2'^2 g_3' + 21 g_2 g_2'^3 g_3 + 120 g_2 g_3'^3 - 108 g_2' g_3 g_3'^2\Big).
\end{aligned}
\tag{55}
$$

Here, $J = J(U)$ is defined as in (50), and $f' \equiv \frac{df}{dU}$ for any $f(U)$. By direct computation, one can check that (54) is equivalent to the following universal PF equation for $\Omega(J)$, a function of the $J$-invariant itself [151, 152]:

$$
\frac{d^2\Omega}{dJ^2} + \frac{1}{J}\frac{d\Omega}{dJ} + \frac{31J - 4}{144J^2(1-J)^2}\Omega = 0.
\tag{56}
$$

This $\Omega$ is related to the geometric period $\omega$ by:

$$
\omega(U) = \sqrt{\frac{g_2(U)}{g_3(U)}}\,\Omega(J(U)).
\tag{57}
$$

The solutions to this universal equation are known. In particular, we then have the following expression for the geometric periods $\omega_a$:

$$
\omega_a = \frac{da}{dU} = C_0 \sqrt{\frac{g_2(U)}{g_3(U)}}\sqrt{\frac{E_6(\tau)}{E_4(\tau)}},
\tag{58}
$$

with $C_0$ some normalization constant to be determined. In practice, one can then obtain a useful expression for $\omega_a(U)$ by writing down $\tau(U)$ around $\tau = i\infty$, at any given order. To do this, consider the series expansion of the $J$-invariant:

$$
j(\tau) = 1728 J(q) = \frac{1}{q} + 744 + 196884\,q + 21493760\,q^2 + \mathcal{O}\!\left(q^3\right),
\tag{59}
$$

with $q \equiv e^{2\pi i \tau}$, and its inverse:

$$
\tau(j) = \frac{1}{2\pi i}\left(-\log(j(U)) + \frac{744}{j(U)} + \frac{473652}{j(U)^2} + \frac{451734080}{j(U)^3} + \mathcal{O}\!\left(\frac{1}{j(U)^4}\right)\right).
\tag{60}
$$

The dual period, $a_D$, can then be written as an expansion in that large-$U$ limit by using (47), at least in principle. Note also that the expression (59) can be re-expanded at large $U$ to obtain an explicit expression for $U(\tau)$. That is often quite useful, as we will see in examples.

## 2.4 Large-volume limit and mirror Calabi-Yau threefold

In the type-IIA description of the $D_{S^1}E_n$ Coulomb branch, the BPS particles are D-branes wrapping holomorphic cycles inside the local del Pezzo geometry, at least semi-classically. (More generally, they are $\Pi$-stable objects in the derived category of coherent sheaves of $\widetilde{\mathbf{X}}_{E_n}$ [153].) The associated 'exact periods' are the 'quantum volumes' of the D0-, D2- and D4-branes. In the large volume limit, we have:

$$
\Pi_{\mathrm{D4}} = \int_{\mathcal{B}_4} e^{(B+iJ)} \mathrm{ch}(L_\varepsilon)\sqrt{\frac{\mathrm{Td}(T\mathcal{B}_4)}{\mathrm{Td}(N\mathcal{B}_4)}} + \mathcal{O}(\alpha'),
\tag{61}
$$

for the wrapped D4-brane. Here $J$ is the Kähler form, which is complexified by the $B$-field, and $L_\varepsilon$ is a (spin$^c$) line bundle, which must often be non-trivial [154]. The period of a D2-brane wrapped on any 2-cycle $\mathcal{C}^k \subset \mathcal{B}_4$ is given by the corresponding complexified Kähler parameter:

$$\Pi_{\mathrm{D2}_{\mathcal{C}^k}} = t_k \equiv \int_{\mathcal{C}^k} (B + iJ).$$ (62)

We also have $\Pi_{\mathrm{D0}} = 1$, the D0-brane being stable at any point on the Kähler moduli space. For $n > 0$, we view $dP_n$ as the blow up of $\mathbb{F}_0 \cong \mathcal{C}_f \times \mathcal{C}_b$ at $n-1$ points, with exceptional curves $E_i$ $(i = 1, \cdots, n-1)$ as mentioned above – see appendix A. We then choose a basis of Kähler parameters:

$$t_f = \int_{\mathcal{C}_f} (B + iJ), \qquad t_b = \int_{\mathcal{C}_b} (B + iJ), \qquad t_{E_i} = \int_{E_i} (B + iJ), \ i = 1, \cdots, n-1.$$ (63)

Note that these parameters are only defined up to shifts by integers, due to large gauge transformations of the $B$-field. For any curve $\mathcal{C}$, we also define the single-valued parameter:

$$Q_{\mathcal{C}} \equiv e^{2\pi i t_{\mathcal{C}}}.$$ (64)

Thus, the large Kähler volume limit for any effective curve $\mathcal{C}$ is equivalent to $Q_{\mathcal{C}} \to 0$.

Let $\{\mathcal{C}^k\}$ be some basis of $H_2(\mathcal{B}_4, \mathbb{Z})$, with the intersection pairing:

$$\mathcal{C}^k \cdot \mathcal{C}^l = \mathcal{I}^{kl}.$$ (65)

We also choose the worldvolume flux on the D4-brane to be:

$$\frac{1}{2\pi} \int_{\mathcal{C}^k} F = \varepsilon_k.$$ (66)

These fluxes must generally be non-zero and half-integer, due to the Freed-Witten anomaly cancellation condition [154]:

$$c_1(L_\varepsilon) + \frac{1}{2} c_1(\mathcal{K}) \in H^2(\mathcal{B}_4, \mathbb{Z}).$$ (67)

On the other hand, any integer-quantized flux on the D4-brane could be set to zero by a large gauge transformation of the $B$-field. The latter transformation corresponds to a large-volume monodromy. We then have:

$$\Pi_{\mathrm{D4}} = \frac{1}{2} \sum_{k,l} (t_k + \varepsilon_k) \mathcal{I}^{kl} (t_l + \varepsilon_l) + \frac{\chi(\mathcal{B}_4)}{24} + \mathcal{O}(\alpha').$$ (68)

Note that the parameters $\varepsilon_k$ just amount to shifting $t_k$ by some half-integers.

For the IIA geometries that are obtained by blowing up $\mathbb{F}_0$,[13] it will be convenient to choose another basis of Kähler parameters, denoted by $a$, $\mu_0$ and $\mu_i$ $(i = 1, \cdots, n-1)$, with:

$$t_f = 2a, \qquad t_b = 2a + \mu_0, \qquad t_{E_i} = a + \mu_i.$$ (69)

The parameter $a$ is the low-energy photon in the 'electric' frame. In the $SU(2)$ gauge-theory limit, the D2-brane wrapped on $\mathcal{C}_f$ is identified with the $W$-boson, and the factor of 2 in (69) corresponds to the '$SU(2)$' normalisation of the electric charge such that it has charge 2; similarly, the other identifications in (69) corresponds to the electric and flavour charges

---

[13]Thus, in all cases except for $E_0$ and $\widetilde{E}_1$, which we can treat separately.

of the other D2 particles, *i.e.* five-dimensional instanton particles and flavour hypermultiplets. Note that the parameters $\mu_0$, $\mu_i$ are pure flavour parameters, in that the corresponding (non-effective) curves $\mathcal{C}_\mu$ have vanishing intersection with the compact four-cycle $\mathcal{B}_4 \subset \widetilde{\mathbf{X}}$. Consequently, they lie along the $E_n^-$ lattice in (A.1). From (68), we then find:

$$\Pi_{\mathrm{D4}} = 2a(2a + \mu_0) - \frac{1}{2} \sum_{i=1}^{n-1} (a + \mu_i + \varepsilon_i)^2 + \frac{n+3}{24} + \mathcal{O}(Q), \tag{70}$$

where we chose $\varepsilon_f = \varepsilon_b = 0$. Once we identify the $W$-boson as coming from a D2-brane wrapping $\mathcal{C}_f$ (and, more generally, the 'electric' particles as being the wrapped D2-branes), then the wrapped D4-brane is identified with the magnetic monopole. We then have:

$$a_D = \frac{\partial \mathcal{F}}{\partial a} = \Pi_{\mathrm{D4}}, \tag{71}$$

and the large volume result (70) then corresponds to a prepotential:

$$\mathcal{F} = \left( \mu_0 - \frac{1}{2} \sum_{i=1}^{n-1} \widetilde{\mu}_i \right) a^2 + \frac{9-n}{6} a^3 + \left( \frac{n+3}{24} - \frac{1}{2} \sum_{i=1}^{n-1} \widetilde{\mu}_i^2 \right) a + \mathcal{O}(Q), \tag{72}$$

where we defined the shifted masses $\widetilde{\mu}_i \equiv \mu_i + \varepsilon_i$. This should be compared to the 5d prepotential for $SU(2)$ with $N_f = n-1$, which reads [43]:

$$\mathcal{F}_{\mathrm{5d}} = m_0 \sigma^2 + \frac{4}{3} \sigma^3 - \frac{1}{6} \sum_{i=1}^{n-1} \sum_{\pm} \Theta(\pm\sigma + m_i)(\pm\sigma + m_i)^3, \tag{73}$$

in the conventions of [121]. We indeed recover the 5d prepotential from (72), in the appropriate limit and in a specific Kähler chamber:

$$\mathcal{F}_{\mathrm{5d}} = \lim_{\beta \to \infty} \frac{i}{\beta^3} \mathcal{F}, \qquad \sigma > |m_j|, \quad j = 0, \cdots, n-1, \tag{74}$$

using the fact that $\mathrm{Im}(a) = \beta\sigma$ and $\mathrm{Im}(\mu_j) = \beta m_j$.

### 2.4.1 Local mirror symmetry for the toric models

For $n \leq 3$, the $E_n$ singularity in Type IIA is also a toric Calabi-Yau threefold. The corresponding toric diagrams are:

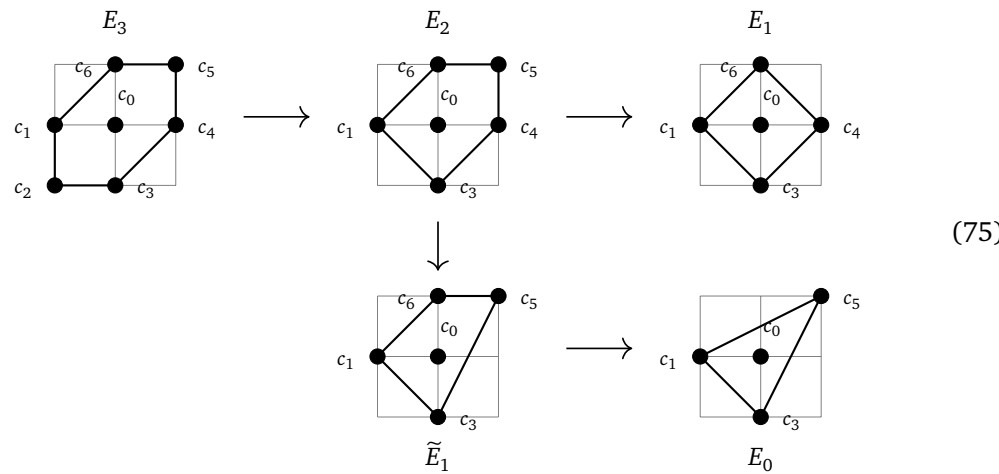

$$\tag{75}$$

Here, the arrows denote the possible partial resolutions of the singularities, which correspond to massive deformations of the 5d SCFTs. Let us then consider the $E_3$ singularity first, since the other toric singularities can be obtained from it by this partial resolution process. The internal point in the toric diagram, indicated by $c_0$ in (75), corresponds to the compact divisor $D_0 \cong \mathcal{B}_4 = dP_3$. Associated to each external point, indicated by $c_i$, $i = 1, \cdots, 6$, we have a non-compact toric divisor $D_i$ of the threefold, which intersects the compact divisor along curves $\mathcal{C}_i$ inside the resolved singularity, $\mathcal{C}_i \cong D_0 \cdot D_i$, The intersection numbers between toric divisors and curves are captured by the following table, which is equivalent to the data of a GLSM [155]:

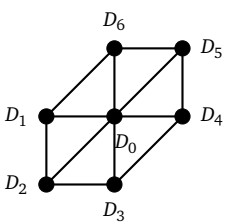

|       | $D_1$ | $D_2$ | $D_3$ | $D_4$ | $D_5$ | $D_6$ | $D_0$ | FI      |
|-------|-------|-------|-------|-------|-------|-------|-------|---------|
| $\mathcal{C}_1$ | $-1$  | $1$   | $0$   | $0$   | $0$   | $1$   | $-1$  | $\xi_1$ |
| $\mathcal{C}_2$ | $1$   | $-1$  | $1$   | $0$   | $0$   | $0$   | $-1$  | $\xi_2$ |
| $\mathcal{C}_3$ | $0$   | $1$   | $-1$  | $1$   | $0$   | $0$   | $-1$  | $\xi_3$ |
| $\mathcal{C}_5$ | $0$   | $0$   | $0$   | $1$   | $-1$  | $1$   | $-1$  | $\xi_5$ |
| $\mathcal{C}_4$ | $0$   | $0$   | $1$   | $-1$  | $1$   | $0$   | $-1$  | $\xi_4$ |
| $\mathcal{C}_6$ | $1$   | $0$   | $0$   | $0$   | $1$   | $-1$  | $-1$  | $\xi_6$ |

(76)

Note the linear equivalences $\mathcal{C}_4 \cong \mathcal{C}_1 + \mathcal{C}_2 - \mathcal{C}_5$ and $\mathcal{C}_6 \cong \mathcal{C}_2 + \mathcal{C}_3 - \mathcal{C}_5$. The triangulated toric diagram shown in (76) corresponds to the smooth local $dP_3$ geometry. The real parameters $\xi_i$ are the Kähler volumes of the curves $\mathcal{C}_i$ in the local threefold – they are the 'FI parameters' in the GLSM language. The Kähler cone is spanned by $(\xi_1, \xi_2, \xi_3, \xi_5) \in \mathbb{R}^4$ satisfying:

$$\xi_1 \geq 0, \quad \xi_2 \geq 0, \quad \xi_3 \geq 0, \quad \xi_5 \geq 0, \quad \xi_1 + \xi_2 - \xi_5 \geq 0, \quad \xi_2 + \xi_3 - \xi_5 \geq 0. \quad (77)$$

Other phases can be obtained by successive flops, therefore moving onto the extended Kähler cone of the singularity, which maps out the full extended Coulomb branch of the 5d SCFT $E_3$ [121]. Viewing $dP_3$ as the blow-up of $\mathbb{F}_0$ at two points, we have the natural basis of curves discussed in subsection 2.1: $\mathcal{C}_f$ and $\mathcal{C}_b$ are the 'fiber' and 'base' curves, respectively, and $E_1$ and $E_2$ are the two exceptional curves. This basis is related to the curves shown in (76) by:

$$\mathcal{C}_f = \mathcal{C}_1 + \mathcal{C}_2, \qquad \mathcal{C}_b = \mathcal{C}_2 + \mathcal{C}_3, \qquad E_1 = \mathcal{C}_5, \qquad E_2 = \mathcal{C}_2. \quad (78)$$

In the 5d $SU(2)$, $N_f = 2$ gauge-theory description, the M2-branes wrapped over $\mathcal{C}_f$ and $\mathcal{C}_b$ give us the W-boson and the instanton particle, respectively, while the M2-branes wrapped over $E_1$ or $E_2$ give rise to hypermultiplets.[14] The $E_1$ fixed point inself has an enhanced $E_3 = SU(3) \times SU(2)$ symmetry. The simple roots of $E_3$ are in one-to-one correspondence with the curves:

$$\mathcal{C}_{\alpha_1} = \mathcal{C}_b - \mathcal{C}_f, \qquad \mathcal{C}_{\alpha_2} = \mathcal{C}_f - E_1 - E_2, \qquad \mathcal{C}_{\alpha_3} = E_1 - E_2, \quad (79)$$

which intersect inside $dP_3$ according to the $E_3$ Cartan matrix. :

$$\mathcal{C}_{\alpha_i} \cdot \mathcal{C}_{\alpha_j} = -A_{ij} = \begin{pmatrix} -2 & 1 & 0 \\ 1 & -2 & 0 \\ 0 & 0 & -2 \end{pmatrix}. \quad (80)$$

Note that, using the 5d gauge-theory parameters (69), the Kähler parameters associated to the roots are:

$$t_{\alpha_1} = \mu_0, \qquad t_{\alpha_2} = -\mu_1 - \mu_2, \qquad t_{\alpha_3} = \mu_1 - \mu_2. \quad (81)$$

See appendix A for more details on the 5d gauge-theory parameterisation.

---

[14]We are following the analysis of [121], where the gauge theory description is read off from a 'vertical reduction' of the toric diagram.

**Local mirror description.** Let us now consider the mirror description of the extended Coulomb branch, as the complex structure deformations of the mirror threefold in IIB:

$$D_{S^1}E_n \quad \longleftrightarrow \quad \text{IIA on } \mathbb{R}^4 \times \widetilde{\mathbf{X}} \quad \longleftrightarrow \quad \text{IIB on } \mathbb{R}^4 \times \widehat{\mathbf{Y}}. \tag{82}$$

For any toric singularity, the local mirror threefold, $\widehat{\mathbf{Y}}$, is given by a hypersurface in $\mathbb{C}^2 \times (\mathbb{C}^*)^2$, with equation [22]:

$$\widehat{\mathbf{Y}} = \left\{ v_1 v_2 + F(t,w) = 0 \,\big|\, (v_1,v_2) \in \mathbb{C}^2, \ (t,w) \in (\mathbb{C}^*)^2 \right\}. \tag{83}$$

Here, $F(t,w)$ is the Newton polynomial associated with the toric diagram, which takes the general form:

$$F(t,w) = \sum_{m \in \Gamma_0} c_m t^{x_m} w^{y_m}, \tag{84}$$

where the sum runs over all the points in the toric diagram $\Gamma_0 \subset \mathbb{Z}^2$, with coordinates $(x_m, y_m)$. The coefficients $c_m$ are the complex structure parameters of the mirror, modulo the gauge equivalences:

$$F(t,w) \sim s_0 F(s_1 t, s_2 w), \qquad (s_0, s_1, s_2) \in (\mathbb{C}^*)^3. \tag{85}$$

Let us associate to each effective curve $\mathcal{C} \subset \widetilde{\mathbf{X}}$ a complexified Kähler parameter $Q_{\mathcal{C}} = e^{2\pi i t_{\mathcal{C}}}$ as in (64). Given a GLSM description of $\widetilde{\mathbf{X}}$, as in (76), the mirror parameter $z_{\mathcal{C}}$ is given by:

$$z_{\mathcal{C}} = \prod_{m \in \Gamma_0} (c_m)^{q^m}, \qquad q^m \equiv \mathcal{C} \cdot D_m. \tag{86}$$

Here, $D_m$ is the toric divisor associated to the point $m \in \Gamma_0$. This is normalized such that we have $z_{\mathcal{C}} \approx Q_f$ in the large volume limit, or equivalently:

$$t_f = \frac{1}{2\pi i} \log(z_f) + \mathcal{O}(z). \tag{87}$$

The hypersurface (83) is a so-called suspension of the affine curve:

$$\Sigma = \{F(t,w) = 0\} \subset (\mathbb{C}^*)^2, \tag{88}$$

and we may focus on the latter. One may view the threefold $\widehat{\mathbf{Y}}$ as a double fibration of $\Sigma$ and $\mathbb{C}^*$ over some complex plane $\{W\} \cong \mathbb{C}$, as we will review in section 3.1.2. The BPS particles arise from D3-branes wrapping supersymmetric 3-cycles which can be constructed explicitly in a standard way [17,23]. The exact periods are then given by the classical periods of the holomorphic 3-form on $\widehat{\mathbf{Y}}$, which can be reduced to a line integral along a one-cycle $\gamma = S_\gamma^3 \cap \Sigma$ on the curve $\Sigma$:

$$\Pi_\gamma = \int_{S_\gamma^3} \Omega = \int_\gamma \lambda_{\text{SW}}. \tag{89}$$

From these considerations, one finds the following Seiberg-Witten differential:

$$\lambda_{\text{SW}} = \log t \, \frac{dw}{w}, \tag{90}$$

up to an overall numerical constant.

**The $E_3$ curve.** The mirror curve for the local $dP_3$ geometry is given by:

$$F_{dP_3}(w, t) = \frac{1}{t}\left(c_1 + \frac{c_2}{w}\right) + \frac{c_3}{w} + c_6 w + c_0 + t\left(c_4 + c_5 w\right). \tag{91}$$

We denote by:

$$z_f = \frac{c_3 c_6}{c_0^2}, \qquad z_b = \frac{c_1 c_4}{c_0^2}, \qquad z_{E_1} = \frac{c_4 c_6}{c_5 c_0}, \qquad z_{E_2} = \frac{c_1 c_3}{c_2 c_0}, \tag{92}$$

the complex-structure parameters mirror to the Kähler volume of the curves (78). We find it useful to introduce the parameters $U$, $\lambda$, $M_1$ and $M_2$ such that:

$$z_f = \frac{1}{U^2}, \qquad z_b = \frac{\lambda}{U^2}, \qquad z_{E_1} = -\frac{1}{UM_1}, \qquad z_{E_2} = -\frac{1}{UM_2}. \tag{93}$$

Using the gauge freedom (85), we may set $c_3 = c_6 = 1$, $c_1 = c_4$, and choose $c_0 = -U$, so that the $E_3$ Seiberg-Witten curve reads:

$$E_3 \; : \qquad \frac{\sqrt{\lambda}}{t}\left(1 + \frac{M_2}{w}\right) + \frac{1}{w} + w - U + t\sqrt{\lambda}\left(1 + M_1 w\right) = 0. \tag{94}$$

The CB parameter $U$ is chosen such that:

$$U \approx \frac{1}{\sqrt{Q_f}} = e^{-2\pi i a}, \tag{95}$$

at large volume, while the mass parameters $\lambda, M_1, M_2$ are related to the 5d gauge parameters as by the mirror map:

$$\lambda = \frac{Q_b}{Q_f} = e^{2\pi i m_0}, \qquad M_i = -\frac{\sqrt{Q_f}}{Q_{E_i}} = e^{-2\pi i \widetilde{\mu}_i} = -e^{-2\pi i \mu_i}, \quad i = 1, 2, \tag{96}$$

setting $\varepsilon_i = \frac{1}{2}$ (mod 1) for the exceptional 2-cycles $E_i$ in $\mathcal{B}_4 \cong \mathrm{Bl}_{n-1}(\mathbb{F}_0)$. Here, $\lambda$ corresponds to the 5d gauge coupling, and $M_1$, $M_2$ correspond to the two hypermultiplet masses. These 'flavour' complex-structure parameters, which we will often call 'the masses' by a slight abuse of terminology, are such that the massless limit corresponds to $\lambda = M_i = 1$. Unlike the relation (95) between $U$ and $a$, which is corrected by worldsheet instantons from the IIA point of view, the large-volume relations (96) are exact in $\alpha'$, as is the case for any Kähler parameter $t_{\mathcal{C}}$ in $\widetilde{\mathbf{X}}$ Poincaré dual to a non-compact divisor.

**The $E_2$ curve.** Let us consider the successive 5d mass deformations shown in (75), to obtain the curves for the other toric $E_n$ singularities. To obtain the $E_2$ geometry, we need to flop the curve $\mathcal{C}_2 \subset dP_3$ and take it to large negative volume. This corresponds to the limit of large negative 5d mass, $m_2 \to -\infty$, which is the limit $M_2 \to 0$. This is equivalent to setting $c_2 = 0$ in (91). We then obtain the curve:

$$E_2 \; : \qquad \frac{\sqrt{\lambda}}{t} + \frac{1}{w} + w - U + t\sqrt{\lambda}\left(1 + M_1 w\right) = 0. \tag{97}$$

The 5d gauge-theory phase is $SU(2)$ with $N_f = 1$. The GLSM description of the $E_2$ toric geometry reads as follows:

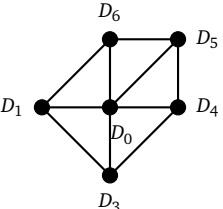

|  | $D_1$ | $D_3$ | $D_4$ | $D_5$ | $D_6$ | $D_0$ | FI |
|---|---|---|---|---|---|---|---|
| $\mathcal{C}_f = \mathcal{C}_1$ | 0 | 1 | 0 | 0 | 1 | $-2$ | $\xi_1$ |
| $\mathcal{C}_b = \mathcal{C}_3$ | 1 | 0 | 1 | 0 | 0 | $-2$ | $\xi_3$ |
| $E_1 = \mathcal{C}_5$ | 0 | 0 | 1 | $-1$ | 1 | $-1$ | $\xi_5$ |
| $\mathcal{C}_4 \cong \mathcal{C}_1 - \mathcal{C}_5$ | 0 | 1 | $-1$ | 1 | 0 | $-1$ | $\xi_4$ |
| $\mathcal{C}_6 \cong \mathcal{C}_3 - \mathcal{C}_5$ | 1 | 0 | 0 | 1 | $-1$ | $-1$ | $\xi_6$ |

(98)

**The $E_1$ curve.** Starting from the $E_2$ theory in its gauge-theory phase, we can integrate out the hypermultiplet with either $m_1 \to -\infty$ or $m_1 \to \infty$ in 5d, which gives us the $SU(2)_0$ or the $SU(2)_\pi$ 5d gauge theory, respectively. In the first limit, we have $M_1 \to 0$ and therefore we find the $E_1$ curve:

$$E_1 : \qquad \sqrt{\lambda}\left(\frac{1}{t} + t\right) + \frac{1}{w} + w - U = 0. \tag{99}$$

For completeness, let us recall the GLSM description of the resolved $E_1$ singularity:

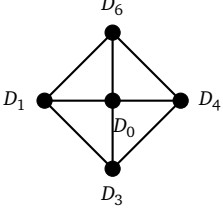

|  | $D_1$ | $D_3$ | $D_4$ | $D_6$ | $D_0$ | FI |
|---|---|---|---|---|---|---|
| $\mathcal{C}_f = \mathcal{C}_1 \cong \mathcal{C}_4$ | 0 | 1 | 0 | 1 | $-2$ | $\xi_1$ |
| $\mathcal{C}_b = \mathcal{C}_3 \cong \mathcal{C}_6$ | 1 | 0 | 1 | 0 | $-2$ | $\xi_3$ |

$$(100)$$

**The $\widetilde{E}_1$ curve.** This case is distinct from the all other $E_n$ with $n > 0$, since $dP_1 \cong \mathbb{F}_1$ is not a blow-up of $\mathbb{F}_0$. Instead, we have the non-trivial fibration:

$$\mathcal{C}_f \to \mathbb{F}_1 \to \mathcal{C}_b. \tag{101}$$

The GLSM description reads:

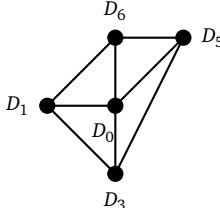

|  | $D_1$ | $D_3$ | $D_5$ | $D_6$ | $D_0$ | FI |
|---|---|---|---|---|---|---|
| $\mathcal{C}_f = \mathcal{C}_1 \cong \mathcal{C}_5$ | 0 | 1 | 0 | 1 | $-2$ | $\xi_1$ |
| $\mathcal{C}_b = \mathcal{C}_3$ | 1 | 1 | 1 | 0 | $-3$ | $\xi_3$ |
| $\mathcal{C}_6 \cong \mathcal{C}_3 - \mathcal{C}_5$ | 1 | 0 | 1 | $-1$ | $-1$ | $\xi_5$ |

$$(102)$$

Let us note that the instanton particle, which is the D2-brane wrapping $\mathcal{C}_b$, has electromagnetic charge $(m, q) = (0, 3)$, since $D_0 \cdot \mathcal{C}_b = -3$. We then have the identification:

$$t_f = 2a, \qquad t_b = 3a + \mu_0, \tag{103}$$

which is distinct from (69). Starting from the $E_2$ curve, we should take the limit $M_1 \to \infty$. Using the gauge freedom (85), we first rescale $t \to t/\sqrt{M_1}$ and redefine $\lambda \to \lambda/\sqrt{M_1}$. We then have:

$$\widetilde{E}_1 : \qquad \sqrt{\lambda}\left(\frac{1}{t} + tw\right) + \frac{1}{w} + w - U = 0. \tag{104}$$

**The $E_0$ curve.** Finally, we can take the limit from $\widetilde{E}_1$ to $E_0$, which corresponds to a 'negative 5d gauge coupling', $\lambda \to \infty$. We should first perform a gauge transformation (85) with $(s_0, s_1, s_2) = (\lambda^{-\frac{1}{3}}, \lambda^{-\frac{1}{3}}, \lambda^{\frac{1}{6}})$, rescale $U \to 3\lambda^{\frac{1}{3}}U$ (the factor 3 being there for future convenience), and then take the limit $\lambda \to \infty$. One then obtains:

$$E_0 : \qquad \frac{1}{t} + \frac{1}{w} + tw - 3U = 0. \tag{105}$$

The local $\mathbb{P}^2$ geometry (also known as $dP_0$) has the GLSM description:

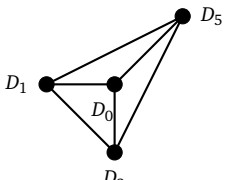

|  | $D_1$ | $D_3$ | $D_5$ | $D_0$ | FI |
|---|---|---|---|---|---|
| $H$ | 1 | 1 | 1 | $-3$ | $\xi$ |

$$(106)$$

**4d limit.** It is also interesting to consider the four-dimensional 'geometric-engineering' limit of the $E_3$ curve (94), given by the small-$\beta$ limit. We pick:

$$w = e^{-2\pi\beta x}\,, \tag{107}$$

for the coordinate $w$, as well as:

$$\lambda = \left(2\pi i\beta\Lambda_{(2)}\right)^2\,, \qquad M_1 = -e^{2\pi\beta m_1}\,, \qquad M_2 = -e^{-2\pi\beta m_2}\,, \tag{108}$$

for the mass parameters,[15] keeping $\Lambda_{(2)}$ fixed. This scale is identified with the dynamical scale of 4d $\mathcal{N} = 2$ $SU(2)$ with two flavours. Recall, that, for $SU(2)$ with $N_f$ flavours, we have:

$$\Lambda_{(N_f)}^{b_0} = \mu^{b_0} e^{2\pi i\tau(\mu)}\,, \qquad b_0 = 4 - N_f\,, \qquad \tau = \frac{\theta}{2\pi} + \frac{4\pi i}{g_{4\mathrm{d}}^2}\,. \tag{109}$$

We identify the 5d and 4d gauge couplings at the threshold scale $\mu \sim \frac{1}{\beta}$, according to $\beta m_0 \propto \frac{1}{g_{4\mathrm{d}}^2}$. The 5d $U$-parameter and the 4d $u$-parameter can be matched as:

$$U = 2 + 4\pi^2 u\beta^2 + O(\beta^3)\,, \qquad u = \langle\mathrm{Tr}(\Phi^2)\rangle \approx -a^2\,, \qquad \Phi = -\frac{i}{\sqrt{2}}\begin{pmatrix} a & 0 \\ 0 & -a \end{pmatrix}\,. \tag{110}$$

We then obtain the 4d curve:

$$\frac{\Lambda(x + m_1)}{t} + \Lambda t(x + m_2) + x^2 - u = 0\,, \tag{111}$$

with the replacement $t \to -it$ done for convenience. Due to the change of coordinate (107), the 4d curve is now a curve in $\mathbb{C} \times \mathbb{C}^*$. The residual $\mathbb{Z}_2$ symmetry of the 4d $u$-plane for the $N_f = 2$ curve is restored by shifting $u$ by an $a$-independent term, namely: $\widetilde{u} = u - \Lambda^2/2$. As pointed out in [36], this leads to $a$-independent terms in the prepotential, which have no effect on the low-energy effective action. From the five-dimensional curve perspective, we can view this as a mixing of the $\mathcal{O}(\beta^2)$ term in (110) with $\lambda$, due to the fact that the parameters $u$ and $\Lambda_{(2)}^2$ have the same scaling dimension. Such mixings will be a general feature of 4d limits. The identification (108) is in agreement with Nekrasov partition function computations in 4d and in 5d, as discussed in section 9 and appendix D. Similar 4d limits can be taken from the $E_2$, $E_1$ and $\widetilde{E}_1$ curves, with:

$$\lambda_{E_2} = -i\left(2\pi i\beta\Lambda_{(1)}\right)^3\,, \qquad \lambda_{E_1} = \lambda_{\widetilde{E}_1} = \left(2\pi i\beta\Lambda_{(0)}\right)^4\,. \tag{112}$$

For both the $E_1$ and the $\widetilde{E}_1$ theory, this gives us the curve corresponding to the pure $SU(2)$ gauge theory in four-dimensions:

$$\frac{\Lambda_{(0)}^2}{t} + \Lambda_{(0)}^2 t + x^2 - u = 0\,. \tag{113}$$

Similarly, the limit on the Weierstrass form of the $E_2$ curve leads to a four-dimensional curve isomorphic to:

$$\frac{\Lambda_{(1)}}{t}(x + m_1) + \Lambda_{(1)}^2 t + x^2 - u = 0\,. \tag{114}$$

We give the Weierstrass form of all these curves in appendix C.

---

[15]We changed the sign of $m_2$ in 4d, in keeping with conventions used later in the text.

### 2.4.2 The $E_n$ Seiberg-Witten curves

While the mirror curves for the local toric $dP_n$ geometries (*i.e.* $n \leq 3$) can be found from the toric data, the curves for the non-toric cases ($n \geq 4$) can be determined as limits of the $E$-string theory SW curve [11, 24, 156], or, alternatively, using toric-like diagrams [157]. These curves are most easily written in terms of the $E_n$ characters:

$$\chi_{\mathcal{R}}^{E_n}(v) = \sum_{\rho \in \mathcal{R}} e^{2\pi i \rho(v)}, \tag{115}$$

for $\rho = (\rho_i)$ the weights of the representation $\mathcal{R}$, and $v = (v_i)$ the $E_n$ flavour parameters, with the index $i \in \{1, \ldots, n\}$. The relation between these parameters and the 5d gauge-theory parameters can be found as explained in appendix A. The curves are written explicitly in Weierstrass form in appendix C. The massless $E_n$ curves correspond to the $S^1$ reduction of the 5d SCFTs, with no mass deformations turned on. The massless limit of these curves is obtained by setting the characters equal to the dimension of the corresponding representation. For the $E_{6,7,8}$ theories, they read:

$$\begin{aligned}
E_8 \ &: \ y^2 = 4x^3 - \frac{1}{12}U^4 x + \frac{1}{216}(U - 864)U^5, \\
E_7 \ &: \ y^2 = 4x^3 - \frac{1}{12}(U - 36)(U + 12)^3 x + \frac{1}{216}(U - 60)(U + 12)^5, \\
E_6 \ &: \ y^2 = 4x^3 - \frac{1}{12}(U - 18)(U + 6)^3 x + \frac{1}{216}(U^2 - 24U + 36)(U + 6)^4,
\end{aligned} \tag{116}$$

with the following singular fibers at finite $U$:

$$\begin{aligned}
E_8 \ &: \ II^* \oplus I_1, \\
E_7 \ &: \ III^* \oplus I_1, \\
E_6 \ &: \ IV^* \oplus I_1.
\end{aligned} \tag{117}$$

This manifestation of the flavour symmetry in the mirror threefold occurs for all $E_n$ theories. Furthermore, from this configuration of singular fibers it is straightforward to obtain the four-dimensional limit of these theories. This is done by identifying the $I_1$ singularities with the KK charge and decoupling it from the bulk by 'zooming in' around the $E_n$ type Kodaira singularity on the $U$-plane. It is well known that these flow in 4d to the Minahan-Nemeschansky (MN) theories [5, 6], which have the following scaling dimensions for the Coulomb-branch parameter: $(\Delta_{E_8}, \Delta_{E_7}, \Delta_{E_6}) = (6, 4, 3)$. Thus, we have:

$$\begin{aligned}
E_8 \ &: \ (U, x, y) \longrightarrow (\beta^6 u, \beta^{10} x, \beta^{15} y), \\
E_7 \ &: \ (U, x, y) \longrightarrow (\beta^4 u - 12, \beta^6 x, \beta^9 y), \\
E_6 \ &: \ (U, x, y) \longrightarrow (\beta^3 u - 6, \beta^4 x, \beta^6 y),
\end{aligned} \tag{118}$$

including constant shifts to bring the relevant singularity to the origin of the $u$-plane. This leads to the massless SW curves for the four-dimensional MN theories:

$$\begin{aligned}
E_8^{(4d)} \ &: \ y^2 = 4x^3 - 4u^5, \\
E_7^{(4d)} \ &: \ y^2 = 4x^3 + 4u^3 x, \\
E_6^{(4d)} \ &: \ y^2 = 4x^3 + u^4,
\end{aligned} \tag{119}$$

which are standard $E_n$ double-point singularities at the origin of $(x, y, u) \in \mathbb{C}^3$. One can also reproduce the deformation patterns of these singularities by keeping track of the various 5d mass parameters [24, 156].

The other massless $E_n$ curves can be analysed in a similar way. One finds that the $U$-plane has the following singularities, in addition to the $I_{9-n}$ singularity at infinity [11]:

$$
\begin{aligned}
E_5 &: \quad I_1^* \oplus I_1 \,, \\
E_4 &: \quad I_5 \oplus I_1 \oplus I_1 \,, \\
E_3 &: \quad I_3 \oplus I_2 \oplus I_1 \,, \\
E_2 &: \quad I_2 \oplus I_1 \oplus I_1 \oplus I_1 \,, \\
E_1 &: \quad I_2 \oplus I_1 \oplus I_1 \,, \\
\widetilde{E}_1 &: \quad I_1 \oplus I_1 \oplus I_1 \oplus I_1 \,, \\
E_0 &: \quad I_1 \oplus I_1 \oplus I_1 \,.
\end{aligned}
\tag{120}
$$

The 4d low-energy effective field theories obtained from the circle compactification of the 5d $E_n$ SCFTs are IR free for $n < 6$. Interestingly, the $E_5$ theory, which has a gauge-theory phase corresponding to $SU(2)$, $N_f = 4$ in five dimensions, becomes an $SU(2)$ theory with $N_f = 5$ upon $S^1$ reduction, which matches the $E_5 = \mathfrak{so}(10)$ symmetry of the UV theory. In some sense, the 'instanton particle' becomes a perturbative hypermultiplet in four-dimensions, but it is more accurate to say that the full IR-free $SU(2)$ description is a magnetic dual description of the UV theory. For the $E_4$ theory, we have an $I_5$ point, corresponding to SQED with five flavours, which again reproduces the $E_4 = \mathfrak{su}(5)$ symmetry. Note that the $E_3$ theory is special in that there are now two distinct points with a non-trivial Higgs branch. This matches with the fact that the Higgs branch of the 5d SCFT $E_3$ is the union of two cones, on which each of the two factors in $E_3 = \mathfrak{su}(3) \oplus \mathfrak{su}(2)$ act independently. In 4d, the instanton corrections separate the two Higgs branch cones along the complexified Coulomb branch. Similarly, the $E_2$ and $E_1$ theories both have an $\mathfrak{su}(2)$ symmetry that is reproduced by an $I_2$ singularity. On the other hand, the abelian part of $E_2 = \mathfrak{su}(2) \oplus \mathfrak{u}(1)$ and $\widetilde{E}_1 = \mathfrak{u}(1)$ is encoded in the Seiberg-Witten geometry in a more subtle manner, which we will discuss in the next section.

## 3 Rational elliptic surfaces, Mordell-Weil group and global symmetries

In the previous section, we mentioned the flavour symmetry *algebra* of various rank-one theories, but it is natural to ask whether one can also determine the global form of the *flavour symmetry group* – that is, the group that acts faithfully on gauge-invariant states – directly from the SW geometry. For the massless $E_n$ theory, the Higgs branch is always isomorphic to the moduli space of one $E_n$-instanton, or equivalently to the minimal nilpotent orbit of $E_n$. (Except for $\widetilde{E}_1$ and $E_2$, which one should discuss separately.) These Higgs branches are consistent with the actual flavour symmetry group of the massless theory being:

$$
G_F = \mathrm{E}_n / \mathrm{Z}(\mathrm{E}_n) \,,
\tag{121}
$$

where $\mathrm{E}_n$ denotes the simply-connected Lie group with Lie algebra $E_n$, and $\mathrm{Z}(\mathrm{E}_n)$ denotes its center – see table 5. Very recently, the flavour symmetry group was determined to be precisely the centerless (121) by looking at the 5d BPS states in M-theory [33] – see also [158] and the index computation in [159]. In this work, we will give two complementary derivations of that same fact, both from the 4d Coulomb-branch point of view. In addition, we will discuss the abelian symmetries, and any flavour symmetry-breaking pattern, in a unified manner, by taking full advantage of the elliptic fibration structure of the rank-one SW geometry.

In order to do so, it is useful to introduce some additional formalism, namely the theory

Table 5: Simply-connected $E_n$ groups and their centers.

| $n$ | 1 | 3 | 4 | 5 | 6 | 7 | 8 |
|---|---|---|---|---|---|---|---|
| $E_n$ | $SU(2)$ | $SU(3) \times SU(2)$ | $SU(5)$ | Spin(10) | $E_6$ | $E_7$ | $E_8$ |
| $Z(E_n)$ | $\mathbb{Z}_2$ | $\mathbb{Z}_3 \times \mathbb{Z}_2 \cong \mathbb{Z}_6$ | $\mathbb{Z}_5$ | $\mathbb{Z}_4$ | $\mathbb{Z}_3$ | $\mathbb{Z}_2$ | 0 |

of rational elliptic surfaces.[16] From that more global perspective, one can study the physics of $D_{S^1}E_n$ throughout its whole parameter space rather systematically and efficiently. This perspective also leads to an improved understanding of the 'well-known' 4d gauge theories and SCFTs, as we will discuss in the next section.

## 3.1 The Seiberg-Witten geometry as a rational elliptic surface

Consider the SW geometry (43) at fixed mass parameters. We write it as an elliptic fibration:

$$E \longrightarrow S \xrightarrow{\pi} \mathbb{P}^1, \tag{122}$$

where the genus-zero base $\overline{\mathcal{M}}_C \cong \mathbb{P}^1$ is the $U$-plane with the point at infinity added, and the fiber $E$ is the Seiberg-Witten curve. Its minimal Weierstrass model reads:

$$y^2 = 4x^3 - g_2(U)x - g_3(U), \tag{123}$$

which is a single equation in the complex variables $(x, y, U)$, thus describing a dimension-two complex variety. By using homogeneous coordinates (as in footnote 10), this can be interpreted as a projective variety. Importantly, this rational elliptic fibration has a section, called the zero *section* $O$, which is given explicitly by the point 'at infinity', $O = (x, y) = (\infty, \infty)$ on each elliptic fiber.[17]

The Weierstrass model (123) has codimension-one singularities along the discriminant locus $\Delta(U) = 0$, as discussed in the last section (see table 4). In the $(x, y, U)$ variables, they look locally like ADE singularities. Each singular Kodaira fiber $F_v$ at $U = U_{*,v}$ can then be resolved in a canonical fashion, giving us smooth reducible fibers:

$$\pi^{-1}(U_{*,v}) = F_v \cong \sum_{i=0}^{m_v-1} \widehat{m}_{v,i} \Theta_{v,i}, \tag{124}$$

where $\Theta_{v,i}$ are the $m_v$ irreducible fiber components, of multiplicity $\widehat{m}_{v,i}$, in $F_v$. If $m_v = 1$, the irreducible fiber $F_v = \Theta_{v,0}$ is a genus-zero curve (a rational curve with a node or with a cusp, for $F_v$ of type $I_1$ or $II$, respectively). In all other cases, $F_v$ is reducible and the exceptional fibers together with $\Theta_{0,v}$ (all of genus zero) intersect according to the *affine* Dynkin diagram of $\mathfrak{g}$, where $\mathfrak{g}$ is the flavour algebra listed in table 4, and $\widehat{m}_{v,i}$ are the Coxeter labels; in particular, every irreducible component $\Theta_{v,i}$ has self-intersection $-2$ and corresponds to a simple root of $\mathfrak{g}_v$. For every resolved fiber $F_v$, the zero section $O$ intersects $F_v$ only through the fiber component $\Theta_{v,0}$ (which corresponds to the affine node in the ADE Dynkin diagram of $F_v$). Some of the relevant affine Dynkin diagrams are shown in figure 2. The resulting smooth surface $\widetilde{S} \to S$, called the Kodaira-Neron model, is birational to the Weierstrass model $S$ of the SW geometry.

---

[16]For further background on this subject, we refer to the very accessible book by Schütt and Shioda [45], from which much of the mathematical discussion in this section is taken.

[17]In the notation of footnote 10, the zero section is $[X, Y, Z] = [1, 1, 0]$. At smooth fibers, this defines the 'origin' of the elliptic curve $E \cong T^2$.

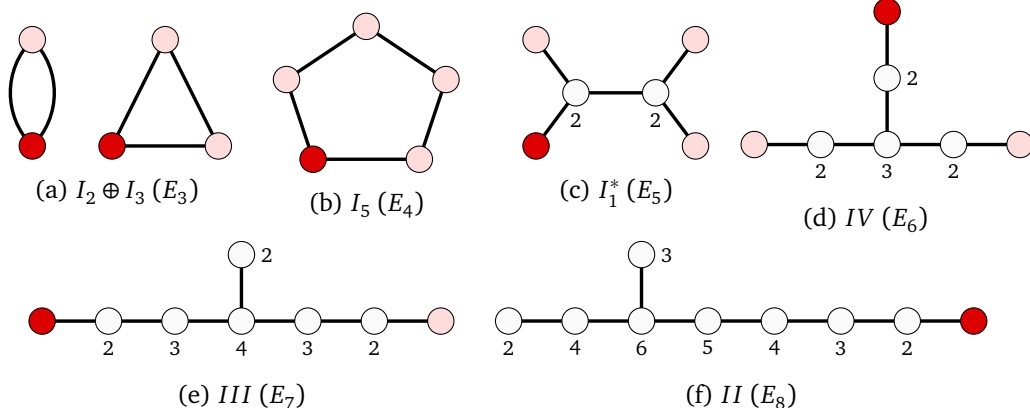

Figure 2: Examples of affine Dynkin diagrams corresponding to resolved Kodaira fibers. These are the ones that correspond to the semi-simple $E_n$ Lie algebras. The affine node $\Theta_{\nu,0}$ is indicated in dark red, and the nodes with unit multiplicity ($\widehat{m}_{\nu,i} = 1$) are all the nodes in (dark or light) red. The multiplicities $\widehat{m}_{\nu,i} > 1$ are indicated next to the nodes.

For future reference, to each reducible fiber $F_\nu$, let us associate the finite abelian group:

$$Z(F_\nu) \equiv R_\nu^\vee / R_\nu \,, \tag{125}$$

where $R_\nu$ is the root lattice of $\mathfrak{g}_\nu$ and $R_\nu^\vee$ is its dual lattice.[18] It is isomorphic to the center $Z(\widetilde{G}_\nu)$ of the simply-connected Lie group $\widetilde{G}_\nu$ associated with that algebra, and it has order:

$$N_\nu = |Z(F_\nu)| = |\det(A_{\mathfrak{g}_\nu})| \,, \tag{126}$$

where $A_{\mathfrak{g}_\nu}$ denotes the Cartan matrix of the Lie algebra $\mathfrak{g}_\nu$. Note that $N_\nu$ is the number of components $\Theta_{\nu,i}$ of $F_\nu$ with $\widehat{m}_{\nu,i} = 1$ in the decomposition (124).

### 3.1.1 Mathematical interlude (I): rational elliptic surfaces

What we have described so far is a rational complex surface that admits an elliptic fibration with a section. By definition,[19] $\mathcal{S}$ (or rather, its resolution $\widetilde{\mathcal{S}}$) is a *rational elliptic surface* (RES). Such surfaces are tightly constrained, and a full classification exists [47, 48]. Any rational elliptic surface can be obtained by blowing up $\mathbb{P}^2$ at nine points – in other words, a RES is an almost del Pezzo surface $dP_9$. In particular, it is also Kähler.

The most important topological fact about $\widetilde{\mathcal{S}}$ is that it is simply-connected and that its topological Euler characteristic $e(\widetilde{\mathcal{S}})$ is equal to 12. We further have that $\chi = 1$ for the holomorphic Euler characteristic, and that the canonical divisor has trivial self-intersection:[20] Thus:

$$b_2 = h^{1,1} = 10 \,, \qquad h^{2,0} = 0 \,, \qquad \mathcal{K}_{\widetilde{\mathcal{S}}} \cdot \mathcal{K}_{\widetilde{\mathcal{S}}} = 0 \,, \tag{127}$$

with the Betti numbers $b_k = \dim H_k(\widetilde{\mathcal{S}}, \mathbb{Z})$ and $h^{p,q} = \dim H^{p,q}(\widetilde{\mathcal{S}})$. We then have:

$$\mathrm{NS}(\widetilde{\mathcal{S}}) \cong H^{1,1}(\widetilde{\mathcal{S}}) \cap H^2(\widetilde{\mathcal{S}}, \mathbb{Z}) \cong \mathbb{Z}^{10} \,, \tag{128}$$

---

[18]In the present case of an ADE algebra, we have $R_\nu^\vee \cong \Lambda_\nu$, with $\Lambda_\nu$ the weight lattice of $\widetilde{G}_\nu$ such that $\mathrm{Lie}(\widetilde{G}_\nu) = \mathfrak{g}_\nu$ and $\pi_1(\widetilde{G}_\nu) = 0$.

[19]To be precise, we should also require that the fibration be relatively minimal, meaning that one should blow down any exceptional curve (*i.e.* any $(-1)$-curve) in the fiber.

[20]Recall that $\mathcal{K} \cdot \mathcal{K} = 9 - n$ for $dP_n$. In the physics literature, RES are sometimes called $\frac{1}{2}$K3 surfaces.

for the Neron-Severi (NS) group of $\widetilde{\mathcal{S}}$ – *i.e.* the group of divisors modulo linear equivalences, in the present context. It is naturally endowed with an integral *lattice structure*, with the bilinear form given by the intersection pairing:

$$\text{NS}(\widetilde{\mathcal{S}}) \times \text{NS}(\widetilde{\mathcal{S}}) \to \mathbb{Z}, \qquad (D, D') \mapsto \langle D, D' \rangle_{\text{NS}} \equiv D \cdot D'. \tag{129}$$

A beautiful mathematical fact, which we will discuss further below, is that the NS lattice of a rational elliptic surface takes the explicit form [45]:

$$\text{NS}(\widetilde{\mathcal{S}}) = U \oplus E_8^-. \tag{130}$$

In particular, it is unimodular and of signature $(1,9)$. Here, $U$ is the dimension-2 lattice generated by the zero section $(O)$ and the generic fiber $F \cong E$, with intersection pairing:

$$U \cong \text{Span}((O), F), \qquad I_U = \begin{pmatrix} -1 & 1 \\ 1 & 0 \end{pmatrix}, \tag{131}$$

and $E_8^-$ is the $E_8$ lattice with an overall minus sign (*i.e.*, $I_{E_8^-} = -A_{E_8}$, with $A_{E_8}$ the $E_8$ Cartan matrix). Note that this generalizes to $dP_9$ the structure of the intersection pairing for the $dP_n$ surfaces $(n \leq 8)$, which have lattices $\mathbb{Z} \oplus E_n^-$.

Another very important set of global constraints is as follows. To each exceptional fiber $F_\nu$, one associates its Euler number, which is given by:

$$e(F_\nu) = \begin{cases} m_\nu = k & \text{if } F_\nu \text{ is of type } I_{k>0} \\ m_\nu + 1 & \text{otherwise} \end{cases} \tag{132}$$
$$= \text{ord}(\Delta) \text{ at } U_{*,\nu},$$

where $\text{ord}(\Delta)$ is as listed in table 4. We also associate an ADE Lie algebra $\mathfrak{g}_\nu$ to each fiber $F_\nu$, including the trivial algebra for $F_\nu$ of type $I_1$ or $II$, with rank:

$$\text{rank}(F_\nu) \equiv \text{rank}(\mathfrak{g}_\nu) = m_\nu - 1. \tag{133}$$

Given these definitions, we have the two conditions:

$$\sum_\nu e(F_\nu) = 12, \qquad \sum_\nu \text{rank}(F_\nu) \leq 8, \tag{134}$$

which severely restrict the possible configurations of singular fibers. Using these and some more subtle geometric constraints (see in particular the discussion in subsection 3.2.1 below), the complete list of all rational elliptic surfaces was first constructed by Persson [47] and further checked by Miranda [48]. There are exactly 289 distinct RES. We will see that a given surface can be interpreted as the Coulomb branch of several distinct $E_n$ theories on a circle; there are then 548 distinct $D_{S^1}E_n$ CB configurations in total, as summarised in table 1 in the introduction.

**Quadratic twist and 'transfer of $^*$' operation.** The allowed coordinate transformations that preserve the Weierstrass form are $(x, y) \to (f^2 x, f^3 y)$, with $f = \mathbb{C}(U)$. On the other hand, a 'quadratic twist' is a rescaling of the form:

$$(x, y) \to (f x, f^{\frac{3}{2}} y), \qquad f \in \mathbb{C}(U), \tag{135}$$

which is equivalent to the rescaling:

$$(g_2, g_3) \to (f^{-2} g_2, f^{-3} g_3), \qquad f \in \mathbb{C}(U). \tag{136}$$

A quadratic twist induces a so-called 'transfer of *' amongst the singular fibers, wherever $\sqrt{f}$ has branch cuts (which can be at a smooth fiber, $I_0$). The corresponding changes of fiber types are:

$$I_k \leftrightarrow I_k^* \ (k \geq 0), \qquad II \leftrightarrow IV^*, \qquad III \leftrightarrow III^*, \qquad IV \leftrightarrow II^*. \qquad (137)$$

This simple operation relates many distinct rational elliptic surfaces amongst themselves [48].

### 3.1.2 Local mirror, rational elliptic surface and the F-theory picture

Recall that the local Calabi-Yau threefold $\widehat{\mathbf{Y}}$ mirror to the local $dP_n$ geometry $\widetilde{\mathbf{X}}_{E_n}$ is a suspension of the $E_n$ Seiberg-Witten curve. In the toric case, in particular, it is given by (83). Let $F(x, y; U) = 0$ denote the SW curve at a particular value of $U \in \mathbb{C}$. By introducing some complex variables $v_1, v_2$ and $W$, one can write down the threefold as a complete intersection in five variables $(x, y, v_1, v_2, W)$ [23]:

$$F(x, y; W) = 0, \qquad v_1 v_2 = U - W. \qquad (138)$$

This describes the mirror threefold as a double fibration over the $W$-plane, at fixed $U$ (and, implicitly, fixed mass parameters $M$):

$$E \times \mathbb{C}^* \to \widehat{\mathbf{Y}} \to \mathbb{C} \cong \{W\}. \qquad (139)$$

The SW curve fibered over the $W$-plane is again our RES $\mathcal{S}$, with $W$ substituted for $U$. The $\mathbb{C}^* \cong \mathbb{R} \times S_*^1$ fiber contains a non-trivial one-cycle $S_*^1$ which degenerates precisely when $W = U$.

The Coulomb-branch BPS states arise from D3-branes wrapping Lagrangian 3-cycles $S_\gamma^3$ calibrated by the holomorphic 3-form. The latter takes the form $\Omega = \Omega_2 \wedge \frac{dv_1}{v_1}$. The 3-cycle $S_\gamma^3$ can be constructed explicitly as follows [23]. Consider a path on the $W$-plane from a singularity $W = W_*$, where the elliptic fiber $E$ degenerates along some one-cycle $\gamma \in E$, to $W = U$, where the $\mathbb{C}^*$ fiber degenerates. By fibering the torus $T^2 \cong \gamma \times S_*^1$ over that path, one spans out the closed 3-cycle $S_\gamma^3$, which is topologically a three-sphere. Let $\Gamma_2 \subset S_\gamma^3$ be the two-chain with boundary along $\gamma \in E_U$ above the fiber at $W = U$, obtained by forgetting the $S_*^1$ fiber. We then have the periods:

$$\Pi_\gamma = \int_{S_\gamma^3 \subset \widehat{\mathbf{Y}}} \Omega = \int_{\Gamma \subset \mathcal{S}} \Omega_2 = \int_{\gamma \in E} \lambda_{\text{SW}}, \qquad (140)$$

with $\partial \Gamma = \gamma$, provided that:

$$\Omega_2 = d\lambda_{\text{SW}}, \qquad (141)$$

inside $\mathcal{S}$. Here, the closed (and exact) 2-form $\Omega_2$ is the holomorphic symplectic 2-form on $\mathcal{S}$ that appears in the integrable-system description of Seiberg-Witten theory [160]. Note that we simply have:

$$\Omega_2 = \boldsymbol{\omega} \wedge dU, \qquad (142)$$

with $\boldsymbol{\omega}$ the holomorphic one-form of the elliptic fiber, as follows from (45); here and in the following, we freely switch back and forth between $W$ and $U$ to describe the '$U$-plane' base of the rational elliptic surface $\mathcal{S}$. It is important to note, however, that $W$ is a coordinate on the IIB geometry while $U$ is a complex structure parameter. It is the double fibration structure (138) that allows us to substitute one for the other in the obvious way. In general, one should also consider more general paths on the $W$-plane to construct supersymmetric 3-cycles. The electro-magnetic charge of the BPS state is fixed by a choice of $\gamma$ at the 'base point' $W = U$, but the path can branch out and meet several Kodaira singularities, as long as the total charge $\gamma$ is conserved. More formally, we may also consider candidate 'pure flavour states', which are

closed 2-cycles $\Gamma$ with $\partial\Gamma = 0$, constructed by connecting directly different Kodaira singularities in the appropriate manner. In all cases, it follows from (140) and (142) that a necessary topological condition for a 2-cycle or 2-chain $\Gamma \subset \mathcal{S}$ to give rise to a BPS state is that is has 'one leg along the base and one leg along the fiber'.

**Correspondence with F-theory.**  Since part of the IIB mirror symmetry appears to have an elliptic fibration, it is useful to think about it in the language of $F$-theory. In our original setup, we have a pure geometry in Type IIB with constant axio-dilaton, which is then 'F-theory' on $\mathbb{R}^4 \times \widehat{\mathbf{Y}} \times T^2$. If we now interpret the elliptic fiber $E$ as the axio-dilaton, instead of the trivial $T^2$ factor, the Kodaira singularities of the Weierstrass model correspond to 7-branes in the standard way. In this picture, the singularity of the $\mathbb{C}^*$ fibration at $W = U$ is interpreted as the position of a probe D3-brane on the $W$-plane [23]. This gives a nice alternative description of the $U$-plane as the geometry seen by a D3-brane in the background of some fixed 7-branes.
The F-theory language offers some additional physical intuition. Firstly, it is clear in this picture that the Kodaira singularities of the SW geometry realize the non-abelian ADE-type *flavour symmetries* of the theory, simply because the 7-branes wrap non-compact cycles $\mathbb{C}^* \times T^2 \subset \widehat{\mathbf{Y}} \times T^2$. The BPS states from the 2-chains $\Gamma \subset \mathcal{S}$ here correspond to *string junctions* on the $W$-plane, which are open-string networks connecting the D3-brane to the 7-branes in a supersymmetric fashion. Such string junctions have been extensively studied in the literature, in this very same context [53, 56, 161–165]. Secondly, it is well-known in F-theory that sections of the elliptic fibration are related to abelian symmetries and to the global form of the 'gauge group' – see *e.g.* the review [166]. In the rest of this section, we will argue, not surprisingly given what we have written so far, that essentially the same conclusions can be reached when interpreting sections of the rank-one Seiberg-Witten geometries in terms of the 4d flavour symmetry.

Let us also recall that the F-theory perspective leads to a nice interpretation of the Higgs branch that emanates from a Kodaira singularity with reducible components [50]. Indeed, moving onto that Higgs branch corresponds to moving the D3-brane probe on top of the 7-brane stack at $W = U_{*,v}$ before 'dissolving' it into the 7-branes, which gives the Higgs branch as the $\widetilde{G}_v$ one-instanton moduli space.[21]

**Fixing $F_\infty$, the fiber at infinity.**  Consider a fixed rank-one 4d $\mathcal{N} = 2$ supersymmetric field theory $\mathcal{T}_{F_\infty}$, which is either a 5d SCFT on a circle, a 4d SCFT, or a 4d $\mathcal{N} = 2$ asymptotically-free theory. For each theory, we are interested in the class of all rational elliptic surfaces with a fixed singularity at $U = \infty$, whose corresponding (resolved) Kodaira fiber is denoted by $F_\infty$. The choice of $F_\infty$ fixes the 'UV definition' of the field theory:[22]

$$\mathcal{T}_{F_\infty} \qquad \longleftrightarrow \qquad \{ \mathcal{S} \mid \pi^{-1}(\infty) = F_\infty \}. \tag{143}$$

For purely four-dimensional theories, this point of view was emphasized in [58]. As we reviewed in the previous section, the SW geometry for the KK theory $D_{S^1}E_n$ has an $I_{9-n}$ fiber at infinity, as determined by the large volume monodromy in Type IIA. We can then obtain the strictly four-dimensional theories by additional limits, thus 'growing' the singularity at infinity. The 4d limits from the 5d $E_n$ SCFT to the 4d $E_n$ MN SCFT for $n = 6, 7, 8$ correspond to the degenerations:

$$F_\infty^{\text{5d}} \to F_\infty^{\text{4d}} : \qquad I_3 \to IV \quad (E_6), \qquad I_2 \to III \quad (E_7), \qquad I_1 \to II \quad (E_8), \tag{144}$$

---

[21]When a perturbative open-string description of this process exists (in particular, for $k$ D7-branes in the case of an $I_k$ singularity), it reproduces exactly the ADHM construction.

[22]With the important exception of $F_\infty = I_8$, which includes both $E_1$ and $\widetilde{E}_1$.

at infinity, wherein one $I_1$ collides with the '5d' fiber at infinity $F_\infty^{5d}$ to give the '4d' fiber $F_\infty^{4d}$. Similarly, the geometric-engineering limit from the $D_{S^1}E_n$ theory with $1 \leq n \leq 5$ to the 4d $SU(2)$ gauge theory with $N_f = n-1$ corresponds to:

$$F_\infty^{5d} \to F_\infty^{4d} : \qquad I_{8-N_f} \to I_{4-N_f}^* \quad (E_{N_f-1}, N_f = 0, 1, 2, 3, 4), \tag{145}$$

wherein two $I_1$'s are brought in to merge with the $I_{8-N_f}$ fiber at infinity. The correspondence between $F_\infty$ and 4d $\mathcal{N} = 2$ theories was summarised in table 1 in the introduction. The remaining choices, $F_\infty = II^*, III^*$ or $IV^*$ correspond to the Argyres-Douglas theories $H_0$, $H_1$ and $H_2$, respectively, as also discussed in [58]. We will discuss the purely 4d theories further in section 4.

Finally, we should mention that one may also consider the 'generic' situation for which the fiber at infinity is trivial. The interpretation of that configuration is that we are considering the 6d $\mathcal{N} = (1,0)$ $E$-string SCFT with $E_8$ symmetry compactified on $T^2$, whose $U$-plane has the singularities [11]:

$$\text{6d } E\text{-string } (F_\infty^{6d} = I_0): \qquad II \oplus I_1 \oplus I_1, \tag{146}$$

in the massless limit. The 5d $E_8$ theory with $F_\infty = I_1$ is obtained from the $E$-string theory by sending one $I_1$ singularity to infinity, which corresponds to shrinking the $T^2$ to $S^1$ [156].

## 3.2 Mordell-Weil group and global symmetries

Let us finally explain how the flavour symmetry group is encoded by the rank-one Seiberg-Witten geometry. This involves reviewing some very interesting mathematical results, following closely [45].

### 3.2.1 Mathematical interlude (II): Mordell-Weil group and Shioda map

**The Mordell-Weil group of sections.** Any elliptic curve famously has the structure of an additive group; viewing the curve as the torus $E \cong \mathbb{C}/(\mathbb{Z} + \tau\mathbb{Z})$, the neutral element is the origin, and the addition operation is simply the addition of complex numbers. This becomes more interesting for an elliptic curve defined over the field $\mathbb{Q}$, in which case the equation $F(x, y) = 0$ for the curve has a finite number of rational solutions, which form a finitely generated abelian group. More generally, we are here considering the equation (123) where $g_2, g_3$ are valued in $\mathbb{C}(U)$, the field of rational functions of $U$. A *rational section* of this elliptic fibration is a rational solution to the equation (123):

$$P = (x(U), y(U)), \qquad \text{with} \quad x(U), y(U) \in \mathbb{C}(U). \tag{147}$$

By the Mordell-Weil theorem, the sections of $\mathcal{S}$ form a finitely generated abelian group, which we denote by either $\text{MW}(\mathcal{S})$ or $\Phi$.[23] We then have:

$$\Phi = \text{MW}(\mathcal{S}) \cong \mathbb{Z}^{\text{rk}(\Phi)} \oplus \mathbb{Z}_{k_1} \oplus \cdots \oplus \mathbb{Z}_{k_t}. \tag{148}$$

Here, $\text{rk}(\Phi)$ is the rank of the MW group – that is, the number of independent generators of the free part of $\Phi$. Note that the point 'at infinity' $O = (\infty, \infty)$ is the neutral element of the MW group, and therefore does not contribute to the rank. The MW group also generally has a torsion component, which we denote by $\Phi_{\text{tor}}$. The addition on sections in $\Phi$ is given by the

---

[23]We will denote the MW *group* by $\Phi$, and use the symbol $\text{MW}(\mathcal{S})$ for the MW *lattice*, to be defined below.

standard addition of rational points of an elliptic curve. Let $P_1 = (x_1, y_1)$ and $P_2 = (x_2, y_2)$ be two distinct points in $\Phi$. Their sum is given by:

$$P = P_1 + P_2 = (x, y), \qquad \begin{cases} x = -(x_1 + x_2) + \frac{1}{4}\left(\frac{y_1 - y_2}{x_1 - x_2}\right)^2, \\ y = -\frac{y_1 - y_2}{x_1 - x_2}(x - x_1) - y_1. \end{cases} \tag{149}$$

For $P_1 = P_2$, we have the duplication formula:

$$P = 2P_1 = (x, y), \qquad \begin{cases} x = -2x_1 + \xi^2, \qquad\qquad \xi \equiv \frac{12x_1^2 - g_2}{4y_1}, \\ y = -2\xi(x - x_1) - y_1. \end{cases} \tag{150}$$

The inverse of a point $P = (x, y)$ is given by $-P = (x, -y)$, so that $P - P = O$. A section $P$ is $\mathbb{Z}_k$ torsion if $kP = P + P + \cdots + P = O$. Each section $P$ defines a divisor $(P) \in \mathrm{NS}(\widetilde{S})$.

**Vertical and horizontal divisors.** One defines the *trivial lattice* of *vertical divisors* in $\widetilde{S}$ as the sublattice $\mathrm{Triv}(\widetilde{S}) \subset \mathrm{NS}(\widetilde{S})$ generated by the zero section, $(O)$, and by the fiber components. We then have:

$$\mathrm{Triv}(\widetilde{S}) \cong U \oplus T^-, \qquad\qquad T \equiv \bigoplus_v R_v, \tag{151}$$

where $R_v$ is the root lattices of the Lie algebra $\mathfrak{g}_v$ associated to the reducible fiber $F_v$ in the Kodaira-Neron model, with the intersection form given by the Cartan matrix. Note that:

$$(I_v)_{ij} = (-A_{\mathfrak{g}_v})_{ij} = \Theta_{v,i} \cdot \Theta_{v,j}, \tag{152}$$

for $T^-$. We will call $T$ 'the 7-brane root lattice', as a nod to the F-theory picture. We have:

$$\mathrm{rank}(T) = \sum_v \mathrm{rank}(\mathfrak{g}_v), \tag{153}$$

with $\mathrm{rank}(\mathfrak{g}_v)$ as in (133). Note that, in accordance to (130), $T$ is a sublattice of the $E_8$ lattice. The 'non-trivial' divisors, or *horizontal divisors*, must then span the complement of $T$ in $E_8$. They are precisely generated by the (non-zero) sections $P$; each divisor $(P)$ decomposes into a horizontal and a vertical component, but there are enough sections to generate all vertical divisors. More precisely, we have the following theorem:

$$\Phi \cong \mathrm{NS}(\widetilde{S})/\mathrm{Triv}(\widetilde{S}), \tag{154}$$

as an isomorphism of abelian groups. It follows, in particular, that:

$$\mathrm{rk}(\Phi) = 8 - \mathrm{rank}(T), \tag{155}$$

which implies the second condition in (134). The simple relation (155) will be important to understand the flavour symmetry on the $U$-plane.

**The Shioda map.** The isomorphism (154) would be more useful if it could be 'split', *i.e.* if we could embed the MW group of sections inside the NS group of divisors. This can be done at the price of tensoring with $\mathbb{Q}$. There exists a unique group homomorphism [167]:

$$\varphi : \Phi \to \mathrm{NS}(\widetilde{S}) \otimes \mathbb{Q}, \tag{156}$$

which maps sections to horizontal divisors with rational coefficients. In other words, we must have that $\varphi(P) = (P) \bmod \mathrm{Triv}(\widetilde{S}) \otimes \mathbb{Q}$ and that:

$$\varphi(P) \cdot (O) = 0, \qquad \varphi(P) \cdot F = 0, \qquad \varphi(P) \cdot \Theta_{v,i} = 0, \ \forall v, i. \tag{157}$$

The map (156), known as the Shioda map, is given explicitly by:

$$\varphi(P) = (P) - (O) - ((P) \cdot (O) + 1)F + \sum_v \sum_{i=1}^{\text{rank}(\mathfrak{g}_v)} \lambda_{v,i}^{(P)} \Theta_{v,i}, \tag{158}$$

with the rational coefficients:

$$\lambda_{v,i}^{(P)} = \sum_{j=1}^{\text{rank}(\mathfrak{g}_v)} (A_{\mathfrak{g}_v}^{-1})_{ij} \Theta_{v,j} \cdot (P), \tag{159}$$

given in terms of the inverse of the Cartan matrix of $\mathfrak{g}_v$. In particular, for each $F_v$, the coefficients $\lambda_{v,i}$ are valued in $\frac{1}{N_v}\mathbb{Z}$, with $N_v$ defined in (126). Note also that $\lambda_{v,i} = 0$, $\forall i$, if $P$ intersects the resolved Kodaira fiber $F_v$ at the 'trivial' affine node $\Theta_{v,0}$.

**The MW lattice and the narrow MW lattice.** Given two sections $P$ and $Q$, define the $\mathbb{Q}$-valued bilinear form:

$$\langle P, Q \rangle = -(\varphi(P) \cdot \varphi(Q)). \tag{160}$$

In this way, the intersection pairing induces a (positive-definite) lattice structure on the free part of the MW group:

$$\text{MW}(\widetilde{\mathcal{S}})_{\text{free}} \equiv \Phi/\Phi_{\text{tor}}. \tag{161}$$

This defines the *Mordell-Weil lattice* (MWL). The intersection pairing on sections is called the height pairing. It is often useful to define some natural sublattices of the MW lattice. In particular, one defines the *narrow Mordell-Weil lattice* $\text{MS}(\mathcal{S})_0$ as:

$$\text{MS}(\widetilde{\mathcal{S}})_0 = \left\{ P \in \text{MW}(\mathcal{S}) \,\middle|\, (P) \text{ intersects } \Theta_{v,0} \text{ for all } F_v \right\}, \tag{162}$$

with the lattice structure defined by the height pairing. Since $\lambda_{v,i} = 0$ for narrow sections, the narrow MW lattice is an integral lattice. One also defines the 'essential sublattice' $L$ as minus the complement of the trivial lattice inside the NS lattice:

$$\text{NS}(\widetilde{\mathcal{S}}) = L^- \oplus \text{Triv}(\widetilde{\mathcal{S}}). \tag{163}$$

Using the fact that the NS lattice of a RES is unimodular, one can show that the essential lattice is isomorphic to the narrow MW lattice and that the MW lattice itself is isomorphic to the dual lattice:

$$\text{MW}(\widetilde{\mathcal{S}})_0 \cong L, \qquad \text{MW}(\widetilde{\mathcal{S}})_{\text{free}} \cong L^\vee. \tag{164}$$

Finally, we have the important fact that, given (130), $L$ is the orthogonal complement of the 7-brane root lattice $T$ inside the $E_8$ lattice:

$$L = T^\perp \text{ in } E_8. \tag{165}$$

While $T = \oplus_v R_v$ is the root lattice of a semi-simple subalgebra $\mathfrak{g}_T = \oplus_v \mathfrak{g}_v$ of $E_8$, the essential sublattice $L$ depends not only on $\mathfrak{g}_T$ as a Lie algebra, but on its particular embedding inside $E_8$. We review some relevant facts about subalgebras of $E_8$ in appendix A.3.

**Torsional sections.** The kernel of the Shioda map is precisely the torsion part of the Mordell-Weil group:

$$\ker(\varphi) = \Phi_{\text{tor}}. \tag{166}$$

Equivalently, a section $P$ is torsion if and only if $\langle P, P \rangle = 0$. It follows that, if $P$ is torsion, we have $\varphi(P) \cdot \Gamma = 0$ for any divisor $\Gamma \in \text{NS}(\widetilde{S})$, and therefore we have the non-trivial integrality condition:

$$\sum_{\nu} \sum_{i=1}^{\text{rank}(\mathfrak{g}_\nu)} \lambda_{\nu,i}^{(P)} \Theta_{\nu,i} \cdot \Gamma \in \mathbb{Z} . \tag{167}$$

Let $T'$ denote the primitive closure of the 7-brane root lattice $T$ inside the $E_8$ lattice:[24]

$$T' = (T \otimes \mathbb{Q}) \cap E_8 . \tag{168}$$

One can prove that:

$$\Phi_{\text{tor}} \cong T'/T . \tag{169}$$

Moreover, since $T'$ is a sublattice of the dual lattice $T^\vee$, we have the important property that the torsion subgroup of the Mordell-Weil group is injective onto the center group $Z(T) = T^\vee/T$:

$$\Phi_{\text{tor}} \hookrightarrow Z(T) = \bigoplus_{\nu} Z(F_\nu) , \tag{170}$$

with $Z(F_\nu)$ defined in (125). This embedding can be determined by explicit computation in the Kodaira-Neron model $\widetilde{S}$.

### 3.2.2 Flavour symmetry group from the SW elliptic fibration

To study the flavour symmetry of a theory $\mathcal{T}_{F_\infty}$ with a Coulomb branch described by a family of rational elliptic surfaces as in (143), it is useful to consider two opposite limits. We first consider the 'massless curve' – in particular, we have then $M_F = 1$ for the theories $D_{S^1}E_n$. In the massless limit, the full flavour symmetry of the UV theory should be manifest. The other limit is the 'maximally massive curve', wherein the UV flavour symmetry $G_F$ is broken explicitly to a maximal torus, $U(1)^f$.

**Structure of the flavour symmetry algebra.** Consider the $U$-plane of a 4d $\mathcal{N} = 2$ theory $\mathcal{T}_{F_\infty}$ with fixed masses (and/or relevant deformations) turned on, which is described by a particular RES $S$ with Kodaira fibers:

$$F_\nu = F_\infty \oplus F_1 \oplus \cdots \oplus F_k . \tag{171}$$

We decompose the 7-brane root lattice in terms of the contribution from infinity and of the contribution from the interior:

$$T = R_\infty \oplus R_F , \qquad R_F = \bigoplus_{\nu=1}^{k} R_\nu . \tag{172}$$

Here, the 'flavour 7-brane root lattice' $R_F$ is the root lattice of the non-abelian flavour algebra of the theory $\mathcal{T}_{F_\infty}$ for some fixed values of the masses:

$$\mathfrak{g}_F^{\text{NA}} = \bigoplus_{\nu=1}^{k} \mathfrak{g}_\nu . \tag{173}$$

On the other hand, the fiber at infinity does not contribute to the flavour symmetry. The reason for this is perhaps easiest to explain in the F-theory picture: BPS states charged under the flavour symmetry are open strings stretched between the probe D3-brane and stacks of

---

[24]A sublattice $M \subset N$ is called primitive if $N/M$ is torsion-free. The primitive closure of any sublattice $N$ in $M$ is the smallest primitive sublattice $N' \subset M$ that contains $N$.

7-branes, which have a mass proportional to the distance between the D3- and the 7-branes. Modes of open strings stretching all the way to infinity have infinite mass, and are therefore not part of the 4d $\mathcal{N} = 2$ theory under consideration.

In addition, the flavour group generally includes abelian factors. They are precisely generated by infinite-order sections, $P \in \Phi_{\text{free}}$. Indeed, that is how $U(1)$ gauge fields arise in F-theory [69, 70]. Consider the $E_n$ theories, for definiteness (the other 4d $\mathcal{N} = 2$ theories being obtained from them in appropriate limits). In the IIB description on $\widehat{\mathbf{Y}}$, we have 3-cycles of the schematic form $\varphi(P) \times S^1_*$, which are mirror to 'flavour' two-cycles in the $E_n$ sublattice of $H_2(\widetilde{\mathbf{X}}, \mathbb{Z})$ [168]. Reducing the $C_4$ RR gauge field of IIB on that 3-cycle, we obtain a background $U(1)$ gauge field in the low-energy description. The horizontality conditions (157) ensure that the abelian gauge field is massless and neutral under the non-abelian flavour symmetry $\mathfrak{g}_F^{\text{NA}}$. The number of abelian factors in the low-energy flavour symmetry is then given by the rank of the Mordell-Weil group, and we have the full flavour algebra:

$$\mathfrak{g}_F = \bigoplus_{s=1}^{\text{rk}(\Phi)} \mathfrak{u}(1)_s \oplus \bigoplus_{v=1}^{k} \mathfrak{g}_v \, , \tag{174}$$

for any extended CB configuration described by a particular RES $\mathcal{S}$. In particular, we see from (155) that:

$$\text{rank}(\mathfrak{g}_F) = 8 - \text{rank}(F_\infty) \, . \tag{175}$$

This equation only depends on the fiber at infinity, and gives the rank of the flavour symmetry $G_F$ of $\mathcal{T}_{F_\infty}$, as indicated. The physical reason for this is clear: as we vary the mass parameters of a given theory $\mathcal{T}_{F_\infty}$, we may break the UV symmetry group $G_F$ to its maximal torus, or to any allowed subgroup, while keeping the rank fixed. This is precisely what being on the extended Coulomb branch, as opposed to the Higgs or mixed branches, means. Such extended CB deformations are realised by 'fusing' or 'splitting' 7-branes by continuously varying the complex structure parameters of the mirror threefold $\widehat{\mathbf{Y}}$ or, equivalently, the parameters of the Weierstrass model $\mathcal{S}$ over the $W$-plane.

**Flavour charges of the BPS states.** Consider any BPS state on the Coulomb branch, corresponding to a 2-chain $\Gamma$ in $\widetilde{\mathcal{S}} \subset \widehat{\mathbf{Y}}$. Its flavour charges under the non-abelian flavour symmetry $\mathfrak{g}_v \subset \mathfrak{g}_F$ associated to the Kodaira fiber $F_v$ are determined by the intersection numbers:

$$w_i^{(\mathfrak{g}_v)}(\Gamma) = \Theta_{v,i} \cdot \Gamma \, . \tag{176}$$

The integers $w_i^{(\mathfrak{g}_v)}$ give us the weight vectors in the Dynkin basis, and thus determine which representations of $\mathfrak{g}_v$ are spanned by the BPS states. Any physical state of the theory $\mathcal{T}_{F_\infty}$ should have finite mass, and therefore its corresponding 2-chain $\Gamma$ should not intersect the fiber at infinity. We then have:

$$\Gamma \text{ physical} \quad \Longleftrightarrow \quad w_i^{(F_\infty)}(\Gamma) = \Theta_{\infty,i} \cdot \Gamma = 0 \, , \tag{177}$$

which can be taken as a 'topological' definition of what we mean by a physical state.

**Massless limit with $G_F$ semi-simple.** Consider a theory $\mathcal{T}_{F_\infty}$ in the massless limit such that $\mathfrak{g}_F$ is semi-simple, and let $\widetilde{G}_F$ denote the corresponding simply-connected group. That is the case, in particular, for all the $E_n$ KK theories with the exception of $\widetilde{E}_1$ and $E_2$. This means that the Mordell-Weil group of $\mathcal{S}$ is purely torsion, $\Phi = \Phi_{\text{tor}}$, and so $\text{rk}(\Phi) = 0$. Such rational elliptic surfaces are called *extremal* – we will discuss them further in subsection 3.3. The flavour

algebra $\mathfrak{g}_F = \mathfrak{g}_F^{\mathrm{NA}}$ is a maximal semi-simple Dynkin sub-algebra of $E_8$ (see appendix A.3). As explained above, $\Phi_{\mathrm{tor}}$ injects into the finite abelian group $Z(T) = T^\vee/T$, which is:

$$\Phi_{\mathrm{tor}} \hookrightarrow Z(T) = Z(F_\infty) \oplus Z(\widetilde{G}_F). \tag{178}$$

In the extremal case, $T' = E_8$ and the torsion group is related to the embedding of the full 7-brane lattice inside the $E_8$ lattice:

$$\Phi_{\mathrm{tor}} \cong E_8/T. \tag{179}$$

Let us denote by $\mathcal{Z}^{[1]}$ the subgroup of sections that are narrow in the interior of the $U$-plane:

$$\mathcal{Z}^{[1]} = \left\{ P \in \Phi_{\mathrm{tor}} \,\middle|\, (P) \text{ intersects } \Theta_{v,0} \text{ for all } F_{v\neq\infty} \right\}, \tag{180}$$

and let us denote by $\mathscr{F}$ the cokernel of the inclusion map $\mathcal{Z}^{[1]} \to \Phi_{\mathrm{tor}}$. In other words, $\mathscr{F}$ is the abelian group defined by the short exact sequence:

$$0 \to \mathcal{Z}^{[1]} \to \Phi_{\mathrm{tor}} \to \mathscr{F} \to 0. \tag{181}$$

Note that $\mathscr{F}$ is a subgroup of $Z(\widetilde{G}_F)$. Given the injection (178), we can write any element of $\Phi_{\mathrm{tor}}$ as $P \sim (z_\infty, z_F)$, where $z_\infty \in Z(F_\infty)$ and $z_F \in Z(\widetilde{G})_F$. The subgroup $\mathcal{Z}^{[1]}$ corresponds to elements of the form $P \sim (z_\infty, 0)$, while the group $\mathscr{F}$ contains all the elements in the image of the projection map $(z_\infty, z_F) \mapsto z_F$. We then claim that the *flavour symmetry group* of the theory $\mathcal{T}_{F_\infty}$ is given by:

$$G_F = \widetilde{G}_F / \mathscr{F}. \tag{182}$$

The argument for (182) is similar to the one given in the F-theory context [52, 71, 72]. One should consider all possible *closed* 2-cycles $\Gamma \in \mathrm{NS}(\widetilde{\mathcal{S}})$, which give rise to formal 'pure flavour' states. The existence of torsion sections $P_{\mathrm{tor}}$ constrains the allowed weights of the pure flavour states due to the integrability condition (167), which gives:

$$\sum_{l=1}^{\mathrm{rank}(F_\infty)} \lambda_{\infty,l}^{(P_{\mathrm{tor}})} w_i^{(F_\infty)} + \sum_{i=1}^{\mathrm{rank}(\mathfrak{g}_F^{\mathrm{NA}})} \lambda_{v,i}^{(P_{\mathrm{tor}})} w_i^{(\mathfrak{g}_F^{\mathrm{NA}})} \in \mathbb{Z}. \tag{183}$$

For the pure flavour states that satisfy the physical state condition (177), we have:

$$\sum_{i=1}^{\mathrm{rank}(\mathfrak{g}_F^{\mathrm{NA}})} \lambda_{v,i}^{(P_{\mathrm{tor}})} w_i^{(\mathfrak{g}_F^{\mathrm{NA}})} \in \mathbb{Z}, \qquad \forall\, P_{\mathrm{tor}} \in \mathscr{F}. \tag{184}$$

The only sections that contribute to the constraint (184) are the elements of $\mathscr{F}$ since, by definition, the 'interior-narrow' sections in $\mathcal{Z}^{[1]} \subset \Phi_{\mathrm{tor}}$ lead to the constraint:

$$\sum_{i=1}^{\mathrm{rank}(F_\infty)} \lambda_{\infty,i}^{(P_{\mathrm{tor}})} w_i^{(F_\infty)} \in \mathbb{Z}, \qquad \forall\, P_{\mathrm{tor}} \in \mathcal{Z}^{[1]}, \tag{185}$$

which is trivial on physical states. This determines (182) as the effectively acting non-abelian group on pure flavour states. We should note that the actual BPS states, which correspond to two-chains ending on the fiber above $W = U$ and thus carry electro-magnetic charge, will typically be charged under the center of $\widetilde{G}_F$, but the heuristic argument above shows that the 'gauge invariant states' are only charged under the smaller group $G_F$. We will also check this claim explicitly in many examples, using a more direct but essentially equivalent argument presented in subsection 3.2.3.

We should also note that the 'interior-narrow' section constraint (185) would be non-trivial when dealing with defect states, which are BPS D3-branes on non-compact 3-cycles stretching all the way to infinity. This leads us to the natural *conjecture* that this group is isomorphic to the one-form symmetry of the field theory:

$$\mathcal{Z}^{[1]} \cong 1\text{-form symmetry of } \mathcal{T}_{F_\infty}. \tag{186}$$

We will show that this agrees with all the known results. For instance, if the conjecture holds, it must be true that, for a fixed $F_\infty$, $\mathcal{Z}^{[1]}$ remains the same for any configuration of the singular fibers $\{F_\nu\}$ in the interior, which is a very strong constraint. We leave a more detailed discussion and derivation of (186) for future work. In particular, it would be important to relate precisely the group $\mathcal{Z}^{[1]}$ with the defect group of the UV theory, which can be computed directly from the mirror threefold $\widehat{\mathbf{Y}}$ [78,79]. In the case of the $D_{S^1}E_1$ theory, we will observe that the full Mordell-Weil torsion encodes the $\mathbb{Z}_4$ two-group symmetry recently discovered in [33]. We then have the more general conjecture:

$$\Phi_{\text{tor}} \cong 2\text{-group symmetry of } \mathcal{T}_{F_\infty} \text{ on its Coulomb branch.} \tag{187}$$

We again leave a deeper discussion of this point for future work.

**Non-abelian flavour symmetry $G_F^{\text{NA}}$ in general.** In any theory $\mathcal{T}_{F_\infty}$ with a flavour algebra (173) for some fixed values of the masses, the same argument as above determines the global form of the non-abelian part of the flavour symmetry group:

$$G_F^{\text{NA}} = \widetilde{G}_F^{\text{NA}}/\mathscr{F}, \tag{188}$$

where $\mathscr{F}$ is defined as in (182) in terms of the torsion part of the Mordell-Weil group. Of course, conjecture (186) should still hold as well.

**Abelian limit with generic masses.** The opposite limit to the extremal limit is when the rank of the Mordell-Weil group is the maximal one allowed by the fiber at infinity:

$$\text{rk}(\Phi) = 8 - \text{rank}(F_\infty) = \text{rank}(G_F). \tag{189}$$

In that limit, the flavour group is abelian and thus entirely generated by sections. The singular fibers in the interior are then irreducible (that is, of type $I_1$ or $II$). This corresponds to the maximal symmetry breaking allowed on the extended CB, *i.e.* with generic masses turned on:

$$G_F \rightarrow \prod_{s=1}^{\text{rank}(G_F)} U(1)_s. \tag{190}$$

Let the sections $P_s$ be the corresponding generators of $\Phi_{\text{free}}$. The divisor dual to $U(1)_s$ is given by $\varphi(P_s)$. Then, the $U(1)_s$ charge of any 'pure flavour' state $\Gamma$ is given by:

$$q_s(\Gamma) \equiv \varphi(P_s) \cdot \Gamma. \tag{191}$$

From the Shioda map, we then obtain an integrality condition:

$$q_s - \sum_{i=1}^{\text{rank}(F_\infty)} \lambda_{\infty,i}^{(P_s)} w_i^{(F_\infty)} \in \mathbb{Z}. \tag{192}$$

On states satisfying the physical condition (177), the second contribution is trivial, and we simply have:

$$q_s(\Gamma) \in \mathbb{Z} \quad \text{if } \Gamma \text{ is 'physical'.} \tag{193}$$

Since there is are no reducible fibers in this abelian configuration, the physical states actually span the narrow Mordell-Weil lattice (162). Let $\Lambda_{\text{phys}}$ denote the weight lattice of flavour charges for the physical states, which is then isomorphic to the narrow MWL – in particular, it is an integral lattice. According to (164) and (165), this physical flavour weight lattice is isomorphic to the complement of the 7-brane lattice at infinity inside the $E_8$ lattice:

$$\Lambda_{\text{phys}} \cong L \cong R_\infty^\perp \quad \text{in } E_8. \tag{194}$$

For $\mathfrak{g}_F$ semi-simple in the UV, we can check in each case (either for the 5d $E_n$ theories or for the 4d theories considered in section 4), according to the general classification results [45, 169], that $\Lambda_{\text{phys}}$ is the root lattice of $\mathfrak{g}_F$. Therefore, since $Z(G_F) \cong \Lambda_{\text{phys}}/\Lambda_r$, the actual flavour group is the centerless group, $G_F = \widetilde{G}_F/Z(\widetilde{G}_F)$. This gives a complementary derivation of (182) which avoids having to carefully compute the intersection of torsion sections with the reducible fibers.[25]

**Symmetry group $G_F$ in the general case.** In the general case of a flavour algebra (174), physical states $\Gamma$ carry both weights under $\mathfrak{g}_F^{\text{NA}}$ and abelian charges:

$$w_i^{(\mathfrak{g}_v)}(\Gamma) = \Theta_{v,i} \cdot \Gamma, \qquad q_s(\Gamma) \equiv \varphi(P_s) \cdot \Gamma. \tag{195}$$

The allowed weights are constrained by torsion sections as in (184), and the abelian charges satisfy the conditions:

$$q_s - \sum_{i=1}^{\text{rank}(\mathfrak{g}_F^{\text{NA}})} \lambda_{v,i}^{(P_s)} w_i^{(\mathfrak{g}_F^{\text{NA}})} \in \mathbb{Z}, \qquad \forall P_s \in \Phi_{\text{free}}. \tag{196}$$

Thus, for any given RES $\widetilde{\mathcal{S}}$ corresponding to an extended CB configuration of $\mathcal{T}_{F_\infty}$, the global form of the IR flavour symmetry takes the schematic form:

$$G_F = \frac{U(1)^{\text{rk}(\Phi)} \times \widetilde{G}_F^{\text{NA}}}{\prod_{s=1}^{\text{rk}(\Phi)} \mathbb{Z}_{m_s} \times \prod_{j=1}^{p} \mathbb{Z}_{k_p}}, \tag{197}$$

where the two factors in the denominator are determined by the conditions (196) and by the torsion sections, respectively. In this work, we will mostly focus on the case of $G_F$ semi-simple. The detailed form of (196) can also be deduced from the general classification of Mordell-Weil lattices [45, 169], in principle, by mass-deforming into a purely abelian flavour phase.

### 3.2.3 Global symmetries from the BPS spectrum

As a consistency check of the above discussion, it is interesting to also compute the flavour group more directly, which can be done if we know the low-energy spectrum $\mathscr{S}$, similarly to the recent discussion in [33]. As a reasonably good approximation of the strong-coupling spectrum, for our purpose, we can often consider $\mathcal{S}$ to be the set of dyons that become massless at the SW singularities $U_*$. This is closely related to the existence of quiver point, which we will briefly discuss in subsection 3.4 and throughout later sections.

At a generic point on the Coulomb branch, there is a $U(1)_m^{[1]} \times U(1)_e^{[1]}$ one-form symmetry in the strict IR limit, which is the one-form symmetry of a pure $U(1)$ gauge theory [31]. One can think of $U(1)_e^{[1]}$ as the group of global gauge transformations in the electric frame, and similarly for $U(1)_m^{[1]}$ in the magnetic frame. This accidental continuous one-form symmetry

---

[25]See [73] for a related argument in the context of F-theory on an elliptically fibered Calabi-Yau threefold.

is broken explicitly to a discrete subgroup (which can be trivial) by the spectrum of charged massive BPS particles $\mathscr{S}$. The one-form symmetry of the full 4d $\mathcal{N} = 2$ theory is then given by that subgroup.[26] See also [94] for further discussion.

Given a theory at fixed masses with a flavour symmetry algebra $\mathfrak{g}_F$ which is non-abelian, for simplicity, let $\widetilde{G}_F$ denote the corresponding simply-connected group, and let $Z(\widetilde{G}_F)$ be its center. For concreteness, let us have $Z(\widetilde{G}_F) = \mathbb{Z}_{n_1} \times \cdots \mathbb{Z}_{n_p}$. The dyons in $\mathscr{S}$ fall in representations $\mathfrak{R}_\psi$ of $\mathfrak{g}_F$. Let us denote these states $\psi$ by the charges:

$$\psi \, : \, (m, q; l_1, \cdots, l_p), \qquad l_1 \in \mathbb{Z}_{n_1}, \cdots, l_p \in \mathbb{Z}_{n_p}, \tag{198}$$

where $(m, q)$ are the electromagnetic charges, and the integers $l_j \bmod n_j$ give the charges of $\psi$ under the center $Z(\widetilde{G}_F)$. Let us define the subgroup:

$$\mathscr{E} \, \subset \, U(1)_m^{[1]} \times U(1)_e^{[1]} \times Z(\widetilde{G}_F), \tag{199}$$

as the maximal subgroup that leaves the spectrum $\mathscr{S}$ invariant. We will denote the generators of $\mathscr{E}$ by:

$$g^{\mathscr{E}} = (k_m, k_q; z_1, \cdots, z_p), \qquad k_m \in \mathbb{Q}, \; k_q \in \mathbb{Q}, \; z_j \in \mathbb{Z}_{n_j}. \tag{200}$$

This is a generator that acts on a state (198) as:

$$g^{\mathscr{E}} \, : \quad \psi \to e^{2\pi i \left( k_m m + k_q q + \sum_{j=1}^{p} \frac{z_i l_i}{n_i} \right)} \psi. \tag{201}$$

Let $\mathcal{Z}^{[1]}$ denote the subgroup of $\mathcal{E}$ generated by:

$$g^{\mathcal{Z}^{[1]}} = (k_m, k_e; 0, \cdots, 0). \tag{202}$$

In addition, the projection $\pi_F : U(1)_m^{[1]} \times U(1)_e^{[1]} \times Z(\widetilde{G}_F) \to Z(\widetilde{G}_F)$ gives a subgroup $\mathscr{F}$ of $Z(\widetilde{G}_F)$ generated by:

$$g^{\mathscr{F}} = (z_1, \cdots, z_p), \tag{203}$$

for each generator (200). These three groups are related by a short exact sequence:

$$0 \to \mathcal{Z}^{[1]} \to \mathcal{E} \to \mathscr{F} \to 0, \tag{204}$$

precisely as in (181). Here, $\mathcal{Z}^{[1]}$ is exactly the one-form symmetry. On the other hand, $\mathscr{F}$ is the subgroup of the flavour center $Z(\widetilde{G}_F)$ that can be compensated by gauge transformations, and therefore the actual non-abelian flavour group of the theory is $G_F = \widetilde{G}_F / \mathscr{F}$, as in (182). In the presence of both a one-form symmetry and a non-trivial flavour symmetry, the group $\mathscr{E}$ itself could be a non-trivial 2-group of the field theory, as shown in [33]. This leads to the conjecture (187). In this work, we will focus on the computation of the global non-abelian symmetry group.

## 3.3 Extremal rational elliptic surfaces and Coulomb branch configurations

A small and particularly interesting subset of all rational elliptic surfaces consists of those with a Mordell-Weil group of rank zero, $\mathrm{rk}(\Phi) = 0$, which are called *extremal*. There are only 16 of them, as classified by Miranda and Persson [46]. We list them in tables 6 and 7. By our general discussion, they correspond to Coulomb branch configurations with a semi-simple flavour symmetry. A given extremal RES generally corresponds to several 4d $\mathcal{N} = 2$ theories $\mathcal{T}_{F_\infty}$, simply by choosing which of the Kodaira fibers sits 'at infinity' on the one-dimensional Coulomb branch.

---

[26]We are very grateful to M. Del Zotto for explaining this to us.

Table 6: Extremal rational elliptic surfaces without multiplicative (*i.e.* $I_k$) fibers.

| $\{F_v\}$ | Notation | $\Phi_{\text{tor}}$ | 4d theory | $\mathfrak{g}_F$ |
|---|---|---|---|---|
| $II^*, II$ | $X_{22}$ | - | AD $H_0$ | - |
| | | | $E_8$ MN | $E_8$ |
| $III^*, III$ | $X_{33}$ | $\mathbb{Z}_2$ | AD $H_1$ | $A_1$ |
| | | | $E_7$ MN | $E_7$ |
| $IV^*, IV$ | $X_{44}$ | $\mathbb{Z}_3$ | AD $H_2$ | $A_2$ |
| | | | $E_8$ MN | $E_6$ |
| $I_0^*, I_0^*$ | $X_{11}(j)$ | $\mathbb{Z}_2 \times \mathbb{Z}_2$ | $SU(2), N_f = 4$ | $D_4$ |

The four surfaces listed in table 6 do not have any multiplicative fibers, and therefore they cannot correspond to the $D_{S^1}E_n$ theories, which have $F_\infty = I_{9-n}$. Instead, they correspond to the seven 'classic' 4d SCFTs associated to the 7 additive Kodaira singularities $II$, $III$, $IV$, $II^*$, $III^*$, $IV^*$ and $I_0^*$ – this was previously discussed in [58]. In each case, the massless curve has a single Kodaira singularity at the origin, and therefore the singularity at infinity is such that $\mathbb{M}_0 \mathbb{M}_\infty = \mathbf{1}$. Thus, the first three extremal surfaces in table 6 describe both the $E_n$ Minahan-Nemeschansky theories [5,6] and the three rank-one AD theories. The last surface, $X_{11}(j)$, describes $SU(2)$ with four flavours. It is the only extremal surface that comes in a one-dimensional family [46] (all the other extremal fibrations are unique), corresponding to the marginal gauge coupling of this 4d SCFT. From the MW torsion of these surfaces, one can also deduce the flavour symmetry group. This will be discussed in section 4.

The remaining 12 extremal RES are listed in table 7. These are also all the extremal RES that have more than 2 singular fibers – in fact, they can only have 3 or 4 singular fibers. The first and second columns in table 7 indicate the singular fibers and the names of the corresponding surfaces in the notation of [46]. The third column gives the MW group of the elliptic fibration, which is purely torsion. The fourth column lists the 4d $\mathcal{N} = 2$ field theories for which this extremal RES describes a CB configuration, while the fifth column gives the unbroken flavour symmetry algebra in each case. The last column in table 7 indicates the modular group of the surface, up to conjugacy. The modular group is identified by working out the expression for $U(\tau)$ from the SW curve as explained in section 2.3.2. We will discuss these modular properties in more detail in the following sections.

All the massless $E_n$ KK theories other than $E_2$ and $\widetilde{E}_1$ appear in table 7. These last two are the exceptions because their flavour group includes one $U(1)$ factor, and therefore the corresponding rational elliptic surfaces have $\text{rk}(\Phi) = 1$. (Similarly so for 4d $SU(2)$ with $N_f = 1$.) All but one of the extremal rational surfaces with more than 2 singularities have interesting modular properties. The monodromy group turns out to be a congruence subgroup, as listed in table 7 up to conjugacy. The first three curves in this table, $X_{211}, X_{321}$ and $X_{431}$, are the massless curves for the $E_8, E_7$ and $E_6$ theories, respectively. Since the corresponding five-dimensional SCFTs flow to SCFTs in four-dimensions, their congruence subgroups must have elliptic points, which is indeed the case for the modular groups $\Gamma^0(2)$ and $\Gamma^0(3)$, respectively. The CB for the massless $D_{S^1}E_8$ theory is not a modular curve. All these CB configurations will be discussed in more detail in section 8. Note that none of the other elliptic curves have elliptic points. Moreover, we observe that the list of modular groups for the extremal elliptic fibrations includes all possible torsion-free congruence subgroups of $\text{PSL}(2, \mathbb{Z})$ up to index 12 – the latter have been classified in [170].

For each five-dimensional $E_n$ theory, we can realise many of the corresponding subalgebras

Table 7: Extremal rational elliptic surfaces with $I_k$ fibers and corresponding field theories.

| $\{F_v\}$ | Notation | $\Phi_{\text{tor}}$ | Field theory | $\mathfrak{g}_F$ | Modularity |
|---|---|---|---|---|---|
| $II^*, I_1, I_1$ | $X_{211}$ | — | $D_{S^1}E_8$ | $E_8$ | — |
| | | | AD $H_0$ | — | |
| $III^*, I_2, I_1$ | $X_{321}$ | $\mathbb{Z}_2$ | $D_{S^1}E_8$ | $E_7 \oplus A_1$ | $\Gamma_0(2)$ |
| | | | $D_{S^1}E_7$ | $E_7$ | |
| | | | AD $H_1$ | $A_1$ | |
| $IV^*, I_3, I_1$ | $X_{431}$ | $\mathbb{Z}_3$ | $D_{S^1}E_8$ | $E_6 \oplus A_2$ | $\Gamma_0(3)$ |
| | | | $D_{S^1}E_6$ | $E_6$ | |
| | | | AD $H_2$ | $A_2$ | |
| $I_4^*, I_1, I_1$ | $X_{411}$ | $\mathbb{Z}_2$ | $D_{S^1}E_8$ | $D_8$ | $\Gamma_0(4)$ |
| | | | 4d pure $SU(2)$ | — | |
| $I_1^*, I_4, I_1$ | $X_{141}$ | $\mathbb{Z}_4$ | $D_{S^1}E_8$ | $D_5 \oplus A_3$ | $\Gamma_0(4)$ |
| | | | $D_{S^1}E_5$ | $D_5$ | |
| | | | 4d $SU(2) N_f = 3$ | $A_3$ | |
| $I_2^*, I_2, I_2$ | $X_{222}$ | $\mathbb{Z}_2 \times \mathbb{Z}_2$ | $D_{S^1}E_7$ | $D_6 \oplus A_1$ | $\Gamma(2)$ |
| | | | 4d $SU(2) N_f = 2$ | $A_1 \oplus A_1$ | |
| $I_9, I_1, I_1, I_1$ | $X_{9111}$ | $\mathbb{Z}_3$ | $D_{S^1}E_8$ | $A_8$ | $\Gamma_0(9)$ |
| | | | $D_{S^1}E_0$ | — | |
| $I_8, I_2, I_1, I_1$ | $X_{8211}$ | $\mathbb{Z}_4$ | $D_{S^1}E_8$ | $A_7 \oplus A_1$ | $\Gamma_0(8)$ |
| | | | $D_{S^1}E_7$ | $A_7$ | |
| | | | $D_{S^1}E_1$ | $A_1$ | |
| $I_5, I_5, I_1, I_1$ | $X_{5511}$ | $\mathbb{Z}_5$ | $D_{S^1}E_8$ | $A_4 \oplus A_4$ | $\Gamma_1(5)$ |
| | | | $D_{S^1}E_4$ | $A_4$ | |
| $I_6, I_3, I_2, I_1$ | $X_{6321}$ | $\mathbb{Z}_6$ | $D_{S^1}E_8$ | $A_5 \oplus A_2 \oplus A_1$ | $\Gamma_0(6)$ |
| | | | $D_{S^1}E_7$ | $A_5 \oplus A_2$ | |
| | | | $D_{S^1}E_6$ | $A_5 \oplus A_1$ | |
| | | | $D_{S^1}E_3$ | $A_2 \oplus A_1$ | |
| $I_4, I_4, I_2, I_2$ | $X_{4422}$ | $\mathbb{Z}_4 \times \mathbb{Z}_2$ | $D_{S^1}E_7$ | $A_3 \oplus A_3 \oplus A_1$ | $\Gamma_0(4) \cap \Gamma(2)$ |
| | | | $D_{S^1}E_5$ | $A_3 \oplus A_1 \oplus A_1$ | |
| $I_3, I_3, I_3, I_3$ | $X_{3333}$ | $\mathbb{Z}_3 \times \mathbb{Z}_3$ | $D_{S^1}E_6$ | $A_2 \oplus A_2 \oplus A_2$ | $\Gamma(3)$ |

of rank $s = n$ (see table 14 in appendix), but not all of them. Instead, for each simple algebra $E_n$, we realise a CB for each of its regular 'maximal' semi-simple subalgebra, whose Dynkin diagram is obtained by deleting a single node of the *affine* $E_n$ Dynkin diagram [171]. For $D_{S^1}E_8$, in particular, we have the 9 distinct CB configurations with flavour algebras indicated in blue in table 7, including $\mathfrak{g}_F = E_8$ itself:

$$E_8 \rightarrow E_8, \ E_7 \oplus A_1, \ E_6 \oplus A_2, \ D_5 \oplus A_3, \ A_4^2, \ A_5 \oplus A_2 \oplus A_1, \ A_8, \ A_7 \oplus A_1, \ D_8. \qquad (205)$$

Two of the remaining 6 subalgebras ($D_4 \oplus A_1^4$ and $A_1^8$) cannot be realised as the 7-brane root lattice of a RES [47]. The remaining 4 cases are:

$$A_2^4 \ (4I_3), \quad D_6 \oplus A_1^2 \ (I_2^* \oplus 2I_2), \quad D_4^2 \ (2I_0^*), \quad A_3^2 \oplus A_1^2 \ (2I_4 \oplus 2I_2), \qquad (206)$$

which are realised by the extremal fibrations $X_{3333}$, $X_{222}$, $X_{11}(j)$ and $X_{4422}$, respectively. These 4 subalgebras of $E_8$ are realised physically as CB configurations of the 6d $E$-string theory on $T^2$ which are obtained as mass deformations of the $E_8$ configuration (146) and do not descent to CB configurations of the 5d KK theory $D_{S^1}E_8$. Similarly, we have the regular semi-simple subalgebras of the simple groups $E_n$:

$$
\begin{aligned}
E_7 &\rightarrow E_7, D_6 \oplus A_1, A_5 \oplus A_2, A_3^2 \oplus A_1, A_7, \\
E_6 &\rightarrow E_6, A_5 \oplus A_1, A_2^3, \\
E_5 &\rightarrow D_5, A_3 \oplus A_1^2, \\
E_4 &\rightarrow A_4,
\end{aligned}
\tag{207}
$$

which are all realised in table 7. Finally, let us note that the last configuration in table 7, $X_{3333}$, gives the so-called $T_3$ description of the $E_6$ theory, in which only an $A_2^3$ algebra is manifest; similarly, the configuration $X_{4422}$ for $E_7$ with $A_3^2 \oplus A_1$ realised, and the configuration $X_{6321}$ for $E_8$ with $A_5 \oplus A_2 \oplus A_1$ realised, can be obtained by Higgsing from the $T_4$ and $T_6$ theories, respectively [172].

### 3.4 Modularity on the $U$-plane and BPS quivers

We end this section with some general comments, before delving into many examples in the rest of this paper. Our approach, in the following, will be to explore the $U$-plane at various special points in parameters space, fixing the $E_n$ masses $M_F$ ($\lambda$ and $M_i$, and similarly for the 4d theories) and then studying the electromagnetic periods $a$, $a_D$ and their monodromies on the resulting $U$-plane. In general, the Picard-Fuchs equation (54) will be unyieldy and an explicit solution will be out of reach with the methods we are using – for instance, for generic mass parameters the $E_n$ theory has $n + 3$ $I_1$ singularities, and the monodromy group will be generated by $n + 3$ monodromy matrices $\mathbb{M}_\nu$, each conjugate to $T$, and such that:

$$
\prod_{\nu=1}^{n+3} \mathbb{M}_\nu = T^{n-9},
\tag{208}
$$

for some appropriate base point and ordering, as in figure 1. We do not attempt to solve for the $\mathbb{M}_\nu$ in that general case. Instead, we first fix some *interesting* values of the masses, such as the massless points discussed above. In such special limits, we can often give an explicit solution for the electromagnetic periods; it is then instructive to compute the monodromies by brute force.

In some interesting special cases, we can use a much more powerful and elegant method, however. It turns out that, in many instances, the $U$-plane is a modular curve – that is, a quotient of the upper half-plane by a subgroup $\Gamma$ of the modular group $\mathrm{PSL}(2, \mathbb{Z})$:

$$
\{U\} \cong \mathbb{H}/\Gamma, \qquad \tau \mapsto U(\tau), \ \forall \tau \in \mathbb{H}.
\tag{209}
$$

In this case, we can describe the $U$-plane equivalently as a fundamental domain for $\Gamma$ in the upper half-plane. Under this map, the fibers of type $I_k$ and $I_k^*$ correspond to cusps of width $k$ of $\Gamma$, while the remaining additive fibers correspond to the elliptic points of order two or three, depending on the value of $\tau$ in table 4. The corresponding monodromies can then be read off directly by conjugating the monodromies for the cusp and elliptic points of the modular group $\mathrm{PSL}(2, \mathbb{Z})$ itself. We will see this simple but powerful approach at work in many examples.

To identify which CB configurations are modular, we employ a combination of methods. Firstly, we can always compute $U(\tau)$ explicitly in the weak-coupling regime, as explained at the end of section 2.3.2, and try to see whether it can be identified with a principal modular

function – a so-called Hauptmodul – for some $\Gamma$. In the cases when the $q$-series expansion of $U(\tau)$ is a McKay-Thompson series of the Monster group [173], this identification is eased by the fact that all such series arising in Moonshine are Hauptmoduls of certain genus zero modular groups. In this paper, we will mainly focus on congruence subgroups, which have been classified in [170, 174]. This classification provides configurations of singular fibers (up to quadratic twists) for each congruence subgroup, which thus allows for the identification of all the modular curves (209) with $\Gamma$ a congruence subgroup.

The last point that we would like to mention concerns a by-product of our computations, which would deserve a more serious investigation. Namely, we can often identify *quiver points* on the $U$-planes. Those are points where the central charges of $n+3$ 'light' BPS particles almost align – for the $D_{S^1}E_n$ theories, they become real – and where, conjecturally, the full BPS spectrum can be obtained as bound-states of the $n+3$ elementary particles. The problem of finding the spectrum, as such a point, can be formulated in terms of a BPS quiver – see *e.g.* [26, 85–87]. The rough intuition for quiver points, and an explicit way to compute the resulting quivers, follows from considering the IIB mirror geometry, $\widehat{\mathbf{Y}}$. We mentioned that BPS particles correspond to D3-branes wrapping Lagrangian 3-cycles. In the IIA description, the full $\mathcal{B}_4$ collapses to zero-volume in the classical picture, and the (derived) category of quiver representations is expected to accurately describe the category of B-branes in that regime. In the mirror IIB description, we have 'light' wrapped D3-branes on the 'small' 3-cycles mirror to the shrinking D0/D2/D4 bound states, that correspond to string junctions connecting a base point $W = U_0$ near the origin of the $W$-plane to the '7-branes' around it. In many cases, the fractional branes are then simply the smallest 'vanishing paths' (in the sense of Picard-Lefschetz theory) on the $W$-plane [23]. In other words, in an ideal situation, the fractional branes are the dyons that become massless at the $U$-plane singularities around the base point. Once we have identified the electromagnetic charge $\gamma_i = (m_i, q_i)$ of these dyons, the BPS quiver is obtained by assigning a quiver node $(i) \sim \mathcal{E}_{\gamma_i}$ to each light dyon, and a number $n_{ij}$ or arrows from node $(i)$ to $(j)$ given by the Dirac pairing, which is also the oriented intersection number between the 3-cycles inside $\widehat{\mathbf{Y}}$:

$$n_{ij} = m_i q_j - q_i m_j = \langle S^3_{\gamma_i}, S^3_{\gamma_j} \rangle. \tag{210}$$

For the $D_{S^1}E_n$ theories, we recover in this way many known 'fractional brane quivers' for $dP_n$, toric and non-toric – as emphasised in [26], fractional-brane quivers *are* 5d BPS quivers. The quivers are best understood in terms of CB configurations with only multiplicative fibers, where each $I_k$ singularity corresponds to a 'block' of quiver nodes; in particular, the cases of a $D_{S^1}E_n$ CB with 3 multiplicative fibers in the interior, corresponding to a configuration $\mathcal{S} \cong (I_{9-n}, I_{k_1}, I_{k_2}, I_{k_3})$ with $k_1 + k_2 + k_3 = n + 3$, reproduce quivers obtained from 3-block exceptional collections on del Pezzo surfaces [175] – see *e.g.* [176–178]. Finally, let us mention that, importantly, a BPS quiver generally comes with a non-trivial superpotential, which should be computed, in principle, by a careful consideration of the disk instantons – see *e.g.* [179] for the case of the mirror to a toric threefold. It would be very interesting, but probably challenging, to compute the superpotential in the non-toric cases discussed below.

# 4 Rank-one 4d $\mathcal{N} = 2$ theories, revisited

In this section, we discuss the well-known case of purely four-dimensional rank-one theories. In particular, we revisit the Coulomb branches of the $SU(2)$ gauge theories with $N_f \le 3$ flavours, which are asymptotically free. This serves to illustrate the general formalism in a familiar setting. Moreover, our observations on the precise interpretation of the Mordell-Weil group of the SW geometry appear to be new. Finally, the 4d theories arise as limits of the 5d theories, and are thus important as such.

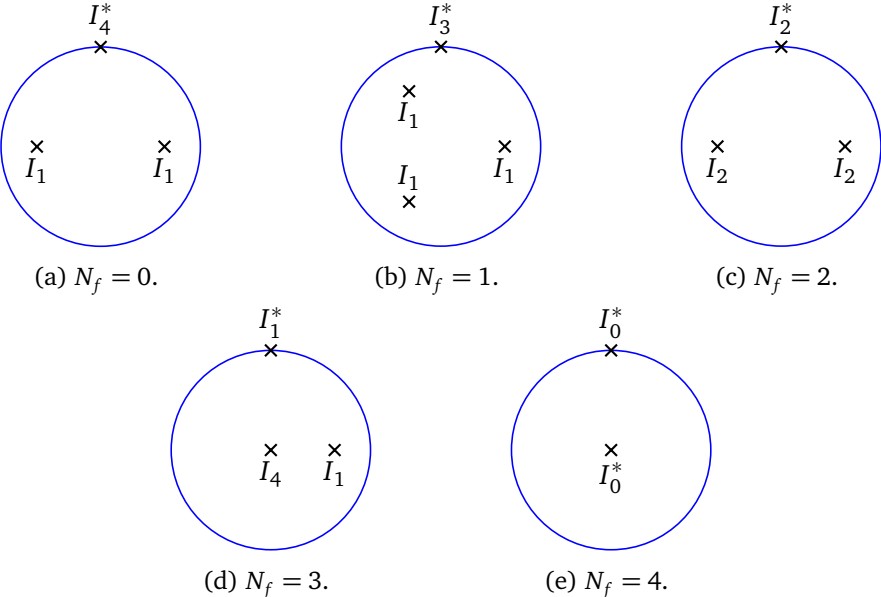

(a) $N_f = 0$.      (b) $N_f = 1$.      (c) $N_f = 2$.

(d) $N_f = 3$.      (e) $N_f = 4$.

Figure 3: The $u$-plane of 4d $\mathcal{N} = 2$ $SU(2)$ with $N_f$ massless flavours.

## 4.1 Four-dimensional theories: A bird's-eye view

Let us start with some general comments, before delving into more detailed computations in the next subsections. As we explained in the previous section, the SW geometry of the 4d $\mathcal{N} = 2$ $SU(2)$ gauge theory coupled to $N_f$ fundamental hypermultiplets is described by rational elliptic surfaces with $F_\infty = I^*_{4-N_f}$. In the limit of vanishing quark masses, the flavour symmetry group for $N_f > 1$ is the quotient of $\text{Spin}(2N_f)$ by its center, namely:

| $N_f$ | 2 | 3 | 4 |
|---|---|---|---|
| $G_F$ | $(SU(2)/\mathbb{Z}_2) \times (SU(2)/\mathbb{Z}_2)$ | $SU(4)/\mathbb{Z}_4$ | $\text{Spin}(8)/(\mathbb{Z}_2 \times \mathbb{Z}_2)$ |

(211)

For $N_f = 1$, the flavour symmetry is abelian. For $N_f = 0$, the flavour symmetry group is trivial and we have a $\mathbb{Z}_2$ electric one-form symmetry, $\mathcal{Z}^{[1]} = \mathbb{Z}_2$. The flavour symmetry groups (211) are easily understood in the free UV description: there is an $SO(2N_f)$ symmetry acting on $2N_f$ half-hypermultiplets in the fundamental of the $SU(2)$ gauge group, but the action of the $\mathbb{Z}_2$ center of $SO(2N_f)$ on the matter fields is equivalent to the action of the center of the gauge $SU(2)$. Therefore, the actual *flavour* symmetry is $SO(2N_f)/\mathbb{Z}_2$, which the same as (211). At first sight, this appears to be in tension with the discussion in [2], where it is shown that various dyons sit in spinors of $\text{Spin}(2N_f)$. These are not gauge-invariant states, however, thus there is no contradiction. In what follows, we will give other derivations of the flavour symmetry groups (211) by using the low-energy description. As a further confirmation, note that this global form of the flavour symmetry group is in perfect agreement with the Schur index as given in [180].

The $u$-planes for the massless 4d $SU(2)$ gauge theories are depicted in figure 3. If $N_f \neq 1$, the corresponding rational surfaces are extremal, in agreement with the fact that the flavour symmetry has no abelian factor. The Mordell-Weil group for all the massless theories are [46, 47]:

| $N_f$ | 0 | 1 | 2 | 3 | 4 |
|---|---|---|---|---|---|
| $\Phi$ | $\mathbb{Z}_2$ | $\mathbb{Z}$ | $\mathbb{Z}_2^2$ | $\mathbb{Z}_4$ | $\mathbb{Z}_2^2$ |

(212)

This agrees with our discussion from section 3.2. For the pure $SU(2)$ theory, there is no

Table 8: Number of distinct rational elliptic surfaces on the extended CB of each 4d theory.

| $SU(2), N_f$ flavours, $\quad F_\infty = I^*_{4-N_f}$ | $N_f = 0$ | $N_f = 1$ | $N_f = 2$ | $N_f = 3$ | $N_f = 4$ |
|---|---|---|---|---|---|
| # $\mathcal{S}$'s | 1 | 2 | 6 | 13 | 19 |
| AD theories, $\quad F_\infty = II^*, III^*, IV^*$ | | $H_0$ | $H_1$ | $H_2$ | |
| # $\mathcal{S}$'s | | 2 | 4 | 8 | |
| MN theories, $\quad F_\infty = IV, III, II$ | | $E_6$ | $E_7$ | $E_8$ | |
| # $\mathcal{S}$'s | | 49 | 93 | 137 | |

reducible Kodaira fiber in the interior and therefore $\mathcal{F} = 0$. Instead, $\Phi = \mathbb{Z}_2$ injects into $Z(F_\infty) = \mathbb{Z}_2^2$. According to the conjecture (186), it is then interpreted as the electric one-form symmetry of the $SU(2)$ gauge theory. For $N_f = 1$, the Mordell-Weil group is free and the flavour symmetry is abelian. For $N_f > 1$, $\Phi = \Phi_{\text{tor}} = \mathcal{F}$, which leads to (211).

For generic masses, we have $N_f + 2$ $I_1$ singularities in the interior of the Coulomb branch, and $\Phi = \mathbb{Z}^{N_f}$. As we vary the mass parameters, we can obtain a number of other singularities. In fact, we can obtain all possible configurations allowed by the classification of rational elliptic surfaces, at fixed $F_\infty$. The exact number of distinct configurations of Kodaira singularities for every 4d theory is given in table 8.

**AD points and flavour symmetry.** Some of these configurations are:

$$I^*_3 \oplus II \oplus I_1, \qquad I^*_2 \oplus III \oplus I_1, \qquad I^*_1 \oplus IV \oplus I_1, \qquad (213)$$

including the fiber at infinity, which are the maximal Argyres-Douglas points on the Coulomb branch of $SU(2)$ with $N_f = 1, 2, 3$. One can 'zoom in' onto the AD point, which amounts to merging the $I_1$ with the $I^*_{4-N_f}$ at infinity. This is the 4d $\mathcal{N} = 2$ SCFT limit, and the corresponding SW geometry is described by an extremal RES as in table 6.

For the $H_1$ and $H_2$ theories, we have $\mathbb{Z}_2$ and $\mathbb{Z}_3$ Mordell-Weil torsion which embeds diagonally into $Z(T) = \mathbb{Z}_2^2$ and $\mathbb{Z}_3^2$, respectively. Following our interpretation of the MW group, we find that the flavour symmetry group of the rank-one AD theories is:

$$G_F[H_0] = 0, \qquad G_F[H_1] = SU(2)/\mathbb{Z}_2, \qquad G_F[H_2] = SU(3)/\mathbb{Z}_3. \qquad (214)$$

This is in agreement with the Schur index computation [180, 181] and with the BPS spectrum, as we will discuss below. Incidentally, we also see that, according to (186), these AD theories do not have 1-form symmetries, in agreement with [78, 79].

**MN theories and flavour symmetry.** The remaining 'classic' 4d SCFTs are the $E_n$ MN theories. They are directly obtained from circle compactification of the 5d theory, as we reviewed in section 2.4.2. In the massless limit, they are described by the same rational elliptic surfaces as the AD theories, simply by sending $U$ to $1/U$ (see table 6). It then follows from our general considerations that:

$$G_F[E_6 \, \text{MN}] = E_6/\mathbb{Z}_3, \qquad G_F[E_7 \, \text{MN}] = E_7/\mathbb{Z}_2, \qquad G_F[E_8 \, \text{MN}] = E_8, \qquad (215)$$

just like their 5d parents. This same result was recently obtained in [34].

## 4.2  The pure $SU(2)$ SW solution

Let us first review the celebrated Seiberg-Witten solution for the four-dimensional $\mathcal{N} = 2$ supersymmetric $SU(2)$ gauge theory [1,2]. The Weierstrass form of the pure $SU(2)$ SW curve is given by [2]:

$$g_2(u) = \frac{4u^2}{3} - 4\Lambda^4, \qquad g_3(u) = -\frac{8u^3}{27} + \frac{4}{3}u\Lambda^4, \qquad (216)$$

with the discriminant $\Delta = 16\Lambda^8\left(u^2 - 4\Lambda^4\right)$. Fixing the dynamical scale such that $\Lambda^4 = \frac{1}{4}$ for convenience, the $u$-plane singularities are at $u_* = \pm 1$ and $\infty$. The $J$-invariant then reads:

$$J(u) = \frac{(4u^2 - 3)^3}{27(u^2 - 1)}, \qquad (217)$$

such that $J \sim u^4$ in the classical limit, while in the strong coupling regime we have $J \sim (u - u_*)^{-1}$. Consequently, the monodromies will be conjugate to $T^4$ and $T$, respectively, in SL$(2, \mathbb{Z})$. Let us note that the $J$-invariant only depends on $z \equiv u^2$.[27] Inverting the expression (217) in the weak coupling regime leads to:

$$u(\tau) = \frac{1}{8}\left(q^{-\frac{1}{4}} + 20q^{\frac{1}{4}} - 62q^{\frac{3}{4}} + 216q^{\frac{5}{4}} - 641q^{\frac{7}{4}} + 1636q^{\frac{9}{4}} + \mathcal{O}\left(q^{\frac{11}{4}}\right)\right). \qquad (218)$$

The coefficients match the McKay-Thompson series of class $4C$ for the Monster group [173], which are reproduced by the exact expression:

$$u(\tau) = \frac{\vartheta_2(\tau)^4 + \vartheta_3(\tau)^4}{2\vartheta_2(\tau)^2\vartheta_3(\tau)^2} = 1 + \frac{1}{8}\left(\frac{\eta(\frac{\tau}{4})}{\eta(\tau)}\right)^8. \qquad (219)$$

In this way, we verify that $u(\tau)$ is a modular function for $\Gamma^0(4)$, which is an index 6 subgroup with three cusps. Using the modular properties of either the $\vartheta$ functions or the Dedekind-$\eta$ function, one can find the $\tau$ values that correspond to the $u$-plane singularities. For instance, under an $S$-transformation:

$$u_D(\tau_D) = 1 + 32q_D + 256q_D^2 + 1408q_D^3 + \mathcal{O}\left(q_D^4\right), \qquad (220)$$

and thus the $u = 1$ singularity corresponds to the $\tau = 0$ cusp. Note that the sign of (219) changes under a $T^2$ transformation and thus the $u = -1$ cusp is mapped to $\tau = 2$ (or, equivalently, to $\tau = -2$). Incidentally, as pointed out in [103], the origin of the Coulomb branch corresponds to the points in the orbit of $\tau = 1 + i$. A fundamental domain for $\Gamma^0(4)$ consistent with these cusps is shown in figure 4a. We can read off the monodromies around the cusps from the choice of the fundamental domain, as discussed in section 3.4 and in appendix B. There are two strong coupling cusps at $\tau = 0, 2$ of width one (since they are $I_1$ type singularities), and one cusp of width 4 at infinity, with the associated monodromies:

$$\mathbb{M}_{u=1} = STS, \qquad \mathbb{M}_{u=-1} = (T^2S)T(T^2S)^{-1}, \qquad \mathbb{M}_\infty = PT^4. \qquad (221)$$

The periods $a$, $a_D$ can be written throughout the whole $u$-plane in terms of hypergeometric functions [182], which involves a choice of non-trivial branch cuts. In this approach, it is

---

[27]This is a manifestation of the spontaneously broken $\mathbb{Z}_2$ symmetry on the Coulomb branch. In terms of $z$, we have:

$$z(\tau) = 1 + \frac{1}{64}\left(\frac{\eta(\frac{\tau}{2})}{\eta(\tau)}\right)^{24} = \frac{1}{64}\left(q^{-\frac{1}{2}} + 40 + 276q^{\frac{1}{2}} - 2048q + \mathcal{O}\left(q^{\frac{1}{2}}\right)\right),$$

with coefficients matching the McKay-Thompson series of class $2B$ for the Monster group. This is a Hauptmodul for $\Gamma^0(2)$, a subgroup of index 3 with two cusps.

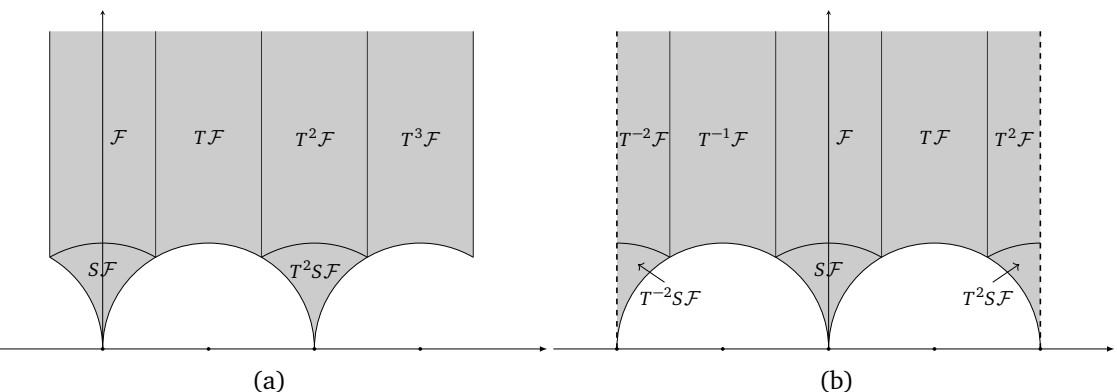

Figure 4: Fundamental domains for $\Gamma^0(4)$. Figure (a) shows a standard choice, with width one cusps at $\tau = 0$ and 2, while in figure (b) the cusp at $\tau = \pm 2$ is split, with the branch cut of the periods indicated by the dashed line.

directly apparent that the monodromies around the strong coupling singularities depend on the base-point in the $u$-plane. The standard choice of cuts for the hypergeometric functions corresponds to a 'splitting' of the fundamental domain as shown in figure 4b, such that the cusps at $\tau = -2$ and $\tau = 2$ are identified. We will come back to this picture when we discuss the 5d $E_1$ theory in section 5. Finally, we can determine the BPS states becoming massless at the $u$-plane singularities from the monodromies as follows. Recall that the monodromy around a singularity where $k$ particles of charge $(m, e)$ become massless is $\mathbb{M}^k_{(m,e)}$, with $\mathbb{M}_{(m,e)}$ given in (41). We thus see that the monopole $(1, 0)$ and dyons $(1, \pm 2)$ are the particles becoming massless at the $\tau = 0$ and $\pm 2$ cusps, respectively.

**Torsion and one-form symmetry.** The pure $SU(2)$ SW geometry has a Mordell-Weil group which is purely torsion, $\Phi = \mathbb{Z}_2$, generated by the section:

$$P = \left(\frac{u}{3}, 0\right), \qquad 2P = O. \tag{222}$$

Indeed, one easily checks that $P$ is a solution to $y^2 = 4x^3 - g_2 x - g_3$ with $g_2$ and $g_3$ given in (216). According to (186), this captures the electric one-form symmetry of the theory. Note that we can see the $\mathbb{Z}_2$ one-form symmetry more directly from the low-energy spectrum [31], following the logic of section 3.2.3. At strong coupling, the full spectrum is generated by the dyons that become massless at the cusps, while at weak coupling we have a tower of dyons and the $W$-boson [1, 182]:

$$\mathscr{S}_S : \ (1, 0), \quad (1, \pm 2), \qquad \mathscr{S}_W : \ (0, 2), \quad (1, 2n), \ n \in \mathbb{Z}. \tag{223}$$

In either regime, the spectrum is left invariant by:

$$g^{\mathscr{E}} = \left(0, \frac{1}{2}\right), \tag{224}$$

following the notation (200). We therefore have an electric one-form symmetry $\mathbb{Z}_2 \subset U(1)^{[1]}_e$, as expected from the UV description [31].

### 4.3 Asymptotically-free $SU(2)$ theories and Argyres-Douglas points

Consider the 4d $SU(2)$ gauge theories with $0 < N_f \leq 3$ flavours. Their SW curves, with all the mass parameters turned on, are given in appendix C, equation (C.1). The monodromy at

Table 9: Properties of the 4d $SU(2)$ theory with $N_f < 3$ flavours. Note that the modular groups for $N_f = 0$, 2 and 3 are in the same $\text{PSL}(2, \mathbb{R})$ conjugacy class.

| Theory | $\Delta(u) = 0$ | $F_{v \neq \infty}$ | $F_\infty$ | Modular Function | Monodromy | Cusps $\tau$ |
|---|---|---|---|---|---|---|
| $N_f = 0$ | $+1, -1$ | $I_1, I_1$ | $I_4^*$ | $u(\tau) = 1 + \frac{1}{8}\left(\frac{\eta(\frac{\tau}{4})}{\eta(\tau)}\right)^8$ | $\Gamma^0(4)$ | $0, 2, i\infty$ |
| $N_f = 1$ | $u^3 = 1$ | $3I_1$ | $I_3^*$ | $u^3 = \frac{2E_4(\tau)^{\frac{3}{2}}}{E_4(\tau)^{\frac{3}{2}} + E_6(\tau)}$ | $\Gamma_{N_f = 1}$ | $0, 1, 2, i\infty$ |
| $N_f = 2$ | $+1, -1$ | $I_2, I_2$ | $I_2^*$ | $u(\tau) = 1 + \frac{1}{8}\left(\frac{\eta(\frac{\tau}{2})}{\eta(2\tau)}\right)^8$ | $\Gamma(2)$ | $0, 1, i\infty$ |
| $N_f = 3$ | $0, 1$ | $I_4, I_1$ | $I_1^*$ | $u(\tau) = -\frac{1}{16}\left(\frac{\eta(\tau)}{\eta(4\tau)}\right)^8$ | $\Gamma_0(4)$ | $0, -\frac{1}{2}, i\infty$ |

infinity is determined by the one-loop $\beta$-function [1,2]:

$$\mathbb{M}_\infty = PT^{4-N_f} = \begin{pmatrix} -1 & N_f - 4 \\ 0 & -1 \end{pmatrix}, \tag{225}$$

which correspond to $F_\infty = I^*_{4-N_f}$. The $u$-planes of the massless theories are depicted in figure 3. The Weierstrass form of the SW curves are given by:

$$
\begin{aligned}
N_f = 1 : \quad & g_2(u) = \frac{4u^2}{3}, & g_3(u) = -\frac{8u^3}{27} + \frac{16}{27}, \\
N_f = 2 : \quad & g_2(u) = \frac{4u^2}{3} + 4, & g_3(u) = -\frac{8u^3}{27} + \frac{8u}{3}, \\
N_f = 3 : \quad & g_2(u) = \frac{4}{3}\left(u^2 - 16u + 16\right), & g_3(u) = -\frac{8}{27}\left(u^3 + 30u^2 - 96u + 64\right),
\end{aligned}
\tag{226}
$$

in the conventions of appendix C.1, where the dynamical scales are set to (C.3) for convenience. The $u$-planes manifest a spontaneously broken $\mathbb{Z}_{4-N_f}$ symmetry [2]. For $N_f = 2$ and $N_f = 3$, we have the non-trivial SW singularities $I_2 \oplus I_2$ and $I_4$, respectively, and the corresponding low-energy descriptions in terms of SQED with 2 or 4 electrons, respectively, reproduce the expected Higgs branches. For $N_f = 2$, the Higgs branch consists of two cones of the form $\mathbb{C}^2/\mathbb{Z}_2$ which arise at two distinct points on the Coulomb branch, and the two factors of the flavour group $G_F = SO(3) \times SO(3)$ act on these two cones independently. Similarly to the $N_f = 0$ case, one can directly compute $u(\tau)$ from the massless curves:

$$
\begin{aligned}
N_f = 1 : \quad & u(\tau) = \frac{1}{2^{\frac{2}{3}} 6}\left(q^{-\frac{1}{3}} + 104 q^{\frac{2}{3}} - 7396 q^{\frac{5}{3}} + \mathcal{O}\left(q^{\frac{8}{3}}\right)\right), \\
N_f = 2 : \quad & u(\tau) = \frac{1}{8}\left(q^{-\frac{1}{2}} + 20 q^{\frac{1}{2}} - 62 q^{\frac{3}{2}} + 216 q^{\frac{5}{2}} + O\left(q^{\frac{7}{2}}\right)\right), \\
N_f = 3 : \quad & u(\tau) = -\frac{1}{16}\left(q^{-1} - 8 + 20q - 62q^3 + 216q^5 + \mathcal{O}\left(q^6\right)\right).
\end{aligned}
\tag{227}
$$

For $N_f = 2, 3$, the coefficients of the $q$-series expansion of $u(\tau)$ are those of the McKay-Thompson series of class 4C, just like in (218). The monodromy group differs in each case, however. The basic global data of the massless $u$-planes are given in table 9. For any $N_f$, the relation $J(u) = J(q)$ gives a polynomial equation for $u(\tau)$ of order 6. It follows that the monodromy group $\Gamma_{N_f}$ for $N_f$ massless flavours should be an index 6 subgroup, with $T^{N_f - 4} \in \Gamma_{N_f}$.

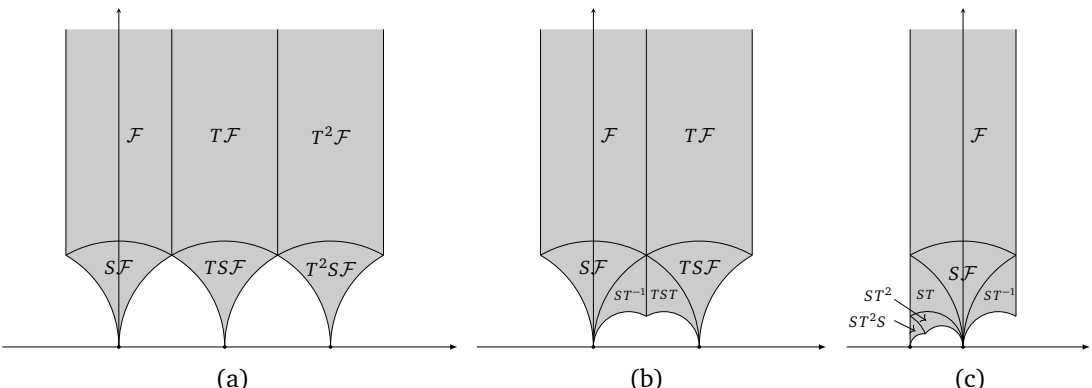

Figure 5: Fundamental domains for 4d $SU(2)$ theories with $N_f = 1, 2, 3$ flavours.

For $N_f = 1$, the theory has three strong coupling singularities, reflecting the residual $\mathbb{Z}_3$ symmetry of the $u$-plane. A similar curve arises in the context of the pure $SU(3)$ SYM theory [103]. In this case, $u(\tau)$ can be written in closed form in terms of fractional powers of Eisenstein series [84]. It is not a modular function for either SL$(2, \mathbb{Z})$ or any of the congruence subgroups. A possible fundamental domain for the monodromy group is shown in figure 5a. For the massless $N_f = 2$ theory, $u(\tau)$ turns out to be a modular function for $\Gamma(2)$. Similarly, the monodromy group for $N_f = 3$ is $\Gamma_0(4)$. Possible choices of fundamental domains are drawn in figures 5b and 5c. In summary, for all the asymptotically free $SU(2)$ gauge theories with $N_f \neq 1$, the $u$-plane is a modular curve, with the modular group as indicated.

We can again use the modular properties of $u(\tau)$ to match the cusps with the $u$-plane singularities in each case. For $N_f = 1$ and $N_f = 2$, the strong-coupling cusps are related by the $\mathbb{Z}_{4-N_f}$ symmetry of the $u$-plane. For $N_f = 3$, we use the transformations:

$$u(\tau) = -\frac{1}{16}\left(\frac{\eta(\tau)}{\eta(4\tau)}\right)^8 \xrightarrow{S} -16\left(\frac{\eta(\tau)}{\eta(\frac{\tau}{4})}\right)^8 \xrightarrow{T^2} 16\left(\frac{\eta(\frac{\tau}{4})\eta(\tau)^2}{\eta(\frac{\tau}{2})^3}\right)^8 \xrightarrow{S} \left(\frac{\eta(4\tau)\eta(\tau)^2}{\eta(2\tau)^3}\right)^8.$$

Here, $\tau$ successively denotes $\gamma\tau$ for the appropriate transformed element $\gamma \in$ SL$(2, \mathbb{Z})$. These relations can be proven using the identities reviewed in appendix B. They imply the identification of the $u = 1$ singularity with the cusp at $\tau = \frac{1}{2}$ (or $\tau = -\frac{1}{2}$). For all $N_f \leq 3$, the monodromies can then be read off from the list of coset representatives, which of course reproduces the well-known results [2].

In the rest of this section, we further comment on the global symmetry groups and on the possible Coulomb branch configurations, in particular the ones that include Argyres-Douglas points [4].

### 4.3.1 Symmetry group and BPS spectrum

Let us consider the global symmetry in each case, in order to check (211) from the infrared perspective. We consider the three cases in turn:

**$N_f = 1$, $G_F = U(1)$.** The flavour group is abelian and of rank one. Correspondingly, there is a single free generator of the Mordell-Weil group:

$$P_1 = \left(\frac{u}{3}, \Lambda^3\right), \tag{228}$$

with $\Lambda^3 = -\frac{4i}{3\sqrt{3}}$ in our conventions. Note that this section generates $\Phi \cong \mathbb{Z}$ also for non-zero mass $m$. As we take the limit $m_1 \to \infty$ with $m_1\Lambda^3$ fixed, $P_1$ becomes the torsion section (222) of the pure $SU(2)$ theory.

$N_f = 2$, $G_F = SO(3) \times SO(3)$.  This massless SW geometry has three torsion sections:

$$P_1 = \left(-\frac{2u}{3}, 0\right), \qquad P_2 = \left(\frac{1}{3}(u-3), 0\right), \qquad P_3 = \left(\frac{1}{3}(u+3), 0\right), \tag{229}$$

which satisfy $2P_i = O$, and $P_i + P_j = P_k$ for $i \neq j$ and $k \neq i, j$, thus spanning $\Phi = \mathbb{Z}_2 \times \mathbb{Z}_2$. Note that $P_2$ ($P_3$) intersects the 'trivial' component $\Theta_{\nu,0}$ of the $I_2$ singular fiber at $u = -1$ (and $u = 1$, respectively). Each of these sections generates a $\mathbb{Z}_2$ subgroup that injects into $\Phi$ according to (181). Since the subgroup of sections that are narrow is trivial and the two subgroups $\mathbb{Z}_2^{(f)} = \Phi/\mathbb{Z}_2$ act on the individual $SU(2)$ factors of the flavour symmetry, we find that $G_F = SO(3) \times SO(3)$, as previously mentioned. This global symmetry can also be understood directly from either the strong- or the weak-coupling spectrum [2, 183]:

$$\begin{aligned} \mathscr{S}_S &: (1, 0; 1, 0), && (\pm 1, 1; 0, 1), \\ \mathscr{S}_W &: (0, 2; 0, 0), && (0, 1; 1, 1), && (1, 2n; 1, 0), && (1, 2n+1; 0, 1). \end{aligned} \tag{230}$$

The charges are $(m, q; 2j_1, 2j_2)$ for a dyon $(m, q)$ in the representation of spin $(j_1, j_2)$ of the universal cover $\widetilde{G}_F = SU(2) \times SU(2)$. Moreover, $(2j_1, 2j_2)$ mod 2 is the charge under the $\mathbb{Z}_2 \times \mathbb{Z}_2$ center of $\widetilde{G}_F$. All these states are left invariant by the $\mathbb{Z}_2 \times \mathbb{Z}_2$ action generated by:

$$g^{\mathscr{E}} = \left(\frac{1}{2}, \frac{1}{2}; 1, 0\right), \quad \left(0, \frac{1}{2}; 0, 1\right), \tag{231}$$

in the notation (200), from which we conclude that the actual flavour group is $SO(3) \times SO(3)$.

$N_f = 3$, $G_F = PSU(4)$.  This massless SW geometry also has three torsion sections:

$$P_1 = \left(\frac{1}{3}(u+4), -4u\right), \quad P_2 = \left(-\frac{2}{3}(u-2), 0\right), \quad P_3 = \left(\frac{1}{3}(u+4), 4u\right), \tag{232}$$

which satisfy $P_k + P_l = P_{k+l \,(\text{mod}\, 4)}$ with $P_0 \equiv O$, thus spanning $\Phi = \mathbb{Z}_4$. Note that all sections intersect non-trivially the $I_4$ singular fiber. We then have the flavour group $G_F = PSU(4) = SU(4)/\mathbb{Z}_4$ by our general argument. This can also be verified at the level of the BPS spectrum. In the weak and strong coupling regions, we have [2, 183]:

$$\begin{aligned} \mathscr{S}_S &: (2, -1; 0), && (1, 0; 1), && (-1, 1; 1), \\ \mathscr{S}_W &: (0, 1; 2), && (0, 2; 0), && (1, 2n; 1), && (1, 2n+1; -1), && (2, 2n+1; 0), \end{aligned} \tag{233}$$

where the last entry $z$ (mod 4) in $(m, q; z)$ denotes the charge of the corresponding dyons under the center $\mathbb{Z}_4$ of $\text{Spin}(6) = SU(4)$. The $\mathbb{Z}_4$ action generated by:

$$g^{\mathscr{E}} = \left(-\frac{1}{4}, \frac{1}{2}; 1\right), \tag{234}$$

leaves all the BPS states invariants. The actual flavour group is therefore $PSU(4)$.

### 4.3.2  Configurations for $SU(2)$, $N_f = 1$

Let us now consider all possible distinct SW geometries for each theory, using the classification of rational elliptic surfaces. With one flavour, there are only two allowed configurations, listed below:

| $\{F_\nu\}$ | $m_1$ | $\mathfrak{g}_F$ | rk$(\Phi)$ | $\Phi_{\text{tor}}$ |
|---|---|---|---|---|
| $I_3^*, 3I_1$ | $m_1$ | $\mathfrak{u}(1)$ | 1 | $-$ |
| $I_3^*, II, I_1$ | $m_1^3 = \frac{27}{16}\Lambda^3$ | $\mathfrak{u}(1)$ | 1 | $-$ |

(235)

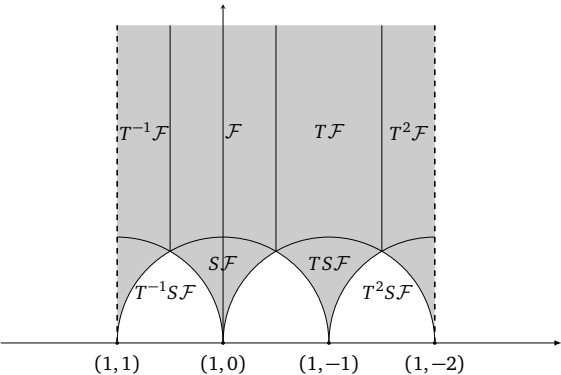

Figure 6: Fundamental domain for the massless 4d $SU(2)$ $N_f = 1$ theory, with the branch cut of the periods indicated by the dashed line. We also indicate the BPS particles becoming massless at the $I_1$ cusps.

The generic configuration $(I_3^*, 3I_1)$ includes the $\mathbb{Z}_3$-symmetric massless curve for $m_1 = 0$. At the three $I_1$ singularities, the dyons $(1,0)$, $(1,-1)$ and $(1,-2)$ become massless. This can be seen from the fundamental domain in figure (5a), as follows:

$$\mathbb{M}_{(\tau=k)} = (T^k S)T(T^k S)^{-1} = \mathbb{M}_{(1,-k)}, \tag{236}$$

for $k = 0, 1, 2$. In fact, keeping track of the branch cuts of the periods, one can 'split' the fundamental domain at $\tau = 2$ as shown in figure 6, so that either the dyon $(1,1)$ or the $(1,-2)$ become massless at that third cusp, depending on the sheet.

The second configuration in (235) is obtained by tuning the mass into the strong coupling region, as indicated. Let us consider the configuration containing the *AD* theory on the CB. Here, we fix $\Lambda = \frac{2^{4/3}}{3}$ so that $m_1 = 1$, for convenience. One then finds:

$$J(u) = -\frac{(3u-4)(3u+4)^3}{64(3u+5)}. \tag{237}$$

One root of $J = J(\tau)$ is:

$$u(\tau) = -\frac{5}{3} - \frac{1}{9}\left(\frac{\eta\left(\frac{\tau}{3}\right)}{\eta(\tau)}\right)^{12}, \tag{238}$$

which is the Hauptmodul of $\Gamma^0(3)$. We note that, in this case, the periods can be expressed in terms of hypergeometric functions. However, for our purposes, it suffices to read the monodromies from the fundamental domain.

The AD theory $H_0$ can be obtained in 3 equivalent ways, by 'colliding' a pair of $I_1$ cusps. At these points, two mutually non-local particles become massless. In terms of $u(\tau)$, we recover the Hauptmodul of $\Gamma^0(3)$ and its $T$ and $T^2$ transformations, respectively, in the three distinct cases. The pairs of BPS particles becoming massless in each case are the 'neighbouring' dyons indicated in figure 6. Let us also note that:

$$\begin{aligned}
\mathbb{M}_{(1,1)}\mathbb{M}_{(1,0)} &= (ST)^{-1}, \\
\mathbb{M}_{(1,0)}\mathbb{M}_{(1,-1)} &= T(ST)^{-1}T^{-1}, \\
\mathbb{M}_{(1,-1)}\mathbb{M}_{(1,-2)} &= T^2(ST)^{-1}T^{-2},
\end{aligned} \tag{239}$$

which is indeed, in each case, the monodromy for a singularity of type *II*. For the case when $u(\tau)$ is given by (238), the remaining $I_1$ cusp is at $\tau = 0$, where the dyon $(1,0)$ becomes massless, and we have the monodromies:

$$\mathbb{M}_{I_1} = STS^{-1}, \qquad \mathbb{M}_{II} = T^2(ST)^{-1}T^{-2}, \qquad \mathbb{M}_{\infty} = PT^3, \tag{240}$$

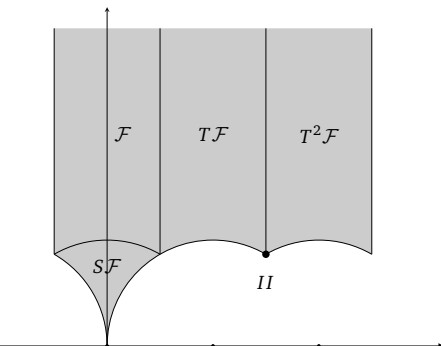

Figure 7: Fundamental domain for $\Gamma^0(3)$ corresponding to the configuration $(I_3^*, I_1, II)$ on the CB of the 4d $SU(2)$, $N_f = 1$ theory. The marked point $\tau = 2 + e^{2i\pi/3}$ is the elliptic point of the congruence subgroup $\Gamma^0(3)$.

which satisfy $\mathbb{M}_{I_1}\mathbb{M}_{II}\mathbb{M}_\infty = \mathbf{1}$. These monodromies are in agreement with the fundamental domain drawn in figure 7, where the AD theory appears at the elliptic point. Similar fundamental domains can be drawn for the other two cases, by shifting the cusp and the elliptic point appropriately.

### 4.3.3 Configurations for $SU(2)$, $N_f = 2$

With two flavours, there are six allowed configurations:

| $\{F_v\}$ | $m_1$ | $m_2$ | $\mathfrak{g}_F$ | rk($\Phi$) | $\Phi_{\text{tor}}$ |
|---|---|---|---|---|---|
| $I_2^*, 2I_2$ | 0 | 0 | $A_1 \oplus A_1$ | 0 | $\mathbb{Z}_2 \times \mathbb{Z}_2$ |
| $I_2^*, I_2, 2I_1$ | $m_1 \neq \pm\Lambda$ | $m_1$ | $A_1 \oplus \mathfrak{u}(1)$ | 1 | $\mathbb{Z}_2$ |
| $I_2^*, III, I_1$ | $m_1 = \pm\Lambda$ | $m_1$ | $A_1 \oplus \mathfrak{u}(1)$ | 1 | $\mathbb{Z}_2$ |
| $I_2^*, 2II$ | $\sqrt{2}e^{i\pi/4}\Lambda$ | $e^{i\pi/2}m_1$ | $2\mathfrak{u}(1)$ | 2 | — |
| $I_2^*, II, 2I_1$ | $\frac{1}{2}e^{3\pi i/4}\Lambda$ | $e^{i\pi/2}m_1$ | $2\mathfrak{u}(1)$ | 2 | — |
| $I_2^*, 4I_1$ | $m_1$ | $m_2$ | $2\mathfrak{u}(1)$ | 2 | — |

(241)

Here, we only listed some particular values of the parameters $m_{1,2}$ for the curve (C.1) which give rise to the corresponding configuration. For generic values of the masses, we have the last configuration, $4I_1$ singularities. The general structure of the extended CB can be understood starting from the large-mass limit [2]. For equal bare masses $m_1 = m_2 \gg \Lambda$, the two $I_1$ singularities corresponding to the quarks collide, giving rise to one $I_2$ singularity, from which emanate a classical Higgs branch $\mathbb{C}^2/\mathbb{Z}_2$, while we have the two $I_1$ singularities of the pure $SU(2)$ theory in the strong-coupling region. This configuration is the one on the second line in (241). Colliding this $I_2$ singularity with one of the other two $I_1$'s (which would correspond to the monopole and dyon of the pure $SU(2)$, in the large mass limit), we can obtain a type-$III$ singularity, as two types of mutually non-local BPS states become massless [4]. This is the third line in (241).

Recall that the massless configuration $(I_2^*, 2I_2)$ corresponds to an extremal rational elliptic surface associated to the congruence subgroup $\Gamma(2)$ with $u(\tau)$ given in (227) and in table 9. By performing an $S$-transformation, one can show that one $I_2$ cusp is at $u(\tau = 0) = 1$. Furthermore, $u(\tau)$ changes sign under a $T$, transformation, so that $u(\tau = \pm 1) = -1$. In the

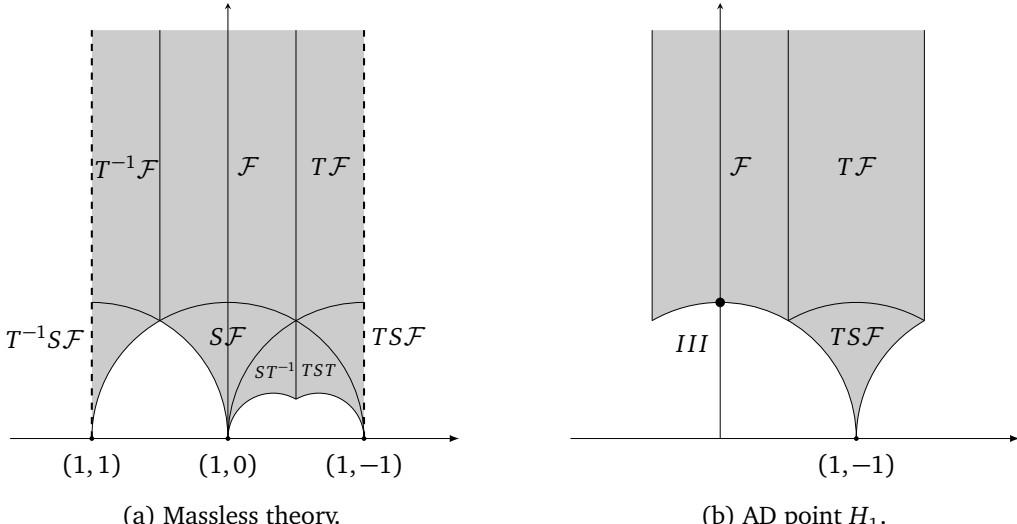

(a) Massless theory.

(b) AD point $H_1$.

Figure 8: Fundamental domains for the 4d $SU(2)$ $N_f = 2$ theory. (a) Choice of fundamental domain for $m_1 = m_2 = 0$ with a branch cut structure indicated by the dotted lines. The $\tau = 0, \pm 1$ singularities are $I_2$ cusps. (b) Choice of fundamental domain for the configuration involving a type-$III$ singularity. Three BPS states become massless at the point at $\tau = i$.

large mass picture, one $I_2$ cusp is obtained from the pair of quarks, while the other $I_2$ is obtained by 'colliding' the monopole and dyon of pure $SU(2)$ after a non-trivial monodromy as $m_1 = m_2 \to 0$ [4]. At this point, the flavour symmetry is enhanced from $\mathfrak{u}(2)$ to $\mathfrak{so}(4)$ and the 'second' $\mathbb{C}^2/\mathbb{Z}_2$ Higgs branch appears. The BPS states becoming massless at the two $I_2$ cusps of the massless theory have charges $(1,0)$ and $(1,\pm 1)$, in agreement with [183]. The corresponding fundamental domain is shown in figure 8a. In particular, the monodromy matrices:

$$\mathbb{M}_{I_2} = \mathbb{M}_{(1,0)}^2 = ST^2 S^{-1}, \quad \mathbb{M}'_{I_2} = \mathbb{M}_{(1,\pm 1)}^2 = (T^{\mp 1}S)T^2(T^{\mp 1}S)^{-1}, \quad \mathbb{M}_\infty = PT^2, \quad (242)$$

can be read off directly from figure 8a. Let us analyse in more detail the configuration containing the type-$III$ singularity. For $m_1 = m_2 = \Lambda$, with $\Lambda = \sqrt{2}$, we find:

$$
\begin{aligned}
u(\tau) &= -5 + \frac{1}{8}\left(\frac{\eta(\tau)^2}{\eta(\frac{\tau}{2})\eta(2\tau)}\right)^{24} \\
&= \frac{1}{8}\left(\frac{1}{q^{\frac{1}{2}}} - 16 + 276 q^{\frac{1}{2}} + 2048 q + 11202 q^{\frac{3}{2}} + \mathcal{O}(q^2)\right),
\end{aligned}
\tag{243}
$$

whose coefficients match the 4$A$ McKay-Thompson series of the monster group [173]. The expression (243) is related by a $T$ transformation to the McKay-Thompson series of class 2$B$, namely:

$$u(\tau) = -5 - \frac{1}{8}\left(\frac{\eta\left(\frac{\tau \pm 1}{2}\right)}{\eta(\tau \pm 1)}\right)^{24}, \tag{244}$$

which is the Hauptmodul of the congruence subgroup $\Gamma^0(2)$. Compared to the analysis for $N_f = 1$, here we split one of the $I_2$ cusps of the massless $N_f = 2$ theory into two $I_1$'s, and collide one of those with the other $I_2$ cusp. Thus, in terms of the BPS particles, one of the two dyons of the split $I_2$, say a dyon $(1,\mp 1)$, will also become massless, together with the two

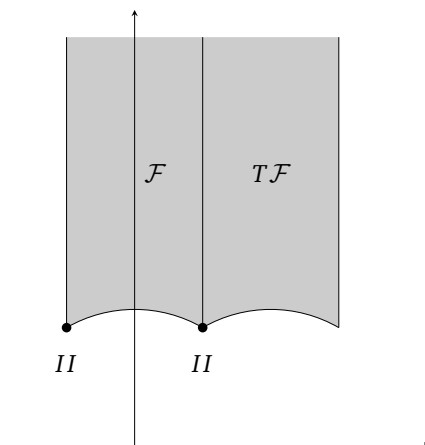

Figure 9: Fundamental domain for the $(I_2^*, 2II)$ configuration on the CB of the 4d $SU(2)$ $N_f = 2$ theory.

monopoles $(1, 0)$ at the other $I_2$ cusp. Indeed, we have:

$$\mathbb{M}_{(1,0)}^2 \mathbb{M}_{(1,-1)} = \mathbb{M}_{(1,1)} \mathbb{M}_{(1,0)}^2 = S^{-1}, \tag{245}$$

which is exactly the monodromy around the type-*III* elliptic point at $\tau = i$ in figure 8b. A similar analysis can be done for the case when the two $(1, \pm 1)$ dyons and a monopole $(1, 0)$ collide to form a type-*III* singularity.

Let us also note that the flavour symmetry group $SO(3)$ of the AD point can be corroborated from the corresponding BPS states. Indeed, at the AD point in the configuration above, we have a single dyon $(1, \mp 1; 0)$ and a doublet of $\widetilde{G}_F = SU(2)$, denoted by $(1, 0; 1)$, where the last entry denotes the charge under the center of $SU(2)$. We then find that the actual flavour symmetry is $SU(2)/\mathbb{Z}_2$, by the same argument as for the massless curve. Moreover, the full spectrum of the 4d gauge theory is compatible with that symmetry, which agrees with the fact that the MW group of this configuration is $\Phi = \mathbb{Z}_2$.

Finally, the remaining possibilities in (241) involve 'colliding' $I_1$'s corresponding to $(1, 0)$ states with the $(1, \pm 1)$ states, forming type-*II* singularities. For instance, we can obtain the $(I_2^*, 2II)$ configuration by tuning the masses to the values given in (241). In that case, we find:

$$J(u) = 1 + \frac{u^2}{27} \qquad \Longleftrightarrow \qquad u^2(\tau) = -27 + 27 \frac{E_4(\tau)^3}{E_4(\tau)^3 - E_6(\tau)^2}, \tag{246}$$

with $u(\tau)$ itself having coefficients matching the McKay-Thompson series of class 2*A*. In this case, the monodromy group is $\Gamma^2$, the group whose elements are the squares of PSL$(2, \mathbb{Z})$, with the fundamental domain drawn in figure 9. Note that the type-*II* singularities are elliptic points of order 3, namely those for which $J(\tau_*) = 0$, which correspond to the zeroes of the Eisenstein series $E_4$ at $\tau_* = e^{2i\pi/3}$. As a result, we expect two type-*II* elliptic points at $\tau_*$ and $1 + \tau_*$. We note that:

$$\mathbb{M}_{(1,1)} \mathbb{M}_{(1,0)} = (ST)^{-1}, \qquad \mathbb{M}_{(1,0)} \mathbb{M}_{(1,-1)} = T(ST)^{-1} T^{-1}, \tag{247}$$

in agreement with the domain shown in figure 9.

### 4.3.4  Configurations for $SU(2)$, $N_f = 3$

With three flavours, there are 13 allowed configurations:

| $\{F_v\}$ | $m_1$ | $m_2$ | $m_3$ | $\mathfrak{g}_F$ | $\mathrm{rk}(\Phi)$ | $\Phi_{\mathrm{tor}}$ |
|---|---|---|---|---|---|---|
| $I_1^*, I_4, I_1$ | $0$ | $0$ | $0$ | $A_3$ | $0$ | $\mathbb{Z}_4$ |
| $I_1^*, I_3, 2I_1$ | $m_1$ | $m_1$ | $m_1$ | $A_2 \oplus \mathfrak{u}(1)$ | $1$ | $-$ |
| $I_1^*, IV, I_1$ | $\Lambda/2$ | $m_1$ | $m_1$ | $A_2 \oplus \mathfrak{u}(1)$ | $1$ | $-$ |
| $I_1^*, I_3, II$ | $-\Lambda/16$ | $m_1$ | $m_1$ | $A_2 \oplus \mathfrak{u}(1)$ | $1$ | $-$ |
| $I_1^*, III, I_2$ | $\Lambda/4$ | $0$ | $0$ | $2A_1 \oplus \mathfrak{u}(1)$ | $1$ | $\mathbb{Z}_2$ |
| $I_1^*, 2I_2, I_1$ | $m_1$ | $0$ | $0$ | $2A_1 \oplus \mathfrak{u}(1)$ | $1$ | $\mathbb{Z}_2$ |
| $I_1^*, III, II$ | $-\frac{7}{4}\Lambda$ | $i\sqrt{2}\Lambda$ | $m_1$ | $A_1 \oplus 2\mathfrak{u}(1)$ | $2$ | $-$ |
| $I_1^*, III, 2I_1$ | $\frac{m_2^2}{\Lambda} + \frac{\Lambda}{4}$ | $m_2$ | $m_1$ | $A_1 \oplus 2\mathfrak{u}(1)$ | $2$ | $-$ |
| $I_1^*, II, I_2, I_1$ | $m_1$ | $\frac{(4m_1 + \Lambda)^{3/2}}{6\sqrt{3\Lambda}}$ | $m_1$ | $A_1 \oplus 2\mathfrak{u}(1)$ | $2$ | $-$ |
| $I_1^*, I_2, 3I_1$ | $m_1$ | $m_2$ | $m_1$ | $A_1 \oplus 2\mathfrak{u}(1)$ | $2$ | $-$ |
| $I_1^*, 2II, I_1$ | $\left(-2T_2\Lambda + \frac{13}{8}\Lambda^3,\ 5T_2\Lambda^2 - \frac{57}{16}\Lambda^4\right)$ | | | $3\mathfrak{u}(1)$ | $3$ | $-$ |
| $I_1^*, II, 3I_1$ | $\left(\frac{1}{4}T_2\Lambda - \frac{1}{16}\Lambda^3,\ \frac{1}{2}T_2\Lambda^2 - \frac{3}{16}\Lambda^4\right)$ | | | $3\mathfrak{u}(1)$ | $3$ | $-$ |
| $I_1^*, 5I_1$ | $m_1$ | $m_2$ | $m_3$ | $3\mathfrak{u}(1)$ | $3$ | $-$ |

(248)

We use the $N_f = 3$ curve given in (C.4), and we again only specified the masses for some simple configurations of interest. For the configurations $(I_1^*, 2II, I_1)$ and $(I_1^*, II, 3I_1)$ we give values for the $SO(6)$ Casimirs $(T_3, T_4)$ defined in (C.2). Note that the $(I_1^*, II, 3I_1)$ configuration can be in fact obtained by only 'fixing' one of the Casimirs. Here we give a subfamily for which this configuration is realized. For generic masses, we have $5I_1$ singularities. For equal bare masses $m_1 = m_2 = m_3$, three of these singularities collide, forming an $I_3$ singularity, with flavour symmetry $\mathfrak{u}(3)$, as in the second line in (248). In the massless limit, the $I_3$ merges with another $I_1$ cusp, leading to the enhanced $\mathfrak{so}(6) \cong \mathfrak{su}(4)$ flavour algebra, and the Higgs branch dimension increases accordingly. This is the first line in (248). As discussed above, the massless configuration $(I_1^*, I_4, I_1)$ corresponds to the extremal rational elliptic surface $X_{141}$, and in that case the $u$-plane is a modular curve for $\Gamma_0(4)$. From the fundamental domain shown in figure 5c, one can read off the monodromies:

$$\mathbb{M}_{I_4} = ST^4 S^{-1} = \mathbb{M}_{(1,0)}^4, \quad \mathbb{M}_{I_1}^\pm = (ST^{\pm 2}S)T(ST^{\pm 2}S)^{-1} = \mathbb{M}_{(2,\pm 1)}, \quad \mathbb{M}_\infty = PT. \quad (249)$$

They satisfy:

$$\mathbb{M}_{I_4}\mathbb{M}_{I_1}^-\mathbb{M}_\infty = \mathbb{M}_{I_1}^+\mathbb{M}_{I_4}\mathbb{M}_\infty = \mathbf{1}, \quad (250)$$

as expected. Therefore, in the massless gauge theory, we have four monopoles $(1,0)$ becoming massless at $\tau = 0$ and one dyon $(2, \mp 1)$ becoming massless at $\tau = \pm\frac{1}{2}$. Note that the BPS quiver

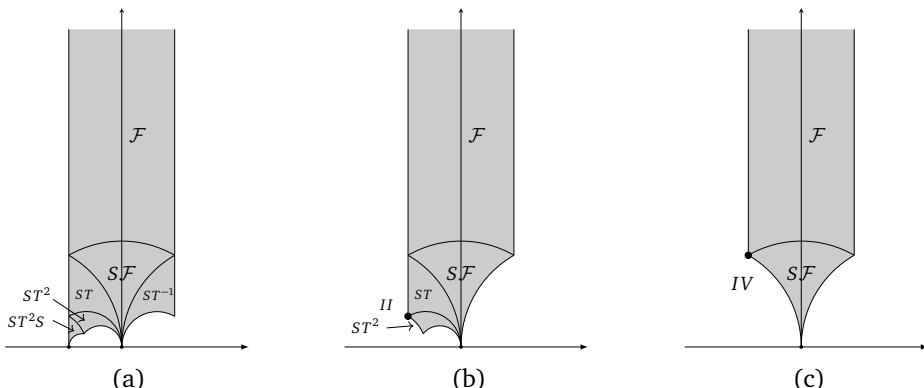

Figure 10: Fundamental domains for $SU(2)$ $N_f = 3$ configurations. (a) $\Gamma_0(4)$ corresponding to $(I_1^*, I_4, I_1)$, the massless configuration. (b) $\Gamma_0(3)$ corresponding to the configuration $(I_1^*, I_3, II)$, involving the $H_0$ AD point. (c) Fundamental domain for the CB configuration $(I_1^*, IV, I_1)$ involving the $H_2$ AD point. This last configuration is not modular. The transitions between these configurations can be seen from the fundamental domains.

for that massless configuration takes the form [86]:

$$
\begin{array}{c}
\mathcal{E}_{\gamma_2 = (1,0)} \\
\big\uparrow \\
\mathcal{E}_{\gamma_1 = (1,0)} \longleftarrow \mathcal{E}_{\gamma_5 = (2,-1)} \longrightarrow \mathcal{E}_{\gamma_3 = (1,0)} \qquad \cong \qquad \mathcal{E}_{\gamma_{1,2,3,4} = (1,0)} \longleftarrow \mathcal{E}_{\gamma_5 = (2,-1)} \\
\big\downarrow \\
\mathcal{E}_{\gamma_4 = (1,0)}
\end{array}
\qquad (251)
$$

Note that, depending on the base point on the $U$-plane, we have either the dyon $\gamma_5 = (2,-1)$ or $\gamma_5' = (2,1)$. The corresponding quivers differ by the orientation of the arrows, and are related by a quiver mutation (see *e.g.* [86]) on the central node. On the right-hand-side of (251), and in the following sections, we use the 'block' notation, wherein several nodes with the same dyonic charge (the four monopoles $(1,0)$, in this case) are written as one node, with the understanding that the arrows between blocks connect each node of one block to each node of the other.

Another interesting limit is obtained by starting with equal masses $m_1 = m_2 = m_3$ and by tuning that equal mass $m_1 = \Lambda/2$, as shown on the third line of (248). In that case, the $I_3$ cusp merges with one $I_1$ cusp, forming an $IV$ singularity, while the Higgs branch remains the same. Setting $\Lambda = 4$ for simplicity, we find:

$$
u(\tau) = -19 - 27 \frac{E_4(\tau)^{3/2} + E_6(\tau)}{E_4(\tau)^{3/2} - E_6(\tau)} . \qquad (252)
$$

The elliptic point of type $IV$ corresponds to the zero of the Eisenstein series $E_4$, namely $\tau_* = e^{2i\pi/3}$. The monodromies are:

$$
\mathbb{M}_{IV} = (ST)^{-2} , \qquad \mathbb{M}_{I_1} = STS^{-1} , \qquad \mathbb{M}_\infty = PT , \qquad (253)
$$

satisfying $\mathbb{M}_{IV} \mathbb{M}_{I_1} \mathbb{M}_\infty = \mathbf{1}$, as expected. Importantly, we have:

$$
\mathbb{M}_{IV} = \mathbb{M}_{(2,1)} \mathbb{M}_{(1,0)}^3 , \qquad (254)
$$

so that the AD point can be interpreted as having three monopoles $(1,0)$ and one dyon $(2,1)$ becoming massless [4]. This corresponds to 'spitting' the $I_4$ cusp of the massless gauge theory into $I_3 \oplus I_1$ before merging the $I_3$ with the other, mutually non-local, $I_1$ singularity. The fundamental domain for this configuration is shown in figure 10c. It follows that the BPS quiver for the AD theory $H_2$ can be obtained from (251) by deleting one of the four 'monopole' nodes, say $\mathcal{E}_{\gamma_1 = (1,0)}$.

Let us also comment on the flavour group at this AD point. The flavour symmetry algebra of the configuration, which can be read off from the SW geometry, is $\mathfrak{su}(3) \oplus \mathfrak{u}(1)$, which corresponds precisely to the splitting $I_4 \to I_3 \oplus I_1$. Moreover, the Mordell-Weil group is torsionless in this case. If we focus on the AD point itself, the relevant BPS particles are $(2,1;0)$ and $(1,0;1)$, and the corresponding flavour symmetry group is $SU(3)/\mathbb{Z}_3$ due to the $\mathbb{Z}_3$ action $(\frac{1}{3}, \frac{1}{3}; 1)$ that leaves the configuration invariant. On the other hand, the full theory includes the additional flavour singlet $(1,0;0)$ which is not invariant under this $\mathbb{Z}_3$, and therefore the flavour symmetry group of the full configuration is $SU(3)$. As we 'zoom in' into the AD point, one sends that additional dyon singularity to infinity, fusing $I_1^* \oplus I_1 \to IV^*$ to obtain the configuration $(IV^*, IV)$ of the 4d SCFT $H_2$. In that limit, we recover the flavour group $SU(3)/\mathbb{Z}_3$, consistently with the non-trivial Mordell-Weil torsion of that limiting configuration.

All the other configurations in (248) can be discussed similarly. For instance, if we set $m_1 = m_2 = m_3 = -\Lambda/16$, we obtain the configuration $(I_1^*, I_3, II)$ on the fourth line in (248). In that case, we find:

$$u(\tau) = -\frac{1}{16}\left(7 + \left(\frac{\eta(\tau)}{\eta(3\tau)}\right)^{12}\right),$$

(255)

which is the Hauptmodul for $\Gamma_0(3)$. The fundamental domain for this configuration is shown in figure 10b. We find the monodromies:

$$\mathbb{M}_{I_3} = ST^3S^{-1} = \mathbb{M}_{(1,0)}^3, \qquad \mathbb{M}_{II} = ST^2(ST)^{-1}(ST^2)^{-1} = \mathbb{M}_{(2,1)}\mathbb{M}_{(1,0)},$$

(256)

which satisfy $\mathbb{M}_{II}\mathbb{M}_{I_3}\mathbb{M}_\infty = \mathbf{1}$.

# 5 The $E_1$ and $\widetilde{E}_1$ theories – 5d $SU(2)_\theta$

In this and the next two sections, we explore the $U$-plane of the $E_n$ theories with $n \leq 3$. The corresponding toric geometries in Type IIA, and their Type IIB mirror, have been well studied in the literature – see e.g. [12, 18, 23, 184]. Here, we focus on the 5d interpretation and conduct a systematic analysis of the possible Coulomb branch configurations. Moreover, we solve for the physical periods as explicitly as possible for some interesting values of the masses, and in particular in the massless limit. We also discuss the modular properties of the $U$-plane as well as aspects of the global symmetries, following the general approach outlined in the previous sections.

## 5.1 The $E_1$ theory – 5d $SU(2)_0$: $\mathbb{Z}_4$ torsion and BPS quivers

Let us first consider the $E_1$ theory, which is the UV completion of the five-dimensional $SU(2)_0$ gauge theory [13]. Its SW curve was first derived and studied in [12,19]. The 'toric' expression (99) for the curve can be brought to the Weierstrass form (48), with:

$$g_2(U) = \frac{1}{12}\left(U^4 - 8(1+\lambda)U^2 + 16(1-\lambda+\lambda^2)\right),$$

$$g_3(U) = -\frac{1}{216}\left(U^6 - 12(1+\lambda)U^4 + 24(2+\lambda+2\lambda^2)U^2 - 32(2-3\lambda-3\lambda^2+2\lambda^3)\right),$$

(257)

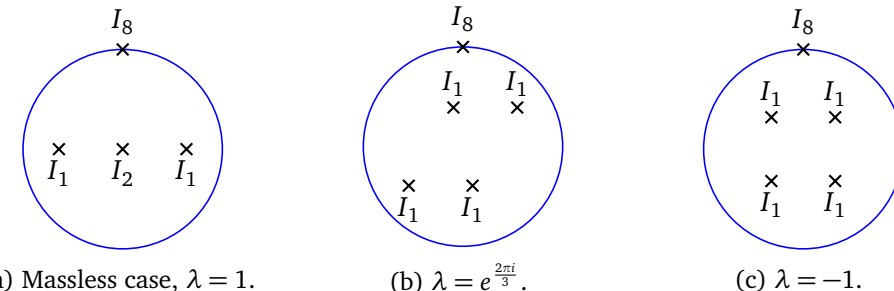

| (a) Massless case, $\lambda = 1$. | (b) $\lambda = e^{\frac{2\pi i}{3}}$. | (c) $\lambda = -1$. |

Figure 11: The $U$-plane of the $D_{S^1}E_1$ theory for some values of $\lambda$. Notice the $\mathbb{Z}_2$ symmetry, which is enhanced to $\mathbb{Z}_4$ at $\lambda = -1$.

Table 10: The two configurations of singular fibers for the $E_1$ theory.

| $\{F_v\}$ | $\lambda$ | $\mathfrak{g}_F$ | $\mathrm{rk}(\Phi)$ | $\Phi_{\mathrm{tor}}$ |
|---|---|---|---|---|
| $I_8, 2I_1, I_2$ | 1 | $A_1$ | 0 | $\mathbb{Z}_4$ |
| $I_8, 4I_1$ | $\lambda \neq 1$ | $\mathfrak{u}(1)$ | 1 | $\mathbb{Z}_2$ |

and with discriminant:

$$\Delta(U) = \lambda^2 \Big( U^4 - 8(1+\lambda)U^2 + 16(1-\lambda)^2 \Big). \tag{258}$$

At generic values of $\lambda$, the discriminant has four distinct roots, and we have four distinct $I_1$ singularities in the interior of the $U$-plane, plus the $I_8$ singularity at infinity – see figure 11. Note that $g_2$ and $g_3$ in (257) depend on $U^2$ instead of $U$, and therefore the $\mathbb{Z}_2$ action:

$$\mathbb{Z}_2 : \quad U \to -U, \tag{259}$$

is a symmetry of the $U$-plane for any value of the complexified 5d gauge coupling, $\lambda$. This symmetry has a simple physical explanation. Recall that $U$ is defined as the expectation value of the five-dimensional fundamental Wilson loop wrapped on $S^1$. Then (259) is precisely the action of the $\mathbb{Z}_2$ one-form symmetry of the $E_1$ theory [76, 77], which gives rise to both a one-form and an ordinary (zero-form) symmetry of the KK theory $D_{S^1}E_1$. Both are spontaneously broken on the Coulomb branch. It is useful to note that the $E_1$ SW geometry is a two-to-one covering of the pure $SU(2)$ geometry [19]:

$$u_{(4d)} \longleftrightarrow \frac{U^2}{4} - 1 - \lambda, \qquad \Lambda^4_{(4d)} \longleftrightarrow \lambda, \tag{260}$$

with the 4d curve given in (216). Let us study the $U$-plane in some detail. There are two configurations of singular fibers depending on the value of the parameter $\lambda$, as shown in table 10. The case $\lambda = 1$ is the massless point, which gives us the low-energy description of the 5d SCFT $\mathbb{R}^4 \times S^1$ with vanishing real masses and without any non-trivial flavour Wilson line. For $\lambda \neq 1$, the corresponding configuration $(I_8, 4I_1)$ is not extremal, with the $\mathfrak{su}(2)$ flavour algebra broken to the Cartan subalgebra. The point $\lambda = -1$ corresponds to setting to zero the fundamental flavour Wilson line for $E_1 = \mathfrak{su}(2)$:

$$\chi_1^{E_1} = \sqrt{\lambda} + \frac{1}{\sqrt{\lambda}} = 0. \tag{261}$$

The corresponding $U$-plane turns out to be $\mathbb{Z}_4$ symmetric, and the periods can be expressed in terms of hypergeometric functions.

### 5.1.1 The $E_1$ massless curve

Let us first consider the massless $E_1$ curve, by fixing $\lambda = 1$. In this case, it will be useful to consider the variable:[28]

$$w = \frac{U^2}{16} = \frac{1}{16 z_f}. \tag{262}$$

Let $(a_D, a)$ be the physical periods, which are related to the D4- and D2-brane periods as discussed in section 2.4. We also introduce the geometric periods $\omega = \frac{d\Pi}{dU}$, namely:

$$\Pi_{D4} = a_D, \qquad \Pi_{D2_f} = 2a, \qquad \omega_D = \frac{da_D}{dU}, \qquad \omega_a = \frac{da}{dU}, \tag{263}$$

with the D4 period as given by (68):

$$\Pi_{D4} = \Pi_{D2_f} \Pi_{D2_b} + \frac{1}{6} = 2a\left(2a + \frac{1}{2\pi i}\log(\lambda)\right) + \frac{1}{6}. \tag{264}$$

These geometric periods satisfy the Picard-Fuchs equation:

$$\frac{d^2\omega}{dU^2} + \frac{3U^2 - 16}{U(U^2 - 16)}\frac{d\omega}{dU} + \frac{1}{U^2 - 16}\omega = 0. \tag{265}$$

One can analyse the solutions to this equation, and their monodromies, rather explicitly. We will first discuss the periods on the $w$-plane, before going back to the physical $U$-plane.

**Geometric periods on the $w$-plane.** In terms of $w$ defined as in (262), the massless $E_1$ curve takes the Weierstrass form:

$$g_2(w) = \frac{4}{3}(16w^2 - 16w + 1), \qquad g_3(w) = -\frac{8}{27}(64w^3 - 96w^2 + 30w + 1), \tag{266}$$

with a discriminant $\Delta(w) = 256(w-1)w$. Thus, this one-parameter family of curves has two $I_1$ singularities (at $w = 0$ and $w = 1$) in the interior of the $w$-plane. One can also check that the fiber at infinity is of type $I_4^*$, which is consistent with the fact that the $w$-plane is isomorphic to the four-dimensional pure $SU(2)$ Coulomb branch. The $w$-plane analysis that follows is then essentially the same as in [182].

Starting from the curve (266), one can consider a distinct curve obtained by a rescaling:

$$(g_2, g_3) \to \left((16w)^2 g_2, (16w)^3 g_3\right). \tag{267}$$

This is a quadratic twist, as explained around (136). The Kodaira singular fibers are then transmuted according to:

$$w = (0, 1, \infty): \qquad (I_1, I_1, I_4^*) \to (I_1^*, I_1, I_4). \tag{268}$$

This operation is equivalent to a rescaling of the coordinates $(x, y) \to \left((16w)^{-1}x, (16w)^{-\frac{3}{2}}y\right)$, which has the effect of rescaling the holomorphic one-form as $\boldsymbol{\omega} \to (16w)^{-\frac{1}{2}}\boldsymbol{\omega}$. In addition, we find it convenient to multiply the geometric periods of this new curve by the regular function $16w$, so that we actually consider:

$$\widetilde{\omega}_\gamma \equiv \sqrt{16w}\,\omega_\gamma = U\frac{d\Pi_\gamma}{dU}. \tag{269}$$

---

[28]This $w$ is distinct from the variable $w \in \mathbb{C}^*$ used to describe the mirror curves $F(t, w) = 0$ in the previous section. We will discuss various distinct '$w$-planes' in the next subsections. This should cause no confusion.

Let us now study these rescaled geometric periods on the $w$-plane. Firstly, they satisfy the following PF equation:

$$(w-1)\frac{d\widetilde{\omega}}{dw^2} + \frac{d\widetilde{\omega}}{dw} - \frac{1}{4w^2}\widetilde{\omega} = 0. \tag{270}$$

This is a standard hypergeometric differential equation, with singularities at $w = 0, 1, \infty$. In particular, at each of the three singularities there is only one regular solution. A convenient basis of solutions is given by:

$$\widetilde{\omega}_a(w) = -\frac{1}{2\pi i}\, {}_2F_1\left(\frac{1}{2}, \frac{1}{2}, 1; \frac{1}{w}\right), \qquad \widetilde{\omega}_D(w) = -\frac{1}{\pi}\, {}_2F_1\left(\frac{1}{2}, \frac{1}{2}, 1; 1-\frac{1}{w}\right). \tag{271}$$

The period $\widetilde{\omega}_a$ is regular in the large volume limit, $w = \infty$, while the 'dual period' $\widetilde{\omega}_D$ is regular at the 'conifold point', $w = 1$. A Gauss-Ramanujan identity for the hypergeometric function provides a way of analytically continuing these solutions past their respective regions of convergence, to unit argument:

$$_2F_1\left(\frac{1}{2}, \frac{1}{2}, 1; x\right) \approx \frac{4}{\pi}\log(2) - \frac{1}{\pi}\log(1-x) + \mathcal{O}\left[(1-x)\log(1-x)\right]. \tag{272}$$

To analytically continue to $w = 0$, we use the Barnes integral representation:

$$_2F_1\left(\frac{1}{2}, \frac{1}{2}, 1; x\right) = \frac{1}{2\pi i \, \Gamma\left(\frac{1}{2}\right)^2} \int_{-i\infty}^{i\infty} \frac{\Gamma\left(\frac{1}{2}+t\right)^2 \Gamma(-t)}{\Gamma(1+t)}(-x)^t dt, \tag{273}$$

for $|\arg(-x)| < \pi$. The integration contour separates the poles of $\Gamma\left(\frac{1}{2}+t\right)$ from those of $\Gamma(-t)$. Consequently, when closing the contour to the right we recover the regular solution for $\widetilde{\omega}_a$ at $w = \infty$, while when closing it to the left we find the asymptotic expansion:

$$_2F_1\left(\frac{1}{2}, \frac{1}{2}, 1; \frac{1}{w}\right) \longrightarrow \frac{w^{\frac{1}{2}}}{\pi i}\, {}_2F_1\left(\frac{1}{2}, \frac{1}{2}, 1; w\right)\left(-\log\left(-\frac{w}{16}\right)\right) - \frac{w^{\frac{3}{2}}}{2\pi i}f(w), \tag{274}$$

where $f(w)$ is given by:[29]

$$f(w) = \sum_{k=1}^{\infty} \kappa_k w^{k-1} = 1 + \frac{21}{32}w + \frac{185}{384}w^2 + \frac{18655}{49152}w^3 + \frac{102501}{327680}w^4 + \mathcal{O}\left(w^5\right). \tag{275}$$

We thus find:

$$\widetilde{\omega}_a(w) = \frac{\epsilon\sqrt{w}}{2\pi^2}\left[{}_2F_1\left(\frac{1}{2}, \frac{1}{2}, 1; w\right)\log\left(-\frac{w}{16}\right) + \frac{w}{2}f(w)\right], \tag{276}$$

at $w = 0$. Here we introduced the sign $\epsilon = \text{sign}(\text{Im}(w))$, corresponding to a choice of branch for the square root. This expression with $\epsilon = 1$ holds on the principal branch of $\widetilde{\omega}_a$. Note that $\widetilde{\omega}_a$ has a branch cut on the interval $w \in (0, 1]$. The same method can be used to analytically

---

[29]The rational coefficients $\kappa_k$ can be written as the following residues:

$$\kappa_k = \frac{2}{\pi}\text{Res}_{s=k}\left[2^{4(s-k)}\Gamma(-s)^2\Gamma\left(\frac{1}{2}+s\right)^2\right].$$

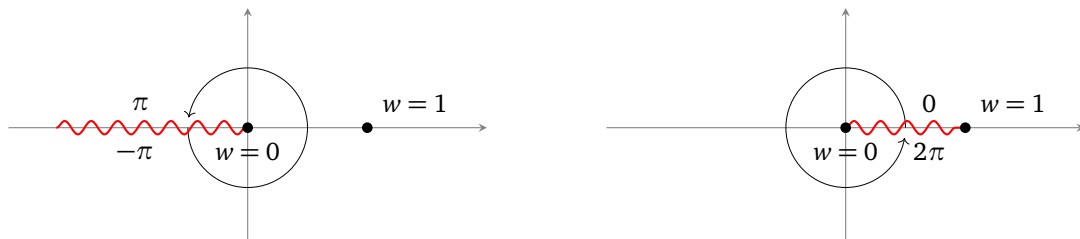

Figure 12: Branch cuts of the geometric periods $\widetilde{\omega}_D$ and $\widetilde{\omega}_a$, respectively.

continue $\widetilde{\omega}_D$ to $w = 0$. The leading asymptotics for the two periods are as follows:

$$
\begin{aligned}
w = \infty \ : \ & \widetilde{\omega}_a(w) \approx -\frac{1}{2\pi i}\left(1 + \frac{1}{4w} + \mathcal{O}\left(\frac{1}{w^2}\right)\right), \\
& \widetilde{\omega}_D(w) \approx -\frac{1}{\pi^2}\log(16w) + \mathcal{O}\left(\frac{1}{w}\right), \\
w = 1 \ : \ & \widetilde{\omega}_a(u) \approx \frac{1}{2\pi^2 i}\log\left(\frac{u}{16}\right) + \mathcal{O}\left(u\log(u)\right), \\
& \widetilde{\omega}_D(u) \approx -\frac{1}{\pi}\left(1 + \frac{1}{4}u + \mathcal{O}(u^2)\right), \\
w = 0 \ : \ & \widetilde{\omega}_a(w) \approx \frac{\sqrt{w}}{2\pi^2}\left(i\pi - \epsilon \log\left(\frac{w}{16}\right)\right) + \mathcal{O}\left(w^{\frac{3}{2}}\right), \\
& \widetilde{\omega}_D(w) \approx \frac{\sqrt{w}}{\pi^2}\log\left(\frac{w}{16}\right) + \mathcal{O}\left(w^{\frac{3}{2}}\right),
\end{aligned}
\tag{277}
$$

where we introduced the coordinate $u = 1 - \frac{1}{w}$. Note that the principal branch of $\widetilde{\omega}_a$ differs from that of $\widetilde{\omega}_D$. The branch cuts of the two periods are shown in figure 12. The factor of $\epsilon$ is then necessary to match the two principal branches, allowing us to consider linear combinations of the periods. As a consistency check, let us compute the monodromies of the geometric periods in the $(\widetilde{\omega}_D, \widetilde{\omega}_a)$ basis. As we go around $w = 0$ ($w \to e^{2\pi i}w$), one finds:

$$
\mathbb{M}_{w=0}^{(g)+} = \begin{pmatrix} -3 & -4 \\ 1 & 1 \end{pmatrix}, \qquad \mathbb{M}_{w=0}^{(g)-} = \begin{pmatrix} 1 & -4 \\ 1 & -3 \end{pmatrix},
\tag{278}
$$

corresponding to a base point in the upper- or lower-half plane, respectively. The monodromies at $w = 1$ and $w = \infty$ (with $u \to e^{2\pi i}u$ and $w \to e^{-2\pi i}w$ at $w = \infty$, respectively) are:

$$
\mathbb{M}_{w=1}^{(g)} = \begin{pmatrix} 1 & 0 \\ -1 & 1 \end{pmatrix}, \qquad \mathbb{M}_{w=\infty}^{(g)} = \begin{pmatrix} 1 & 4 \\ 0 & 1 \end{pmatrix}.
\tag{279}
$$

These matrices satisfy:

$$
\mathbb{M}_{w=1}\mathbb{M}_{w=0}^-\mathbb{M}_{w=\infty} = \mathbf{1} = \mathbb{M}_{w=0}^+\mathbb{M}_{w=1}\mathbb{M}_{w=\infty},
\tag{280}
$$

as expected. Let us note also that these can be written as:

$$
\mathbb{M}_{w=0}^{(g)\epsilon} = (T^{-2\epsilon}S)(-T)(T^{-2\epsilon}S)^{-1}, \qquad \mathbb{M}_{w=1}^{(g)} = STS^{-1}, \qquad \mathbb{M}_{w=\infty}^{(g)} = T^4,
\tag{281}
$$

for $\epsilon = \pm 1$. This is of course consistent with the global analysis, wherein we have the Kodaira fibers $(I_1^*, I_1, I_4)$ as in (268). In particular, the point at $w = 0$ is the $I_1^*$ singularity.

Interestingly, the $w$-plane can be viewed as a modular curve for the congruence subgroup $\Gamma^0(4)$ of $SL(2,\mathbb{Z})$, exactly like the Coulomb branch of the pure $SU(2)$ SW geometry [2]. The dependence of $\mathbb{M}_{w=0}$ on the base point is due to our choices of branch cuts [182], and it can be interpreted as a 'splitting' of the fundamental domain, as shown in figure 4b, such that the cusps at $\tau = -2$ and $\tau = 2$ are identified.

**Physical periods on the $w$-plane.** The physical periods on the $w$-plane can now be obtained, in principle, from the geometric periods, using the fact that:

$$\widetilde{\omega}_\gamma = U \frac{d}{dU} \Pi_\gamma. \tag{282}$$

Let us also note that they satisfy the following third order differential equation:

$$\left( z \left( \theta_z - \frac{1}{2} \right)^2 - \theta_z^2 \right) \theta_z \Pi_\gamma = 0, \tag{283}$$

with $z = \frac{1}{w}$ and $\theta = z \frac{d}{dz}$. This is a Meijer equation, and therefore the solutions can be written in terms of Meijer $G$-functions. In order to fix a basis that corresponds to the physical periods $(a_D, a)$, let us first consider the asymptotics of the periods obtained by integrating the geometric periods. These are:

$$
\begin{aligned}
w = \infty \;:\; & a(w) \approx \alpha_\infty - \frac{1}{4\pi i} \log(w) + \mathcal{O}\left( \frac{1}{w} \right), \\
& a_D(w) \approx \beta_\infty + \frac{1}{(2\pi i)^2} \log^2\left( \frac{1}{16w} \right) + \mathcal{O}\left( \frac{1}{w} \right), \\
w = 1 \;:\; & a(u) \approx \alpha_1 - \frac{1}{4\pi^2 i} \left( 1 - \log\left( \frac{u}{16} \right) \right) u + \mathcal{O}\left( u^2 \right), \\
& a_D(u) \approx \beta_1 - \frac{u}{2\pi} + \mathcal{O}(u^2), \\
w = 0 \;:\; & a(w) \approx \alpha_0 + \frac{\epsilon \sqrt{w}}{2\pi^2} \left( 2 - \log\left( -\frac{w}{16} \right) \right) + \mathcal{O}\left( w^{\frac{3}{2}} \right), \\
& a_D(w) \approx \beta_0 - \frac{\sqrt{w}}{\pi^2} \left( 2 - \log\left( \frac{w}{16} \right) \right) + \mathcal{O}\left( w^{\frac{3}{2}} \right),
\end{aligned}
\tag{284}
$$

with $\alpha_*$, $\beta_*$ some integration constants to be determined. We fix two of the constants, namely:

$$\alpha_\infty = -\frac{1}{4\pi i} \log(16), \qquad \beta_1 = 0, \tag{285}$$

such that $2a(w)$ matches with the D2-brane period on $\mathcal{C}_f$ at large volume $a \approx \frac{1}{4\pi i} \log z_f$, and such that $a_D(w)$ vanishes at $w = 1$. Note that, a priori, this $a_D(w)$ might not match with the D4-brane period at large volume. Comparing to the D4-brane period (264), this fixes $\beta_\infty = \frac{1}{6}$, and we will check that this is indeed consistent. In order to fix $\beta_0$, we proceed as in [56]. Using the connection formula:

$$_2F_1\left( \frac{1}{2}, \frac{1}{2}, 1; 1 - x \right) = x^{-\frac{1}{2}} \, _2F_1\left( \frac{1}{2}, \frac{1}{2}, 1; 1 - \frac{1}{x} \right), \tag{286}$$

which analytically continues the $a_D$ period from the region $|u| < 1$ towards $w = 0$ (excluding the point $w = 0$ point) in the $|w| < 1$ region, we have:

$$
\begin{aligned}
a_D(w = 1) &= -\frac{1}{2\pi} \int_0^1 \frac{1}{\sqrt{w}} \, _2F_1\left( \frac{1}{2}, \frac{1}{2}, 1; 1 - w \right) dw + \beta_0 \\
&= -\frac{1}{2\pi^2} \int_0^1 \frac{1}{\sqrt{w}} \sum_{n=0}^\infty \frac{1}{n!} \frac{\Gamma\left( \frac{1}{2} + n \right)^2}{\Gamma(1 + n)} (1 - w)^n \, dw + \beta_0 \\
&= -\frac{1}{2\pi^2} \pi^2 + \beta_0.
\end{aligned}
\tag{287}
$$

Given that $a_D$ vanishes at $w = 1$, it follows that $\beta_0 = \frac{1}{2}$. Having fixed this constant, we can write the periods in terms of Meijer $G$-functions:

$$
\begin{aligned}
a(w) &= -\frac{1}{4\pi^2 i} G_{3,3}^{2,2}\left(\begin{matrix} \frac{1}{2} & \frac{1}{2} & 1 \\ 0 & 0 & 0 \end{matrix} \middle| -\frac{1}{w}\right) + \frac{\widetilde{\epsilon}}{4}, \\
a_D(w) &= -\frac{1}{2\pi^3} G_{3,3}^{3,2}\left(\begin{matrix} \frac{1}{2} & \frac{1}{2} & 1 \\ 0 & 0 & 0 \end{matrix} \middle| \frac{1}{w}\right) + \frac{1}{2},
\end{aligned}
\tag{288}
$$

where $\widetilde{\epsilon} = \pm 1$ (modulo even integers) to match the large volume limit of $a(w)$ on a given branch. Using the Barnes-type integral representation of the $G$-functions,[30] one can check that $\beta_\infty = \frac{1}{6}$, and one also finds $\alpha_0 = \frac{\widetilde{\epsilon}}{4}$. We thus have:

$$
a(w = 0) = \frac{\widetilde{\epsilon}}{4}, \qquad a_D(w = 0) = \frac{1}{2}.
\tag{289}
$$

Finally, to fix the remaining constant $\alpha_1$ at $w = 1$, one can directly evaluate the Meijer $G$-function at unit argument. This gives the 'quantum volume' of the curve $\mathcal{C}_f$ at the conifold point:

$$
\Pi_{\mathrm{D2}_f}(w = 1) = 2a(w = 1) = 2\alpha_1 = \frac{1 + \widetilde{\epsilon}}{2} + i\frac{4G}{\pi^2},
\tag{290}
$$

where $G \approx 0.916$ is the Catalan constant. This agrees with the recent discussion in [185].

**Geometric periods on the $U$-plane.** Having obtained explicit expressions for the periods on the $w$-plane, the next step is to analytically continue the periods on the $U$-plane, which is a double-cover of the $w$-plane. Let us first consider the geometric periods. We introduce the functions:

$$
f_a(w) = -\frac{1}{2\pi i}\, {}_2F_1\left(\frac{1}{2}, \frac{1}{2}; 1; \frac{16}{U^2}\right), \qquad f_D(w) = -\frac{1}{\pi}\, {}_2F_1\left(\frac{1}{2}, \frac{1}{2}; 1; 1 - \frac{16}{U^2}\right).
\tag{291}
$$

Since the map $U \mapsto w$ is 2 to 1, we will split the $U$-plane into two regions separated by the imaginary axis, which we denote by $A$ (for $\mathrm{Re}(U) > 0$) and $B$ (for $\mathrm{Re}(U) < 0$). The above functions have branch cuts inherited from the hypergeometric function. Thus, it is not directly obvious what their expressions throughout the whole $U$-plane are. Here, we will interpret $f_D$ as a *local* function, which is well defined only around one of the two 'conifold' singularities at $U^2 = 16$. The branch cuts of $f_a$ connect the singularities at $U = \pm 4$ to the $U = 0$ singularity. We choose the branch cut of $f_D$ to run along $U \in [0, i\infty)$, in agreement with the $w$-plane branch cut. Recall also that the large volume asymptotics on the $w$ plane gives:

$$
f_a(w) \approx -\frac{1}{2\pi i} + \mathcal{O}\left(\frac{1}{w}\right), \qquad f_D(w) \approx -\frac{1}{\pi^2}\log(16w) + \mathcal{O}\left(\frac{1}{w}\right).
\tag{292}
$$

The geometric periods in the $A$ and $B$ regions will be linear combinations of $f_a$ and $f_D$, with the large volume ($U \to \infty$) asymptotics:

$$
\widetilde{\omega}_a \approx -\frac{1}{2\pi i}, \qquad \widetilde{\omega}_D \approx \frac{2}{\pi^2}\log\left(\frac{1}{U}\right),
\tag{293}
$$

---

[30]We have:

$$
G_{3,3}^{2,2}\left(\begin{matrix} \frac{1}{2} & \frac{1}{2} & 1 \\ 0 & 0 & 0 \end{matrix} \middle| x\right) = \frac{1}{2\pi i}\int \frac{\Gamma(-s)^2\,\Gamma\left(\frac{1}{2}+s\right)^2}{\Gamma(1+s)\Gamma(1-s)} x^s ds, \qquad G_{3,3}^{3,2}\left(\begin{matrix} \frac{1}{2} & \frac{1}{2} & 1 \\ 0 & 0 & 0 \end{matrix} \middle| x\right) = \frac{1}{2\pi i}\int \frac{\Gamma(-s)^3\,\Gamma\left(\frac{1}{2}+s\right)^2}{\Gamma(1-s)} x^s ds.
$$

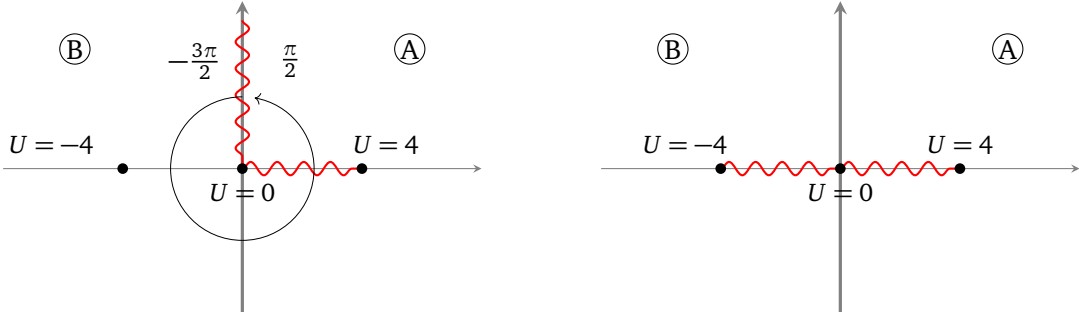

Figure 13: Branch cuts for $\widetilde{\omega}_D(U)$ and $\widetilde{\omega}_a(U)$, respectively.

which reproduce the large volume monodromy:

$$\mathbb{M}_{U=\infty} = \begin{pmatrix} 1 & 8 \\ 0 & 1 \end{pmatrix} = T^8. \tag{294}$$

We first note that $f_a$ can be used for the $a$ period in both regions of the $U$-plane, since the large volume expression is a regular function. The $\widetilde{\omega}_a$ period will thus have two branch cuts, running along $U \in (0_+, 4]$ and $U \in [-4, 0_-)$. We can choose $U_* = -4$ to be the cusp where $a_D(U_*) = 0$. Thus, in region $B$, the dual period $\widetilde{\omega}_D$ will be given by $f_D$. The mapping of the angles between the $w$-sheets and the $U$-plane is:

$$\begin{aligned} U &: -\frac{3\pi}{2} \to -\pi, \quad -\pi \to -\frac{\pi}{2}, \quad -\frac{\pi}{2} \to 0, \quad 0 \to \frac{\pi}{2}, \\ w &: -3\pi \to -2\pi, \quad -2\pi \to -\pi, \quad -\pi \to 0, \quad 0 \to \pi. \end{aligned} \tag{295}$$

Recall that $\arg(w) \in (-\pi, \pi)$ was the principal branch of $\widetilde{\omega}_D$ in the $w$-plane. Now, consider the function $f_D$ in region $A$. Analytic continuation to $U \to \infty$ leads to:

$$f_D^{(A)} \approx -\frac{1}{\pi^2} \log(16 w^{(A)}) = -\frac{1}{\pi^2}\Big( \log(16 w^{(B)}) - 2\pi i \Big). \tag{296}$$

In order for this to match with the asymptotic expansion of $\widetilde{\omega}_D$ in all regions, we must subtract a factor of $4 f_a$. We then have:

$$\begin{aligned} A &: \quad \widetilde{\omega}_D(U) = f_D(U) + 4 f_a(U), \\ B &: \quad \widetilde{\omega}_D(U) = f_D(U), \end{aligned} \tag{297}$$

while:

$$\widetilde{\omega}_a(U) = f_a(U), \tag{298}$$

for the entire $U$-plane. The branch cuts of the geometric periods $\widetilde{\omega}_D$ and $\widetilde{\omega}_a$ are shown in figure 13. The monodromies around the two singularities at $U = \pm 4$ read:

$$\mathbb{M}_{U=-4} = \begin{pmatrix} 1 & 0 \\ -1 & 1 \end{pmatrix} = STS, \qquad \mathbb{M}_{U=4} = \begin{pmatrix} -3 & 16 \\ -1 & 5 \end{pmatrix} = (T^4 S) T (T^4 S)^{-1}. \tag{299}$$

For the series expansion around $U = 0$, one needs to take into account the various branch cuts of $\widetilde{\omega}_a$ and $\widetilde{\omega}_D$. In the region where $\mathrm{Re}(U) > 0$, and $\mathrm{Im}(U) < 0$, for instance, the asymptotics are:

$$\widetilde{\omega}_a(U) \approx -\frac{U}{2\pi^2} \log \frac{U}{4} + i \frac{U}{2\pi} + \mathcal{O}\left(U^3\right), \qquad \widetilde{\omega}_D(U) \approx -\frac{U}{4\pi^2} \log \frac{U}{4} + i \frac{U}{8\pi} + \mathcal{O}\left(U^3\right), \tag{300}$$

leading to the monodromy:

$$\mathbb{M}_{U=0} = \begin{pmatrix} -3 & 8 \\ -2 & 5 \end{pmatrix} = (T^2 S) T^2 (T^2 S),$$ (301)

which, in particular, agrees with the fact that $U = 0$ is an $I_2$ singularity. These monodromies satisfy the consistency condition:

$$\mathbb{M}_{U=-4} \mathbb{M}_{U=0} \mathbb{M}_{U=4} = \mathbb{M}_{U=\infty}^{-1}.$$ (302)

Our explicit analysis of the $U$-plane periods thus recovers the Kodaira singularities expected from the discriminant of the Seiberg-Witten curve:

$$(I_1, I_2, I_1, I_8) \qquad \text{at} \qquad U_* = -4, 0, 4, \infty.$$ (303)

Finally, by integrating the geometric periods once, we can obtain the physical periods on the $U$-plane, similarly to the analysis on the $w$-plane. One can determine in that way which BPS particles become massless at which points. This can also be understood, more simply, from the explicit monodromy matrices that we just derived.

**Massless dyons and 5d BPS quiver.** One finds that the following dyons of the KK theory $D_{S^1} E_1$ become massless at these points:

$$
\begin{array}{llll}
U = -4 & (I_1) & : & \text{a monopole of charge } (1,0), \text{ becomes massless,} \\
U = 0 & (I_2) & : & \text{two dyons of charge } (-1, 2), \text{ become massless,} \\
U = 4 & (I_1) & : & \text{a dyon of charge } (1, -4), \text{ becomes massless.}
\end{array}
$$ (304)

Here, we fixed the overall signs of the electromagnetic charges such that the total charge vanishes. Interestingly, the point $U = 0$ is also a *quiver point*, where these BPS particles are mutually BPS. Using the fact that $a = \frac{1}{4}$ and $a_D = \frac{1}{2}$ at the origin, we find the central charge:

$$\mathcal{Z}_{\gamma=(1,0)}(U=0) = \frac{1}{2}, \qquad \mathcal{Z}_{\gamma=(-1,2)}(U=0) = 0, \qquad \mathcal{Z}_{\gamma=(1,-4)}(U=0) = \frac{1}{2}.$$ (305)

The central charges of the $\gamma = (1, -4)$ also carries a contribution from one unit of KK momentum (D0-brane charge) [26]. The associated 5d BPS quiver is obtained by assigning one node $\mathcal{E}_\gamma$ to each of the four dyons, and by drawing a net number of arrows $n_{ij}$ from $\mathcal{E}_{\gamma_i}$ to $\mathcal{E}_{\gamma_j}$ given by the Dirac pairing, $n_{ij} = \det(\gamma_i, \gamma_j)$. The resulting quiver reads:

$$
\begin{array}{ccc}
\mathcal{E}_{\gamma_1=(1,0)} & \twoheadrightarrow & \mathcal{E}_{\gamma_2=(-1,2)} \\
\downarrow & \diagdown & \downarrow \\
\mathcal{E}_{\gamma_3=(-1,2)} & \twoheadrightarrow & \mathcal{E}_{\gamma_4=(1,-4)}
\end{array}
$$ (306)

This is a well-known 'toric' quiver for local $\mathbb{F}_0$ – see *e.g.* [186].

**Modularity.** The Coulomb branch of the massless $E_1$ theory on a circle can be written as a modular curve for the congruence subgroup $\Gamma^0(8)$. To see this explicitly, we should look at the explicit map $U = U(\tau)$, which can be worked out as explained in section 2.3. We find:

$$U(\tau) = \frac{\eta\left(\frac{\tau}{2}\right)^{12}}{\eta\left(\frac{\tau}{4}\right)^4 \eta(\tau)^8} = q^{-\frac{1}{8}} + 4q^{\frac{1}{8}} + 2q^{\frac{3}{8}} - 8q^{\frac{5}{8}} - q^{\frac{7}{8}} + \mathcal{O}(q).$$ (307)

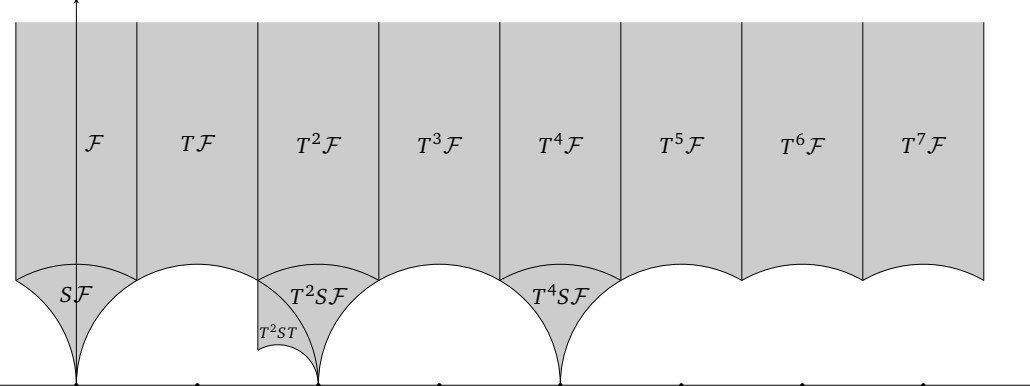

Figure 14: A fundamental domain for $\Gamma^0(8)$ on the upper half-plane. The four cusps at $\tau = 0, 2, 4, i\infty$ have widths 1, 2, 1 and 8, respectively. The modular curve $\mathbb{H}/\Gamma^0(8)$ is isomorphic to the Coulomb branch of the massless $D_{S^1}E_1$ theory.

We refer to appendix B for some background on modular functions. The $\eta$-quotient (307) is the Hauptmodul for $\Gamma^0(8)$. Note that this expression is obviously invariant under the action of $T^8$. Its series expansion reproduces the coefficients of the McKay-Thompson series of class $8E$ of the Monster group [173]. Using the $S$-transformation properties of the Dedekind $\eta$ function:

$$\eta\left(\frac{\tau}{\alpha}\right) = i^{-\frac{1}{2}} \sqrt{\alpha \, \tau_D} \, \eta(\alpha \, \tau_D), \tag{308}$$

for $\tau_D = -\frac{1}{\tau}$, we find the $q$-series expansion around $\tau_D = 0$:

$$U(\tau_D) = 4 \frac{\eta(2\tau_D)^{12}}{\eta(4\tau_D)^4 \, \eta(\tau_D)^8} = 4 + 32q_D + 128q_D^2 + 384q_D^3 + \mathcal{O}(q_D^4). \tag{309}$$

Therefore, the cusp at $\tau = 0$ corresponds to the type-$I_1$ Kodaira singularity at $U = 4$ on the $U$-plane. Similarly, under a $T^2$ transformation $\tau \to \tilde{\tau} = \tau + 2$, (307) becomes:

$$U(\tilde{\tau}) = -i\left(\frac{\eta\left(\frac{\tilde{\tau}}{4}\right)}{\eta(\tilde{\tau})}\right)^4 = -i\left(\tilde{q}^{-\frac{1}{8}} - 4\tilde{q}^{\frac{1}{8}} + 2\tilde{q}^{\frac{3}{8}} + 8\tilde{q}^{\frac{5}{8}} + \mathcal{O}\left(\tilde{q}^{\frac{7}{8}}\right)\right). \tag{310}$$

Under an additional $S$-transformation, we find the following series expansion for $\tilde{\tau}_D = \tau_D + 2$:

$$U(\tilde{\tau}_D) = -16i\left(\frac{\eta(4\tilde{\tau}_D)}{\eta(\tilde{\tau}_D)}\right)^4 = -16i\left(\tilde{q}_D^{\frac{1}{2}} + 4\tilde{q}_D^{\frac{3}{2}} + 14\tilde{q}_D^{\frac{5}{2}} + \mathcal{O}\left(\tilde{q}_D^{\frac{7}{2}}\right)\right). \tag{311}$$

Therefore, the cusp at $\tau = 2$ corresponds to the $I_2$ type singularity located at $U = 0$. Finally, note that (307) changes sign under a $T^4$ transformation, and thus the cusp at $\tau = 4$ corresponds to $U = -4$. There is therefore a one-to-one mapping between the cusps of $\Gamma^0(8)$ and the $U$-plane singularities.

The periods themselves have interesting modular properties. For instance, the geometric period $\omega_a$ turns out to be a modular form of weight 1, which can be expressed in terms of the Jacobi $\vartheta$-function:

$$\frac{da}{dU}(\tau) = -\frac{1}{8\pi i}\vartheta_2\left(\frac{\tau}{2}\right)^2 = -\frac{1}{2\pi i}\left(q^{\frac{1}{8}} + 2q^{\frac{5}{8}} + q^{\frac{9}{8}} + 2q^{\frac{13}{8}} + \mathcal{O}\left(q^{\frac{17}{8}}\right)\right). \tag{312}$$

We will see similar modular properties appear for most of the massless $E_n$ theories.

**Global flavour symmetry.** The massless $E_1$ curve has three non-trivial sections:

$$P_1 = \left(\frac{1}{12}(U^2 + 4), -U\right), \quad P_2 = \left(\frac{1}{12}(U^2 - 8), 0\right), \quad P_3 = \left(\frac{1}{12}(U^2 + 4), U\right), \quad (313)$$

which span a $\mathbb{Z}_4$ torsion group with $P_k + P_l = P_{k+l \pmod 4}$. Let us note that the sections $P_1$ and $P_3$ intersect non-trivially the $I_2$ singular fiber at $U = 0$. The remaining section, $P_2$, only intersects the 'trivial' component of this fiber and therefore generates a $\mathbb{Z}_2^{[1]}$ subgroup which injects in the torsion group $\mathbb{Z}_4$ according to (181). The group $\mathscr{F} = \mathbb{Z}_2^{(f)} = \mathbb{Z}_4/\mathbb{Z}_2^{[1]}$ then constraints the global form of the flavour group to be:

$$G_F = SU(2)/\mathbb{Z}_2^{(f)} \cong SO(3), \quad (314)$$

in agreement with [33]. We also identify the $\mathbb{Z}_2^{[1]}$ subgroup as the one-form symmetry of the $E_1$ theory. Indeed, the same structure can be derived using the BPS spectrum, which can be deduced in principle from the quiver (306). We have the states:

$$\mathscr{S} \; : \; (1, 0; 0), \quad (-1, 2; 1), \quad (1, -4; 0), \quad (315)$$

where the charges $(m, q; l)$ are given as in (198), with $l \in \mathbb{Z}_2$ the charge under the center of the flavour $\widetilde{G}_F \cong SU(2)$. The spectrum is left invariant by a group $\mathscr{E} = \mathbb{Z}_4$ generated by:

$$g^{\mathscr{E}} = \left(0, \frac{1}{4}; 1\right). \quad (316)$$

This $\mathbb{Z}_4$ contains a $\mathbb{Z}_2^{[1]}$ subgroup:

$$g^{\mathscr{Z}^{[1]}} = \left(0, \frac{1}{2}; 0\right), \quad (317)$$

which implies that the theory has an electric one-form symmetry $\mathbb{Z}_2^{[1]}$, as expected [76, 77]. We also have the cokernel $\mathscr{F} = \mathbb{Z}_2^{(f)}$ as above, which implies (314).

### 5.1.2 The $E_1$ curve with $\lambda = -1$

The Picard-Fuchs equations for the periods can be solved for generic values of $\lambda$ using Frobenius' method. However, there is a second value of the $\lambda$ parameter for which the geometric periods are standard hypergeometric functions, namely at $\lambda = -1$. In that limit, the $U$-plane has a $\mathbb{Z}_4$ symmetry, rather than the generic $\mathbb{Z}_2$ symmetry of the $E_1$ theory. Furthermore, the associated rational elliptic surface has $\Phi_{\text{tor}} = \mathbb{Z}_2$ for $\lambda \neq 1$. Let us introduce the coordinate:

$$w = -\frac{U^4}{64} = e^{i\pi s}\frac{U^4}{64}, \quad (318)$$

and start by discussing the periods on this $w$-plane. Here the parameter $s = \pm 1 \pmod 2$ is introduced in order to keep track of logarithmic ambiguities.

**Geometric periods on the $w$-plane.** In terms of $w$, the PF equation for the $\widetilde{\omega}$ periods reads:

$$(w - 1)\frac{d^2\widetilde{\omega}}{dw^2} + \frac{d\widetilde{\omega}}{dw} - \frac{3}{16w^2}\widetilde{\omega} = 0. \quad (319)$$

This is again a standard hypergeometric differential equation, as for $\lambda = 1$. However, there are now two regular solutions at $w = 0$, as the singularity at $w = 0$ is in fact an elliptic point rather than a cusp. We would like to normalize the periods in the large volume limit as:

$$\widetilde{\omega}_a \approx -\frac{1}{2\pi i}, \qquad \widetilde{\omega}_D \approx \frac{1}{2\pi^2}\left(-6\log(2) + \log\left(\frac{e^{i\pi s}}{w}\right) + \log(\lambda)\right), \quad (320)$$

such that $\widetilde{\omega}_D$ includes the $\log(\lambda)$ contribution of the $D4$ brane in (264). It will become clear momentarily that, in order for the $D4$ period to vanish at the 'conifold' point $w = 1$, we must impose:

$$\log(\lambda) = -i\pi s. \tag{321}$$

As a result, it is convenient to set $s = -1$. Thus, we use the following geometric periods:

$$\widetilde{\omega}_a = -\frac{1}{2\pi i}\, {}_2F_1\left(\frac{1}{4}, \frac{3}{4}; 1; \frac{1}{w}\right), \qquad \widetilde{\omega}_D = -\frac{1}{\sqrt{2}\pi}\, {}_2F_1\left(\frac{1}{4}, \frac{3}{4}; 1; 1 - \frac{1}{w}\right). \tag{322}$$

These functions have the same branch cuts as the $\lambda = 1$ geometric periods shown in figure 12, and their asymptotics can be found as before. For the expressions at $w = 0$, one can either use Barnes' type integrals or Kummer's connection formulae. We find the following monodromies for $u \to e^{2\pi i}u$ around $w = 1$, and $w \to e^{-2\pi i}w$ around the point at infinity, respectively:

$$\mathbb{M}_{w=1} = \begin{pmatrix} 1 & 0 \\ -1 & 1 \end{pmatrix} = STS^{-1}, \qquad \mathbb{M}_{w=\infty} = \begin{pmatrix} 1 & 2 \\ 0 & 1 \end{pmatrix} = T^2. \tag{323}$$

Similarly, for the monodromy around $w = 0$ (with $w \to e^{2\pi i}w$) we have:

$$\mathbb{M}_{w=0}^{\pm} = T^{-\epsilon}ST^{\epsilon}, \tag{324}$$

with $\epsilon = \text{sign}(\text{Im}(w))$, by keeping track of the base-point. As a result, the three singularities are $(I_2, I_1, III^*)$, which correspond to the singular points of the $\Gamma^0(2)$ elliptic curve[31]. Note also that both 'orbifold' monodromies (324) satisfy $(\mathbb{M}_{w=0})^4 = \mathbf{1}$, which will be useful below.

**Physical periods on the $w$-plane.** Recall that the geometric periods $\widetilde{\omega}$ are the logarithmic derivative of the physical periods. Due to the minus sign in the definition of $w$, one needs to take additional care with the logarithmic ambiguities. The analysis is similar to the $\lambda = 1$ case and we only outline the main steps of the computation. Integrating the asymptotics of the geometric periods, we find that the physical periods can be expressed in terms of Meijer-G functions, namely:

$$a(w) \approx -\frac{1}{8\sqrt{2}\pi^2 i}G_{3,3}^{2,2}\left(\begin{matrix} \frac{1}{4} & \frac{3}{4} & 1 \\ 0 & 0 & 0 \end{matrix}\middle| -\frac{1}{w}\right) + \frac{s-1}{8},$$
$$a_D(w) \approx -\frac{1}{8\sqrt{2}\pi^3}G_{3,3}^{3,2}\left(\begin{matrix} \frac{1}{4} & \frac{3}{4} & 1 \\ 0 & 0 & 0 \end{matrix}\middle| \frac{1}{w}\right) + \frac{1}{4}, \tag{325}$$

where the above expression of $a_D$ is chosen such that it vanishes at $w = 1$. It now becomes clear from the asymptotics of $a_D$ that for this to satisfy the asymptotics of the $D4$ period, one must impose the constraint (321) introduced before. In this setting, we find that at $w = 0$ the periods become:

$$a_D(w=0) = \frac{1}{4}, \qquad a(w=0) = \frac{s-1}{8} = -\frac{1}{4}. \tag{326}$$

Note that, compared to the $\lambda = 1$ case, we can now evaluate both the values of $a(w = 1)$ and $a_D(w = 1)$, without the use of the Meijer-G function. Using the methods of [56], we find:

$$a(w=1) \approx -\frac{s}{8} + 0.15257\, i, \tag{327}$$

in agreement with [97].

---

[31]The congruence subgroup $\Gamma^0(2)$ has two cusps of widths 1 and 2 and one elliptic point of order 2.

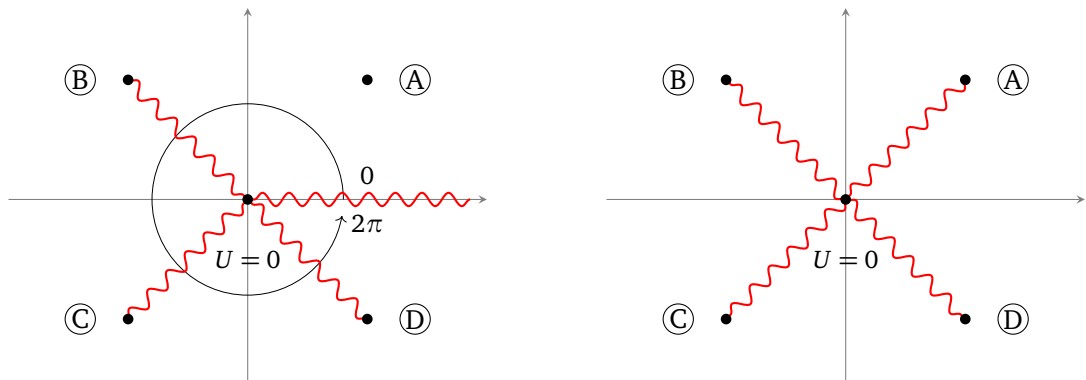

Figure 15: Possible choices of branch cuts for the physical periods $a_D(U)$ and $a(U)$ for the $E_1$ theory with $\lambda = -1$.

**Periods on the $U$-plane.** Consider now the geometric periods on the $U$-plane. In analogy to the $\lambda = 1$ case, we first introduce the functions:

$$f_a = -\frac{1}{2\pi i} {}_2F\left(\frac{1}{4}, \frac{3}{4}; 1; -\frac{64}{U^4}\right), \qquad f_D = -\frac{1}{\sqrt{2}\pi} {}_2F\left(\frac{1}{4}, \frac{3}{4}; 1; 1 + \frac{64}{U^4}\right). \tag{328}$$

Since $f_a$ is a regular function at $U \to \infty$, we will use this to define $\widetilde{\omega}_a$ throughout the $U$-plane. The map $U \mapsto w$ is 4 to 1, and thus we will split the $U$-plane in four quadrants, labelled $A$ to $D$, as shown in figure 15. Recall also that on the $w$-plane, $f_D$ has the following behaviour at large volume ($w \to \infty$):

$$f_D(w) \approx -\frac{1}{2\pi^2}\Big(6\log(2) + \log(w)\Big). \tag{329}$$

Given that the principal branch of the $w$-plane matches with region $A$ of the $U$-plane, let us choose $U = 2\sqrt{2}e^{i\pi/4}$ as the point where $\widetilde{w}_D = 0$. Hence, $f_D$ can be used to describe the $\widetilde{\omega}_D$ period in region $A$. However, since the other regions correspond to other sheets of the $w$-plane, analytic continuation of $f_D$ (which we take to be well defined around the relevant $U^4 = -64$ singularity) to large volume differs by the large volume expression obtained in region $A$. In particular:

$$f_D^{(n)}(U) \approx -\frac{1}{2\pi^2}\Big(6\log(2) + \log(w^{(A)}) + 2\pi i\, n\Big), \tag{330}$$

with $n = 1, 2, 3$ for regions $B, C$ and $D$, respectively. By $\log(w^{(A)})$ here, we mean that the underlying $\log(U)$ term will have the same principal branch as the logarithm from region $A$. Because of this, we then find:

$$\widetilde{\omega}_D(u) = f_D(u) - 2n f_a(u), \qquad \widetilde{\omega}_a(U) = f_a(U), \tag{331}$$

where we introduce $u = 1 - U_i$, with $U_i$ the four singularities. The branch cuts of these periods are shown in figure 15. Given the behaviour of $f_a$ around the $U^4 = -64$ singularities for $u \to e^{2\pi i} u$, namely: $f_a \to f_a - f_D$, we can readily find the monodromies:

$$\mathbb{M}_{U_n} = \begin{pmatrix} 2n+1 & 4n^2 \\ -1 & -2n+1 \end{pmatrix} = (T^{-2n}S)T(T^{-2n}S)^{-1}, \tag{332}$$

which correspond to $I_1$ cusps. Furthermore, the monodromy at large volume (for $w \to e^{-2\pi i} w$) becomes:

$$\mathbb{M}_{U=\infty} = \begin{pmatrix} 1 & 8 \\ 0 & 1 \end{pmatrix} = T^8, \tag{333}$$

corresponding to an $I_8$ singularity.

**Modularity.** Let us now comment on the modular properties of this configuration. The $\mathbb{Z}_4$ symmetry of the $U$-plane becomes manifest at the level of the $J$-invariant, which only depends on $X = U^4$. Thus, we can start by solving the cubic[32]:

$$(64 + X)j(\tau) = (48 + X)^3. \tag{334}$$

One root of the above equation is given by:

$$X(\tau) = U(\tau)^4 = -\left(64 + \left(\frac{\eta(\frac{\tau}{2})}{\eta(\tau)}\right)^{24}\right), \tag{335}$$

whose $q$-series expansion is the McKay-Thompson series of class 2$B$. Another root can be obtained from a $T$ transformation and turns out to be the McKay-Thompson series of class 4$A$. We can further show that the above $\eta$-quotient is in fact a modular function for the $\Gamma^0(2)$ congruence subgroup, in agreement with the $w$-plane monodromies previously found. Taking the fourth root of $X(\tau)$, we can then obtain an expression for $U(\tau)$ – we can then view the monodromy group for the $U$-plane as a 4-to-1 cover of $\Gamma^0(2)$. As a result, the monodromies around the 'conifold' singularities can be read from the coset representatives $T^{2n}S$, for $n \in \{0, 1, 2, 3\}$, for instance, which are also the monodromies we derived using the physical periods (332). Let us also note that, the derivative of the $a$ period can be expressed in terms of the Jacobi theta function:

$$\frac{da}{dU}(\tau) = \frac{e^{\frac{5\pi i}{4}}}{8\pi i} \vartheta_2\left(\frac{\tau}{2}\right)^2. \tag{336}$$

**Massless dyons and 5d BPS quiver.** The previous prescription for the periods on the $U$-plane implies that the KK theory has the dyons $(1, 0)$, $(1, -2)$, $(1, -4)$ and $(1, -6)$ that become massless at the four $I_1$ singularities, respectively. These states do not appear to correspond to physical states, however. Instead, we interpret them as defects corresponding D3-branes wrapping non-compact 3-cycles that 'end' at the $I_1$ cusps. The dual compact 3-cycles should be identified with the 'fractional branes' in that case – see *e.g.* [23, 176]. It is not entirely clear why our particular computation gives us this basis of 'external states'. It appears to be related to the fact that we implicitly chose a base point at large volume, instead of choosing it at $U = 0$; we hope to clarify this point in future work.

To identify the actual dynamical BPS particles that vanish at the four points $U^4 = -64$, which corresponds to the usual 'fractional brane states' on the local $\mathbb{F}_0$ singularity, it is safer to proceed as follows. We know that one of the BPS states is the D4-brane in IIA, which is the monopole $(1, 0)$ that vanishes at the 'conifold point' on the $U$-plane, with a corresponding conifold monodromy $\mathbb{M}_{w=1}$ given in (323). Then, the other 3 singularities on the $U$-plane can be obtained by conjugating the conifold monodromy with the $\mathbb{Z}_4$ 'orbifold' monodromy at the origin:

$$\mathbb{M}_k = \mathbb{M}_{\text{orb}}^k \mathbb{M}_0 \mathbb{M}_{\text{orb}}^{-k}, \qquad \mathbb{M}_0 = STS^{-1}, \quad \mathbb{M}_{\text{orb}} = TST^{-1}, \tag{337}$$

for $k = 0, 1, 2, 3$. The corresponding BPS states agree with the known results about fractional branes, and give us the $\mathbb{Z}_4$-symmetric quiver:

$$
\begin{array}{ccc}
\mathcal{E}_{\gamma_1=(1,0)} & \longrightarrow\!\!\!\!\longrightarrow & \mathcal{E}_{\gamma_2=(-1,2)} \\
\Big\downarrow\Big\uparrow & & \Big\downarrow\Big\uparrow \\
\mathcal{E}_{\gamma_4=(1,-2)} & \longleftarrow\!\!\!\!\longleftarrow & \mathcal{E}_{\gamma_3=(-1,0)}
\end{array}
\tag{338}
$$

---

[32]This is in fact a degree 12 equation in $U$.

This is the well-known second 'toric phase' for the 5d BPS quiver of the $E_1$ theory, as discussed in detail in [26, 187]. Using the known brane charges of the fractional branes and the exact periods (326) at the origin of the $U$-plane, one can check that the 4 fractional branes have a real central charge, $\mathcal{Z} = \frac{1}{4}$, so that we indeed have a quiver point at $U = 0$.[33]

**Torsion section and one-form symmetry.** Finally, let us discuss the global symmetries in the general case $\lambda \neq 1$. The MW group for the $(I_8, 4I_1)$ configuration is $\Phi = \mathbb{Z} \oplus \mathbb{Z}_2$. The $U(1)$ symmetry is generated by the horizontal divisor $\varphi(P)$ associated to the section:

$$P = \left( \frac{U^2 + 4(2 - \lambda)}{12}, -U \right), \tag{339}$$

which generates the free part of $\Phi$, and reduces to the $\mathbb{Z}_4$ generator $P_1$ in (313) when $\lambda = 1$. We have a $\mathbb{Z}_2$ torsion section:

$$P_{\text{tor}} = \left( \frac{U^2 - 4(1 + \lambda)}{12}, 0 \right), \tag{340}$$

which reduces to $P_2$ in (313) when $\lambda = 1$. For any $\lambda \neq 1$, we have $\Phi_{\text{tor}} = \mathcal{Z}^{[1]} = \mathbb{Z}_2^{[1]}$, consistent with our identification of $\mathcal{Z}^{[1]}$ with the one-form symmetry of the field theory. This is also confirmed by the dyonic charges in (338), which are left invariant by (317).

## 5.2 The $\widetilde{E}_1$ theory – 5d $SU(2)_\pi$: an Argyres-Douglas point on the $U$-plane

The $\widetilde{E}_1$ theory is the UV completion of the parity-violating $SU(2)_\pi$ gauge theory in 5d. Let us briefly discuss its $U$-plane. The Weierstrass form of the curve (104) read:

$$
\begin{aligned}
g_2(U) &= \frac{1}{12} \left( U^4 - 8U^2 - 24\lambda U + 16 \right), \\
g_3(U) &= -\frac{1}{216} \left( U^6 - 12U^4 - 36\lambda U^3 + 48U^2 + 216\lambda^2 - 64 \right),
\end{aligned}
\tag{341}
$$

with the massless limit corresponding to $\lambda = 1$. By direct inspection, we find the following allowed configurations of singular fibers:

$$
\begin{array}{|c||c|c|c|c|}
\hline
\{F_\nu\} & \lambda & \mathfrak{g}_F & \text{rk}(\Phi) & \Phi_{\text{tor}} \\
\hline\hline
I_8, 2I_1, II & \pm\frac{16i}{3\sqrt{3}} & \mathfrak{u}(1) & 1 & - \\
\hline
I_8, 4I_1 & \lambda & \mathfrak{u}(1) & 1 & - \\
\hline
\end{array}
\tag{342}
$$

This is of course in agreement with the Persson classification [47]. As for $E_1$, the generic point on the Coulomb branch of $\widetilde{E}_1$ has $4I_1$ type singularities. It is worth pointing out that the classification of rational elliptic surfaces includes two distinct configurations with singular fibers $(I_8, 4I_1)$, which are distinguished by their MW torsion. That mathematical fact dovetails nicely with the existence of two distinct theories with $T^8$ monodromy at large volume, $E_1$ and $\widetilde{E}_1$, with only the former having a non-trivial one-form symmetry [76, 77].

One can write down the Picard-Fuchs equation for the geometric periods as in (54), but its explicit form is not particularly illuminating. In particular, we do not find any modular

---

[33]There are some ambiguities in this identification, which we hope to discuss elsewhere. Importantly, for this computation, one must use the Chern characters of the branes, which include induced D0-charge from worldvolume flux. Then, if we use for instance the 'dictionary' of equation (5.26) in [187] together with our result $\Pi_{D4} = \frac{1}{4}$, $\Pi_{D2_f} = -\frac{1}{2}$ and $\Pi_{D2_b} = 0$ from this section, we indeed obtain $\mathcal{Z} = \frac{1}{4}$ for the four fractional branes.

properties in terms of congruence subgroups, nor any configuration for which the periods can be expressed in terms of hypergeometric functions. Around any given point, the periods can always be found as series expansions using the Frobenius method.

The $D_{S^1}\widetilde{E}_1$ Coulomb branch exhibits a feature that did not appear on the CB of the $E_1$ theory, however: there exists an allowed configuration with a singularity of type $II$, whose low-energy description is in terms of $H_0$, the Argyres-Douglas theory without flavour symmetry. As we reviewed in section 4.3.2, $H_0$ also appears on the Coulomb branch of the $SU(2)$ theory with $N_f = 1$. In fact, as we mentioned in the introduction, it is clear from the classification of RES that $H_0$ appears rather generically on rank-one Coulomb branches of theories with enough parameters – we 'simply' need to tune the parameters such that two mutually non-local BPS particles $\mathcal{E}_{1,2}$ with Dirac pairing $\langle \mathcal{E}_1, \mathcal{E}_2 \rangle = 1$ become massless at the same point.

Recall that the Coulomb branch operator of the $H_0$ fixed point has scaling dimension $\Delta_u = \frac{6}{5}$. We can then 'zoom-in' around the type-$II$ singularity as we have done for the $E_n$ MN theories in section 2.4.2, as follows. Setting $\lambda = \frac{16i}{3\sqrt{3}}$, the type-$II$ singularity sits at $U = -2i\sqrt{3}$, and we thus consider:

$$U \longrightarrow \beta^{\frac{6}{5}} u - 2i\sqrt{3}, \qquad (x, y) \longrightarrow \left( \beta^{\frac{2}{5}} x, \ \beta^{\frac{3}{5}} y \right), \tag{343}$$

in the $\beta \to 0$ limit, which leads to the 4d curve:

$$y^2 = 4x^3 - \frac{64i}{9\sqrt{3}} u. \tag{344}$$

We can also study the deformation pattern of this curve by introducing the parameter $c$ of the $H_0$ theory, with scaling dimension $\Delta_c = 2 - \Delta_u = \frac{4}{5}$. For this purpose, we consider the following limit:

$$U \longrightarrow u\, \beta^{\frac{6}{5}} - 2i\sqrt{3} - \frac{3c}{4} \beta^{\frac{4}{5}}, \qquad \lambda \longrightarrow \frac{16i}{3\sqrt{3}} + c\, \beta^{\frac{4}{5}}, \tag{345}$$

such that, under the same rescaling of $(x, y)$ as in (343), we obtain the curve:

$$y^2 = 4x^3 + \frac{4i}{\sqrt{3}} c\, x - \frac{64i}{9\sqrt{3}} u, \tag{346}$$

which, up to a rescaling of the parameters, is precisely the curve for $H_0$ with the relevant coupling turned on [3]. We have thus found an RG flow from the $\widetilde{E}_1$ theory to the Argyres-Douglas theory. In the above limit viewed from the point of view of the RES, the fiber at infinity, $F_\infty = I_8$, becomes an $II^*$ fiber after merging with two $I_1$ singularities from the interior.

# 6 The $E_0$ fixed point: $\mathbb{Z}_3$ symmetry on the $U$-plane

Let us now consider the $E_0$ theory. This is a 5d SCFT without flavour symmetry, and therefore it does not have any relevant deformations, nor any gauge-theory phase. It can be obtained as a deformation of the $\widetilde{E}_1$ theory [14], since it is realised in M-theory on a local $\mathbb{P}^2$, which is obtained from $dP_1$ by blowing down the exceptional curve. The $D_{S^1} E_0$ theory is then realised in Type IIA on that same geometry, which is the simplest example of a local Calabi-Yau threefold with an exceptional divisor. The origin of the Kähler cone, where the $\mathbb{P}^2$ shrinks to zero size, is the orbifold singularity $\mathbb{C}^3/\mathbb{Z}_3$. This geometry has obviously been studied in much detail in the literature – see in particular [16, 23, 188, 189]. The $E_0$ SW curve (105) takes the Weierstrass form:

$$g_2(U) = \frac{3}{4} U \left( 9U^3 - 8 \right), \qquad g_3(U) = -\frac{1}{8} \left( 27U^6 - 36U^3 + 8 \right), \tag{347}$$

with the discriminant:

$$\Delta(U) = 27(U^3 - 1). \tag{348}$$

This has three distinct roots, leading to three $I_1$ cusps in the interior of the $U$-plane, with an additional $I_9$ singularity at infinity. The $J$-invariant:

$$J(U) = \frac{U^3(9U^3 - 8)^3}{64(U^3 - 1)}, \tag{349}$$

only depends on $U^3$, which reflects the spontaneously broken $\mathbb{Z}_3$ symmetry of the $U$-plane. This $\mathbb{Z}_3$ is inherited from the $\mathbb{Z}_3$ one-form symmetry of the 5d $E_0$ theory [76, 77]. The total space of the SW fibration is the RES with singular fibers $(I_9, 3I_1)$, which has a MW group $\Phi = \mathbb{Z}_3$ torsion, in full agreement with the conjecture (186). Let us introduce the useful variable:

$$w = U^3. \tag{350}$$

We will first compute the periods on the $w$-plane, similarly to the $E_1$ theory. In the large volume limit in IIA, the physical periods correspond to the D2-brane wrapping the hyperplane class $H \cong \mathbb{P}^1$ inside $\mathbb{P}^2$ and to the D4-brane wrapping the $\mathbb{P}^2$. Let us denote the single Kähler parameter by:

$$T = \frac{1}{2\pi i} \int_H (B + iJ) = \frac{1}{2\pi i} \log\left(\frac{1}{-27w}\right) = \frac{1}{2\pi i} \log\left(\frac{e^{-i\pi s}}{27w}\right), \tag{351}$$

with the parameter $s$ introduced to keep track of logarithmic ambiguities, as for the $E_1$ theory. Then, the 'naive' brane periods are:

$$t_{D2} = T + \mathcal{O}\left(\frac{1}{w}\right), \qquad t_{D4} = \frac{1}{2}T^2 + \frac{1}{8} + \mathcal{O}\left(\frac{1}{w}\right). \tag{352}$$

However, due to the Freed-Witten anomaly [154], the D4-brane period must carry half a unit of world-volume flux. This induces $-\frac{1}{2}$ unit of D2-brane charge and $\frac{1}{8}$ of a D0-brane charge [190]. The physical periods are then:

$$\Pi_{D2} = t_{D2}, \qquad \Pi_{D4} = t_{D4} - \frac{1}{2}t_{D2} + \frac{1}{8}. \tag{353}$$

We will work in the basis $(a_D, a)$, with:

$$\Pi_{D2} = 3a, \qquad \Pi_{D4} = a_D. \tag{354}$$

Note the factor of 3, which is the electric charge of a wrapped D2-brane, since $[H] \cdot [\mathbb{P}^2] = -3$ inside $\widetilde{\mathbf{X}}$. The geometric periods $\omega = \frac{d\Pi}{dU}$ satisfy the PF equation:

$$\frac{d^2\omega}{dU^2} + \frac{3U^2}{U^3 - 1}\frac{d\omega}{dU} + \frac{U}{U^3 - 1}\omega = 0. \tag{355}$$

**Geometric periods on the $w$-plane.** Upon rescaling $(g_2, g_3) \to (U^2 g_2, U^3 g_3)$, the Weierstrass form of the curve only depends on $w = U^3$, with:

$$g_2(w) = \frac{3}{4}w(9w - 8), \qquad g_3(w) = -\frac{1}{8}w(27w^2 - 36w + 8), \tag{356}$$

while the discriminant becomes:

$$\Delta(w) = 27w^2(w - 1). \tag{357}$$

This curve has one $I_1$ singularity at $w = 1$, one type-$II$ elliptic point at $w = 0$ and one $I_3^*$ singularity at infinity. By performing a quadratic twist, these can be mapped to $(I_1, IV^*, I_3)$, with the $I_3$ at infinity, to match the large volume monodromy in IIA. These transformations amount to a rescaling of the holomorphic one-form as $\boldsymbol{\omega} \to (w)^{-\frac{1}{3}}\boldsymbol{\omega}$. We then consider the rescaled geometric periods:

$$\widetilde{\omega} = w^{\frac{1}{3}}\omega = U\frac{d\Pi}{dU}, \tag{358}$$

which satisfy the following PF equation:

$$(w-1)\frac{d^2\widetilde{\omega}}{dw^2} + \frac{d\widetilde{\omega}}{dw} - \frac{2}{9w^2}\widetilde{\omega} = 0. \tag{359}$$

This is a standard hypergeometric differential equation with singularities at $w = 0, 1, \infty$, with two regular solutions at $w = 0$ and only one regular solution at each of the other two singularities. A convenient basis of solutions is given by:

$$\widetilde{\omega}_a(w) = -\frac{1}{2\pi i}\,{}_2F_1\left(\frac{1}{3}, \frac{2}{3}; 1; \frac{1}{w}\right), \qquad \widetilde{\omega}_D(w) = -\frac{\sqrt{3}}{2\pi}\,{}_2F_1\left(\frac{1}{3}, \frac{2}{3}; 1; 1 - \frac{1}{w}\right). \tag{360}$$

Note that we will need to set $s = -1$ in (351) so that $\widetilde{\omega}_D$ agrees with the D4-brane asymptotics. We can also check that this function is the 'correct' geometric period because it leads to a vanishing D4-brane period at $w = 1$, the conifold point. Analytic continuation to $w = 0$ can be done using the Barnes' integral representations of the hypergeometric function, as well as Kummer's connection formulae. Additionally, the Gauss-Ramanujan formula can be used to obtain expressions at unit argument. A basis of solutions at $w = 0$ is given by:

$$w^{\frac{1}{3}}\,{}_2F_1\left(\frac{1}{3}, \frac{1}{3}; \frac{1}{3}; w\right), \qquad w^{\frac{2}{3}}\,{}_2F_1\left(\frac{2}{3}, \frac{2}{3}; \frac{4}{3}; w\right). \tag{361}$$

Note the cubic roots, which give rise to a $\mathbb{Z}_3$ monodromy. With a bit of work, we can then explicitly compute all the monodromies. At large volume and at the conifold point, we find:

$$\mathbb{M}_{w=\infty} = \begin{pmatrix} 1 & 3 \\ 0 & 1 \end{pmatrix} = T^3, \qquad \mathbb{M}_{w=1} = \begin{pmatrix} 1 & 0 \\ -1 & 1 \end{pmatrix} = STS^{-1}, \tag{362}$$

while the $\mathbb{Z}_3$ orbifold point monodromy reads:

$$\mathbb{M}_{w=0}^+ = T^{-1}(ST)^2 T, \qquad \mathbb{M}_{w=0}^- = T^2(ST)^2 T^{-2}, \tag{363}$$

depending on the base point being in the upper- or lower-half-plane. These monodromies do indeed correspond to the singularities $I_3$, $I_1$ and $IV^*$, respectively, and span the monodromy group $\Gamma^0(3)$. The dependence on the base-point in (363) can be visualised as a splitting of the fundamental domain for the modular group $\Gamma^0(3)$, as in previous examples.

**Physical periods on the $w$-plane.** Recall that the physical periods are related to the geometric periods by (358). Integrating the asymptotics of the geometric periods, we only need to fix the remaining constants of integration. The solutions of the Picard-Fuchs equation satisfied by the physical periods can be expressed in terms of Meijer-G functions. Fixing the integration constants such that $a_D(w = 1) = 0$ and $\Pi_{D2} = 3a$ in the large volume limit, we have:

$$a(w) = -\frac{1}{4\sqrt{3}\pi^2 i}G_{3,3}^{2,2}\left(\begin{matrix}\frac{1}{3} & \frac{2}{3} & 1 \\ 0 & 0 & 0\end{matrix}\middle| -\frac{1}{w}\right) - \frac{s+1}{6},$$

$$a_D(w) = -\frac{\sqrt{3}}{8\pi^3}G_{3,3}^{3,2}\left(\begin{matrix}\frac{1}{3} & \frac{2}{3} & 1 \\ 0 & 0 & 0\end{matrix}\middle| \frac{1}{w}\right) + \frac{1}{3}, \tag{364}$$

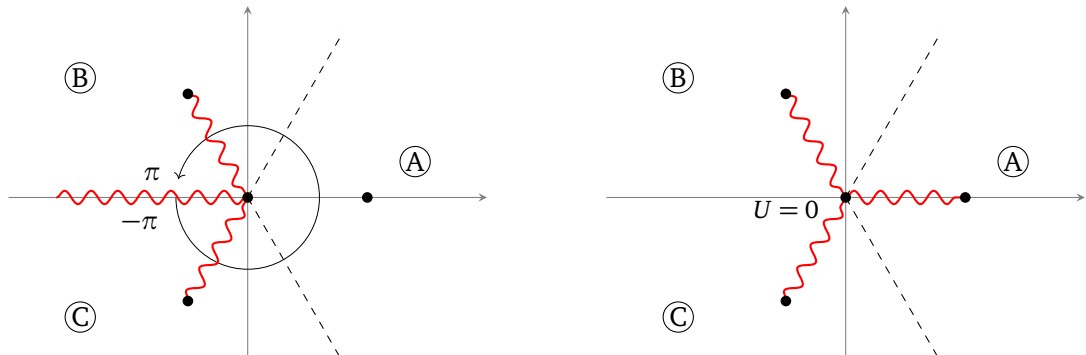

Figure 16: Choice of branch cuts for the physical periods $a_D(U)$ and $a(U)$ of the $E_0$ theory.

with $s$ as introduced in (351). Then, the large volume asymptotics of $a_D$ is fixed to:

$$a_D(w) = -\frac{1}{8\pi^2}\log\left(\frac{1}{27w}\right)^2 + \frac{1}{8}.$$ (365)

Thus, for this to be consistent with D-brane periods, we must 'set' $s = -1$, as previously mentioned. We then find:

$$a_D(w = 0) = \frac{1}{3}, \qquad a(w = 0) = 0,$$ (366)

at the orbifold point. Lastly, we need to fix the integration constant for $a$ at the conifold point, which we can evaluate numerically using the methods of [56]:

$$a(w = 1) = \frac{1}{6} + i\frac{3}{2\pi^2}\text{Im}\left(\text{Li}_2(e^{\frac{\pi i}{3}})\right) = \frac{1}{6} + 0.1543i.$$ (367)

This is consistent with the results of [97, 189] and with [185], which gives the analytic result quoted here.

**Geometric periods on the $U$-plane.** Consider next the geometric periods on the $U$-plane. As before, we introduce:

$$f_a = -\frac{1}{2\pi i}\,_2F_1\left(\frac{1}{3}, \frac{2}{3}; 1; \frac{1}{U^3}\right), \qquad f_D = -\frac{\sqrt{3}}{2\pi}\,_2F_1\left(\frac{1}{3}, \frac{2}{3}; 1; 1 - \frac{1}{U^3}\right).$$ (368)

The function $f_a$ is regular at $U \to \infty$ and can be used as $\widetilde{\omega}_a$ for the entire $U$-plane. The map $U \mapsto w$ is 3 to 1 and we thus split the $U$-plane in three regions labelled by $A, B$ and $C$, each containing one of the $U^3 = 1$ singularities and covering a third of the $U$-plane, namely: $\theta_A \in (-\pi/3, \pi/3)$, $\theta_B \in (\pi/3, \pi)$ and $\theta_C \in (-\pi, -\pi/3)$, as shown in figure 16. The angles are mapped as:

$$\begin{aligned} U: & \quad -\pi \to -\frac{\pi}{3}, \quad -\frac{\pi}{3} \to \frac{\pi}{3}, \quad \frac{\pi}{3} \to \pi, \\ w: & \quad -3\pi \to -\pi, \quad -\pi \to \pi, \quad \pi \to 3\pi. \end{aligned}$$ (369)

Recall that the large volume asymptotics of the function $f_D$ on the $w$-plane are given by:

$$f_D(w) \approx -\frac{3}{4\pi^2}\left(3\log(3) + \log(w)\right).$$ (370)

Consequently, choosing region $A$ to be the one that contains the singularity ($U = 1$) where $\widetilde{\omega}_D \to 0$, we must have:

$$\widetilde{\omega}_D(U) = f_D(U) + 3n\,f_a(U),$$ (371)

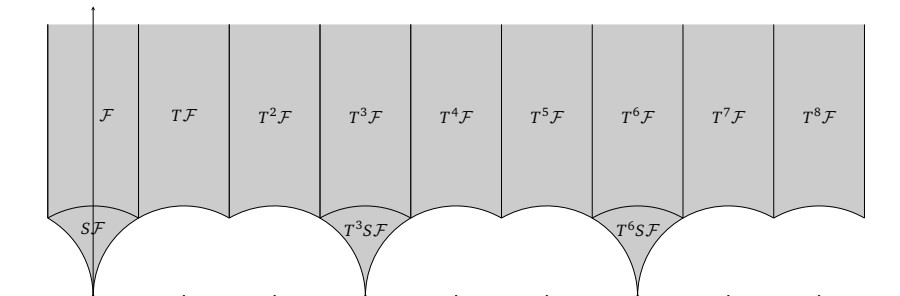

Figure 17: 'Standard' fundamental domain for $\Gamma^0(9)$, with $I_1$ cusps at $\tau = 0, 3$ and 6.

for $n = 0, 1, -1$ in regions $A$, $B$ and $C$, respectively. The monodromies can be computed by making use of the behaviour of $f_a$ around the $U^3 = 1$ singularities, namely: $f_a \to f_a - f_D$ under $u \to e^{2\pi i} u$, for $u = U - U_n$. Thus, we find:

$$\mathbb{M}_{(n)} = \begin{pmatrix} 1 - 3n & 9n^2 \\ -1 & 1 + 3n \end{pmatrix} = (T^{3n}S)T(T^{3n}S)^{-1} \,, \tag{372}$$

with $n = 0, 1, -1$, as before. Furthermore, the monodromy at $\infty$ becomes $T^9$, and thus we find the cusps $(I_9, 3I_1)$, with an obvious $\mathbb{Z}_3$ symmetry. Note also that:

$$\mathbb{M}_{(n)}\mathbb{M}_{(n+1)}\mathbb{M}_{(n+2)} = T^{-9} \,, \tag{373}$$

as expected. These monodromies generate $\Gamma^0(9)$ and agree with the 'simplest' fundamental domain for this congruence subgroup, which is depicted in figure 17. We will show momentarily that $U(\tau)$ is indeed the Hauptmodul for $\Gamma^0(9)$, supporting the claim that this is the monodromy group of the theory.

**Fractional branes and BPS quiver.** The $\mathbb{Z}_3$ symmetry of the $U$-plane is a $\mathbb{Z}_3$ orbifold symmetry in string theory, and the corresponding fractional branes are well understood – see *e.g.* [188]. The $\mathbb{Z}_3$ orbifold monodromy on the $w$-plane is given by (363):

$$\mathbb{M}_{\text{orb}}^+ = \begin{pmatrix} -2 & -3 \\ 1 & 1 \end{pmatrix}, \qquad \mathbb{M}_{\text{orb}}^- = \begin{pmatrix} 1 & -3 \\ 1 & -2 \end{pmatrix}, \qquad \left(\mathbb{M}_{\text{orb}}^{\pm}\right)^3 = 1 \,, \tag{374}$$

for a base point in the upper- or lower-half-plane, respectively. The $U$-plane monodromies with a base point near the origin can be constructed by conjugating the conifold monodromy:

$$\mathbb{M}_{C_k} = (\mathbb{M}_{\text{orb}})^k \mathbb{M}_{w=1}(\mathbb{M}_{\text{orb}})^{-k} \,, \tag{375}$$

for $k = 0, 1, 2$. We will use $\mathbb{M}_{\text{orb}} = \mathbb{M}_{\text{orb}}^-$ in the following discussion. Note that we have:

$$\mathbb{M}_{C_0}\mathbb{M}_{C_1}\mathbb{M}_{C_2} = T^{-9} \,. \tag{376}$$

In terms of the matrices $\mathbb{M}_{(n)}$ in (372), these monodromies are:

$$\mathbb{M}_{C_0} = \mathbb{M}_{(n=0)} = STS^{-1} \,, \qquad \mathbb{M}_{C_2} = \mathbb{M}_{(n=3)} = (T^3 S)T(T^3 S)^{-1} \,, \tag{377}$$

while $\mathbb{M}_{C_1}$ takes the more complicated form:

$$\mathbb{M}_{C_1} = \mathbb{M}_* T \mathbb{M}_*^{-1} \,, \qquad \mathbb{M}_* = \begin{pmatrix} 3 & -2 \\ 2 & -1 \end{pmatrix} = T^2 S T^2 S \,. \tag{378}$$

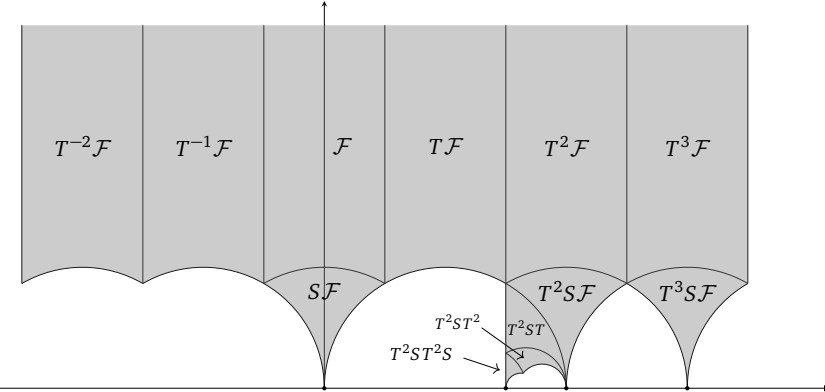

Figure 18: Different fundamental domain for $\Gamma^0(9)$, with $I_1$ cusps at $\tau = 0, \frac{3}{2}$ and 3.

We thus have $3I_1$ cusps, as expected, but with monodromies that are distinct from (372). The monodromies (372) are associated to the massless BPS states:

$$\mathscr{S} : \quad (1,0), \qquad (-1,3), \qquad (1,-6), \tag{379}$$

respectively. The corresponding BPS quiver reads:

$$
\begin{array}{c}
\mathcal{E}_{\gamma_1=(1,0)} \quad\longleftarrow\!\!\!\!\longleftarrow\quad \mathcal{E}_{\gamma_3=(1,-6)} \\[2mm]
\Big\downarrow \qquad\qquad \nearrow \\[2mm]
\mathcal{E}_{\gamma_2=(-1,3)}
\end{array}
\tag{380}
$$

On the other hand, the monodromies (375) are associated to the massless BPS states:

$$\mathscr{S} : \quad (1,0), \qquad (-2,3), \qquad (1,-3), \tag{381}$$

respectively. The corresponding BPS quiver is the well-known $\mathbb{C}^3/\mathbb{Z}_3$ orbifold quiver [188]:

$$
\begin{array}{c}
\mathcal{E}_{\gamma'_1=(1,0)} \quad\longrightarrow\!\!\!\!\longrightarrow\quad \mathcal{E}_{\gamma'_2=(-2,3)} \\[2mm]
\Big\uparrow \qquad\qquad \nearrow \\[2mm]
\mathcal{E}_{\gamma'_3=(1,-3)}
\end{array}
\tag{382}
$$

These two quivers are related by a mutation on the 'bottom' node, in which case the electromagnetic charges transform as [86]:

$$\gamma'_1 = \gamma_1, \qquad \gamma'_3 = -\gamma_2, \qquad \gamma'_2 = \gamma_3 + 3\gamma_2. \tag{383}$$

The BPS states (381) suggest a different choice of the fundamental domain for $\Gamma^0(9)$, which is shown in figure 18. Note, however, that for this choice of fundamental domain it is not immediately obvious that the width of the cusp at $\tau = i\infty$ is 9.

**Modularity.** Let us analyse the modular properties of the $E_0$ curve. Recall that the $J$-invariant only depends on $w = U^3$:

$$J(w) = \frac{w(9w-8)^3}{64(w-1)}, \tag{384}$$

from which we find:

$$w(\tau) = \frac{1}{27}\left(q^{-\frac{1}{3}} + 15 + 54q^{\frac{1}{3}} - 76q^{\frac{2}{3}} - 243q + \mathcal{O}(q^{\frac{4}{3}})\right), \tag{385}$$

which is the McKay-Thompson series of class 3B for the Monster group [173]. We can thus write this in closed form as:

$$w(\tau) = 1 + \frac{1}{27}\left(\frac{\eta(\frac{\tau}{3})}{\eta(\tau)}\right)^{12}, \tag{386}$$

which is the Hauptmodul for $\Gamma^0(3)$, as expected. Under an $S$-transformation, this becomes:

$$w(\tau_D) = 1 + 27\left(\frac{\eta(3\tau_D)}{\eta(\tau_D)}\right)^{12} = 1 + 27q_D + 324q_D^2 + \mathcal{O}(q_D^3), \tag{387}$$

where $\tau_D = -1/\tau$. We thus find that the cusps at $w = \infty, 1$ correspond to $\tau = i\infty$ and $\tau = 0$, respectively. Furthermore, $w(\tau)$ vanishes for $\tau = 2 + e^{2i\pi/3}$ or $\tau = -1 + e^{2i\pi/3}$, in agreement with the splitting of the fundamental domain observed in (363). To obtain the $U$-plane modular function, we take a cubic root of (386), to find:

$$U(\tau) = \frac{1}{3}\left(q^{-\frac{1}{9}} + 5q^{\frac{2}{9}} - 7q^{\frac{5}{9}} + 3q^{\frac{8}{9}} + \mathcal{O}(q)\right) = 1 + \frac{1}{3}\left(\frac{\eta(\frac{\tau}{9})}{\eta(\tau)}\right)^3. \tag{388}$$

This is the Hauptmodul for $\Gamma^0(9)$, and it is obviously invariant under $T^9$ transformations. Its series expansion reproduces the McKay-Thompson series of class 9B. Under a $T^{3n}$ transformation, this expression changes by a phase factor:

$$T^{3n} : U(\widetilde{\tau}) = e^{\frac{4n\pi i}{3}}\left(1 + \frac{1}{3}\left(\frac{\eta(\frac{\widetilde{\tau}}{9})}{\eta(\widetilde{\tau})}\right)^3\right), \tag{389}$$

for $\widetilde{\tau} = \tau + 3n$. Furthermore, under an $S$-transformation, we have:

$$S : U(\tau_D) = 1 + 9\left(\frac{\eta(9\tau_D)}{\eta(\tau_D)}\right)^3 = 1 + 9q_D + 27q_D^2 + \mathcal{O}(q_D^3), \tag{390}$$

for $\tau_D = -\frac{1}{\tau}$. We thus see that the singularity at $U = 1$ corresponds to the $\tau = 0$ cusp. Using the fact that $U(\tau)$ picks up phases under $T^3$ and $T^6$ transformations, we find that the cusps at $U = e^{\frac{4\pi i}{3}}$ and $U = e^{\frac{2\pi i}{3}}$ are in the $\Gamma^0(9)$ orbits of $\tau = 3$ and $\tau = 6$ (or $\tau = -3$), respectively. Let us further note that under the transformation:

$$\gamma : \tau_* \mapsto \frac{-\tau_* + 9}{-\tau_* + 8}, \tag{391}$$

the cusp at $\tau_* = 6$ is mapped to $\tau = \frac{3}{2}$. The element of $\mathrm{PSL}(2, \mathbb{Z})$ that corresponds to this transformation is:

$$\gamma = \begin{pmatrix} -1 & 9 \\ -1 & 8 \end{pmatrix} \in \Gamma^0(9). \tag{392}$$

As a result, $\tau = 6, -3$ and $\frac{3}{2}$ are all in the same orbit of $\Gamma^0(9)$. This shows that the two fundamental domains shown in figures 17 and 18 are indeed fundamental domains of $\Gamma^0(9)$. Finally, let us note that the $a$ period satisfies:

$$\frac{da}{dU}(\tau) = -\frac{3}{2\pi i}\frac{\eta(\tau)^3}{\eta(\frac{\tau}{3})}. \tag{393}$$

**Torsion and one-form symmetry.** We already mentioned that the $\mathbb{Z}_3$ symmetry of the $U$-plane is inherited from the one-form symmetry in 5d. The $D_{S^1}E_0$ SW geometry is an extremal RES with singular fibers $(I_9, 3I_1)$ and with a MW group $\Phi = \mathbb{Z}_3$ spanned by the sections:

$$P_1 = \left(\frac{3}{4}U^2, -1\right), \qquad P_2 = \left(\frac{3}{4}U^2, 1\right) = -P_1. \tag{394}$$

We thus identify this $\mathbb{Z}_3^{[1]}$ group with the one-form symmetry of the $D_{S^1}E_0$ theory, according to our conjecture (186). The one-form symmetry can also be seen directly from the BPS spectrum generated by the 'fractional branes' states (381), which are left invariant by the generator $g^{\mathbb{Z}^{[1]}} = \left(0, \frac{1}{3}\right)$ of the $\mathbb{Z}_3$ group.

# 7 The $E_2$ and $E_3$ theories – 5d $SU(2)$ with one or two flavours

In this section, we discuss the $U$-plane of the $E_2$ and $E_3$ theories, which are the UV-completion of the 5d $SU(2)$ gauge theory with one and two flavours, respectively.

## 7.1 CB configurations for the $E_2$ theory

The Weierstrass form of the $D_{S^1}E_2$ curve (97) reads:

$$
\begin{aligned}
g_2 &= \frac{1}{12}\Big(U^4 - 8(1+\lambda)U^2 - 24\lambda M_1 U + 16\left(1 - \lambda + \lambda^2\right)\Big), \\
g_3 &= -\frac{1}{216}\Big(U^6 - 12(1+\lambda)U^4 - 36\lambda M_1 U^3 + 144\lambda M_1(1+\lambda)U + \\
&\quad + 24\left(2 + \lambda + 2\lambda^2\right)U^2 - 8\left(8 - 12\lambda - 3\left(4 + 9M_1^2\right)\lambda^2 + 8\lambda^3\right)\Big).
\end{aligned}
\tag{395}
$$

The massless curve is obtained by setting $\lambda = M_1 = 1$. However, similarly to the $\widetilde{E}_1$ case, it is not extremal, nor modular, because of the $\mathfrak{u}(1)$ factor in the flavour symmetry algebra. As per the classification of rational elliptic surfaces, the allowed CB configurations are:

| $\{F_v\}$ | $\lambda$ | $M_1$ | $\mathfrak{g}_F$ | $\mathrm{rk}(\Phi)$ | $\Phi_{\mathrm{tor}}$ |
|:---:|:---:|:---:|:---:|:---:|:---:|
| $I_7, I_2, 3I_1$ | $1$ | $1$ | $A_1 \oplus \mathfrak{u}(1)$ | $1$ | — |
| $I_7, III, 2I_1$ | $1$ | $2i$ | $A_1 \oplus \mathfrak{u}(1)$ | $1$ | — |
| $I_7, I_2, II, I_1$ | $1$ | $\pm\frac{2}{3\sqrt{3}}$ | $A_1 \oplus \mathfrak{u}(1)$ | $1$ | — |
| $I_7, 2II, I_1$ | $e^{i\theta_0}$ | $2^{\frac{7}{4}}e^{i\theta_1}$ | $2\mathfrak{u}(1)$ | $2$ | — |
| $I_7, II, 3I_1$ | $e^{\frac{\pi i}{3}}$ | $\frac{2}{3^{3/4}}e^{\frac{5\pi i}{12}}$ | $2\mathfrak{u}(1)$ | $2$ | — |
| $I_7, 5I_1$ | $\lambda$ | $M_1$ | $2\mathfrak{u}(1)$ | $2$ | — |

$$\tag{396}$$

Here, $\theta_0 = -\pi + \tan^{-1}\left(\frac{91\sqrt{7}}{87}\right)$ and $\theta_1 = \frac{1}{2}\tan^{-1}\left(\frac{7\sqrt{7}}{13}\right)$. Note that some of the choices of parameters are not unique. For the first three configurations, setting $\lambda = 1$ leads to an $I_2$ singularity in the bulk. As in the case of the $\widetilde{E}_1$ theory, we see that there are configurations that include the AD theories $H_0$ and $H_1$, corresponding to the type-$II$ and $III$ singularities, respectively. Since the $E_2$ theory admits a gauge-theory deformation to the 5d $SU(2)$ $N_f = 1$ theory, we did expect the existence of at least one $H_0$ point, but we can bypass the 4d gauge-theory limit entirely. For instance, consider the configuration $(I_7, I_2, II, I_1)$ with $M_1 = \frac{2}{3\sqrt{3}}$.

The type-*II* singularity sits at $U = -2\sqrt{3}$, so we consider the limit:

$$(I_7, I_2, II, I_1) : \quad U \to u\,\beta^{\frac{6}{5}} - 2\sqrt{3} - \frac{c}{\sqrt{3}}\,\beta^{\frac{4}{5}}, \quad \lambda \to 1 + c\,\beta^{\frac{4}{5}}, \quad M_1 \to \frac{2}{3\sqrt{3}}, \tag{397}$$

with $\beta \to 0$, leading to the curve:

$$y^2 = 4x^3 + \frac{8}{9}\,c\,x - \frac{16}{9\sqrt{3}}\,u, \tag{398}$$

which gives us directly the SW curve of the $H_0$ theory [3]. Similarly, the $H_1$ AD theory can be recovered from the CB configuration $(I_7, III, 2I_1)$, with the type-*III* singularity sitting at at $U = -2i$ for the values of the parameters in (396). Recalling that we have $\Delta_u = \frac{4}{3}$ and that the 4d parameters $(c, \mu)$ have scaling dimensions $\left(\frac{2}{3}, 1\right)$, we consider the 4d limit:

$$(I_7, III, 2I_1) : \quad \lambda \to 1 + \mu\,\beta, \quad M_1 \to 2i + c\,\beta^{\frac{2}{3}}, \tag{399}$$
$$U \to u\,\beta^{\frac{4}{3}} - 2i - c\,\beta^{\frac{2}{3}} - i\,\mu\,\beta.$$

The resulting four-dimensional curve reads:

$$y^2 = 4x^3 - \left(-\frac{4c^2}{3} + 4i\,u\right)x - \left(\frac{8i}{27}\,c^3 + \frac{4c}{3}\,u + \mu^2\right). \tag{400}$$

Then, upon the redefinition:

$$u \longrightarrow -i\,u - \frac{i}{4}\,c^2, \tag{401}$$

followed by the rescalings $(x, y) \to \left(e^{\frac{\pi i}{2}}\,x, e^{\frac{3\pi i}{4}}\,y\right)$ and $u \to 4u$, $c \to 2c$, $\mu \to 2e^{-\frac{\pi i}{4}}\mu$, we recover the curve:

$$y^2 = 4x^3 - \left(\frac{4c^2}{3} - 16u\right)x + \left(\frac{8c^3}{27} + \frac{32c}{3}\,u + 4\mu^2\right), \tag{402}$$

which is the Weiertrass form of the $H_1$ curve [4, 191]:

$$y^2 + u = x^4 + c\,x^2 + \mu\,x. \tag{403}$$

In this limit, note that it is $\lambda$ that plays the role of the flavour parameter in the Argyres-Douglas theory, while $M$ is related to the parameter conjugate to $u$. This is because the $\mathfrak{su}(2)$ flavour symmetry of $H_1$ is here inherited from the $\mathfrak{su}(2)$ flavour symmetry of the $E_2$ theory, under which the 5d W-boson and instanton particle transform as a doublet. This can be contrasted with the physics of the 4d $SU(2)$ $N_f = 2$ CB, where the $\mathfrak{su}(2)$ comes from a pair of quarks, which are then combined with a light dyon [4].

**Modularity and AD configuration.** We already mentioned that the massless $E_2$ curve is not modular, but one may wonder whether any of the other configurations in (396) could have some interesting modular properties. The genus-zero congruence subgroups of $\mathrm{PSL}(2, \mathbb{Z})$ have been completely classified in [170, 174]. Using their lists, we find that most of the $D_{S^1}E_2$ CB configurations of $D_{S^1}E_2$ cannot be modular.[34] This leaves us with the $(I_7, 2II, I_1)$ configuration, which could be modular under $\Gamma^0(7)$. Note that this was one of the 'missing groups' in the naive pattern that would assign $\Gamma^0(9-n)$ to $D_{S^1}E_n$.

---

[34]At least with respect to a congruence subgroup. In fact, two other configurations are modular with respect to non-congruence subgroups.

To check this explicitly, let us first relate the SW curve (395), which was obtained from the 'toric' curve (97), to the $D_{S^1}E_2$ curve obtained as a limit of the $E$-string curve [24]. This is simply done by finding the map between the gauge theory parameters and the $E_2 = SU(2) \times U(1)$ characters, as explained in appendix A, which gives:

$$\chi_1^{E_2} = \sqrt{\lambda} + \frac{1}{\sqrt{\lambda}}, \qquad \chi_{U(1)}^{E_2} = \lambda^{\frac{1}{4}} M_1. \tag{404}$$

Additionally, the $U$ parameter differs by an overall normalization $\lambda^{\frac{1}{4}}$. In terms of the $E_2$ characters, the $(I_7, 2II, I_1)$ configuration occurs for:

$$\chi_1^{E_2} = \frac{13}{8\sqrt{2}}, \qquad \chi_{U(1)}^{E_2} = 2^{\frac{7}{4}}. \tag{405}$$

We then find that:

$$\lambda^{-\frac{1}{4}} U(\tau) = 2^{\frac{1}{4}} \left( \frac{9}{2} + \left( \frac{\eta\left(\frac{\tau}{7}\right)}{\eta(\tau)} \right)^4 \right), \tag{406}$$

which is indeed the Hauptmodul for $\Gamma^0(7)$. One can check that the $I_1$ singularity is at $\tau = 0$ and that the type-$II$ elliptic points are in the $\Gamma^0(7)$ orbit of $\tau_1 = 3 + e^{\frac{2\pi i}{3}}$ and $\tau_2 = 5 + e^{\frac{2\pi i}{3}}$, respectively.

## 7.2 CB configurations for the $E_3$ theory

Let us now consider the $E_3$ theory, which is the UV fixed point of the 5d $SU(2)$ gauge theory with $N_f = 2$. The Weierstrass form of the toric curve (94) reads:

$$
\begin{aligned}
g_2(U) = \frac{1}{12} \Big( & U^4 - 8(1 + (1+p)\lambda) U^2 - 24\lambda s U \\
& + 16 \left( 1 - (1+p)\lambda + (1-p+p^2)\lambda^2 \right) \Big), \\
g_3(U) = -\frac{1}{216} \Big( & U^6 - 12(1+(1+p)\lambda) U^4 - 36\lambda s U^3 \\
& + 24 \left( 2 + (1+p)\lambda + (2+p+p^2)\lambda^2 \right) U^2 + 144\lambda s (1+(1+p)\lambda) U \\
& + 4\lambda^2 \left( 8 - 96p + 216p^2 + 54s^2 + \left( 22 + 27p + 3p^2 - 2p^3 \right) \lambda \right) \Big),
\end{aligned}
\tag{407}
$$

where we introduced $s = M_1 + M_2$ and $p = M_1 M_2$. Additionally, the $E_3$ characters can be worked out as described in appendix A. One finds:

$$\chi_1^{E_3} = \frac{1 + \lambda(1+p)}{\kappa_{E_3}^2}, \qquad \chi_2^{E_3} = \lambda \frac{1 + (1+\lambda)p}{\kappa_{E_3}^4}, \qquad \chi_3^{E_3} = \frac{s\lambda}{\kappa_{E_3}^3}, \tag{408}$$

for $\kappa_{E_3} \equiv p^{\frac{1}{6}} \lambda^{\frac{1}{3}}$. Here, $\chi_1$ and $\chi_2$ are the $SU(3)$ characters, and $\chi_3$ is the $SU(2)$ character. The 16 allowed configurations of singular fibers are listed in table 11. Some of the configurations shown in the table appear in one- or multi-parameter families, being particular examples which exhibit enhanced $U$-plane symmetry. It is implicitly understood that, for the families with explicit free parameters, the given configuration appears for generic values of the parameter, with finitely many exceptions. It should be pointed out that two configurations appear more than once in Persson's classification [47]. These are:

$$\mathcal{S}_{7 \text{ or } 8} \cong (I_6, III, 3I_1), \qquad \mathcal{S}_{11 \text{ or } 12} \cong (I_6, I_2, 4I_1), \tag{409}$$

which both appear with two possible choices of MW torsion groups, $\mathbb{Z}_2$ or trivial. We will discuss the difference between the two momentarily.

Table 11: Allowed configurations of singular fibres on the Coulomb branch of the $D_{S^1}E_3$ theory. Here, we have $\delta = \frac{3^{3/4}}{2}e^{\frac{5\pi i}{12}}$, $\alpha_\pm = \frac{1}{9}(7\sqrt{6} \pm 5\sqrt{15})$ and $\kappa_\pm = 2 \pm \sqrt{3}$ and $b \in \mathbb{C}$.

| $\mathcal{S}_\#$ | $\{F_\nu\}$ | $\lambda$ | $M_1$ | $M_2$ | $\mathfrak{g}_F$ | rk($\Phi$) | $\Phi_{\text{tor}}$ |
|---|---|---|---|---|---|---|---|
| 1 | $I_6, I_3, I_2, I_1$ | 1 | 1 | 1 | $A_2 \oplus A_1$ | 0 | $\mathbb{Z}_6$ |
| 2 | $I_6, IV, 2I_1$ | 1 | $i$ | $-i$ | $A_2 \oplus \mathfrak{u}(1)$ | 1 | $\mathbb{Z}_3$ |
| 3 | $I_6, I_3, 3I_1$ | 1 | $e^{\frac{2\pi i}{3}}$ | $e^{\frac{4\pi i}{3}}$ | $A_2 \oplus \mathfrak{u}(1)$ | 1 | $\mathbb{Z}_3$ |
| 4 | $I_6, 2I_2, 2I_1$ | 1 | $M$ | $M$ | $2A_1 \oplus \mathfrak{u}(1)$ | 1 | $\mathbb{Z}_2$ |
| 5 | $I_6, III, I_2, I_1$ | 1 | $\frac{1}{2}$ | $\frac{1}{2}$ | $2A_1 \oplus \mathfrak{u}(1)$ | 1 | $\mathbb{Z}_2$ |
| 6 | $I_6, III, II, I_1$ | 1 | $\delta - i$ | $\delta + i$ | $A_1 \oplus 2\mathfrak{u}(1)$ | 2 | $-$ |
| 7 | $I_6, III, 3I_1$ | $\frac{(1\pm M)^2}{M^2}$ | $M$ | $M$ | $A_1 \oplus 2\mathfrak{u}(1)$ | 2 | $\mathbb{Z}_2$ |
| 8 | $I_6, III, 3I_1$ | $-\frac{(1-M^2)^2}{4M^2}$ | $M$ | $\frac{1}{M}$ | $A_1 \oplus 2\mathfrak{u}(1)$ | 2 | $-$ |
| 9 | $I_6, I_2, 2II$ | 1 | $\frac{1}{2\sqrt{2}}$ | $-\frac{1}{2\sqrt{2}}$ | $A_1 \oplus 2\mathfrak{u}(1)$ | 2 | $-$ |
| 10 | $I_6, I_2, II, 2I_1$ | $-1$ | $\alpha_+$ | $\alpha_-$ | $A_1 \oplus 2\mathfrak{u}(1)$ | 2 | $-$ |
| 11 | $I_6, I_2, 4I_1$ | $\lambda$ | $M$ | $M$ | $A_1 \oplus 2\mathfrak{u}(1)$ | 2 | $\mathbb{Z}_2$ |
| 12 | $I_6, I_2, 4I_1$ | $\lambda$ | $M$ | $\frac{1}{M}$ | $A_1 \oplus 2\mathfrak{u}(1)$ | 2 | $-$ |
| 13 | $I_6, 3II$ | $e^{\frac{2\pi i}{3}}$ | $\kappa_- e^{\frac{5\pi i}{6}}$ | $\kappa_+ e^{\frac{11\pi i}{6}}$ | $3\mathfrak{u}(1)$ | 3 | $-$ |
| 14 | $I_6, 2II, 2I_1$ | $\chi = \left(-\frac{3}{8}, 3, 0\right)$ | | | $3\mathfrak{u}(1)$ | 3 | $-$ |
| 15 | $I_6, II, 4I_1$ | $\chi = \left(-6 + b^2, 12 - 6b, 6\right)$ | | | $3\mathfrak{u}(1)$ | 3 | $-$ |
| 16 | $I_6, 6I_1$ | $\lambda$ | $M_1$ | $M_2$ | $3\mathfrak{u}(1)$ | 3 | $-$ |

It is interesting to explore how various configurations can be connected, similarly to the 4d gauge-theory analysis. For instance, one can first start by setting the masses equal, $M_1 = M_2 = M$, which automatically leads to an $I_2$ singularity, namely the $\mathcal{S}_{11}$ configuration. Imposing $\lambda = \frac{1}{M^2}$ (or $\lambda = 1$), one finds a one-parameter family for the configuration $\mathcal{S}_4 \cong (I_6, 2I_2, 2I_1)$. Furthermore, tuning $M = \pm 2$ (or $M = \pm\frac{1}{2}$, respectively) leads to the configuration $\mathcal{S}_5 \cong (I_6, III, I_2, I_1)$, as one of the $I_2$ singularities combines with a mutually nonlocal $I_1$. Alternatively, starting from $\mathcal{S}_{11} \cong (I_6, I_2, 4I_1)$, one can set $\lambda = \frac{(1\pm M)^2}{M^2}$ instead, which leads to the $\mathcal{S}_7 \cong (I_6, III, 3I_1)$ configuration. Note that from the family $\mathcal{S}_7$, it is not possible to further tune the parameter $M$ in such a way as to obtain the type-$IV$ singularity in the configuration $\mathcal{S}_2$.

Another starting point would be to set the masses $M_1 = \frac{1}{M_2} = M$, which, in the 4d gauge-theory limit, would still correspond to $|m_1^{(4d)}| = |m_2^{(4d)}|$. In this case we also an $I_2$ fiber along that family, which we will argue to be $\mathcal{S}_{12}$. Compared to the previous case, it is now possible to obtain a type-$IV$ singular fiber, as follows. One can first set $\lambda = 1$, leading to $\mathcal{S}_3 = (I_6, I_3, 3I_1)$, and then subsequently set $M = i$ to reach $\mathcal{S}_2 \cong (I_6, IV, 2I_1)$. Alternatively, one can tune $\lambda = -\frac{(1-M^2)^2}{4M^2}$ to find $\mathcal{S}_8 \cong (I_6, III, 3I_1)$, which is further 'enhanced' to $\mathcal{S}_2 \cong (I_6, IV, 2I_1)$ by setting again $M = i$. In other words, if we start from the massless configuration $\mathcal{S}_1 \cong (I_6, I_3, I_2, I_1)$, the type-$IV$ singularity is formed when the $I_2$ fiber 'splits' into two $I_1$ singular fibers, with one

of them merging with the $I_3$. From these considerations, we see that the two configurations $\mathcal{S}_7$ and $\mathcal{S}_8$, despite having the same singular fiber types, have different physics. The different torsion groups reflect that fact.[35] A similar reasoning can be applied to the two distinct $(I_6, I_2, 4I_1)$ configurations.

It is apparent from table 11 that we can find interesting configurations on the Coulomb branch of the $D_{S^1}E_3$ theory even for $|\lambda| = |M_i| = 1$. In field theory language, this corresponds to turning on flavour Wilson lines along the $S^1$ in an otherwise massless five-dimensional theory. We analyse some of these configurations in the following, starting with the massless curve.

### 7.2.1 Massless curve, modularity and flavour symmetry group

The massless configuration corresponds to the extremal rational elliptic surface $X_{6321}$, which has a $\mathbb{Z}_6$ MW group. The Weierstrass form of the massless curve reads:

$$
\begin{aligned}
g_2(U) &= \frac{1}{12} U\left(U^3 - 24U - 48\right), \\
g_3(U) &= -\frac{1}{216}\left(U^6 - 36U^4 - 72U^3 + 216U^2 + 864U + 864\right),
\end{aligned}
\tag{410}
$$

with the discriminant:

$$
\Delta(U) = (U - 6)(U + 2)^3(U + 3)^2.
\tag{411}
$$

Consequently, the $U$-plane has three singularities in the bulk, $(I_1, I_3, I_2)$, and one $I_6$ cusp at infinity. The geometric periods:

$$
\omega = \frac{d\Pi}{dU},
\tag{412}
$$

satisfy the Picard-Fuchs equation:

$$
(U - 6)(U + 2)(U + 3)\frac{d^2\omega}{dU^2} + (3U^2 - 2U - 24)\frac{d\omega}{dU} + U\omega = 0.
\tag{413}
$$

Compared to the previous examples, this has four singular points. We will not solve (413) directly, but rather use the modular properties of the curve [192]. One can first compute $U(\tau)$, to find:

$$
U(\tau) = q^{-\frac{1}{6}} + 1 + 6q^{\frac{1}{6}} + 4q^{\frac{1}{3}} - 3q^{\frac{1}{2}} - 12q^{\frac{2}{3}} - 8q^{\frac{5}{6}} + \mathcal{O}(q) = 6 + \frac{\eta\left(\frac{\tau}{6}\right)^5 \eta\left(\frac{\tau}{2}\right)}{\eta(\tau)^5 \eta\left(\frac{\tau}{3}\right)},
\tag{414}
$$

which gives the McKay-Thompson series of class $6E$ of the Monster group and the Hauptmodul for $\Gamma^0(6)$. The massless $D_{S^1}E_3$ SW geometry is therefore modular for that congruence subgroup. This expression is invariant under a $T^6$ transformation, while under an $S$ transformation it becomes:

$$
U(\tau_D) = 6 + 72\frac{\eta(6\tau_D)^5\eta(2\tau_D)}{2\eta(\tau_D)^5\eta(3\tau_D)} = 6 + 72q_D + 360q_D^2 + \mathcal{O}(q_D^3),
\tag{415}
$$

where $\tau_D = -\frac{1}{\tau}$. Thus, the $I_1$ singularity at $U = 6$ corresponds to an $I_1$ cusp at $\tau = 0$. To find the other cusps, we use the properties of the $\eta$-function listed in appendix B. One can show that, under an $ST^3$ transformation, we have:

$$
U(\widetilde{\tau}_D) = 6 - \frac{9\eta\left(3\widetilde{\tau}_D\right)^{14}}{\eta\left(6\widetilde{\tau}_D\right)^5 \eta\left(2\widetilde{\tau}_D\right) \eta\left(\frac{3\widetilde{\tau}_D}{2}\right)^5 \eta\left(\widetilde{\tau}_D\right)^2 \eta\left(\frac{\widetilde{\tau}_D}{2}\right)} = -3 - 9\widetilde{q}_D^{\frac{1}{2}} + \mathcal{O}(\widetilde{q}_D),
\tag{416}
$$

---

[35]For instance, the fact that $\mathcal{S}_8$ contains $\mathcal{S}_2$ as a limit implies it cannot have $\mathbb{Z}_2$ torsion, because $\mathcal{S}_2$ has $\mathbb{Z}_3$ torsion and $\mathbb{Z}_2$ is not a subgroup of $\mathbb{Z}_3$.

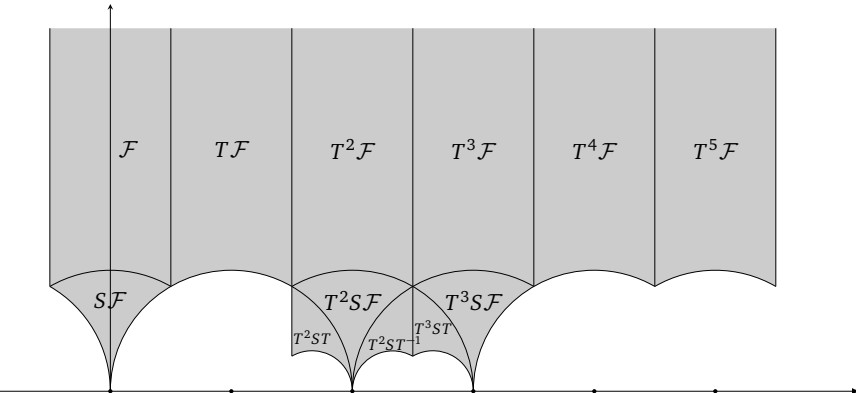

Figure 19: Fundamental domain for $\Gamma^0(6)$ with cusps at $\tau = 0, 2, 3$ and $i\infty$ of widths $1, 3, 2$ and $6$, respectively.

where $\widetilde{\tau}_D = -\frac{1}{\tau+3}$. As a result, the $I_2$ cusp at $\tau = 3$ corresponds to the $U = -3$ singularity. Finally, an $ST^2$ transformation leads to a more involved closed form expression, with the power series:

$$U(\widetilde{\tau}_D) = -2 - 8e^{\frac{2\pi i}{3}} \widetilde{q}_D^{\frac{1}{3}} - 24e^{\frac{\pi i}{3}} \widetilde{q}_D^{\frac{2}{3}} + \mathcal{O}(\widetilde{q}_D), \tag{417}$$

where here $\widetilde{\tau}_D = -\frac{1}{\tau+2}$, and thus the $U = -2$ singularity corresponds to the $I_3$ cusp at $\tau = 2$. The corresponding fundamental domain of $\Gamma^0(6)$ is shown in figure 19. We also find that the $a$ period satisfies:

$$\frac{da}{dU}(\tau) = -\frac{1}{2\pi i} \frac{\eta\left(\frac{\tau}{6}\right)\eta(\tau)^6}{\eta\left(\frac{\tau}{3}\right)^2 \eta\left(\frac{\tau}{2}\right)^3}. \tag{418}$$

**Flavour symmetry group, BPS states and BPS quiver.** From the fundamental domain shown in figure 19, we can directly read off the monodromies around the singularities of the massless $E_3$ curve. We find that the following dyons of the KK theory $D_{S^1}E_3$ become massless at these points:

$$
\begin{array}{llll}
U = & 6\,(\tau = 0) & (I_1): & \text{a monopole of charge } (1, 0; 0, 0) \text{ becomes massless,} \\
U = & -2\,(\tau = 2) & (I_3): & \text{three dyons of charge } (-1, 2; 0, 1) \text{ become massless,} \\
U = & -3\,(\tau = 3) & (I_2): & \text{two dyons of charge } (1, -3; 1, 0) \text{ become massless,}
\end{array}
\tag{419}
$$

in the notation $(m, q; l_1, l_2)$ of (198), with $(l_1, l_2) \in \mathbb{Z}_2 \times \mathbb{Z}_3$ the charge under the center of the $SU(2) \times SU(3)$ flavour group associated to the $I_2$ and $I_3$ cusps. where the states are charged under $U(1)_m^{[1]} \times U(1)_e^{[1]} \times \mathbb{Z}_2 \times \mathbb{Z}_3$, with $\mathbb{Z}_k$ the center of the flavour symmetry associated to the $I_k$ cusps. In particular let us also note that the group $\mathscr{E} = \mathbb{Z}_2 \times \mathbb{Z}_3 \cong \mathbb{Z}_6$, generated by:

$$g^{\mathscr{E}} = \left(0, \frac{1}{3}; 0, 1\right), \quad \left(0, \frac{1}{2}; 1, 0\right), \tag{420}$$

leaves these states invariants. We can then conclude that the actual flavour group is:

$$G_F = E_3/\mathbb{Z}_6 = PSU(3) \times SO(3), \tag{421}$$

as expected. The BPS quiver corresponding to the states (419) takes the form of a 3-blocks quiver, for a total of 6 nodes:

$$\begin{array}{ccc}
\mathcal{E}_{\gamma_1=(1,0)} & \longrightarrow\!\!\!\!\to & \mathcal{E}_{\gamma_{2,3,4}=(-1,2)} \\[2pt]
\Big\Uparrow & \nearrow & \\[2pt]
\mathcal{E}_{\gamma_{5,6}=(1,-3)} & &
\end{array} \tag{422}$$

which makes the $\mathfrak{su}(3) \oplus \mathfrak{su}(2)$ symmetry – or rather, its Weyl group $S_3 \times S_2$ – apparent. This quiver is a well-known 'toric quiver' for local $dP_3$ – see for example the 'Model 10d' in [193].

**Torsion sections for the massless curve.** The global form of the flavour group of massless $D_{S^1}E_3$ can also be understood from the MW group $\Phi \cong \mathbb{Z}_6$. The sections of the massless $E_3$ curve are given explicitly by:

$$\begin{aligned}
P_1 &= \left( \frac{1}{12}(U^2 + 12U + 24), -(U+2)(U+3) \right), \\
P_2 &= \left( \frac{1}{12}U^2, -(U+2) \right), \\
P_3 &= \left( \frac{1}{12}(U^2 - 12), 0 \right), \\
P_4 &= \left( \frac{1}{12}U^2, U+2 \right), \\
P_5 &= \left( \frac{1}{12}(U^2 + 12U + 24), (U+2)(U+3) \right),
\end{aligned} \tag{423}$$

which indeed span a $\mathbb{Z}_6 \cong \mathbb{Z}_2 \times \mathbb{Z}_3$ torsion group, with $P_k + P_l = P_{k+l \ (\mathrm{mod}\ 6)}$. One can then check that $P_2$ and $P_4$ only intersect the $\Theta_{v,0}$ component of the $I_2$ fiber, so that these sections generate a $\mathbb{Z}_3$ subgroup which injects into $\mathbb{Z}_3 = Z(SU(3))$ at the $I_3$ singularity. Similarly, the $P_3$ section intersects the $\Theta_{v,0}$ component of the $I_3$ fiber, and generates a $\mathbb{Z}_2$ subgroup that injects into $\mathbb{Z}_2 = Z(SU(2))$ at the $I_2$ singularity. As a result, the global form of the flavour symmetry is indeed $(SU(3) \times SU(2))/\mathbb{Z}_6$.

### 7.2.2 Other interesting $D_{S^1}E_3$ CB configurations

$\mathcal{S}_{16} \cong (I_6, 6I_1)$ **with** $\mathbb{Z}_6$ **symmetry.** The configuration $\mathcal{S}_{16}$ appears at generic points on the Coulomb branch of $E_3$. We can, however, tune the parameters to obtain a $\mathbb{Z}_6$-symmetric configuration, with the $U$-plane singularities organized as the roots of $U^6 = 432$. This occurs for $(\lambda, M_1, M_2) = (e^{\frac{4\pi i}{3}}, e^{\frac{7\pi i}{6}}, e^{\frac{i\pi}{6}})$. Note that this corresponds to setting the three $E_3$ characters to zero. In this case, $U(\tau)$ is a solution to:

$$(X(\tau) - 432)j(\tau) = X(\tau)^2, \tag{424}$$

for $X(\tau) = U(\tau)^6$. This leads to the solutions:

$$X(\tau) = 864 \, \frac{E_4(\tau)^{\frac{3}{2}}}{E_4(\tau)^{\frac{3}{2}} \pm E_6(\tau)}. \tag{425}$$

This function appeared for the massless 4d $SU(2)$ $N_f = 1$ theory in table 9, and it is not a modular function for any of the congruence subgroups. Nonetheless, the $\mathbb{Z}_6$ symmetry leads

to a major simplification of the PF equation, and the geometric periods can be expressed in terms of hypergeometric functions. In particular, we find that:

$$\frac{da}{dU} = -\frac{1}{2\pi i}\frac{1}{U}\,_2F_1\left(\frac{1}{6},\frac{5}{6},1;\frac{432}{U^6}\right), \qquad \Pi_f = 2a\,. \tag{426}$$

The periods can thus be determined explicitly on the $U$-plane, exactly as for the $E_1$ theory. Using the $\mathbb{Z}_6$ symmetry of the $U$-plane one can first determine the geometric periods $\widetilde{\omega} = \theta_U \Pi$ on the $w = \frac{U^6}{432}$ plane. In this case, the $\mathbb{Z}_6$ 'orbifold' monodromy corresponds to a type-$II^*$ singularity at the origin of the $w$-plane. It reads:

$$\mathbb{M}^+_{w=0} = \begin{pmatrix} 0 & -1 \\ 1 & 1 \end{pmatrix} = ST\,, \qquad \mathbb{M}^-_{w=0} = \begin{pmatrix} 1 & -1 \\ 1 & 0 \end{pmatrix} = T(ST)T^{-1}\,, \tag{427}$$

for a base point in the upper/lower half-plane, respectively. Consequently, we can find the $U$-plane monodromies by conjugating the 'conifold' monodromy, exactly as in previous examples:

$$\mathbb{M}_k = \left(\mathbb{M}^-_{w=0}\right)^k \mathbb{M}_{w=1}\left(\mathbb{M}^-_{w=0}\right)^{-k}\,, \qquad \mathbb{M}_{w=1} = STS^{-1}\,, \tag{428}$$

for $k = 0,\dots,5$. We then find that the BPS states becoming massless at the six $I_1$ cusps are:

$$\pm(1,0)\,, \qquad \pm(0,1)\,, \qquad \pm(1,-1)\,. \tag{429}$$

The corresponding $\mathbb{Z}_6$-symmetric quiver takes the form:

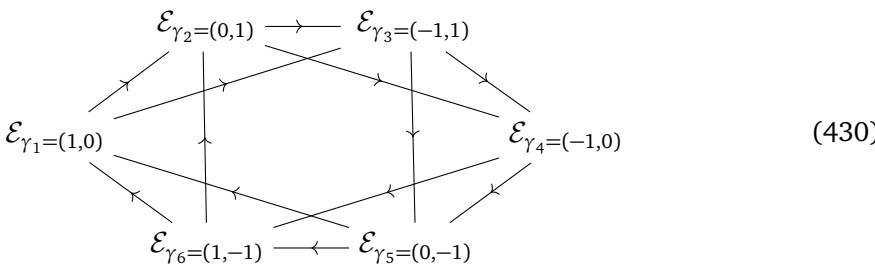

$$\tag{430}$$

This is again a well-known quiver for $dP_3$, called 'Model 10a' in [193]. Let us also mention that the $a$ period satisfies the simple relation:

$$U(\tau)\frac{da}{dU}(\tau) = -\frac{1}{2\pi i}E_4(\tau)^{\frac{1}{4}}\,. \tag{431}$$

With a bit more work, one can also establish the existence of a quiver point at $U = 0$, where the central charges of the 6 fractional branes align to $\mathcal{Z} = \frac{1}{6}$.

**$\mathcal{S}_7 \cong (I_6, III, 3I_1)$ with $\mathbb{Z}_3$ symmetry.** This configuration can be obtained for unit absolute values of the parameters from the one-parameter family $\mathcal{S}_7$ in table 11 by setting $M = e^{\frac{2\pi i}{3}}$, for instance. This is equivalent to $(\lambda, M_1, M_2) = (e^{-\frac{2\pi i}{3}}, e^{\frac{2\pi i}{3}}, e^{\frac{2\pi i}{3}})$, or to setting the $E_3$ characters to $\chi_1^{E_3} = \chi_2^{E_3} = 0$ and $\chi_3^{E_3} = 2$. In that case, the discriminant becomes:

$$\Delta(U) = U^3(U^3 - 64)\,, \tag{432}$$

with three $I_1$ type singularities at $U^3 = 64$ and an elliptic point (a type-$III$ singularity) at $U = 0$, where we have a $H_1$ AD theory at low energy. As in previous examples, we can flow directly to this Argyres-Douglas fixed point, bypassing the 4d gauge theory. Before taking the

4d limit explicitly, let us look at the modular properties of this $D_{S^1}E_3$ curve. $U(\tau)$ can be obtained by solving the cubic equation:

$$(X(\tau) - 64)j(\tau) = (X(\tau) - 48)^3, \tag{433}$$

for $X(\tau) = U(\tau)^3$. It turns out that $X(\tau)$ is a modular function for the $\Gamma^0(2)$ congruence subgroup, with a root:

$$X(\tau) = 64 + \left( \frac{\eta\left(\frac{\tau}{2}\right)}{\eta(\tau)} \right)^{24}, \tag{434}$$

which is the Hauptmodul for $\Gamma^0(2)$. Furthermore, the $a$-period satisfies:

$$\frac{da}{dU} = -\frac{1}{2\pi i}\frac{1}{U}\,{}_2F_1\left(\frac{1}{4}, \frac{3}{4}, 1; \frac{64}{U^3}\right), \qquad U(\tau)\frac{da}{dU}(\tau) = -\frac{\sqrt{2}}{4\pi i}\left(\vartheta_3(\tau)^4 + \vartheta_4(\tau)^4\right)^{\frac{1}{2}}. \tag{435}$$

As a result, one can find the analytic continuation of the periods throughout the whole $U$-plane, as before, and one can determine which BPS states become massless at the $U$-plane singularities.

In order to better understand the difference between this configuration, $\mathcal{S}_7 \cong (I_6, III, 3I_1)$, and the configuration $\mathcal{S}_8$ to be discussed momentarily, let us look at the BPS states from the point of view of the massless theory. In the configuration at hand, where $M_1 = M_2$, the type-$III$ singularity is obtained when the two dyons $(1, -3)$ of the $I_2$ singularity in (419) as well as one of the dyons $(-1, 2)$ or $(1, 0)$ become massless at the same point on the $U$-plane, depending on the deformation pattern. In the first case, for instance, we have:

$$\mathbb{M}_{III} = \mathbb{M}_{(1,-2)}\mathbb{M}_{(1,-3)}^2 = T^3 S^{-1} T^{-3}. \tag{436}$$

In this configuration, we have the light BPS states:

$$\mathscr{S}: \underbrace{(1,-3;1),\ (1,-2;0)}_{H_1},\quad 2(-1,2;0),\quad (1,0;0), \tag{437}$$

where the third entry is the $\mathbb{Z}_2$ center charge associated with the $\mathfrak{su}(2)$ symmetry from the type-$III$ singularity. Here, we see that all the particles are left invariant by a $\mathbb{Z}_2$ generated by:

$$g^{\mathscr{E}} = \left(0, \frac{1}{2}; 1\right), \tag{438}$$

and therefore the non-abelian part of the flavour symmetry of this CB configuration is $SU(2)/\mathbb{Z}_2$, in agreement with the MW torsion. We can 'zoom in' onto the AD theory by taking a 4d limit, as in other examples. In order to keep track of the parameters of the $H_1$ theory, we consider the limit:

$$U \to u\,\beta^{\frac{4}{3}} + 2c\,\beta^{\frac{2}{3}}, \quad \lambda \to e^{\frac{4\pi i}{3}}\left(1 - 2c\,\beta^{\frac{2}{3}}\right), \quad M_{1,2} \to e^{\frac{2\pi i}{3}}\left(1 \pm \mu\,\beta\right). \tag{439}$$

Using the scaling $(x, y) \to (\beta^{\frac{2}{3}}x, \beta y)$, the curve reduces to:

$$y^2 = 4x^3 - \left(\frac{16c^2}{3} - 4u\right)x - \left(\frac{64c^3}{27} - \frac{8c}{3}u - 4\mu^2\right), \tag{440}$$

which, upon the redefinition $u \to 4u + c^2$ becomes the Weierstrass form of the usual curve (403) for the $H_1$ AD theory. This shows that there exists an RG flow from the $E_3$ theory to the $H_1$ Argyres-Douglas theory, which moreover preserves the full symmetry group $SO(3) \subset E_3/\mathbb{Z}_6$ along the flow. We emphasise again that, in this setup, the starting point is the massless $E_3$ theory with certain Wilson lines turned on along the $S^1$.

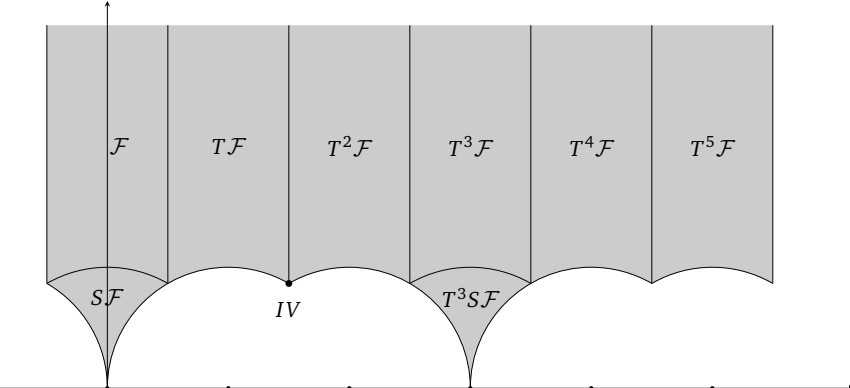

Figure 20: Fundamental domain for the $(I_6, IV, 2I_1)$ configuration on the CB of the $D_{S^1}E_3$ theory.

$\mathcal{S}_8 \cong (I_6, III, 3I_1)$. This other $(I_6, III, 3I_1)$ configuration can be obtained from the massless configuration by first 'splitting' the $I_2$ singularity of the massless curve through setting $M_1 = 1/M_2$, and then merging one of the resulting $I_1$ fibers with an $I_2$ subset of the $I_3$ fiber. In terms of the BPS states (419), we then have the monodromy:

$$\mathbb{M}_{III} = \mathbb{M}^2_{(1,-2)}\mathbb{M}_{(1,-3)} = T^2 S^{-1} T^{-2}. \tag{441}$$

It is interesting to contrast the charges of the light BPS states to (437). In the present case, we have:

$$\mathscr{S} : \underbrace{(1,-3;0), (-1,2;1)}_{H_1}, \quad (-1,2;0), \quad (1,-3,0), \quad (1,0;0). \tag{442}$$

In this case, the $\mathbb{Z}_2$ action $g^{\mathscr{E}} = (\frac{1}{2}, \frac{1}{2}; 1)$ leaves invariant the states involved in the $H_1$ point, but not the full spectrum of the larger theory. This is similar to the discussion at the end of section 4.3.4.

$\mathcal{S}_2 \cong (I_6, IV, 2I_1)$ and the $H_2$ AD point. This configuration shows perhaps the most 'unexpected' structure of the $U$ plane. For the values of the parameters displayed in table 11, the discriminant becomes:

$$\Delta(U) = U^4(U^2 - 27), \tag{443}$$

with two $I_1$ singularities at $U^2 = 27$ and a type-$IV$ singularity at the origin. In this configuration, the $E_3$ characters are:

$$\chi_1^{E_3} = 3, \qquad \chi_2^{E_3} = 3, \qquad \chi_3^{E_3} = 0. \tag{444}$$

The low-energy physics of this singularity is the AD theory $H_2$, which appears on the Coulomb branch of the 4d $SU(2)$ theory with $N_f = 3$ flavours, while the gauge theory phase of the $E_3$ theory only has $N_f = 2$. Let us first analyse the modular properties of this curve. $U(\tau)$ can be obtained by solving:

$$(X(\tau) - 27)\, j(\tau) = X(\tau)(X(\tau) - 24)^3, \tag{445}$$

with $X(\tau) = U(\tau)^2$. While the $\mathcal{S}_2$ curve is not modular, we find that $X(\tau)$ itself is a modular function for the $\Gamma^0(3)$ congruence subgroup, being generated by:

$$X(\tau) = 27 + \left(\frac{\eta\left(\frac{\tau}{3}\right)}{\eta(\tau)}\right)^{12}. \tag{446}$$

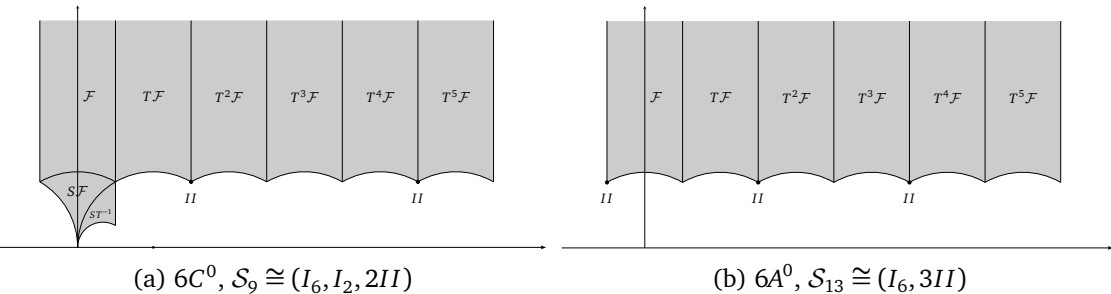

(a) $6C^0$, $\mathcal{S}_9 \cong (I_6, I_2, 2II)$

(b) $6A^0$, $\mathcal{S}_{13} \cong (I_6, 3II)$

Figure 21: Fundamental domains for two modular configurations on the CB of $D_{S^1}E_3$.

Furthermore, the $a$ period satisfies:

$$\frac{da}{dU} = -\frac{1}{2\pi i} \frac{1}{U} \, {}_2F_1\left(\frac{1}{3}, \frac{2}{3}; 1; \frac{27}{U^2}\right), \tag{447}$$

which can be thus solved explicitly on the $U$-plane. The way this configuration appears can be also understood from the corresponding fundamental domain, which is shown in figure 20. Note that despite the fact that the configuration $\mathcal{S}_2$ is not modular, we can draw a fundamental domain by making use of the fundamental domain of $\Gamma^0(3)$. Starting with the BPS states of the massless theory (419), the type-$IV$ singularity appears when the three dyons $(-1, 2)$ of the $I_3$ cusp become massless at the same point with one of the dyons $(1, -3)$ of the $I_2$ cusp, with the monodromy:

$$\mathbb{M}_{IV} = \mathbb{M}^3_{(-1,2)}\mathbb{M}_{(1,-3)} = T^2(ST)^{-2}T^{-2}. \tag{448}$$

This is consistent with the $\Gamma^0(3)$ Hauptmodul given in (446), as one can check that $X(\tau_*) = 0$, for $\tau_* = 2 + e^{\frac{2\pi i}{3}}$. We can again obtain the full 4d curve, including all the 4d parameters, from the full $D_{S^1}E_3$ curve. Recall that the $H_2$ theory has a Coulomb branch parameter $u$ of scaling dimension $\Delta_u = \frac{3}{2}$, together with the parameters $(c, \mu_1, \mu_2)$, of scaling dimensions $\left(\frac{1}{2}, 1, 1\right)$. We then consider:

$$U \longrightarrow u \, \beta^{\frac{3}{2}} + 2c \, \beta^{\frac{1}{2}}, \quad \lambda \longrightarrow 1 - \mu_1 \beta, \qquad M_{1,2} \to \pm i - c \, \beta^{\frac{1}{2}} \pm \frac{i \, \mu_2}{2}\beta, \tag{449}$$

together with $(x, y) \to \left(\beta x, \beta^{\frac{3}{2}} y\right)$. Upon the redefinition $u \to u - c \, \mu_1$, the curve becomes:

$$\begin{aligned}
g_2(u) &= \frac{4}{3}\left(\mu_1^2 - \mu_1\mu_2 + \mu_2^2 - 3cu\right), \\
g_3(u) &= \frac{1}{27}\left(8(\mu_1^3 + \mu_2^3) - 12\mu_1\mu_2(\mu_1 + \mu_2) - 27c^2\mu_1^2 + 18c(\mu_1 - 2\mu_2)u - 27u^2\right).
\end{aligned} \tag{450}$$

This is the Weierstrass form of the curve:

$$F(x, t) = \mu_2 t + \mu_1 x + u + tx(t + x) + t^2 c = 0, \tag{451}$$

which is precisely the SW curve for the AD theory $H_2$ [4, 191].

**Modular configurations.** Let us briefly comment on the remaining configurations. As for $D_{S^1}E_2$, we can use the classification of the genus-zero congruence subgroups [174] to check which configurations are modular. We find that the configurations $\mathcal{S}_9 \cong (I_6, I_2, 2II)$ and $\mathcal{S}_{13} \cong (I_6, 3II)$ correspond to the congruence subgroups $6C^0$ and $6A^0$, respectively, in their

notation. For the first configuration, for the values of the parameters listed in table 11, the $U$-plane is $\mathbb{Z}_2$ symmetric and we find that:

$$U(\tau) = \frac{i}{\sqrt{2}} \left( \frac{\eta\left(\frac{\tau}{3}\right)}{\eta(\tau)} \right)^6 , \tag{452}$$

which reproduces the completely replicable function of class 6c [194]. One can further check that the type-$II$ elliptic points correspond to $\tau_1 = 2 + e^{\frac{2\pi i}{3}}$ and $\tau_2 = 5 + e^{\frac{2\pi i}{3}}$, while the $I_2$ cusp sits at $\tau = 0$. Finally, for the $\mathbb{Z}_3$ symmetric configuration $(I_6, 3II)$ listed in table 11, the $J$-invariant is a quadratic polynomial in $w = U^3$ and the monodromy group on the $w$-plane will be the subgroup of square elements of PSL$(2, \mathbb{Z})$, usually denoted by $\Gamma^2$. This has a cusp of width two and two elliptic points of order two, one of which is the 'orbifold' point on the $w$-plane. Going back to the $U$-plane, the orbifold singularity will be resolved, and thus, $\Gamma^2$ can be viewed as a triple cover of the monodromy group on the $U$-plane. We draw fundamental domains for these configurations in figure 21.

# 8 The non-toric $E_n$ theories – 5d $SU(2)$, $3 \leq N_f \leq 7$

In this section, we discuss various configurations of singular fibers of rational elliptic surfaces that correspond to the non-toric $D_{S^1} E_n$ theories. An important subset that we will focus on here consists of the extremal rational elliptic surfaces, which we introduced in section 3.3.

## 8.1 Massless curves and modularity

The Seiberg-Witten curves mirror to the non-toric local $dP_n$ geometries can be determined as limits of the $E$-string theory curve [11, 24, 156], and are usually expressed in terms of the $E_n$ characters, as we reviewed in section 2.4.2. The massless curves are obtained when the characters are set to the dimension of the corresponding $E_n$ representations. The modular properties of the curves for $n > 4$ were discussed in [102], while the periods for the $D_{S^1} E_n$ theories with $n > 5$ have been explicitly computed in *e.g.* [95, 98, 151]. For completeness, we summarise some relevant results below. Additionally, we list the torsion sections and discuss the global form of the flavour symmetry, which confirms that the flavour group is the centerless $E_n/Z(E_n)$, as anticipated in (121).

   We will also comment on the modular properties of these curves, which we anticipated in section 3.3. In table 7, and more specifically in table 2 in the introduction, we see the obvious pattern $\Gamma = \Gamma^0(9 - n)$ for $E_n$, with $n = 4$ being the important exception.[36] The massless $E_4$ curve corresponds to the configuration $\mathcal{S} \cong (2I_5, 2I_1)$ while, on the other hand, the congruence subgroup $\Gamma^0(5)$ has only two cusps[37] and thus cannot be the correct modular group. By direct computation, we find that $U(\tau)$ is in fact a modular function for $\Gamma^1(5)$, in this case. Note, however, that $\overline{\Gamma}^0(n) = \overline{\Gamma}^1(n)$ for $n = 2, 3, 4, 6$, where we use the $\overline{\Gamma}$ notation to emphasise that these are subgroups of PSL$(2, \mathbb{Z})$ rather than the full SL$(2, \mathbb{Z})$, and we can thus view the pattern $\Gamma_{E_n} = \Gamma^1(9 - n)$ as valid for $n > 2$ instead.[38]

---

[36]Another exception is $n = 8$: $\Gamma^0(1) = $ PSL$(2, \mathbb{Z})$ is indeed the monodromy group on the massless $E_8$ CB, but that CB configuration is not modular, as we will see.

[37]More generally, the congruence subgroup $\Gamma^0(p)$ for prime $p$ has only two cusps. This also 'explains' why $\Gamma^0(7)$ could not be a modular group for massless $E_2$.

[38]By a slight abuse of notation, we used $\Gamma(N), \Gamma^0(N), \cdots$ to denote the corresponding congruence subgroups of PSL$(2, \mathbb{Z})$ instead of SL$(2, \mathbb{Z})$.

### 8.1.1 The massless $D_{S^1}E_4$ theory

The massless curve for the $D_{S^1}E_4$ theory is obtained from the mass-deformed curve by setting the characters to $\chi = \{5, 10, 5, 10\}$. In our conventions, it reads:

$$
\begin{aligned}
g_2(U) &= \frac{1}{12}\left(U^4 - 40U^2 - 120U - 80\right), \\
g_3(U) &= -\frac{1}{216}\left(U^6 - 60U^4 - 180U^3 + 480U^2 + 2736U + 3160\right),
\end{aligned}
\tag{453}
$$

with the discriminant:

$$
\Delta(U) = (U^2 - 5U - 25)(U + 3)^5,
\tag{454}
$$

from which we see the $I_1, I_1, I_5$ singularities in the bulk, and the $I_5$ at infinity.

**Modular properties.** The PF equation (54) for this theory becomes:

$$
(U + 3)(U^2 - 5U - 25)\frac{d^2\omega}{dU^2} + (3U^2 - 4U - 40)\frac{d\omega}{dU} + U\omega = 0.
\tag{455}
$$

This differential equation has four singular points, and the solutions can be expressed in terms of the local Heun function [195]. Here, we instead use the modular properties of the curve to analyse the light BPS states appearing in the massless theory. We first find that:

$$
U(\tau) = q^{-\frac{1}{5}} + 2 + 10q^{\frac{1}{5}} + 5q^{\frac{2}{5}} - 15q^{\frac{3}{5}} - 24q^{\frac{4}{5}} + \mathcal{O}(q).
\tag{456}
$$

This turns out the be the Hauptmodul for $\Gamma^1(5)$, given explicitly in [170]:

$$
U(\tau) = -3 + q^{-\frac{1}{5}}\prod_{n=1}^{\infty}\left(1 - q^{\frac{n}{5}}\right)^{-5\left(\frac{n}{5}\right)},
\tag{457}
$$

where $\left(\frac{n}{5}\right)$ denotes the Legendre symbol. It is convenient to rewrite the above expression in terms of the Hauptmodul of $\Gamma^0(5)$:

$$
f(\tau) = \left(\frac{\eta\left(\frac{\tau}{5}\right)}{\eta(\tau)}\right)^6,
\tag{458}
$$

as described in [170]. We find that, around the cusp at infinity, we have:

$$
U(\tau) = \frac{1}{2}\left(5 + f(\tau) + \sqrt{125 + f(\tau)(22 + f(\tau))}\right).
\tag{459}
$$

A fundamental domain for $\Gamma^1(5)$ is shown in figure 22.

**BPS states and flavour group.** Using the fundamental domain in figure 22, one finds the following BPS states:

$$
\begin{array}{llll}
\tau = 0 & (I_1) & : & \text{a monopole of charge } (1, 0; 0), \\
\tau = \frac{5}{2} & (I_1) & : & \text{a dyon of charge } (-2, 5; 0), \\
\tau = 3 & (I_5) & : & \text{five dyons of charge } (1, -3; 1),
\end{array}
\tag{460}
$$

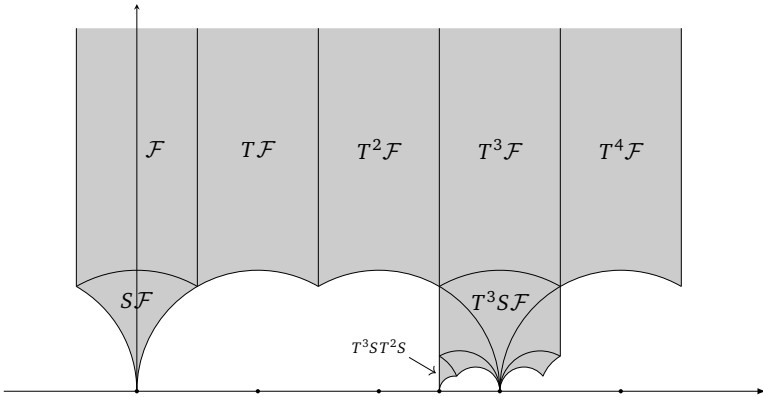

Figure 22: Fundamental domain for $\Gamma^1(5)$, with $(I_1, I_1, I_5, I_5)$ cusps at $\tau = 0, \frac{5}{2}, 3$ and $i\infty$.

where $(m, q; l)$ includes $l \in \mathbb{Z}_5$, indicating thus the charge under the center of the flavour $SU(5)$. The corresponding BPS quiver is a 3-blocks quiver with 7 nodes:

$$
\begin{array}{ccc}
\mathcal{E}_{\gamma_1=(1,0)} & \text{\ggg} & \mathcal{E}_{\gamma_2=(-2,5)} \\
\downarrow & \diagup & \\
\mathcal{E}_{\gamma_{3,4,5,6,7}=(1,-3)} & &
\end{array}
\tag{461}
$$

This is a known quiver for $dP_4$ [177]. Note that the D0-brane state corresponds to the ranks $(1; 3; 1)$ for the three blocks of this quiver. The BPS states (460) are left invariant by $\mathcal{E} = \mathbb{Z}_5$ generated by:

$$
g^{\mathcal{E}} = \left( 0, \frac{2}{5}; 1 \right),
\tag{462}
$$

which confirms the global form of the flavour group, $G_F = PSU(5)$.

**Torsion sections.** Another way to see the global form of the flavour group is from the MW group. In our conventions, the sections of the extremal rational elliptic surface are:

$$
\begin{aligned}
P_1 &= \left( \frac{1}{12}(U^2 - 8), U + 3 \right), \\
P_2 &= \left( \frac{1}{12}(U^2 + 12U + 28), -(U + 3)^2 \right), \\
P_3 &= \left( \frac{1}{12}(U^2 + 12U + 28), (U + 3)^2 \right), \\
P_4 &= \left( \frac{1}{12}(U^2 - 8), -(U + 3) \right),
\end{aligned}
\tag{463}
$$

with $P_k + P_l = P_{k+l \pmod 5}$. These sections intersect the $I_5$ fiber non-trivially, and thus the full $\Phi_{\text{tor}} = \mathbb{Z}_5$ restricts the global form of the flavour symmetry, as expected.

### 8.1.2 The massless $D_{S^1}E_5$ theory

The other massless theories can be treated similarly. The massless $E_5$ curve is obtained by fixing the characters $\chi^{E_5} = \{10, 45, 16, 120, 16\}$, which gives:

$$
\begin{aligned}
g_2(U) &= \frac{1}{12}(U+4)^2(U^2-8U-32), \\
g_3(U) &= -\frac{1}{216}(U+4)^3(U^3-12U^2-24U+224),
\end{aligned}
\tag{464}
$$

with the discriminant:

$$
\Delta(U) = (U+4)^7(U-12).
\tag{465}
$$

This is the $(I_4, I_1^*, I_1)$ configuration of singular fibers. Note that this configuration is related to the one for the pure 4d $SU(2)$ theory by a quadratic twist.

**Picard-Fuchs equation and modularity.** The Picard-Fuchs equation (54) for the geometric periods $\omega = d\Pi/dU$ reduces to:

$$
(U+4)^2(U-12)\frac{d^2\omega}{dU^2} + (U+4)(3U-20)\frac{d\omega}{dU} + U\omega = 0.
\tag{466}
$$

This is a hypergeometric differential equation, as can be easily seen by introducing $w = \frac{U+4}{16}$. A convenient basis of solutions for this is given by:

$$
\begin{aligned}
\omega_a &= -\frac{1}{2\pi i}\frac{1}{U+4}\,{}_2F_1\left(\frac{1}{2},\frac{1}{2};1;\frac{16}{U+4}\right), \\
\omega_D &= -\frac{1}{\pi}\frac{1}{U+4}\,{}_2F_1\left(\frac{1}{2},\frac{1}{2};1;1-\frac{16}{U+4}\right),
\end{aligned}
\tag{467}
$$

where $\omega_D$ is chosen such that it reproduces the $T^4$ monodromy at infinity. Then, the monodromies around the other cusps read:

$$
\mathbb{M}_{U=12} = STS^{-1}, \qquad \mathbb{M}_{U=-4} = (T^{\pm 2}S)(-T)(T^{\pm 2}S)^{-1},
\tag{468}
$$

where, for $U = -4$, one needs to specify the base point. The above monodromies can also be found from the modular properties of the curve. Solving the equation $J = J(\tau)$, we find that:

$$
U(\tau) = 12 + \left(\frac{\eta\left(\frac{\tau}{4}\right)}{\eta(\tau)}\right)^8,
\tag{469}
$$

which is the Hauptmodul for $\Gamma^0(4)$. Using the properties of the $\eta$-function, one can show that the $(I_1, I_1^*)$ cusps at $\tau = 0, \pm 2$ correspond to the singularities at $U = 12, -4$, as expected. The fundamental domain consistent with these values is shown in figure 23b below.

**Torsion sections and flavour group.** The discussion of the BPS states at the massless points of the $D_{S^1}E_n$ theories with $n > 4$ is more subtle, and we postpone it to the next subsection, and to future work. We can still determine the flavour symmetry group from our general approach using the MW torsion. In this case, we an extremal RES with $\Phi = \mathbb{Z}_4$, as shown in table 7. The sections are given explicitly by:

$$
\begin{aligned}
P_1 &= \left(\frac{1}{12}(U+4)(U+8), (U+4)^2\right), \\
P_2 &= \left(\frac{1}{12}(U^2-16), 0\right), \\
P_3 &= \left(\frac{1}{12}(U+4)(U+8), -(U+4)^2\right),
\end{aligned}
\tag{470}
$$

with $P_k + P_l = P_{k+l \pmod 4}$. They intersect non-trivially the $I_1^*$ fiber, and thus we can again argue that the flavour symmetry is given by $G_F = \mathrm{Spin}(10)/\mathbb{Z}_4$.

### 8.1.3 The massless $D_{S^1}E_6$ theory

The massless curve of the $D_{S^1}E_6$ theory takes the simple form:

$$g_2(U) = \frac{1}{12}(U+6)^3(U-18), \qquad g_3(U) = -\frac{1}{216}(U+6)^4(U^2-24U+36), \qquad (471)$$

with the discriminant:

$$\Delta(U) = (U+6)^8(U-21). \qquad (472)$$

This corresponds to the configuration $(I_3, IV^*, I_1)$, which is an extremal RES with $\Phi = \mathbb{Z}_3$.

**Picard-Fuchs equation and modularity.** The PF equation (54) for the geometric periods reads:

$$(U+6)^2(U-21)\frac{d^2\omega}{dU^2} + 3(U+6)(U-12)\frac{d\omega}{dU} + U\omega = 0, \qquad (473)$$

which is again a hypergeometric differential equation, similar to that of $D_{S^1}E_0$ on the $w = U^3$ plane. A convenient basis of solutions is given by:

$$\begin{aligned}
\omega_a &= -\frac{1}{2\pi i}\frac{1}{U+6}\,{}_2F_1\left(\frac{1}{3},\frac{2}{3};1;\frac{27}{U+6}\right), \\
\omega_D &= -\frac{\sqrt{3}}{2\pi}\frac{1}{U+6}\,{}_2F_1\left(\frac{1}{3},\frac{2}{3};1;1-\frac{27}{U+6}\right),
\end{aligned} \qquad (474)$$

with $\omega_D$ chosen such that the monodromy at infinity is $T^3$. The monodromies around the other singularities are:

$$\mathbb{M}_{U=21} = STS^{-1}, \qquad \mathbb{M}_{U=-6} = T^k(ST)^2T^{-k}, \qquad (475)$$

for $k = -1$ or $k = 2$, depending on the base point. These monodromies can be also recovered from the modular properties of the curve. We find that:

$$U(\tau) = 21 + \left(\frac{\eta\left(\frac{\tau}{3}\right)}{\eta(\tau)}\right)^{12}, \qquad (476)$$

which is the Hauptmodul for $\Gamma^0(3)$. One can then easily show that $U(\tau = 0) = 21$. Furthermore, it can be checked that $U(-1 + \tau_*) = U(2 + \tau_*) = -6$, with $\tau_* = e^{\frac{2\pi i}{3}}$, in agreement with the monodromies found from the geometric periods.

**Torsion sections and flavour group.** The SW geometry for the massless $D_{S^1}E_6$ theory is an extremal RES with $\Phi = \mathbb{Z}_3$. The sections read:

$$P_1 = \left(\frac{1}{12}(U+6)^2, (U+6)^2\right), \qquad P_2 = \left(\frac{1}{12}(U+6)^2, -(U+6)^2\right), \qquad (477)$$

with $P_1 + P_2 = O$. They intersect non-trivially the $IV^*$ singular fiber, and thus we argue that the global form of the flavour symmetry is $G_F = E_6/\mathbb{Z}_3$.

### 8.1.4  The massless $D_{S^1}E_7$ theory

The curve of the massless $D_{S^1}E_7$ theory is given by:

$$g_2(U) = \frac{1}{12}(U + 12)^3(U - 36), \qquad g_3(U) = -\frac{1}{216}(U + 12)^5(U - 60), \tag{478}$$

with the discriminant:

$$\Delta(U) = (U + 12)^9(U - 52). \tag{479}$$

This corresponds to the extremal configuration $(I_2, III^*, I_1)$, which has $\Phi = \mathbb{Z}_2$.

**Picard-Fuchs equation and modularity.**  The geometric periods satisfy the hypergeometric differential equation:

$$(U + 12)^2(U - 52)\frac{d^2\omega}{dU^2} + (U + 12)(3U - 92)\frac{d\omega}{dU} + U\omega = 0, \tag{480}$$

which is similar to that of the $\lambda = 1$ configuration for the $D_{S^1}E_1$ theory (on the $w = U^4$ plane of that theory). A convenient basis is given by:

$$
\begin{aligned}
\omega_a &= -\frac{1}{2\pi i}\frac{1}{U + 12}\,_2F_1\left(\frac{1}{4}, \frac{3}{4}; 1; \frac{64}{U + 12}\right), \\
\omega_D &= -\frac{1}{\sqrt{2}\pi}\frac{1}{U + 12}\,_2F_1\left(\frac{1}{4}, \frac{3}{4}; 1; 1 - \frac{64}{U + 12}\right),
\end{aligned}
\tag{481}
$$

with $\omega_D$ chosen such that the monodromy at infinity is $T^2$. The monodromies around the other singularities are:

$$\mathbb{M}_{U=52} = STS^{-1}, \qquad \mathbb{M}_{U=-12} = T^k S T^{-k}, \tag{482}$$

for $k = \pm 1$, depending on the base point. We can again show that these monodromies are consistent with the modular properties of the curve. We first find that:

$$U(\tau) = 52 + \left(\frac{\eta\left(\frac{\tau}{2}\right)}{\eta(\tau)}\right)^{24}, \tag{483}$$

which is the Hauptmodul for $\Gamma^0(2)$. This congruence subgroup only has two cusps of widths 1 and 2, respectively, and an elliptic point of order 2. One can check that $U(\tau = 0) = 52$ and, additionally, that $U(\pm 1 + i) = -12$. Thus, the elliptic point of $\Gamma^0(2)$ is precisely the type-$III^*$ singularity of the massless $D_{S^1}E_7$ curve.

**Torsion sections and flavour group.**  It is straightforward to check that the non-trivial section of the massless $D_{S^1}E_7$ curve is:

$$P_1 = \left(\frac{1}{12}(U + 12)^2, 0\right), \tag{484}$$

which spans a $\mathbb{Z}_2$ torsion group. It intersects non-trivially the type-$III^*$ singular fiber, and therefore the flavour symmetry group is $G_F = \mathrm{E}_7/\mathbb{Z}_2$.

### 8.1.5 The massless $D_{S^1}E_8$ theory

Finally, the massless curve for the $D_{S^1}E_8$ theory reads:

$$g_2(U) = \frac{1}{12}U^4, \qquad g_3(U) = -\frac{1}{216}(U-864)U^5, \qquad (485)$$

with the discriminant and $j$-invariant:

$$\Delta(U) = (U-432)U^{10}, \qquad j(U) = \frac{U^2}{(U-432)}. \qquad (486)$$

The rational elliptic surface associated to this Seiberg-Witten curve is extremal, with the singular fibers $(2I_1, II^*)$ and no torsion. The geometric periods can be determined explicitly, as they again satisfy a hypergeometric differential equation:

$$U^2(U-432)\frac{d^2\omega}{dU^2} + 3U(U-288)\frac{d\omega}{dU} + (U-60)\omega = 0, \qquad (487)$$

which is similar to that of the $\mathbb{Z}_6$ symmetric configuration of the $D_{S^1}E_3$ theory. We choose the periods:

$$\omega_a = -\frac{1}{2\pi i}\frac{1}{U}\,{}_2F_1\left(\frac{1}{6},\frac{5}{6};1;\frac{432}{U}\right), \qquad \omega_D = -\frac{1}{2\pi}\frac{1}{U}\,{}_2F_1\left(\frac{1}{6},\frac{5}{6};1;1-\frac{432}{U}\right), \qquad (488)$$

with $\omega_D$ chosen such that the monodromy at infinity is $T$. The other monodromies read:

$$\mathbb{M}_{U=432} = STS^{-1}, \qquad \mathbb{M}_{U=0} = T^k(ST)T^{-k}, \qquad (489)$$

for $k=0$ or $1$, depending on the base point. As for the modular properties, solving $J = J(\tau)$, the root corresponding to the cusp at $\tau = i\infty$ is given by:

$$U(\tau) = 864\frac{E_4(\tau)^3 + E_4(\tau)^{\frac{3}{2}}E_6(\tau)}{E_4(\tau)^3 - E_6(\tau)^2}, \qquad (490)$$

with its $S$-transformation:

$$U(\tau_D) = 864\frac{E_4(\tau)^3 - E_4(\tau)^{\frac{3}{2}}E_6(\tau)}{E_4(\tau)^3 - E_6(\tau)^2}. \qquad (491)$$

From the zeroes of the Eisenstein series $E_4$, it follows that the elliptic point of type $II^*$ corresponds to $\tau_* = e^{\frac{2\pi i}{3}}$ (or $e^{\frac{\pi i}{3}}$), in agreement with the monodromies found above. A fundamental domain can be chosen as in figure 10c, by replacing $IV$ with $II^*$ (a quadratic twist relates that configuration $(I_1^*, IV, I_1)$ to the $(I_1, II^*, I_1)$ configuration of interest here). Note that the monodromies (489) generate the full PSL$(2,\mathbb{Z})$ while the fundamental domain for the CB configuration consists of two copies of the PSL$(2,\mathbb{Z})$ fundamental domain, therefore the massless $D_{S^1}E_8$ CB is not a modular curve. This is consistent with the fact that the unique index 2 subgroup of PSL$(2,\mathbb{Z})$ only has one $I_2$ cusp [170, 174].

## 8.2 Other configurations: modular curves and 5d BPS quivers

In the rest of this section, we discuss some other interesting CB configurations for the non-toric $D_{S^1}E_n$ theories. Let us start with some general comments on the Higgs branch enhancement as we vary the parameters, using the 5d gauge-theory intuition as a guide, in analogy with the 4d gauge theory analysis [2, 4]. We begin by setting the 5d hypermultiplet mass parameters equal, $M_i = M$. This can be done explicitly by working out the map between the characters

Table 12: Some $M_i = M$ configurations for the $D_{S^1} E_n$ theories. Only the singularities in the interior of the $U$-plane are indicated.

| Theory | Generic | $M_i = M$, $\lambda \neq 1$ | $M_i = 1$, $\lambda \neq 1$ | $M_i = M$, $\lambda = 1$ | $M_i = 1$, $\lambda = 1$ |
|---|---|---|---|---|---|
| $D_{S^1} E_4$ | $7I_1$ | $4I_1, I_3$ | $3I_1, I_4$ | $2I_1, I_2, I_3$ | $2I_1, I_5$ |
| $D_{S^1} E_5$ | $8I_1$ | $4I_1, I_4$ | $2I_1, I_0^*$ | $2I_1, I_2, I_4$ | $I_1, I_1^*$ |
| $D_{S^1} E_6$ | $9I_1$ | $4I_1, I_5$ | $2I_1, I_1^*$ | $2I_1, I_2, I_5$ | $I_1, IV^*$ |
| $D_{S^1} E_7$ | $10I_1$ | $4I_1, I_6$ | $2I_1, I_2^*$ | $2I_1, I_2, I_6$ | $I_1, III^*$ |
| $D_{S^1} E_8$ | $11I_1$ | $4I_1, I_7$ | $2I_1, I_3^*$ | $2I_1, I_2, I_7$ | $I_1, II^*$ |

of $E_n$ and the gauge theory parameters $(\lambda, M_i)$, as discussed in appendix A. In this equal-mass setting and for generic values of the '5d gauge coupling' $\lambda$, $N_f = n - 1$ of the $I_1$ cusps will merge together into an $I_{N_f}$ singularity. The corresponding Higgs branch is the one associated classically with $N_f$ massive fundamental hypermultiplets of $SU(2)$. The flavour symmetry of these theories will thus be $\mathfrak{su}(N_f)$. As the mass is turned off, i.e. for $M \to 1$ with $\lambda \neq 1$, this enhances to $\mathfrak{so}(2N_f)$ – see table 12.

One can instead set $\lambda = 1$ first, in which case the $U$-plane singularities are $(2I_1, I_2, I_{N_f})$. In the large-mass limit ($M \gg 1$ or $M \ll 1$), the $(2I_1, I_2)$ cusps can be viewed as the bulk singularities of the 'massless' $D_{S^1} E_1$ theory, with the BPS states becoming massless at the various cusps listed in (304). Then, as $M \to 1$, the Higgs branch of the $D_{S^1} E_n$ theories changes as follows. For the $D_{S^1} E_4$ theory, the $I_{N_f=3}$ and the $I_2$ fibers merge, forming an $I_5$ singular fiber. For the other cases ($D_{S^1} E_{n>4}$), the $I_{N_f=n-1}$ and the $I_2$ singularity also merge with an $I_1$ singularity, leading to the massless configurations. We note that this discussion exactly parallels the F-theory analysis of the combinations of 7-branes needed to produce the $E_n$ 7-brane [49, 161, 196], which is of course no coincidence.

For $n = 6, 7, 8$, in the fully massless limit $M = \lambda = 1$, the 'elliptic' singularities $IV^*$, $III^*$ and $II^*$, respectively, that appear on the $U$-plane have a low-energy description in terms of the 4d MN theories [11], as we reviewed in section 2.4.2. It is worth remarking that this embedding of the 4d MN theories into the $U$-plane is qualitatively different from the way the AD points often appear (either on 4d $u$-planes or on the $U$-plane). In the latter case, the AD fixed points correspond to points where singularities merge without affecting flavour symmetry algebra nor the Higgs branch. On the other hand, at these $E_n$ MN singularities, the flavour algebra enhances and the Higgs branch dimension increases dramatically – see e.g. [30] for a discussion of the corresponding five-dimensional physics.

### 8.2.1 $D_{S^1} E_4$ configurations

Persson's list for the allowed configurations of singular fibers [47] contains 26 configurations with an $I_5$ fiber, which should all be achievable on the extended Coulomb branch of the $D_{S^1} E_4$ theory. The only configuration with non-trivial torsion turns out to be the massless one. Let us briefly comment on some of these configurations which show interesting symmetries or modular properties.

One such configuration is obtained by setting all the $E_4$ characters to zero. This is the $(I_5, 2I_1, II)$ configuration, with $\text{rk}(\Phi) = 4$, with the $U$-plane showing a $\mathbb{Z}_5$ symmetry. The geometric periods $\tilde{\omega} = \theta_U \Pi$ can be solved in closed form on the $w = U^5$ plane and are similar to those of the massless $E_8$ configuration. Another non-trivial configuration is the $(I_5, 3I_1, IV)$ configuration, which occurs for $(M_i) = (1, e^{\frac{i\pi}{3}}, e^{-\frac{i\pi}{3}})$. In this case the masses are not equal, and thus merging singularities correspond to non-local BPS states becoming massless at the

same point.

Based on the classification of genus-zero congruence subgroups [174], we can also list all $D_{S^1}E_4$ configurations for which the monodromy group is a congruence subgroup. These are given below:

$$
\begin{array}{|c||c|c|c|c|}
\hline
\{F_v\} & \mathrm{rk}(\Phi) & \Phi_{\mathrm{tor}} & \mathfrak{g}_F & \Gamma \in \mathrm{PSL}(2,\mathbb{Z}) \\
\hline
2I_5, 2I_1 & 0 & \mathbb{Z}_5 & A_4 & \Gamma^1(5) \\
\hline
I_5, I_1, 2III & 2 & - & 2A_1 & \Gamma^0(5) \\
\hline
I_5, III, 2II & 3 & - & A_1 & 5A^0 \\
\hline
\end{array}
\tag{492}
$$

where $5A^0$ is an index 5 congruence subgroup, with only one cusp of width 5. We leave a detailed study of the corresponding $U$-planes for future work.

### 8.2.2 $D_{S^1}E_5$ configurations

For the $D_{S^1}E_5$ theory there are 51 allowed configurations, some of which already appear in table 12. Consider first the case where all characters vanish, leading to the generic configuration $(I_4, 8I_1)$, but with a $\mathbb{Z}_4$ symmetry on the $U$-plane. In fact, tuning to an odd looking value, $\chi_2 = 37 + 24\sqrt{3}$, we find that the $U$-plane is $\mathbb{Z}_8$ symmetric instead.

As before, we can also list all the $D_{S^1}E_5$ configurations for which the monodromy group is a congruence subgroup:

$$
\begin{array}{|c||c|c|c|c|}
\hline
\{F_v\} & \mathrm{rk}(\Phi) & \Phi_{\mathrm{tor}} & \mathfrak{g}_F & \Gamma \in \mathrm{PSL}(2,\mathbb{Z}) \\
\hline
I_4, I_1^*, I_1 & 0 & \mathbb{Z}_4 & D_5 & \Gamma^0(4) \\
\hline
2I_4, 2I_2 & 0 & \mathbb{Z}_4 \times \mathbb{Z}_2 & A_3 \oplus 2A_1 & \Gamma^0(4) \cap \Gamma(2) \\
\hline
I_4, I_0^*, 2I_1 & 1 & \mathbb{Z}_2 & D_4 & \Gamma^0(4) \\
\hline
2I_4, 2II & 2 & - & A_3 & 4D^0 \\
\hline
I_4, 2III, I_2 & 2 & \mathbb{Z}_2 & 3A_1 & 4C^0 \\
\hline
I_4, 2III, II & 3 & - & 2A_1 & 4A^0 \\
\hline
\end{array}
\tag{493}
$$

where $4D^0$, $4C^0$ and $4A^0$ are congruence subgroups of index 8, 6 and 4, respectively. As pointed out in table 7, there is another rational elliptic surface associated to a configuration of the $U$-plane that is extremal other than the massless one, namely the $(2I_4, 2I_2)$ configuration. This can be obtained by setting $\chi_1 = -2$, $\chi_2 = -3$, $\chi_4 = 8$, with the other characters set to zero. In this case, we find:

$$
U(\tau) = \left( \frac{\eta\left(\frac{\tau}{2}\right)}{\eta(2\tau)} \right)^4 \xrightarrow{T} -\frac{i\eta(\tau)^{12}}{\eta(2\tau)^8 \eta\left(\frac{\tau}{2}\right)^4},
\tag{494}
$$

which is the Hauptmodul for $\Gamma^0(4) \cap \Gamma(2)$ [170]. From the corresponding fundamental domain shown in figure 23, we read off the following BPS states:

$$
I_2 : \ 2\,(1,0;1,0,0), \quad I_4 : \ 4\,(-1,1;0,1,0), \quad I_2 : \ 2\,(1,-2;0,0,1),
\tag{495}
$$

where we also indicated charges under the $\mathbb{Z}_2 \times \mathbb{Z}_4 \times \mathbb{Z}_2$ center of the flavour group. The corresponding 3-blocks 5d BPS quiver reads:

$$
\begin{array}{c}
\mathcal{E}_{\gamma_{1,2}=(1,0)} \longrightarrow \mathcal{E}_{\gamma_{3,4,5,6}=(-1,1)} \\[2ex]
\Big\downarrow \qquad \qquad \nearrow \\[2ex]
\mathcal{E}_{\gamma_{7,8}=(1,-2)}
\end{array}
\tag{496}
$$

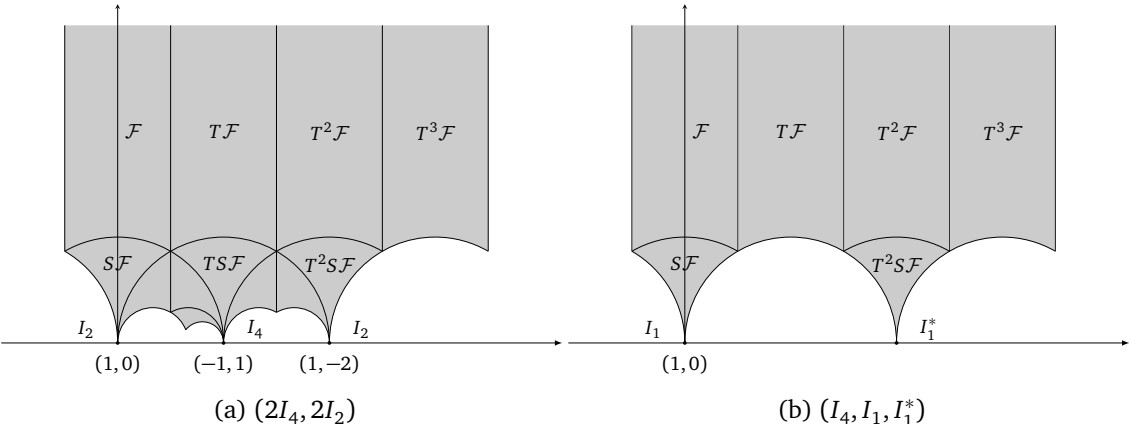

(a) $(2I_4, 2I_2)$          (b) $(I_4, I_1, I_1^*)$

Figure 23: Fundamental domains for configurations on the CB of $D_{S^1}E_5$.

This is a known quiver for local $dP_5$ – in fact, this is also a quiver for a particular degenerate toric limit of local $dP_5$ into an orbifold of the conifold, see *e.g.* 'Model 4d' in [193]. From the fractional-brane basis (495), we can also compute the global form of the flavour group in this extremal configuration. The BPS states are left invariant by a $\mathbb{Z}_4 \times \mathbb{Z}_2$ generated by:

$$g^{\mathscr{E}}_{\mathbb{Z}_4} = \left(\frac{1}{2}, \frac{1}{4}; 1, 1, 0\right), \qquad g^{\mathscr{E}}_{\mathbb{Z}_2} = \left(\frac{1}{2}, \frac{1}{2}; 1, 0, 1\right). \tag{497}$$

From this perspective, we find the flavour group:

$$G_F = SU(2) \times SU(4) \times SU(2) \big/ \mathbb{Z}_4 \times \mathbb{Z}_2, \tag{498}$$

with the non-trivial quotient determined from (497). This also agrees with a direct analysis of the sections in the MW group $\Phi = \mathbb{Z}_4 \times \mathbb{Z}_2$. Note that, unlike other examples for the massless theories, this flavour group is semi-simple but with a non-trivial center, $Z(G_F) = \mathbb{Z}_2$.

    This extremal configuration can be also used to understand the BPS spectrum of the massless theory, as follows. Recall that, setting $M_i = M$ and $\lambda = 1$, we obtain the $(2I_4, I_2, 2I_1)$ configuration. By further setting $M_i = e^{\pm i\pi/2}$, we recover the above $(2I_4, 2I_2)$ configuration. Thus, we can obtain the massless limit by 'breaking' one of the $I_2$ fibers and merging the $I_1$, $I_2$ and $I_4$ fibers into an $I_1^*$ singularity – this can also be understood as an embedding of $A_3 \oplus A_1 \oplus \mathfrak{u}(1)$ inside $D_5$. In terms of the monodromies associated to these massless BPS states, this is:

$$\mathbb{M}_{I_1^*} = \mathbb{M}_{(1,0)}\mathbb{M}^4_{(-1,1)}\mathbb{M}^2_{(1,-2)} = \begin{pmatrix} 1 & -4 \\ 1 & -3 \end{pmatrix} = (T^2 S)(-T)(T^2 S)^{-1}, \tag{499}$$

which agrees with the $I_1^*$ monodromy determined directly from the periods in (468). The fundamental domains for these two configurations are shown in figure 23.

    There are multiple other $D_{S^1}E_5$ configurations for which we can find closed form expressions for the periods. One such example is the $(I_4, 4I_2)$ configuration, which can be obtained for $\chi_2 = -3$, with the other characters vanishing. In terms of the gauge theory parameters, the configuration is obtained by setting $M_1 = M_2 = e^{\pi i/4}$, $M_3 = M_4 = e^{3\pi i/4}$, and thus two of the $I_2$ cusps correspond to two semi-classical 'flavour' cusps. Additionally, we also set $\lambda = 1$, which results into pairing the remaining singularities. In this case, the $U$-plane is $\mathbb{Z}_4$ symmetric and, the monodromy group on the $w = U^4$ plane is $\Gamma_0(2)$. A similar analysis as that for $\lambda = 1$ for $E_1$ leads to the orbifold monodromy:

$$\mathbb{M}_{\text{orb}} = (ST^{\pm\epsilon})S(ST^{\pm\epsilon})^{-1}, \tag{500}$$

with $\epsilon = \pm 1$. Since $\mathbb{M}_{\text{orb}}^4 = \mathbb{I}$, this can then be used to determine the $U$-plane monodromies, leading to the BPS states $\pm(1,0)$ and $\pm(1,-1)$, with two identical states becoming massless at each of the $I_2$ cusps. The corresponding 5d BPS quiver is a 8-nodes, 4-blocks quiver:

$$
\begin{array}{ccc}
\mathcal{E}_{\gamma_{1,2}=(1,0)} & \longrightarrow & \mathcal{E}_{\gamma_{3,4}=(-1,1)} \\
\big\uparrow & & \big\downarrow \\
\mathcal{E}_{\gamma_{7,8}=(1,-1)} & \longleftarrow & \mathcal{E}_{\gamma_{5,6}=(-1,0)}
\end{array}
\tag{501}
$$

This is also a known quiver for $dP_5$ and for its toric limit [177,193]. We can also check that the flavour group acting effectively on the BPS spectrum takes the form $G_F = U(1) \times SU(2)^4/\mathbb{Z}_2^2$, in agreement with the MW group $\Phi \cong \mathbb{Z} \oplus \mathbb{Z}_2^2$.

### 8.2.3 $D_{S^1}E_6$ configurations

For the $D_{S^1}E_6$ theory, there are 77 allowed configurations, three of which are extremal. Let us list the cases corresponding to rational elliptic surfaces associated to congruence subgroups, in the notation of [174]:

$$
\begin{array}{|c||c|c|c|c|}
\hline
\{F_v\} & \text{rk}(\Phi) & \Phi_{\text{tor}} & \mathfrak{g}_F & \Gamma \in \text{PSL}(2,\mathbb{Z}) \\
\hline\hline
I_3, I_1, IV^* & 0 & \mathbb{Z}_3 & E_6 & \Gamma^0(3) \\
\hline
4I_3 & 0 & \mathbb{Z}_3 \times \mathbb{Z}_3 & 3A_2 & \Gamma(3) \\
\hline
I_3, I_6, I_2, I_1 & 0 & \mathbb{Z}_6 & A_5 \oplus A_1 & \Gamma^0(3) \cap \Gamma_0(2) \\
\hline
I_3, I_1^*, II & 1 & - & D_5 & \Gamma^0(3) \\
\hline
I_3, I_0^*, I_1, II & 2 & - & D_4 & \Gamma^0(3) \\
\hline
2I_3, 2III & 2 & - & A_2 \oplus 2A_1 & 3C^0 \\
\hline
I_3, 3III & 3 & - & 3A_1 & \Gamma^3 \\
\hline
\end{array}
\tag{502}
$$

We have already discussed the massless configuration $(I_3, IV^*, I_1)$ in the previous section, so let us discuss the other two extremal cases. The first one is $(4I_3)$, which can be obtained for $\chi_3 = -3$, $\chi_4 = 9$, with the other characters set to zero. Note that this configuration can only appear on the Coulomb branch of the $D_{S^1}E_6$ theory. We find that:

$$
U(\tau) = 3 + \left( \frac{\eta\left(\frac{\tau}{3}\right)}{\eta(3\tau)} \right)^3 ,
\tag{503}
$$

which is the Hauptmodul for $\Gamma(3)$. The map between the $U$-plane and the $\tau$-plane can be found easily, as the above Hauptmodul only changes by an overall factor of $e^{-2\pi i/3}$ upon $T$ transformations. To obtain a quiver description, let us first determine the periods on the $w = U^3$ plane, where the monodromy group is $\Gamma_0(3)$, with the orbifold monodromy:

$$
\mathbb{M}_{\text{orb}} = (ST^k)(ST)^2(ST^k)^{-1} ,
\tag{504}
$$

with $k = -1$ or 2, depending on the base point. Then, by our usual trick, the BPS states associated to the $I_3$ cusps become $(1,0)$, $(-2,1)$ and $(1,-1)$, each of multiplicity 3. The corresponding quiver is variously known as the $T_3$ quiver or as the $\mathbb{C}^3/(\mathbb{Z}_3 \times \mathbb{Z}_3)$ quiver, this orbifold being a degenerate limit of the local $dP_6$:

$$
\begin{array}{ccc}
\mathcal{E}_{\gamma_{1,2,3}=(1,0)} & \longrightarrow & \mathcal{E}_{\gamma_{4,5,6}=(-2,1)} \\
\big\uparrow & \nearrow & \\
\mathcal{E}_{\gamma_{7,8,9}=(1,-1)} & &
\end{array}
\tag{505}
$$

We can analyse the BPS states and the MW torsion section, as in other examples, to conclude that the flavour group of this configuration is $SU(3)^3/\mathbb{Z}_3^2$.

The other extremal configuration is $(I_3, I_6, I_2, I_1)$, which can be obtained by setting the characters $\chi^{E_6}$ to $\{3, -9, -2, -35, -9, 3\}$, for instance. In this case, we have:

$$U(\tau) = 6 + \frac{\eta\left(\frac{\tau}{3}\right)^5 \eta(\tau)}{\eta\left(\frac{2\tau}{3}\right) \eta(2\tau)^5}. \tag{506}$$

This is the Hauptmodul for $\Gamma^0(3) \cap \Gamma_0(2)$, which is a congruence subgroup conjugate to $\Gamma^0(6)$. This is consistent with the fact that this configuration is also the massless $D_{S^1}E_3$ configuration. The $(I_2, I_6, I_1)$ cusps will then be at $\tau = 0, 1$ and $\frac{3}{2}$, respectively, with the BPS states becoming massless at these singularities being:

$$\mathscr{S}: \quad I_2: 2(1,0), \qquad I_6: 6(-1,1), \qquad I_1: (2,-3). \tag{507}$$

The corresponding 5d BPS quiver:

$$\tag{508}$$

is of course another $dP_6$ quiver, which can be obtained from (505) by a series of quiver mutations. Here the D0-brane representation has quiver rank $(1; 1; 2)$. The flavour symmetry group of this configuration is $SU(6) \times SU(2)/\mathbb{Z}_6$, by our usual arguments. Furthermore, we can use the light BPS states of this configuration to understand how the $IV^*$ singularity appears in the massless limit, similarly to the previous examples. For instance, one can fuse the $I_6$ with two other mutually non-local particles, to obtain the monodromy:

$$\mathbb{M}_{(1,0)} \mathbb{M}_{(-1,1)}^6 \mathbb{M}_{(2,-3)} = T^2 (ST)^2 T^{-2}, \tag{509}$$

which is precisely the one around the $IV^*$ singularity found from the periods in (482). This can be understood as an embedding of $A_5 \oplus 2\mathfrak{u}(1)$ inside $E_6$. Incidentally, this gives a first principle derivation of a BPS quiver for the four-dimensional $E_6$ MN theory (although not of the superpotential) [86] – according to this particular realisation of the type-$IV^*$ singularity, the BPS quiver of the 4d MN theory can be simply obtained from (508) by removing the node corresponding to the dyon $\gamma_1 = (1,0)$.

Another $D_{S^1}E_6$ configuration that is modular is the $(I_3, 3III)$ configuration, obtained for $\chi_3 = 78$, $\chi_4 = -1935$ and $\chi_{i\neq 3,4} = 0$ for the other characters. In this case the $J$-invariant is simply given by:

$$j(U) = U^3, \tag{510}$$

and thus $U^3$ is a modular function for $PSL(2, \mathbb{Z})$ itself. As a result, $U(\tau)$ will be a modular function for $\Gamma^3$, which is the subgroup generated by the cubes of the elements of $PSL(2, \mathbb{Z})$. Let us finally mention the configuration $(I_3, 6I_1, III)$ which is obtained for $\chi_i = 0$. In this case the $U$-plane is $\mathbb{Z}_3$ symmetric and the periods can be determined explicitly on the $w = U^3$ plane.

### 8.2.4 $D_{S^1}E_7$ configurations

For the $D_{S^1}E_7$ theory, Persson's classification contains 140 allowed configurations, five of which are extremal. Based on [174], the following cases are modular for a congruence subgroup:

| $\{F_v\}$ | rk($\Phi$) | $\Phi_{\text{tor}}$ | $\mathfrak{g}_F$ | $\Gamma \in \text{PSL}(2,\mathbb{Z})$ |
|---|---|---|---|---|
| $I_2, I_1, III^*$ | 0 | $\mathbb{Z}_2$ | $E_7$ | $\Gamma^0(2)$ |
| $2I_2, I_2^*$ | 0 | $\mathbb{Z}_2 \times \mathbb{Z}_2$ | $D_6 \oplus A_1$ | $\Gamma(2)$ |
| $I_2, I_8, 2I_1$ | 0 | $\mathbb{Z}_4$ | $A_7$ | $\Gamma^0(2) \cap \Gamma_0(4)$ |
| $I_2, I_6, I_3, I_1$ | 0 | $\mathbb{Z}_6$ | $A_5 \oplus A_2$ | $\Gamma^0(2) \cap \Gamma_0(3)$ |
| $2I_2, 2I_4$ | 0 | $\mathbb{Z}_4 \times \mathbb{Z}_2$ | $2A_3 \oplus A_1$ | $\Gamma_0(4) \cap \Gamma(2)$ |
| $I_2, I_1^*, III$ | 1 | $\mathbb{Z}_2$ | $D_5 \oplus A_1$ | $\Gamma^0(2)$ |
| $I_2, IV^*, II$ | 1 | $-$ | $E_6$ | $\Gamma^2$ |
| $3I_2, I_0^*$ | 1 | $\mathbb{Z}_2 \times \mathbb{Z}_2$ | $D_4 \oplus 2A_1$ | $\Gamma(2)$ |
| $I_2, I_0^*, III, I_1$ | 2 | $\mathbb{Z}_2$ | $D_4 \oplus A_1$ | $\Gamma^0(2)$ |
| $I_2, I_4, 2III$ | 2 | $\mathbb{Z}_2$ | $A_3 \oplus 2A_1$ | $4C^0$ |
| $I_2, I_6, 2II$ | 2 | $-$ | $A_5$ | $6C^0$ |
| $I_2, I_0^*, 2II$ | 3 | $-$ | $D_4$ | $\Gamma^2$ |

(511)

The first configuration is the massless configuration, which we have already discussed in the previous subsection. Additionally, this is related to the $(I_2, I_1^*, III)$ configuration by a quadratic twist, and thus the two have the same $J$-invariant. The $(2I_2, I_2^*)$ configuration is also the configuration of the massless 4d $SU(2)$ $N_f = 2$ theory, discussed in section 4. This can be achieved for $\chi^{E_7} = \{5, -59, -16, -330, -144, 3, 8\}$, leading to:

$$U(\tau) = 12 + \left( \frac{\eta\left(\frac{\tau}{2}\right)}{\eta(2\tau)} \right)^8 , \tag{512}$$

which is the Hauptmodul for $\Gamma(2)$. Note that this subgroup is conjugate to $\Gamma^0(4)$. Let us also discuss the $(2I_2, 2I_4)$ configuration, which also appeared on the CB of the $D_{S^1}E_5$ theory. Here, this can be obtained for instance for $\chi_1 = -3$, $\chi_2 = 5$, $\chi_4 = -10$ and $\chi_6 = 3$, with the other characters set to zero. We find that:

$$U(\tau) = \frac{\eta(2\tau)^{12}}{\eta(4\tau)^8 \eta(\tau)^4} , \tag{513}$$

which is the Hauptmodul for $\Gamma_0(4) \cap \Gamma(2)$, as outlined in [170]. Using the properties of the $\eta$-function, we find that the $(I_4, I_4, I_2)$ cusps correspond to $\tau = 0, 1$ and $\frac{1}{2}$, respectively, leading to the BPS states:

$$\mathscr{S} : I_4 : 4(1,0), \qquad I_2 : 2(-2,1), \qquad I_4 : 4(1,-1). \tag{514}$$

This gives a 3-blocks BPS quiver:

$$\begin{array}{ccc} \mathcal{E}_{\gamma_{1,2,3,4}=(1,0)} & \longrightarrow & \mathcal{E}_{\gamma_{5,6}=(-2,1)} \\ \uparrow & \nearrow & \\ \mathcal{E}_{\gamma_{7,8,9,10}=(1,-1)} & & \end{array} \tag{515}$$

Table 13: Configurations on the Coulomb branch of the $D_{S^1}E_8$ theory that are modular with respect to a congruence subgroup. The flavour algebra excludes the abelian $\mathfrak{u}(1)$ factors.

| $\{F_\nu\}$ | $\mathrm{rk}(\Phi)$ | $\Phi_{\mathrm{tor}}$ | $\mathfrak{g}_F$ | $\Gamma \in \mathrm{PSL}(2,\mathbb{Z})$ |
|---|---|---|---|---|
| $I_1, I_2, III^*$ | 0 | $\mathbb{Z}_2$ | $E_7 \oplus A_1$ | $\Gamma_0(2)$ |
| $I_1, I_3, IV^*$ | 0 | $\mathbb{Z}_3$ | $E_6 \oplus A_2$ | $\Gamma_0(3)$ |
| $2I_1, I_4^*$ | 0 | $\mathbb{Z}_2$ | $D_8$ | $\Gamma_0(4)$ |
| $I_1, I_4, I_1^*$ | 0 | $\mathbb{Z}_4$ | $D_5 \oplus A_3$ | $\Gamma_0(4)$ |
| $2I_1, 2I_5$ | 0 | $\mathbb{Z}_5$ | $A_4 \oplus A_4$ | $\Gamma_1(5)$ |
| $I_1, I_6, I_3, I_2$ | 0 | $\mathbb{Z}_6$ | $A_5 \oplus A_2 \oplus A_1$ | $\Gamma_0(6)$ |
| $2I_1, I_8, I_2$ | 0 | $\mathbb{Z}_4$ | $A_7 \oplus A_1$ | $\Gamma_0(8)$ |
| $3I_1, I_9$ | 0 | $\mathbb{Z}_3$ | $A_8$ | $\Gamma_0(9)$ |
| $I_1, III^*, II$ | 1 | — | $E_7$ | $PLS(2,\mathbb{Z})$ |
| $I_1, III, IV^*$ | 1 | — | $E_6 \oplus A_1$ | $PLS(2,\mathbb{Z})$ |
| $I_1, I_2^*, III$ | 1 | $\mathbb{Z}_2$ | $D_6 \oplus A_1$ | $\Gamma_0(2)$ |
| $I_1, I_3^*, II$ | 1 | — | $D_7$ | $\Gamma_0(3)$ |
| $2I_1, I_0^*, I_4$ | 1 | $\mathbb{Z}_2$ | $D_4 \oplus 2A_1$ | $\Gamma_0(4)$ |
| $I_1, I_0^*, III, I_2$ | 2 | $\mathbb{Z}_2$ | $D_4 \oplus 2A_1$ | $\Gamma_0(2)$ |
| $I_1, I_0^*, I_3, II$ | 2 | — | $D_4 \oplus A_2$ | $\Gamma_0(3)$ |
| $I_1, I_5, 2III$ | 2 | — | $A_4 \oplus 2A_1$ | $\Gamma_0(5)$ |
| $I_1, I_7, 2II$ | 2 | — | $A_6$ | $\Gamma_0(7)$ |
| $I_1, I_0^*, III, II$ | 3 | — | $D_4 \oplus A_1$ | $PSL(2,\mathbb{Z})$ |

This is a known $dP_7$ quiver [177], with the D0-brane representation having rank $(1;2;1)$. The details of the flavour symmetry group can be worked out as in other examples. Starting from this configuration, the massless configuration $(I_2, I_1, III^*)$ can be obtained by the recombination:

$$\mathbb{M}_{(1,0)}^3 \mathbb{M}_{(-2,1)}^2 \mathbb{M}_{(1,-1)}^4 = TST^{-1}, \tag{516}$$

which indeed is the monodromy around the type-$III^*$ singularity as determined in (482). This can be viewed as an embedding of $A_2 \oplus A_1 \oplus A_3$ inside $E_7$. Note that this also gives us a BPS quiver for the 4d $E_7$ MN theory, simply by removing node $\gamma_1 = (1, 0)$ in (515).

Let us also briefly comment on the $(I_2, IV^*, II)$ configuration. It turns out that in this case the $J$ invariant is a degree 2 polynomial in $U$, and thus the monodromy group is $\Gamma^2$, *i.e.* the subgroup containing the squares of the elements of $\mathrm{PSL}(2,\mathbb{Z})$. A fundamental domain can be easily drawn for this subgroup, with the coset representatives $\{\mathbb{I}, T\}$. In this case, the order-2 elliptic points will be at $\tau = e^{\frac{\pi i}{3}}$ and $e^{\frac{2\pi i}{3}}$.

### 8.2.5 $D_{S^1}E_8$ configurations

Finally, for the $D_{S^1}E_8$ theory, there are 227 allowed configurations in Persson's classification, nine of which are extremal. Based on [174], the configurations that are modular for a congruence subgroup are shown in table 13. Many of these configurations have already appeared on the Coulomb branch of the other $D_{S^1}E_n$ theories, some of them being also analysed in [156].

Let us consider one of the extremal configurations that could be useful in visualising how the type-$II^*$ singularity appears in the massless limit. For simplicity, take again the $(I_1, I_6, I_3, I_2)$ configuration[39], where now the monodromy group is $\Gamma_0(6)$, with the cusps $(I_6, I_2, I_3)$ at $\tau = 0$, $\frac{1}{3}$ and $\frac{1}{2}$, respectively. The associated BPS states are:

$$\mathscr{S}: \quad I_6: 6(1,0), \quad I_2: 2(-3,1), \quad I_3: 3(2,-1), \tag{517}$$

which gives the 5d BPS quiver:

$$
\begin{array}{ccc}
\mathcal{E}_{\gamma_{1,2,3,4,5,6}=(1,0)} & \longrightarrow & \mathcal{E}_{\gamma_{7,8}=(-3,1)} \\
\downarrow & \nearrow & \\
\mathcal{E}_{\gamma_{9,10,11}=(2,-1)} & &
\end{array}
\tag{518}
$$

This is a correct 3-blocks quiver for $dP_8$ [177], with the D0-brane representation having rank $(1; 3; 2)$. Starting from this configuration, the $II^*$ monodromy in (489) can be realised as:

$$\mathbb{M}_{(1,0)}^5 \mathbb{M}_{(-3,1)}^2 \mathbb{M}_{(2,-1)}^3 = T(ST)T^{-1}. \tag{519}$$

As in the $E_6$ and $E_7$ examples, this construction also gives us a derivation of a BPS quiver for the 4d $E_8$ MN theory, which is obtained from (518) by removing the dyon $\gamma_1 = (1,0)$. We hope to return to this important point in future work.

## 9 Gravitational couplings on the $U$-plane for the $D_{S^1}E_n$ theories

One can consider any 4d $\mathcal{N} = 2$ field theory on a compact 4-manifold, $\mathcal{M}_4$, with the topological twist [197]. The low-energy effective field theory then includes effective gravitational couplings of the form [37, 38, 108]:

$$S_{\text{grav}} = \frac{i}{16\pi} \int_{\mathcal{M}_4} \text{Tr}(R \wedge *R)\,\mathcal{A}(a) + \frac{i}{12\pi} \int_{\mathcal{M}_4} \text{Tr}(R \wedge R)\,\mathcal{B}(a), \tag{520}$$

where $a$ is the low energy photon and $R$ the Riemann curvature. On the Coulomb branch, the field $a$ is constant and we simply have:

$$S_{\text{grav}} = 2\pi i \left( e(\mathcal{M}_4)\mathcal{A}(a) + \sigma(\mathcal{M}_4)\mathcal{B}(a) \right), \tag{521}$$

with $e$ and $\sigma$ the topological Euler characteristic and the signature of $\mathcal{M}_4$, respectively. In general, the topological twist data must also include a choice of spin$^c$ line bundle, which affects the path integral in a subtle way – see *e.g.* [39, 198, 199]. Here, we focus on the type-$A$ and type-$B$ gravitational couplings on the $U$-plane, defined as:

$$e^{-S_{\text{grav}}} = A^e B^\sigma, \quad A(U) = e^{-2\pi i \mathcal{A}}, \quad B(U) = e^{-2\pi i \mathcal{B}}, \tag{522}$$

using the 'mirror map' $a = a(U)$. A general infrared prescription for these couplings was given in [37, 38, 108] based on the S-duality of the low-energy theory. Given any rank-one SW geometry, in particular, we should have:

$$A(U) = \alpha \left( \frac{dU}{da} \right)^{\frac{1}{2}}, \quad B(U) = \beta \left( \Delta^{\text{phys}} \right)^{\frac{1}{8}}, \tag{523}$$

---

[39]This can be obtained for $\chi^{E_8} = \{3, 15, -8, 19, -10, -1, 5, -3\}$, for instance.

with $\Delta^{\text{phys}}$ the so-called 'physical discriminant', and $\alpha$, $\beta$ some prefactors to be determined. In our examples, the physical discriminant will be equal to the geometric discriminant $\Delta$ of the $D_{S^1}E_n$ curves written in Weierstrass normal form, up to an overall factor.

The gravitational couplings (522) can also be extracted from the microscopic calculation provided by the Nekrasov partition function on the $\Omega$-background [109–111, 113, 200]. In this section, we consider the $\mathbb{R}^4 \times S^1$ Nekrasov partition functions for the $D_{S^1}E_n$ theories with $n \leq 3$ in order to extract the gravitational couplings, and we match that result to the infrared expectation (523). We will focus on the $E_1$ and $E_3$ theories in the following.[40] We find perfect agreement between the UV and IR prescriptions up to three-instantons. This computation can be seen as a 5d generalisation of recent computations in [36], where the same IR/UV matching was investigated for rank-one 4d theories; our results reproduce theirs in the 4d gauge-theory limit. We will discuss some other interesting aspects of the gravitational couplings in [40].

### 9.1 Instanton partition functions and gravitational couplings

The $\mathbb{R}^4 \times S^1$ ($K$-theoretic) Nekrasov partition function for $U(N)$ gauge theories is computed via a simple prescription involving Young diagrams [109, 110, 113, 200], which can be adapted to $SU(N)$ gauge theories by carefully decoupling the additional $U(1)$ contribution. For 4d gauge theories, this $U(1)$ factor was first discussed in the context of the AGT correspondence. For the 5d 'toric' theories, the partition function can be determined from the (refined) topological vertex formalism [201,202], which allows for the identification of the correct $SU(N)$ instanton counting prescription. We review the relevant formulas and set our conventions in appendix D.

Consider the Coulomb branch of a 5d SCFT on the $\Omega$-background. As described in section 2, such theories are engineered in M-theory on:

$$\mathbb{C}^2_{\tau_1, \tau_2} \times S^1 \times \mathbf{X}_{E_n}, \qquad (524)$$

where $\mathbf{X}_{E_n}$ is a canonical singularity that admits a crepant resolution. We consider the local del Pezzo geometries $\widetilde{\mathbf{X}} = \text{Tot}(\mathcal{K} \to \mathcal{B}_4)$ engineering the $D_{S^1}E_n$ theories. Let $\tau_i = \beta \epsilon_i$, $i = 1, 2$, be the dimensionless $\Omega$-background parameters, and let us introduce the notation:

$$q = e^{2\pi i \tau_1}, \qquad t = e^{-2\pi i \tau_2}. \qquad (525)$$

Recall also that, for the complexified Kähler parameters associated to curves $\mathcal{C} \in H_2(\mathcal{B}_4, \mathbb{Z})$, we introduced the single-valued parameters:

$$Q_{\mathcal{C}} = e^{2\pi i t_{\mathcal{C}}}, \qquad t_{\mathcal{C}} = \int_{\mathcal{C}} (B + iJ). \qquad (526)$$

The instanton counting prescription for the partition function is valid in the gauge theory phase of the $D_{S^1}E_n$ theories, where it leads to a power series in the instanton counting parameter:

$$\lambda = \frac{Q_{\mathcal{C}_b}}{Q_{\mathcal{C}_f}}, \qquad (527)$$

in our conventions, for $\mathcal{C}_{f,b}$ the fiber and base curves of the Fano surface $\mathcal{B}_4$. In the non-equivariant limit, $\tau_{1,2} \to 0$, the Nekrasov partition function has the asymptotic expansion:

$$-\log(Z_{\mathbb{C}^2 \times S^1}(Q, \tau_1, \tau_2)) = \frac{2\pi i}{\tau_1 \tau_2} \left( \mathcal{F} + (\tau_1 + \tau_2)\mathcal{H} + \tau_1 \tau_2 \mathcal{A} + \frac{\tau_1^2 + \tau_2^2}{3} \mathcal{B} \right) + \mathcal{O}(\tau), \quad (528)$$

---

[40]The other two toric theories with a gauge theory interpretation, $\widetilde{E}_1$ and $\widetilde{E}_2$, can be treated similarly, either intrinsically or as limits of the $E_3$ theory, but there are a few subtleties related to the 5d parity anomaly, which we hope to discuss elsewhere.

from which one can extract the prepotential $\mathcal{F}$ and the gravitational corrections $\mathcal{A}$ and $\mathcal{B}$ [109]. Closed-form expressions for the latter two can be determined from holomorphic anomaly equations – see *e.g.* [203, 204]. Here, we will determine $\mathcal{A}$ and $\mathcal{B}$ from the Seiberg-Witten curve and from (528), leading to consistent results. In four dimensions, this same computation was recently discussed in [36]. The 4d limit of our computation agrees with [36], once we fix our conventions appropriately. We also note that the term $\mathcal{H}$ in (528) is expected to vanish on general grounds, because there are no gravitational terms that it could correspond to on the standard $\Omega$-background, where only the $SU(2)_R$ gauge field is turned on – see *e.g.* [205].

Recall that the gauge-theory phase of the 5d theory is obtained in a special limit where the fiber curve $\mathcal{C}_f$ shrinks to zero while the base curve $\mathcal{C}_b$ remains large, in which case $\lambda$ is very small. A more natural limit, however, is the large volume limit in the local Calabi-Yau $\widetilde{\mathbf{X}}$, in which all effective curves are on equal footing, with their corresponding parameters $Q_{\mathcal{C}} \to 0$. In that limit, the $\mathbb{R}^4 \times S^1$ partition function can be written as a product over contributions of some (generally infinite) number of five-dimensional massive BPS particles, with masses set by the Kähler parameters $Q_{\mathcal{C}}$, and with spins given by the corresponding representations of the little group $SO(4) = SU(2)_L \times SU(2)_R$. That partition function captures the (refined) Gopakumar-Vafa invariants [206] of $\widetilde{\mathbf{X}}$, and can be computed in terms of the refined topological string partition function [202]. For our present purposes, however, it will suffice to compute the $\mathbb{R}^4 \times S^1$ partition function in the 'gauge-theory phase', order by order in the instanton-counting parameter (527), and to compare with the quantities obtained from the Seiberg-Witten geometry in that same limit.

### 9.1.1 $Z_{\mathbb{R}^4 \times S^1}$ partition function of the $E_1$ theory

Let us first review the computation of the Nekrasov partition function for the 5d pure $SU(2)$ gauge theory, which is the gauge-theory phase of the $E_1$ SCFT. This can be obtained directly from the $U(2)$ gauge-theory partition function by imposing the traceless condition on the VEV of the 5d scalars. Using the conventions summarised in appendix D, we thus write the partition function as:

$$Z_{\mathbb{C}^2 \times S^1}(Q, \tau_1, \tau_2) = Z_{\text{vector}}^{\text{class}}(Q, \tau_1, \tau_2) Z_{\text{vector}}^{\text{pert}}(Q, \tau_1, \tau_2) Z_{\text{vector}}^{\text{inst}}(Q, \tau_1, \tau_2), \tag{529}$$

where $Z^{\text{class}}$, $Z^{\text{pert}}$, $Z^{\text{inst}}$ are the classical, perturbative and non-perturbative contributions, respectively. Here we introduced $Q \equiv e^{4\pi i a}$, with $a$ the dimensionless scalar defined in (18). Note that the gravitational corrections do not receive any classical contributions. As argued in (23), the perturbative contribution to the prepotential reads [19]:

$$\mathcal{F}^{\text{pert}} = \frac{2}{(2\pi i)^3}\text{Li}_3(Q) + \frac{2}{3}a^3. \tag{530}$$

The perturbative contribution to the gravitational couplings is discussed briefly in appendix D.1. For the pure $SU(2)$ theory, this reduces to the following contributions to the gravitational corrections:

$$\mathcal{A}^{\text{pert}} = \mathcal{B}^{\text{pert}} = -\frac{1}{2\pi i}\left(\frac{1}{2}\log(1-Q) - \frac{1}{4}\log Q\right). \tag{531}$$

The complete expression for the prepotential is of the form:

$$\mathcal{F} = \mathcal{F}^{\text{pert}} + \frac{1}{(2\pi i)^3}\sum_{k=1}^{\infty} \mathfrak{q}_{E_1}^k \mathcal{F}_k, \tag{532}$$

where $\mathfrak{q}$ is the instanton counting parameter and $\mathcal{F}_k$ are the $k$-instanton contributions. From the prescription outlined in appendix D, we find the following instanton contributions $\mathcal{F}_k$ to

the prepotential:

$$k = 1: \quad \frac{2Q}{(1-Q)^2} = 2Q + 4Q^2 + 6Q^3 + 8Q^4 + 10Q^5 + 12Q^6 + \mathcal{O}(Q^7),$$

$$k = 2: \quad \frac{Q^2(1 + 18Q + Q^2)}{4(1-Q)^6} = \frac{Q^2}{4} + 6Q^3 + \frac{65Q^4}{2} + 110Q^5 + \frac{1155}{4}Q^6 + \mathcal{O}(Q^7), \quad (533)$$

$$k = 3: \quad \frac{2Q^3\left(1 + 98Q + 450Q^2 + 98Q^3 + Q^4\right)}{27(1-Q)^{10}} = \frac{2}{27}Q^3 + 8Q^4 + 110Q^5 + \mathcal{O}(Q^6),$$

where the series expansion in $Q$ is done in order to relate this result to the large volume computations from the Seiberg-Witten curve. Additionally, we find that the instanton corrections to the order $1/\tau$ term in (528) vanishes, as expected. For the gravitational coupling $\mathcal{A}$, the first instanton contributions are:

$$
\begin{aligned}
2\pi i\,\mathcal{A}^{\text{inst}} = {} & \mathfrak{q}_{E_1}\left(\frac{Q}{2} + 5Q^2 + \frac{35Q^3}{2} + 42Q^4 + \frac{165Q^5}{2} + \mathcal{O}(Q^6)\right) \\
& + \mathfrak{q}_{E_1}^2\left(\frac{Q^2}{4} + \frac{35Q^3}{2} + \frac{355Q^4}{2} + \frac{1919Q^5}{2} + \mathcal{O}(Q^6)\right) \\
& + \mathfrak{q}_{E_1}^3\left(\frac{Q^3}{6} + 42Q^4 + \frac{1919}{2}Q^5 + \mathcal{O}(Q^6)\right) + \mathcal{O}\left(\mathfrak{q}_{E_1}^4\right).
\end{aligned}
\quad (534)
$$

Similarly, the first few instanton contributions to $\mathcal{B}$ read:

$$
\begin{aligned}
2\pi i\,\mathcal{B}^{\text{inst}} = {} & \mathfrak{q}_{E_1}\left(\frac{Q}{2} + 7Q^2 + \frac{51Q^3}{2} + 62Q^4 + \frac{145Q^5}{2} + \mathcal{O}(Q^6)\right) \\
& + \mathfrak{q}_{E_1}^2\left(\frac{Q^2}{4} + \frac{51Q^3}{2} + \frac{551Q^4}{2} + \frac{3055Q^5}{2} + \mathcal{O}(Q^6)\right) \\
& + \mathfrak{q}_{E_1}^3\left(\frac{Q^3}{6} + 62Q^4 + \frac{3055Q^5}{2} + \mathcal{O}(Q^6)\right) + \mathcal{O}(\mathfrak{q}_{E_1}^4).
\end{aligned}
\quad (535)
$$

### 9.1.2 $Z_{\mathbb{R}^4 \times S^1}$ partiton function of the $E_3$ theory

As another example, let us consider the partition function of the $E_3$ theory in the $\Omega$-background. The corresponding gauge-theory phase is the 5d $SU(2)$ gauge theory with $N_f = 2$. The partition function that corresponds to the $E_3$ Seiberg-Witten geometry (91) is given by [207]:

$$
\begin{aligned}
Z_{E_3}^{\text{inst}}(a, \mu_1, \mu_2, \tau_1, \tau_2) = {} & \\
& \sum_Y \left(\frac{t}{q}\mathfrak{q}_{E_3}\right)^{|Y|} Z_Y^{\text{vector}}(a, \tau_1, \tau_2) Z_Y^{\text{fund}}(a, \mu_1, \tau_1, \tau_2) Z_Y^{\text{a-fund}}(a, \mu_2, \tau_1, \tau_2),
\end{aligned}
\quad (536)
$$

where $\mu_{1,2}$ are the complex masses for the two flavours, with the various factors defined in appendix D. In order to simplify our expressions, let us define:

$$s = -e^{-2\pi i \mu_1} - e^{-2\pi i \mu_2}, \qquad p = e^{-2\pi i(\mu_1 + \mu_2)}. \quad (537)$$

As we will see below, these parameters can be matched to the parameters $s, p$ introduced to describe the $E_3$ curve in (407). As was the case for the $E_1$ theory, the instanton partition function will be a series in the instanton counting parameter $\mathfrak{q}_{E_3}$. We consider a further limit $Q \to 0$, in order to compare to the large volume limit. The first non-perturbative corrections

to the prepotential then read:

$$(2\pi i)^3 \mathcal{F}_{E_3}^{\text{inst}} = \mathfrak{q}_{E_3} \left( sQ^{\frac{1}{2}} + 2(1+p)Q + 3sQ^{\frac{3}{2}} + 4(1+p)Q^2 + 5sQ^{\frac{5}{2}} + 6(1+p)Q^3 \right) +$$

$$+ \mathfrak{q}_{E_3}^2 \left( \frac{2p - s^2}{8}Q + \frac{1 + 16p + p^2}{4}Q^2 + 5s(1+p)Q^{\frac{5}{2}} + \frac{48(1+p^2) + 198p + 45s^2}{8}Q^3 \right) \tag{538}$$

$$+ \mathfrak{q}_{E_3}^3 \left( \frac{s(s^2 - 3p)}{27}Q^{\frac{5}{2}} + \frac{2(1 + 81p + 81p^2 + p^3)}{27}Q^3 \right) + \mathcal{O}(\mathfrak{q}_{E_3}^4),$$

where we suppress the terms of order $\mathcal{O}(Q^{\frac{7}{2}})$. As was the case for the $E_1$ partition function, we again find that the order $\mathcal{O}(\tau^{-1})$ terms vanish. For $\mathcal{A}$, we find:

$$(2\pi i)\mathcal{A}^{\text{inst}} = \mathfrak{q}_{E_3} \left( \frac{1+p}{2}Q + 2sQ^{\frac{3}{2}} + 5(1+p)Q^2 + 10sQ^{\frac{5}{2}} + \frac{35(1+p)}{2}Q^3 \right)$$

$$+ \mathfrak{q}_{E_3}^2 \left( \frac{1 + 20p + p^2}{4}Q^2 + 10s(1+p)Q^{\frac{5}{2}} + \frac{35(1+p^2) + 154p + 33s}{2}Q^3 \right) \tag{539}$$

$$+ \mathfrak{q}_{E_3}^3 \frac{1}{6} \left( 1 + 105p + 105p^2 + p^3 \right)Q^3 + \mathcal{O}(\mathfrak{q}_{E_3}^4),$$

and similarly for $\mathcal{B}$, we have:

$$(2\pi i)\mathcal{B}^{\text{inst}} = \mathfrak{q}_{E_3} \left( -\frac{1}{8}sQ^{\frac{1}{2}} + \frac{1+p}{2}Q + \frac{21s}{8}Q^{\frac{3}{2}} + 7(1+p)Q^2 + \frac{115s}{8}Q^{\frac{5}{2}} + \frac{51(1+p)}{2}Q^3 \right)$$

$$+ \mathfrak{q}_{E_3}^2 \left( \frac{s^2 - 2p}{16}Q + \frac{1 + 28p + p^2}{4}Q^2 + \frac{115s(1+p)}{8}Q^{\frac{5}{2}} + \frac{408(1+p^2) + 9(218p + 43s^2)}{16}Q^3 \right)$$

$$+ \mathfrak{q}_{E_3}^3 \left( \frac{s(3p - s^2)}{24}Q^{\frac{3}{2}} + \frac{1 + 153p + 153p^2 + p^3}{6}Q^3 \right) + \mathcal{O}(\mathfrak{q}_{E_3}^4),$$

$$\tag{540}$$

where we again suppress the terms of order $\mathcal{O}(Q^{\frac{7}{2}})$.

## 9.2 Seiberg-Witten geometry computations of $\mathcal{F}$, $\mathcal{A}$ and $\mathcal{B}$

We now turn our attention to the toric mirror curves (91). We first show how to compute the quantities of interest from the $E_1$ curve, and then also present the explicit results for the $E_3$ theory. The expressions for the other toric theories follow by flavour decoupling.

### 9.2.1 Instanton expansion from the $D_{S^1}E_1$ curve

Consider first the $E_1$ curve (99), and introduce the dimensionless parameters associated to the complexified Kähler parameters of the IIA geometry:

$$Q_f \equiv Q = e^{2\pi i t_f} = e^{4\pi i a}, \qquad Q_b = e^{2\pi i t_b} = e^{2\pi i(2a + \mu_0)}, \tag{541}$$

where $t_{f,b}$ are the periods corresponding to D2-branes wrapping the curves $\mathcal{C}_f$ and $\mathcal{C}_b$, respectively, with $Q_b = \lambda Q_f$. The D4-brane period is given by $\Pi_{\text{D4}} = a_D$, and the prepotential can be written as:

$$\mathcal{F} = \frac{1}{2} \int \Pi_{\text{D4}} \, dt_f = \frac{1}{4} \int dt_f \int dt_f \, \tau, \qquad \tau = \frac{da_D}{da} = 2\frac{d\Pi_{\text{D4}}}{dt_f}. \tag{542}$$

Let us first consider the 'classical' contribution to the prepotential. This is obtained from the large-volume analysis of the D-brane periods, leading to:

$$\mathcal{F}^{\text{class}} = \frac{4}{3}a^3 + \frac{1}{2\pi i}\log(\lambda)a^2 + \frac{1}{6}a, \tag{543}$$

as in (72). Note that, in the strict 5d limit, the instanton corrections are suppressed, and thus we reproduce the real prepotential for the $E_1$ theory from [13,19]:

$$\mathcal{F}^{(5d)} = \lim_{\beta \to \infty} \frac{i}{\beta^3} \mathcal{F}^{\text{class}} = \frac{4}{3}\sigma^3 + m_0 \sigma^2, \tag{544}$$

for $\lambda \to e^{-2\pi\beta m_0}$, with $\sigma$ the real scalar of the 5d $\mathcal{N} = 1$ vector multiplet and $m_0$ the inverse gauge coupling. The Picard-Fuchs equation (54) for the periods can be solved in the large volume limit for generic values of the parameters, using Frobenius' method. However, this is equivalent to the 'universal' PF equation (56) for the geometric periods, with one of the solutions given by (58). Choosing the appropriate normalization constant, one can match this period with the $a$ period. For the $E_1$ theory, this is:

$$2\pi i \frac{d}{dU} a(U, \lambda) = -\frac{1}{U} - \frac{2(1+\lambda)}{U^3} - \frac{6(1+4\lambda+\lambda^2)}{U^5} - \frac{20(1+9\lambda+9\lambda^2+1)}{U^7} + \mathcal{O}\left(\frac{1}{U^9}\right). \tag{545}$$

It is then straightforward to invert this series expansion to find:

$$U(Q, \lambda) = Q^{-\frac{1}{2}} + (1+\lambda)Q^{\frac{1}{2}} + 3\lambda Q^{\frac{3}{2}} + 5\lambda(1+\lambda)Q^{\frac{5}{2}} + \mathcal{O}\left(Q^{\frac{7}{2}}\right), \tag{546}$$

where we again used $Q = e^{4\pi i a}$. Finally, combining this with the expression $\tau = \tau(j)$ obtained by inverting (59) we obtain the prepotential:

$$
\begin{aligned}
(2\pi i)^3 \mathcal{F} = (2\pi i)^3 \mathcal{F}_{\text{class}} &+ \left(2Q + \frac{1}{4}Q^2 + \frac{2}{27}Q^3 + \frac{1}{32}Q^4\right) + \lambda\left(2Q + 4Q^2 + 6Q^3 + 8Q^4\right) \\
&+ \lambda^2\left(\frac{1}{4}Q^2 + 6Q^3 + \frac{65}{2}Q^4\right) + \lambda^3\left(\frac{2}{27}Q^3 + 8Q^4\right) + \mathcal{O}(\lambda^4),
\end{aligned} \tag{547}
$$

where terms of order $\mathcal{O}(Q^5)$ are suppressed. We thus immediately notice that this agrees with the instanton counting results (533), upon the identification:

$$\mathfrak{q}_{E_1} = \lambda. \tag{548}$$

This comes as no surprise, since we already observed that the 4d limit of the SW curve involves taking $\lambda \to (2\pi i \beta \Lambda)^4$, which is the instanton counting parameter of the resulting four-dimensional theory. Let us also note that the perturbative part of the above expression reproduces the series expansion of the trilogarithm, and thus agrees with (530). The remaining task is to identify the correct expressions for the gravitational corrections $\mathcal{A}$ and $\mathcal{B}$. Following closely the four-dimensional computation in [36], we first consider the quantity:

$$
\begin{aligned}
-\frac{1}{2}\log\left(-Q\frac{dU}{dQ}\right) = a_0 &+ \lambda\left(\frac{1}{2}Q + 5Q^2 + \frac{35}{2}Q^3 + 42Q^4 + \frac{165}{2}Q^5\right) \\
&+ \lambda^2\left(\frac{1}{4}Q^2 + \frac{35}{2}Q^3 + \frac{355}{2}Q^4 + \frac{1919}{2}Q^5\right) + \lambda^3\left(\frac{1}{6}Q^3 + 42Q^4 + \frac{1919}{2}Q^5\right),
\end{aligned} \tag{549}
$$

up to orders $\mathcal{O}(Q^6)$ and $\mathcal{O}(\lambda^4)$, with $a_0$ encoding the perturbative contribution:

$$a_0 = -\frac{1}{2}\log(1-Q) + \frac{1}{4}\log(Q) + \frac{1}{2}\log(2). \tag{550}$$

The perturbative and non-perturbative corrections are in perfect agreement with (531) and (534), respectively. We then find that:

$$A = \sqrt{2}\left(-\frac{1}{4\pi i}\frac{dU}{da}\right)^{\frac{1}{2}}, \tag{551}$$

with $A$ given in (522). For the $\mathcal{B}$ gravitational coupling, let us first define the 'physical' discriminant as:

$$\Delta^{\text{phys}}(U) = \lambda^{-2}\Delta_{E_1}(U), \tag{552}$$

where $\Delta_{E_1}$ is the discriminant of the $E_1$ curve in (258). Then, we find that:

$$
\begin{aligned}
-\frac{1}{8}\log\left(\Delta^{\text{phys}}\right) = {} & b_0 + \lambda\left(\frac{1}{2}Q + 7Q^2 + \frac{51}{2}Q^3 + 62Q^4 + \frac{245}{2}Q^5\right) \\
& + \lambda^2\left(\frac{1}{4}Q^2 + \frac{51}{2}Q^3 + \frac{551}{2}Q^4 + \frac{3055}{2}Q^5\right) + \lambda^3\left(\frac{1}{6}Q^3 + 62Q^4 + \frac{3055}{2}Q^5\right),
\end{aligned}
\tag{553}
$$

with the terms of orders $\mathcal{O}(Q^6)$ and $\mathcal{O}(\lambda^4)$ suppressed, and with:

$$b_0 = -\frac{1}{2}\log(1-Q) + \frac{1}{4}\log(Q). \tag{554}$$

This matches the contributions to the $\mathcal{B}$ gravitational correction in (531) and (535) and, as a result, we find:

$$B = \left(\Delta^{\text{phys}}\right)^{\frac{1}{8}}. \tag{555}$$

It is also instructive to consider the 4d limit of these expressions. First, the perturbative part (531) of these quantities becomes:

$$-\frac{1}{2}\log(1-Q) \approx -\frac{1}{2}\log\left(-4\pi i\beta a\right) + \mathcal{O}(\beta) \approx -\frac{1}{2}\log\left(-\frac{2a}{\Lambda}\right) + \mathcal{O}(\beta), \tag{556}$$

where we introduce the dynamical scale $\Lambda$ as $(2\pi i\beta)^{-1}$. The K-theoretic Nekrasov partition function reduces to its 4d counterpart in the 4d limit, by definition, and it is then not difficult to see that the 4d limits of the expressions for $\mathcal{A}$ and $\mathcal{B}$ are in agreement with the expressions given in [36]. Note that, in order to take the 4d limit at each order in the instanton expansion, one needs to use the exact expression for the $k$-instanton correction instead of the above $Q$-series. Let us show this explicitly for the 1-instanton correction. The 1-instanton corrections to the gravitational couplings which reproduce the series (534), (535) are:

$$
\begin{aligned}
\mathcal{A}: \quad & \mathfrak{q}_{E1}\frac{Q(1+6Q+Q^2)}{2(1-Q)^4} \approx \frac{\Lambda^4}{4a^4} + \mathcal{O}(\beta^2), \\
\mathcal{B}: \quad & \mathfrak{q}_{E1}\frac{Q(1+10Q+Q^2)}{2(1-Q)^4} \approx \frac{3\Lambda^4}{8a^4} + \mathcal{O}(\beta^2).
\end{aligned}
\tag{557}
$$

The physical discriminant reduces to $\Delta^{\text{phys}} \approx 16(2\pi i\beta)^4\Delta^{(4d)}$ in the 4d limit, with $\Delta^{(4d)} = (u^2 - 4\Lambda^4)$, and we find:

$$A = \frac{1}{\sqrt{\Lambda}}\left(-\frac{du}{da}\right)^{\frac{1}{2}}, \qquad B = \frac{\sqrt{2}}{\sqrt{\Lambda}}\left(\Delta^{(4d)}\right)^{\frac{1}{8}}, \tag{558}$$

in good agreement with [36].

### 9.2.2 Instanton expansion from the $D_{S^1}E_3$ curve

The same computation can be carried out for the $E_3$ curve (91), with the parameters $s = M_1 + M_2$ and $p = M_1M_2$. For generic values of the mass parameters, we find:

$$U = Q^{-\frac{1}{2}} + (1+\lambda(1+p))Q^{\frac{1}{2}} + 2s\lambda Q + 3\lambda(1+(1+\lambda)p)Q^{\frac{3}{2}} + 4s\lambda(1+(1+\lambda)p)Q^2 + \mathcal{O}\left(Q^{\frac{5}{2}}\right).$$

We should again arrange our expressions in terms of the parameter $\lambda$. Up to the first instanton correction, the prepotential reads:

$$
\begin{aligned}
(2\pi i)^3 \mathcal{F}_{E_3} = {}& \\
\mathcal{F}_0 + \lambda \Big( & sQ^{\frac{1}{2}} + 2(1+p)Q + 3sQ^{\frac{3}{2}} + 4(1+p)Q^2 + 5sQ^{\frac{5}{2}} + 6(1+p)Q^3 + \mathcal{O}(Q^{\frac{7}{2}}) \Big),
\end{aligned}
\tag{559}
$$

which agrees with (538) upon the identifications $M_i = -e^{-2\pi i \mu_i}$, as in (96). This shows that the parameters $s$ and $p$ defined in (537) are identical to the ones introduced above. We checked explicitly that the IR and UV computations of the prepotential agree up to the three-instanton correction. The $\mathcal{F}_0$ contribution is essentially the perturbative 1-loop contribution:

$$
\begin{aligned}
(2\pi i)^3 \mathcal{F}^{\text{pert}} = {}& 2\text{Li}_3(Q) - \sum_{j=1}^{2} \left( \text{Li}_3\left(-M_j\sqrt{Q}\right) + \text{Li}_3\left(-\frac{\sqrt{Q}}{M_j}\right) \right) \\
= {}& 2\text{Li}_3(Q) - \sum_{j=1}^{2} \left( \text{Li}_3\left(e^{2\pi i(a-\mu_j)}\right) + \text{Li}_3\left(e^{2\pi i(a+\mu_j)}\right) \right),
\end{aligned}
\tag{560}
$$

where we omit a cubic polynomial in $a$. We define the physical discriminant as:

$$
\Delta^{\text{phys}}(U) = \frac{1}{\lambda^2 M_1 M_2} \Delta_{E_3}(U),
\tag{561}
$$

such that the coefficient of the highest power in $U$ in the physical discriminant is unity. We then checked the following relations up to the three-instanton correction:

$$
\begin{aligned}
-\frac{1}{2} \log\left( -\frac{1}{4\pi i} \frac{dU}{da} \right) &= a_0(Q) + 2\pi i \mathcal{A}^{\text{inst}}, \\
-\frac{1}{8} \log\left( \Delta^{\text{phys}}(U) \right) &= b_0(Q) + 2\pi i \mathcal{B}^{\text{inst}},
\end{aligned}
\tag{562}
$$

where $a_0$ and $b_0$ are given by:

$$
\begin{aligned}
a_0 &= -\frac{1}{2} \log(1-Q) + \frac{1}{4} \log(Q) + \frac{1}{2} \log(2), \\
b_0 &= -\frac{1}{2} \log(1-Q) + \frac{3}{8} \log(Q) - \frac{1}{8} \sum_{j=1}^{2} \left( \log\left(1 + M_j\sqrt{Q}\right) + \log\left(1 + \frac{\sqrt{Q}}{M_j}\right) \right).
\end{aligned}
\tag{563}
$$

These factors are the correct perturbative contributions for the 5d $SU(2)$ gauge theory with $N_f = 2$. We then find perfect agreement between the IR and UV computations of the effective gravitational couplings for the $D_{S^1}E_3$ theory.

## Acknowledgements

We are grateful to Fabio Apruzzi, Lakshya Bhardwaj, Matthew Buican, Stefano Cremonesi, Michele Del Zotto, Alba Grassi, Simone Giacomelli, Max Hübner, Heeyeon Kim, Pietro Longhi, Joseph McGovern, Boris Pioline, and especially Sakura Schäfer-Nameki for interesting discussions, feedback and correspondences. The work of CC is supported by a University Research Fellowship 2017, "Supersymmetric gauge theories across dimensions", of the Royal Society. CC is also a Birmingham Fellow. The work of HM is supported by a Royal Society Research Grant for Research Fellows.

# A  Del Pezzo surfaces, $E_n$ characters and 5d gauge-theory variables

In this appendix, we review some well-known facts about how the $E_n$ root system arises from the second homology lattice of the del Pezzo surface $dP_n$ – see *e.g.* [208]. We also explain our choice of 5d gauge-theory parameters, and how they are related to the IIA Kähler parameters naturally associated to the decomposition:

$$H_2(dP_n, \mathbb{Z}) \cong \Lambda_{-\mathcal{K}} \oplus E_n^-. \tag{A.1}$$

We also mention some useful facts about subalgebra of $E_8$.

## A.1  $E_n$ roots and characters from $dP_n$

Consider the $E_n$ algebra, with the $n$ simple roots labelled according to:

$$E_n : \tag{A.2}$$

for $n \geq 4$.[41] Note that the 5d mass deformation of the $E_n$ theory leading to the $E_{n-1}$ theory in the IR corresponds to 'removing' the node $\alpha_n$. For $n < 4$, we have:

$$E_3 : \qquad E_2 : \qquad E_1 : \tag{A.3}$$

In particular, for $E_3 = \mathfrak{su}(3) \oplus \mathfrak{su}(2)$, we denote by $\alpha_1, \alpha_2$ the simple roots of $\mathfrak{su}(3)$. For $n \geq 2$, we view $dP_n$ as the blow-up of $\mathbb{F}_0 \cong \mathcal{C}_b \times \mathcal{C}_f$ at $n-1$ generic points. We then have the basis of curves:

$$\mathcal{C}_b, \qquad \mathcal{C}_f, \qquad E_i, \quad i = 1, \cdots, n-1, \tag{A.4}$$

with the only non-zero intersection numbers inside $dP_n$ being:

$$\mathcal{C}_b \cdot \mathcal{C}_f = 1, \qquad E_i \cdot E_j = -\delta_{ij}. \tag{A.5}$$

This basis is related to the basis $(H, \widetilde{E}_i)$, $i = 0, \cdots, n$, for $dP_n$ viewed as the blow-up of $\mathbb{P}^2$ at $n$ points, with $H$ the hyperplane class in $\mathbb{P}^2$, by:

$$H = \mathcal{C}_f + \mathcal{C}_b - E_1, \quad \widetilde{E}_0 = \mathcal{C}_b - E_1, \quad \widetilde{E}_1 = \mathcal{C}_f - E_1, \quad \widetilde{E}_i = E_i \text{ for } i = 2, \cdots, n-1. \tag{A.6}$$

The canonical class of $dP_n$ is given by:

$$\mathcal{K} = -2\mathcal{C}_b - 2\mathcal{C}_f + \sum_{i=1}^{n} E_i = -3H + \sum_{i=0}^{n} \widetilde{E}_i. \tag{A.7}$$

---

[41]Notice that for $n = 4$, we have the $A_4$ Dynkin diagram with an unusual ordering of the simple roots, $(\alpha_1, \alpha_2, \alpha_4, \alpha_3)$.

$E_n$ **roots.**    Let us consider the following basis of curves orthogonal to $\mathcal{K}$:

$$
\begin{aligned}
\mathcal{C}_{\alpha_1} &= \mathcal{C}_b - \mathcal{C}_f \,, \\
\mathcal{C}_{\alpha_2} &= \mathcal{C}_f - E_1 - E_2 \,, \\
\mathcal{C}_{\alpha_3} &= E_1 - E_2 \,, \\
&\;\;\vdots \\
\mathcal{C}_{\alpha_n} &= E_{n-2} - E_{n-1} \,.
\end{aligned}
\tag{A.8}
$$

These curves intersect according to the Dynkin diagram (A.2), namely:

$$
\mathcal{C}_{\alpha_i} \cdot \mathcal{C}_{\alpha_j} = -A_{ij} \,,
\tag{A.9}
$$

where $(A_{ij})$ is the Cartan matrix of $E_n$, in our choice of basis. (In particular, each $\mathcal{C}_\alpha$ has self-intersection $-2$.) Therefore, M2-brane wrapping these curves will correspond to the $E_n$ W-bosons, in the standard way. The $E_n$ symmetry is a flavour symmetry in 5d, instead of a gauge symmetry, because $E_n$ curves are dual to non-compact divisors.

$E_n$ **characters.**    Let us also define $n$ (formal) curves, denoted by $\mathcal{C}_{e_i}$ corresponding to the fundamental weights of $E_n$, according to:

$$
\mathcal{C}_{\alpha_i} = \sum_{j=1}^{n} A_{ij} \mathcal{C}_{e_j} \,.
\tag{A.10}
$$

Let us denote by $d_n$ the order of the center of the simply-connected group $E_n$:

$$
d_n = |Z(E_n)| = \det(A_{ij}) = \mathcal{K} \cdot \mathcal{K} = 9 - n \,.
\tag{A.11}
$$

It is equal to the degree of the del Pezzo surface (for $n > 2$), as indicated. We see that $d_n \mathcal{C}_{e_i}$ gives us an element of $H_2(dP_n, \mathbb{Z})$.[42] One can use this simple fact to argue that the flavour symmetry group of the 5d $E_n$ theory is $E_n / Z(E_n)$, since formal 'pure flavour' states should wrap integral curves [33].

To each 'pure flavour' curve $\mathcal{C}$, we associate a fugacity $z_{\mathcal{C}} = Q_{\mathcal{C}}$. In particular, let $z_{\alpha_i}$ denote the fugacities associated to the $E_n$ roots (A.8). We then define the fugacities:

$$
t_i = \prod_{j=1}^{n} z_{\alpha_j}^{A_{ij}^{-1}} \,,
\tag{A.12}
$$

which are associated to the fundamental weights $e_i$. Then, for any representation $\mathfrak{R}$ of $E_n$, we can compute the character:

$$
\chi_{\mathfrak{R}} = \sum_{\rho \in \mathfrak{R}} t^\rho \,, \qquad t^\rho \equiv \prod_{i=1}^{n} t_i^{\rho_i} \,,
\tag{A.13}
$$

where $\rho = (\rho_i)$ are the weights of $\mathfrak{R}$ in the fundamental weight basis.

---

[42]For $E_3 = SU(3) \times SU(2)$, we can treat the two factors separately, and therefore $3\mathcal{C}_{e_1}, 3\mathcal{C}_{e_2}$ and $2\mathcal{C}_{e_3}$ are integral cohomology classes.

## A.2   Geometric $E_n$ parameters versus gauge-theory parameters

Let $\widetilde{\mathbf{X}}$ denote the total space of the canonical line bundle over $dP_n$, and let $D_0 \cong [dP_n]$ be the compact divisor. Given any 'flavour' curve $\mathcal{C}_F \subset dP_n$, we have a non-compact divisor $D_F$ such that $\mathcal{C}_F \cong D_F \cdot D_0$. It is then natural to expand the (complexified) Kähler class in terms of parameters $\widehat{a}$ and $\nu_i$ such that:

$$B + iJ = \widehat{a}D_0 + \sum_{i=1}^{n} \nu_i D_{\alpha_i}, \tag{A.14}$$

and similarly for the $C_3$ gauge field in M-theory or Type IIA. We then have a dynamical $U(1)_g$ vector multiplet with scalar field $\widehat{a}$ and $n$ background $U(1)_i$ vector multiplets with mass parameters $\nu_i$. In this basis, the $U(1)_g$ electric charge $\widehat{q}$ and the $U(1)_i$ flavour charges $\widehat{q}_i^F$ of any M2- or D2-brane wrapping a curve $\mathcal{C}$ are given by the intersection numbers:

$$\widehat{q}_0 = -E_0 \cdot \mathcal{C}, \qquad \widehat{q}_i^F = -D_{\alpha_i} \cdot \mathcal{C}. \tag{A.15}$$

In particular, we see that, for $n > 2$, the Kähler parameters in the basis (A.4) read:

$$\begin{aligned}
t_f &= 2\widehat{a} - \nu_1, \\
t_b &= 2\widehat{a} + \nu_1 - \nu_2, \\
t_{E_1} &= \widehat{a} - \nu_2 + \nu_3, \\
t_{E_2} &= \widehat{a} - \nu_2 - \nu_3 + \nu_4, \\
t_{E_j} &= \widehat{a} - \nu_{j+1} + \nu_{j+2}, \qquad j = 3, \cdots, n-1.
\end{aligned} \tag{A.16}$$

This should be compared to the choice (69), namely:

$$t_f = 2a, \qquad t_b = 2a + \mu_0, \qquad t_{E_i} = a + \mu_i, \tag{A.17}$$

which defines the '5d gauge-theory parameters' used throughout the main text. One can easily work out the map between the 'geometric' and '5d gauge-theory' parameters by comparing (A.17) to (A.16). In particular, we have:

$$\widehat{a} = a + \frac{1}{9-n}\left(2\mu_0 - \sum_{j=1}^{n-1} \mu_j\right). \tag{A.18}$$

Of course, the mixing of the $U(1)$ gauge field with background vector multiplets does not affect the physics. We use the parameter $a$ because it directly relates to the 5d (and 4d) gauge theory, wherein the M2-brane wrapped over $\mathcal{C}_f$ is the W-boson of an $SU(2)$ gauge group – that parameterisation is just a convenient choice, which breaks explicitly the permutation symmetry between $\mathcal{C}_f$ and $\mathcal{C}_b$ in $\mathbb{F}_0$. Note that, given the mixing (A.18), we should also define a corresponding complex structure parameter in the mirror geometry $\widehat{\mathbf{Y}}$:

$$\widehat{U} = \left(\lambda^2 \prod_{j=1}^{n-1} M_j\right)^{n-9} U, \tag{A.19}$$

in the notation of section 2.4.1. In fact, $\widehat{U}$ corresponds precisely to the parameter $u$ in [24].

Table 14: Root lattices of rank $s$ that embed into $E_8$. The sublattices that admit two inequivalent embeddings are shown in [blue], while those that cannot be associated with rational elliptic surfaces are in (grey).

| | |
|---|---|
| $s = 8$ | $A_8$, $D_8$, $A_7 \oplus A_1$, $A_5 \oplus A_2 \oplus A_1$, $A_4^2$, $A_2^4$, $E_6 \oplus A_2$, $E_7 \oplus A_1$, $D_6 \oplus A_1^2$, $D_5 \oplus A_3$, $D_4^2$, $(D_4 \oplus A_1^4)$, $A_3 \oplus A_1^2$, $(A_1^8)$. |
| $s = 7$ | $A_6 \oplus A_1$, $A_4 \oplus A_2 \oplus A_1$, $A_5 \oplus A_2$, $A_2^3 \oplus A_1$, $E_6 \oplus A_1$, $E_7$, $D_7$, $D_5 \oplus A_1^2$, $D_4 \oplus A_1^3$, $A_3^2 \oplus A_1$, $(A_1^7)$, $D_6 \oplus A_1$, $D_5 \oplus A_2$, $A_3 \oplus A_2 \oplus A_1^2$, $D_4 \oplus A_3$, $A_3 \oplus A_1^4$, $A_4 \oplus A_3$, $A_5 \oplus A_1^2$, $[A_7]$. |
| $s = 6$ | $A_2^3$, $E_6$, $D_6$, $D_4 \oplus A_1^2$, $[A_3^2]$, $D_5 \oplus A_1$, $A_3 \oplus A_1^3$, $D_4 \oplus A_2$, $A_1^6$, $A_2 \oplus A_1^4$, $A_4 \oplus A_1^2$, $A_6$, $A_3 \oplus A_2 \oplus A_1$, $[A_5 \oplus A_1]$, $A_4 \oplus A_2$, $A_2^2 \oplus A_1^2$. |
| $s = 5$ | $D_5$, $[A_3 \oplus A_1^2]$, $A_3 \oplus A_2$, $A_5$, $A_1^5$, $A_4 \oplus A_1$, $D_4 \oplus A_1$, $A_2 \oplus A_1^3$, $A_2^2 \oplus A_1$. |
| $s = 4$ | $D_4$, $[A_1^4]$, $A_2 \oplus A_1^2$, $A_2^2$, $A_3 \oplus A_1$, $A_4$. |
| $s = 3$ | $A_3$, $A_2 \oplus A_1$, $A_1^3$. |
| $s = 2$ | $A_2$, $A_1^2$. |
| $s = 1$ | $A_1$. |

### A.3 Root lattices embedded into $E_8$

Consider $\mathfrak{g} \subset E_8$ a 'Dynkin subalgebra' of rank $s$ [171], namely an ADE-type subalgebra of $E_8$. The root lattice of $\mathfrak{g}$ embeds into the $E_8$ lattice. It is then one of the root lattices given in table 14. Most of these have a single embedding into $E_8$, while the five root lattices:

$$A_7, \quad A_3^2, \quad A_5 \oplus A_1, \quad A_3 \oplus A_1^2, \quad A_1^4, \tag{A.20}$$

shown in blue in the table, have two distinct embeddings, one being primitive and the other not. Let us also note that, in the language of section 3, the three subalgebras shown in grey in table 14 cannot arise as the 7-brane root lattice of a rational elliptic surface [45].

## B Congruence subgroups of $\mathrm{PSL}(2, \mathbb{Z})$

In this appendix, we review some useful features of the congruence subgroups of $\mathrm{PSL}(2, \mathbb{Z})$. For more details, see for example [209]. The modular group $\mathrm{PSL}(2, \mathbb{Z})$ is generated by:

$$S = \begin{pmatrix} 0 & -1 \\ 1 & 0 \end{pmatrix}, \qquad T = \begin{pmatrix} 1 & 1 \\ 0 & 1 \end{pmatrix}, \tag{B.1}$$

modulo the center of $\mathrm{SL}(2, \mathbb{Z})$, $P = S^2 = -\mathbf{1}$. We first introduce the principal congruence subgroup:

$$\Gamma(N) = \left\{ \begin{pmatrix} a & b \\ c & d \end{pmatrix} \in \mathrm{PSL}(2, \mathbb{Z}) : \begin{pmatrix} a & b \\ c & d \end{pmatrix} = \begin{pmatrix} 1 & 0 \\ 0 & 1 \end{pmatrix} \bmod N \right\}, \tag{B.2}$$

which can be viewed as the kernel of the group homomorphism $\mathrm{PSL}(2, \mathbb{Z}) \to \mathrm{PSL}(2, \mathbb{Z}_N)$. The subgroups $\Gamma$ of $\mathrm{PSL}(2, \mathbb{Z})$ containing the principal congruence subgroup $\Gamma(N)$ are called *congruence subgroups*, with the *level* being the smallest such positive integer $N$. The level-$N$ congruence subgroups encountered in this paper are:

$$\Gamma_0(N) = \left\{ \begin{pmatrix} a & b \\ c & d \end{pmatrix} \in \mathrm{PSL}(2, \mathbb{Z}) : c = 0 \bmod N \right\}.$$

$$\Gamma_1(N) = \left\{ \begin{pmatrix} a & b \\ c & d \end{pmatrix} \in \mathrm{PSL}(2, \mathbb{Z}) : \begin{pmatrix} a & b \\ c & d \end{pmatrix} = \begin{pmatrix} 1 & b \\ 0 & 1 \end{pmatrix} \bmod N \right\}. \tag{B.3}$$

In a similar fashion, we can introduce the groups $\Gamma^0(N)$ and $\Gamma^1(N)$, by requiring $b = 0 \mod N$ instead, which will be related by conjugation to the $\Gamma_0(N)$ and $\Gamma_1(N)$ groups, respectively.

The index[43] $n_\Gamma$ of $\Gamma$ in $\mathrm{PSL}(2,\mathbb{Z})$ is finite for all congruence subgroups. As a result, we have:

$$\mathrm{PSL}(2,\mathbb{Z}) = \bigsqcup_{i=1}^{n_\Gamma} \Gamma\,\alpha_i\,, \qquad \alpha_i \in \mathrm{PSL}(2,\mathbb{Z})\,, \tag{B.4}$$

for a list of coset representatives $\{\alpha_i\}$. The elements of the modular group act on the upper half-plane $\mathcal{H}$ as:

$$\tau \mapsto \frac{a\tau + b}{c\tau + d}\,, \qquad \gamma = \begin{pmatrix} a & b \\ c & d \end{pmatrix} \in \mathrm{PSL}(2,\mathbb{Z})\,, \qquad \forall \tau \in \mathcal{H}\,. \tag{B.5}$$

It then follows that a fundamental domain for the subgroup $\Gamma$ is defined as an open subset $\mathcal{F}_\Gamma \subset \mathcal{H}$ of the upper half-plane, such that no two distinct points are equivalent under the action of $\Gamma$, unless they are on the boundary of $\mathcal{F}_\Gamma$; furthermore, under the action of $\Gamma$, any point of $\mathcal{H}$ is mapped to the closure of $\mathcal{F}_\Gamma$. Let us denote the fundamental domain of $\mathrm{PSL}(2,\mathbb{Z})$ by $\mathcal{F}_0$. The upper half-plane $\mathcal{H}$ is then obtained by the action of the modular group as:

$$\mathcal{H} = \mathrm{PSL}(2,\mathbb{Z})\mathcal{F}_0\,. \tag{B.6}$$

The fundamental domain of $\Gamma \subset \mathrm{PSL}(2,\mathbb{Z})$ can be obtained from a list of coset representatives $\{\alpha_i\}$, since:

$$\mathcal{H} = \left( \bigsqcup_{i=1}^{n_\Gamma} \Gamma\,\alpha_i \right)\mathcal{F}_0 = \bigsqcup_{i=1}^{n_\Gamma} \Gamma\left(\alpha_i \mathcal{F}_0\right)\,. \tag{B.7}$$

Thus, the fundamental domain of $\Gamma$ is $\mathcal{F}_\Gamma = \bigsqcup \alpha_i^{-1}\mathcal{F}_0$, with the coset representatives chosen such that $\mathcal{F}_\Gamma$ has a connected interior.[44]

A *cusp* of $\Gamma$ is defined as an equivalence class in $\mathbb{Q} \cup \{\infty\}$ under the action of $\Gamma$. The $\mathrm{PSL}(2,\mathbb{Z})$ group has only one cusp, with the representative usually chosen as $\tau_\infty = i\infty$. The *width* of the cusp $\tau_\infty$ in $\Gamma$ is the smallest integer $w$ such that $T^w \in \Gamma$. More generally, for a cusp $\tilde{\tau} = \gamma\tau_\infty$, the width is defined as the width of $\tau_\infty$ for the group $\gamma^{-1}\Gamma\gamma$. The cusps other than $\tau_\infty$ are typically chosen as the points of intersection of the fundamental domain with the real axis.

The other special points in the fundamental domain are the *elliptic* points, which are those points with non-trivial stabilizer, *i.e.* $\gamma\tau = \tau$ for some non-trivial element $\gamma \in \Gamma$. The elements $\gamma$ are called the elliptic elements of $\Gamma$. It can be shown that the elliptic points always lie on the boundary of the fundamental domain. Finally, the order of an elliptic point $\tau$ is the order of the stabilizing subgroup of $\tau$ in $\Gamma$. For $\mathrm{PSL}(2,\mathbb{Z})$ the only elliptic points are $\tau_0 \in \{i, e^{\frac{2\pi i}{3}}\}$, with stabilizers $\langle S \rangle$ and $\langle ST \rangle$, of order 2 and 3, respectively. One can prove that, for a given finite index subgroup $\Gamma$ with fundamental domain $\mathcal{F}$, the elliptic points $\tau \in \mathcal{F}$ are always in the $\mathrm{SL}(2,\mathbb{Z})$ orbit of the above elliptic points, *i.e.* $\tau = \gamma\tau_0$, and thus must have orders 2 or 3.

**Elliptic modular surfaces.**   A major simplification in the computation of periods and monodromies occurs when the elliptic surface associated to the Seiberg-Witten curve is modular. These surfaces were first discussed by Shioda [211] and are defined as follows. For a finite index subgroup $\Gamma \subset \mathrm{PSL}(2,\mathbb{Z})$, the quotient $\mathcal{H}/\Gamma$, together with a finite number of cusps, forms a compact Riemann surface $X_\Gamma$, of genus:

$$g_\Gamma = 1 + \frac{n}{12} - \frac{e_2}{4} - \frac{e_3}{3} - \frac{c}{2}\,, \tag{B.8}$$

---

[43]That is, the number of right-cosets of $\Gamma$ in $\mathrm{PSL}(2,\mathbb{Z})$.

[44]The fundamental domains of the congruence subgroups introduced in this section can be drawn using H. Verill's Java Applet [210].

where $n$ is the index of $\Gamma$ in $\mathrm{PSL}(2,\mathbb{Z})$, $e_i$ the number of elliptic points of order $i$ and $c$ the number of cusps. There exists a holomorphic map on $\mathbb{P}^1$:

$$J_\Gamma : X_\Gamma \to \mathbb{P}^1 , \tag{B.9}$$

which reduces to the usual modular function $j(\tau)$ for $\Gamma = \mathrm{PSL}(2,\mathbb{Z})$, where $X_\Gamma$ becomes $\mathbb{P}^1$. The elliptic modular surface corresponding to $\Gamma$ is then the elliptic surface over the Riemann surface $X_\Gamma$. From the above map, it becomes clear that these surfaces are rational.

If the associated elliptic surface to a given Seiberg-Witten curve is modular, one can read the monodromies around the cusps and elliptic points of $\Gamma$ from a set of coset representatives of $\mathcal{H}/\Gamma$ as follows. For a cusp of width $w$ in the $\Gamma$-orbit of $\tau = \gamma\tau_\infty$, with $\gamma \in \mathrm{SL}(2,\mathbb{Z})$, the monodromy matrix is conjugate to $T^w$. In a conveniently chosen basis, this monodromy is given by $\gamma T^w \gamma^{-1}$. The matrices $\gamma$ are part of the coset representatives as one requires that the fundamental domain of $\Gamma$ has connected interior. Note that the choice of $\gamma$ for a given cusp is not unique. Similarly, the monodromies around elliptic points are conjugate to some matrix $\mathbb{M} \in \mathrm{SL}(2,\mathbb{Z})$, as shown in table 4, with the conjugacy matrix $\gamma$ being given by $\tau = \gamma\tau_0$, for $\tau_0 \in \{i, e^{2\pi i/3}\}$.

**Modular forms.** A modular form of integer weight $k$ for a subgroup $\Gamma$ is a holomorphic function $f : \mathcal{H} \to \mathbb{C}$, satisfying:

$$f\left(\frac{a\tau + b}{c\tau + d}\right) = (c\tau + d)^k f(\tau), \qquad \begin{pmatrix} a & b \\ c & d \end{pmatrix} \in \Gamma . \tag{B.10}$$

The ring of modular forms for the $\mathrm{SL}(2,\mathbb{Z})$ group is generated by the $E_4$ and $E_6$ Eisenstein series, defined as:

$$E_4(\tau) = 1 + 240 \sum_{n=1}^\infty \frac{n^3 q^n}{1 - q^n}, \qquad E_6(\tau) = 1 - 504 \sum_{n=1}^\infty \frac{n^5 q^n}{1 - q^n}, \tag{B.11}$$

for $q = e^{2\pi i\tau}$. Additionally, these satisfy:

$$E_k(\tau + 1) = E_k(\tau), \qquad E_k\left(-\frac{1}{\tau}\right) = \tau^k E_k(\tau). \tag{B.12}$$

The modular $j$-function in $\Gamma(1) = \mathrm{PSL}(2,\mathbb{Z})$ defined as:

$$j(\tau) = 1728 \frac{E_4(\tau)^3}{E_4(\tau)^3 - E_6(\tau)^2} , \tag{B.13}$$

is a bijection between $\mathcal{H}/\Gamma(1)$ and $\mathbb{C}$, which parameterizes isomorphism classes of elliptic curves. Using the zeroes of the Eisenstein series $E_{4,6}$, we find that:

$$J(i) = \frac{1}{1728} j(i) = 1, \qquad J\left(e^{\frac{2\pi i}{3}}\right) = \frac{1}{1728} j\left(e^{\frac{2\pi i}{3}}\right) = 0 . \tag{B.14}$$

The rings of modular forms $M_\star(\Gamma)$ for the congruence subgroups usually have more generators. In order to describe those, we first introduce the Dedekind $\eta$-function:

$$\eta(\tau) = q^{\frac{1}{24}} \prod_{j=1}^\infty (1 - q^j) , \tag{B.15}$$

which has the $T$ and $S$ transformations:

$$\eta(\tau + 1) = e^{\frac{i\pi}{12}} \eta(\tau), \qquad \eta(-1/\tau) = \sqrt{-i\tau}\,\eta(\tau) . \tag{B.16}$$

Additionally, we have [192]:

$$\eta\left(\tau + \frac{1}{2}\right) = e^{\frac{i\pi}{24}} \frac{\eta^3(2\tau)}{\eta(\tau)\eta(4\tau)},$$

$$\eta^3\left(\tau + \frac{1}{3}\right) = e^{\frac{i\pi}{12}}\eta^3(\tau) + 3\sqrt{3}e^{-\frac{i\pi}{12}}\eta^3(9\tau). \tag{B.17}$$

A useful theorem used throughout the text states that the $\eta$-quotient $f(\tau) = \prod_{\delta|N} \eta(\delta\tau)^{r_\delta}$ satisfying:

$$\sum_{\delta|N} \delta r_\delta = 0 \bmod 24, \qquad \sum_{\delta|N} \frac{N}{\delta} r_\delta = 0 \bmod 24, \tag{B.18}$$

with $k = \frac{1}{2}\sum_{\delta|N} r_\delta \in \mathbb{Z}$ is a weakly holomorphic weight $k$ modular form for $\Gamma_0(N)$, namely:

$$f\left(\frac{a\tau + b}{c\tau + d}\right) = \chi(d)(c\tau + d)^k f(\tau), \tag{B.19}$$

with the Dirichlet character $\chi(d) = \left(\frac{(-1)^k s}{d}\right)$, where $s = \prod_{\delta|N} \delta^{r_\delta}$. When the associated elliptic curve $X_\Gamma$ has genus zero, there is only one modular form of weight 0, called the Hauptmodul of $\Gamma$. For the congruence subgroups of interest in the main text, these modular functions appear as McKay-Thompson series of the Monster group [173]. One Hauptmodul that does not appear in these lists is the Hauptmodul for $\Gamma_1(5)$. Its expression is given by [170]:

$$f(\tau) = \frac{1}{q}\prod_{n=1}^{\infty}(1-q^n)^{-5\left(\frac{n}{5}\right)}, \tag{B.20}$$

where the notation $\left(\frac{n}{p}\right)$ denotes the Legendre symbol. Note also that this is the fifth power of the Hauptmodul of $\Gamma(5)$.

# C  Seiberg-Witten curves

In this appendix, we list various Seiberg-Witten curves used throughout the main text.

## C.1  Curves for the 4d $SU(2)$ gauge theories

The Weierstrass form of the four-dimensional $SU(2)$ SYM theories we use are given by [2,203]:

$$
\begin{aligned}
N_f = 0 \;:\; & g_2(u) = \frac{4u^2}{3} - 4\Lambda^4, \qquad g_3(u) = -\frac{8u^3}{27} + \frac{4}{3}u\Lambda^4, \\[4pt]
N_f = 1 \;:\; & g_2(u) = \frac{4u^2}{3} - 4m_1\Lambda^3, \qquad g_3(u) = -\frac{8u^3}{27} + \frac{4}{3}m_1 u\Lambda^3 - \Lambda^6, \\[4pt]
N_f = 2 \;:\; & g_2(u) = \frac{4u^2}{3} - 4m_1 m_2\Lambda^2 + \Lambda^4, \\[4pt]
& g_3(u) = -\frac{8u^3}{27} + \frac{4}{3}m_1 m_2 u\Lambda^2 - (m_1^2 + m_2^2)\Lambda^4 + \frac{2}{3}u\Lambda^4, \\[4pt]
N_f = 3 \;:\; & g_2(u) = \frac{4u^2}{3} - \frac{4u\Lambda^2}{3} - 4T_3\Lambda + T_2\Lambda^2 + \frac{\Lambda^4}{12}, \\[4pt]
& g_3(u) = -\frac{8u^3}{27} - \frac{5u^2\Lambda^2}{9} + \frac{u\Lambda}{9}\left(12T_3 + 6T_2\Lambda + \Lambda^3\right) \\[4pt]
& \qquad\quad - T_4\Lambda^2 + \frac{1}{3}T_3\Lambda^3 - \frac{1}{12}T_2\Lambda^4 - \frac{1}{216}\Lambda^6,
\end{aligned}
\tag{C.1}
$$

where in the last line we introduce the $SO(6)$ Casimirs:

$$T_2 = \sum_i^3 m_i^2, \qquad T_4 = \sum_{i<j} m_i^2 m_j^2, \qquad T_3 = \prod_i^3 m_i. \tag{C.2}$$

These conventions are chosen such that the curves agree with the 4d Nekrasov partition function computations. We review the latter in appendix D. Note that in the massless limit, it is convenient to set the dynamical scales to:

$$\Lambda_0 = 2^{-\frac{1}{2}}, \qquad \Lambda_1 = 2^{\frac{2}{3}} 3^{-\frac{1}{2}} i, \qquad \Lambda_2 = 2^{\frac{1}{2}}, \qquad \Lambda_3 = 4. \tag{C.3}$$

These curves are isomorphic to:

$$
\begin{aligned}
N_f = 0 \;:\; & \frac{\Lambda^2}{t} + \Lambda^2 t + x^2 - u = 0, \\
N_f = 1 \;:\; & \frac{\Lambda}{t}(x + m_1) + \Lambda^2 t + x^2 - u = 0, \\
N_f = 2 \;:\; & \frac{\Lambda}{t}(x + m_1) + \Lambda t(x + m_2) + x^2 - \tilde{u} = 0,
\end{aligned}
\tag{C.4}
$$

where for $N_f = 2$ we have:

$$N_f = 2 \;:\; \tilde{u} = u - \frac{\Lambda^2}{2}. \tag{C.5}$$

The CB parameter $\tilde{u}$ in the above notation, breaks the $\mathbb{Z}_2$ symmetry, but agrees with Nekrasov partition function considerations. These $a$-independent shifts do not change the low-energy effective action, as discussed in [36]. Finally, for the $N_f = 3$ theory, the curve:

$$\frac{1}{t}(x + \tilde{m}_1)(x + \tilde{m}_3) + \Lambda t(x + \tilde{m}_2) + x^2 - \tilde{u} = 0, \tag{C.6}$$

has the same Weierstrass form as in (C.1), upon the identifications:

$$N_f = 3 \;:\; \tilde{m}_i = m_i - \frac{\Lambda}{2}, \qquad \tilde{u} = u - (m_1 + m_2 + m_3)\frac{\Lambda}{2} + \frac{\Lambda^2}{4}. \tag{C.7}$$

## C.2 Seiberg-Witten curves for the $E_n$ theories

In this section, we review the Seiberg-Witten curves for the non-toric (rank one) $E_n$ theories, which are obtained as limit of the $E$-string theory SW curve. The fully mass deformed curves were derived in [24, 156], and more recently reviewed in [157, 204]. In terms of the flavour characters $\chi_i$, the $E_8$ curve can be written in Weierstrass form as:

$$
\begin{aligned}
g_2(\widehat{U}, \chi) = {} & \\
& \frac{\widehat{U}^4}{12} - \left(\frac{2}{3}\chi_1 - \frac{50}{3}\chi_8 + 1550\right)\widehat{U}^2 - \left(-70\chi_1 + 2\chi_3 - 12\chi_7 + 1840\chi_8 - 115010\right)\widehat{U} \\
& + \frac{4}{3}\chi_1\chi_1 - \frac{8}{3}\chi_1\chi_8 - 1824\chi_1 + 112\chi_3 - 4\chi_2 + 4\chi_6 - 680\chi_7 + \frac{28}{3}\chi_8\chi_8 + 50744\chi_8 \\
& - 2399276,
\end{aligned}
\tag{C.8}
$$

and:

$$
\begin{aligned}
g_3(\widehat{U}, \chi) = & \\
& -\frac{\widehat{U}^6}{216} + 4\widehat{U}^5 + \left(\frac{1}{18}\chi_1 + \frac{47}{18}\chi_8 - \frac{5177}{6}\right)\widehat{U}^4 \\
& + \left(-\frac{107}{6}\chi_1 + \frac{1}{6}\chi_3 + 3\chi_7 - \frac{1580}{3}\chi_8 + \frac{504215}{6}\right)\widehat{U}^3 + \left(-\frac{2}{9}\chi_1\chi_1 - \frac{20}{9}\chi_1\chi_8\right. \\
& + \frac{5866}{3}\chi_1 - \frac{112}{3}\chi_3 + \frac{1}{3}\chi_2 + \frac{11}{3}\chi_6 - \frac{1450}{3}\chi_7 + \frac{196}{6}\chi_8\chi_8 + 39296\chi_8 - \frac{12673792}{3}\Big)\widehat{U}^2 \\
& + \left(\frac{94}{3}\chi_1\chi_1 - \frac{2}{3}\chi_1\chi_3 + \frac{718}{3}\chi_1\chi_8 - \frac{270736}{3}\chi_1 - \frac{10}{3}\chi_3\chi_8 + 2630\chi_3 - 52\chi_2 + 4\chi_5\right. \\
& - 416\chi_6 + 16\chi_7\chi_8 + 25880\chi_7 - \frac{7328}{3}\chi_8\chi_8 - \frac{3841382}{3}\chi_8 + 107263286\Big)\widehat{U} + \frac{8}{27}\chi_1\chi_1\chi_1 \\
& + \frac{28}{9}\chi_1\chi_1\chi_8 - 1065\chi_1\chi_1 + \frac{118}{3}\chi_1\chi_3 - \frac{4}{3}\chi_1\chi_2 + \frac{4}{3}\chi_1\chi_6 - \frac{8}{3}\chi_1\chi_7 - \frac{40}{9}\chi_1\chi_8\chi_8 \\
& - \frac{19264}{3}\chi_1\chi_8 + \frac{4521802}{3}\chi_1 - \chi_3\chi_3 + \frac{572}{3}\chi_3\chi_8 - 59482\chi_3 - \frac{20}{3}\chi_2\chi_8 + 1880\chi_2 + 4\chi_4 \\
& - 232\chi_5 + \frac{8}{3}\chi_6\chi_8 + 11808\chi_6 - \frac{2740}{3}\chi_7\chi_8 - 460388\chi_7 + \frac{136}{27}\chi_8\chi_8\chi_8 + \frac{205492}{3}\chi_8\chi_8 \\
& + \frac{45856940}{3}\chi_8 - 1091057493\,.
\end{aligned}
$$
(C.9)

The other $E_n$ curves are recovered iteratively. Starting from the $E_8$ curve, one should rescale the variables as:

$$
(\widehat{U}, x, y) \longrightarrow (\alpha\widehat{U}, \alpha^2 x, \alpha^3 x)\,,
$$
(C.10)

and the characters as:

$$
(\chi_1, \chi_2, \chi_3, \chi_4, \chi_5, \chi_6, \chi_7, \chi_8) \to (\alpha^2\chi_1, \alpha^4\chi_2, \alpha^3\chi_3, \alpha^6\chi_4, \alpha^5\chi_5, \alpha^4\chi_6, \alpha^3\chi_7, \alpha^2\chi_8)\,.
$$
(C.11)

Then, taking $\alpha \to \infty$ and setting $\chi_8 = 1$, one obtains the $E_7$ curve, and similarly for the other $E_n$ theories. Note that this statement is equivalent to decomposing $E_n$ into $E_{n-1} \times U(1)$ and decoupling the $U(1)$ factor [156]. For the toric theories, we can also find the map between the $\widehat{U}$, $\chi$ variables used here and the $U$, $\lambda$, $M_i$ variables used in the main text, by explicit comparison with the Weierstrass form of the 'toric' curves, and one finds perfect agreement with the discussion of appendix A. For instance, for the $E_3$ curve, we find:

$$
\chi_1^{E_3} = \frac{1 + \lambda + M_1 M_2 \lambda}{\kappa_{E_3}^2}\,, \qquad \chi_2^{E_3} = \frac{\lambda + \lambda M_1 M_2 + \lambda^2 M_1 M_2}{\kappa_{E_3}^4}\,, \qquad \chi_3^{E_3} = \frac{\lambda(M_1 + M_2)}{\kappa_{E_3}^3}\,,
$$

$$
\widehat{U} = \frac{1}{\kappa_{E_3}} U\,, \qquad\qquad \kappa_{E_3} \equiv \lambda^{\frac{1}{3}} M_1^{\frac{1}{6}} M_2^{\frac{1}{6}}\,.
$$

In the massless case, we always have $\widehat{U} = U$, as we see from (A.19). For generic masses, we can always rescale the characters and the coordinates so that this equality is maintained; this is what we do implicitly in section 8.

# D $SU(2)$ instanton partition functions in 5d and 4d

In this section, we review the Nekrasov partition functions of the 5d rank-one toric $E_n$ theories on $\mathbb{R}^4 \times S^1$ with the $\Omega$-background, as well as their 4d gauge theory limits. These can be determined using the refined topological vertex formalism [202, 212], which is equivalent to the K-theoretic Nekrasov partition functions of the 5d $SU(2)$ gauge theories [113].

## D.1 Perturbative contributions

For the 5d theories, we introduce the notation:

$$q = e^{2\pi i \tau_1}, \qquad p = t^{-1} = e^{2\pi i \tau_2}, \qquad x = e^{2\pi i a}, \tag{D.1}$$

where $\tau_i = \beta \epsilon_i$ are the dimensionless $\Omega$-background parameters and $a$ is the scalar of some $U(1)$ vector multiplet. The perturbative part of the Nekrasov partition function for a gauge theory with gauge group $G$ and matter transforming in some representation $\mathfrak{R}$ of the Lie algebra $\mathfrak{g}$ is:

$$Z^{\text{pert}}_{\mathbb{C}^2 \times S^1}(a, \tau_1, \tau_2) = \prod_{\alpha \in \mathfrak{g}} (x^\alpha; q, p)_\infty \prod_{\rho \in \mathfrak{R}} \frac{1}{(x^\rho y_F; q, p)_\infty}, \tag{D.2}$$

where $\alpha$ are the roots of $\mathfrak{g}$ and $\rho$ are the weights of $\mathfrak{R}$, while the parameters $y_F$ keep track of the flavour symmetries. Hence, for a single hypermultiplet of charge one under a $U(1)$ gauge field, we have:

$$\mathcal{Z}^{\text{hyper}}_{\mathbb{C}^2 \times S^1}(a, \tau_1, \tau_2) = (x; q, p)_\infty^{-1}, \tag{D.3}$$

with the double-Pocchammer symbol defined as:

$$(x; q, p)_\infty = \prod_{j,k=0}^{\infty} (1 - x q^j p^k). \tag{D.4}$$

Note that this product converges for $\text{Im}(\tau_i) > 0$. We define the 'quantum trilog' as:

$$\text{Li}_3(x; q, p) = -\log(x; q, p)_\infty, \tag{D.5}$$

which, in the limit $\tau_i \to 0$, becomes:

$$\text{Li}_3(x; q, p) \sim$$
$$\frac{1}{(2\pi i)^2} \frac{1}{\tau_1 \tau_2} \text{Li}_3(x) - \frac{1}{2\pi i} \frac{\tau_1 + \tau_2}{2\tau_1 \tau_2} \text{Li}_2(x) - \frac{1}{12} \left( 3 + \frac{\tau_1}{\tau_2} + \frac{\tau_2}{\tau_1} \right) \log(1-x) + \mathcal{O}(\tau_1, \tau_2). \tag{D.6}$$

We should mention that the perturbative contribution as written here is given in a somewhat unconventional parity-violating quantisation scheme (see the discussion in [121]). We must generally add additional 5d (flavour) Chern-Simons contribution to restore parity (in parity-preserving theories). This is what we did in (530) and (531), in particular.

## D.2 Instanton contributions

In what follows, we give our conventions for the K-theoretic Nekrasov partition functions.

**5d Partition function.** Following [200], given a Young tableaux $Y = (\nu_i)$, with transpose $Y^t = (\nu_j^t)$, and a box $s = (i, j)$ in $Y$, we define the arm and leg lengths:

$$A_Y(s) = \nu_i - j, \qquad L_Y(s) = \nu_j^t - i. \tag{D.7}$$

Then, for $\tau_i = \beta \epsilon_i$, the non-perturbative part of the Nekrasov parition function of the five-dimensional pure $SU(2)$ theory is given by:

$$Z_Y^{\text{vector}}(a, \tau_1, \tau_2) = \prod_{\alpha, \beta=1}^{2} \frac{1}{N_{\alpha\beta}(a, t, q)}, \tag{D.8}$$

with q the instanton counting parameter, and $Y = (Y_1, Y_2)$. We defined the product:

$$N_{\alpha\beta}(a, \tau_1, \tau_2) = \prod_{s \in Y_\alpha} \left( 1 - e^{-2\pi i(a_\beta - a_\alpha)} e^{2\pi i L_{Y_\alpha}(s)\tau_1} e^{-2\pi i(A_{Y_\beta}(s)+1)\tau_2} \right)$$
$$\prod_{\widetilde{s} \in Y_\beta} \left( 1 - e^{-2\pi i(a_\beta - a_\alpha)} e^{-2\pi i(L_{Y_\beta}(\widetilde{s})+1)\tau_1} e^{2\pi i A_{Y_\alpha}(\widetilde{s})\tau_2} \right), \tag{D.9}$$

with $a_1 = -a_2 = -a$. Introducing $Q_\alpha = e^{-2\pi i a_\alpha}$, as well as $Q_{\beta\alpha} = Q_\beta Q_\alpha^{-1}$, we have:

$$N_{\alpha\beta}(Q, q, t) = \prod_{s \in Y_\alpha} \left( 1 - Q_{\beta\alpha} q^{L_{Y_\alpha}(s)} t^{A_{Y_\beta}(s)+1} \right) \prod_{\widetilde{s} \in Y_\beta} \left( 1 - Q_{\beta\alpha} q^{-L_{Y_\beta}(\widetilde{s})-1} t^{-A_{Y_\alpha}(\widetilde{s})} \right),$$
$$= \prod_{i,j=1}^{\infty} \frac{1 - Q_{\beta\alpha} q^{-i+\nu_{\alpha,j}} t^{-j+\nu_{\beta,i}^t+1}}{1 - Q_{\beta\alpha} q^{-i} t^{-j+1}}, \tag{D.10}$$

with $q, t$ defined in (D.1). Note that the above prescription in fact applies to the partition functions of $U(N)$ gauge theories, rather than $SU(N)$ or $Sp(N)$. In five dimensions, we will need to account for non-zero Chern-Simons level and fundamental or anti-fundamental matter contributions. The Chern-Simons contribution can be expressed as [213]:

$$\mathcal{Z}_{Y, k}^{CS}(a, \tau_1, \tau_2) = e^{-2\pi i \frac{k}{2}(\tau_1 + \tau_2)} \prod_{\alpha=1}^{2} \prod_{s \in Y_\alpha} e^{-2\pi i k a_\alpha} e^{-2\pi i k(i-1)\tau_1} e^{-2\pi i k(j-1)\tau_2}$$
$$= \prod_{\alpha=1}^{2} e^{-2\pi i k a_\alpha |Y_\alpha|} e^{-2\pi i \frac{k}{2} \|Y_\alpha^t\|^2 \tau_1} e^{-2\pi i \frac{k}{2} \|Y_\alpha\|^2 \tau_2} \tag{D.11}$$
$$= \prod_{\alpha=1}^{2} Q_\alpha^{k|Y_\alpha|} q^{-\frac{k}{2}\|Y_\alpha^t\|^2} t^{\frac{k}{2}\|Y_\alpha\|^2},$$

for which we used the identities [202]:

$$\sum_{s \in Y} (i-1) = \frac{1}{2} \left( \|Y\|^2 - |Y| \right), \qquad \sum_{s \in Y} (j-1) = \frac{1}{2} \left( \|Y^t\|^2 - |Y| \right), \tag{D.12}$$

with $|Y| = \sum_i \nu_i$ and $\|Y\|^2 = \sum_i \nu_i^2$. Finally, the building blocks for fundamental and anti-fundamental matter can be expressed as [214]:

$$Z_Y^{\text{fund}}(a, \mu, \tau_1, \tau_2) =$$
$$\prod_{s \in Y_1} \left( 1 - e^{-2\pi i(\mu+a)} q^{L_{Y_1}(s)+\frac{1}{2}} t^{-j+\frac{1}{2}} \right) \prod_{\widetilde{s} \in Y_2} \left( 1 - e^{-2\pi i(\mu-a)} q^{L_{Y_2}(\widetilde{s})+\frac{1}{2}} t^{-\widetilde{j}+\frac{1}{2}} \right),$$
$$Z_Y^{\text{a-fund}}(a, \mu, \tau_1, \tau_2) = \tag{D.13}$$
$$\prod_{s \in Y_2} \left( 1 - e^{-2\pi i(\mu+a)} q^{-L_{Y_2}(s)-\frac{1}{2}} t^{j-\frac{1}{2}} \right) \prod_{\widetilde{s} \in Y_1} \left( 1 - e^{-2\pi i(\mu-a)} q^{-L_{Y_1}(\widetilde{s})-\frac{1}{2}} t^{\widetilde{j}-\frac{1}{2}} \right),$$

where $s = (i, j)$ and $\widetilde{s} = (\widetilde{i}, \widetilde{j})$. These can also be expressed in terms of the function $N_{\alpha\beta}$ defined before. It is known that the Nekrasov partition function for the toric $E_n$ theories agrees with results from topological vertex. For a review of the topological vertex formalism for all the

toric cases, see [207, 214]. In our notation, the $E_n$ partition functions are given by:

$$E_1 \; : \; Z_{E_1}^{\text{inst}}(a, \tau_1, \tau_2) = \sum_Y \left(\mathfrak{q}e^{-2\pi i(\tau_1+\tau_2)}\right)^{|Y|} Z_Y^{\text{vector}}(a, \tau_1, \tau_2),$$

$$\widetilde{E}_1 \; : \; Z_{\widetilde{E}_1}^{\text{inst}}(a, \tau_1, \tau_2) = \sum_Y \left(\mathfrak{q}e^{-2\pi i(\tau_1+\tau_2)}\right)^{|Y|} Z_Y^{\text{vector}}(a, \tau_1, \tau_2)Z_{Y,\, k=1}^{CS}(a, \tau_1, \tau_2),$$

$$E_2 \; : \; Z_{E_2}^{\text{inst}}(a, \mu_1, \tau_1, \tau_2) = \sum_Y \left(\mathfrak{q}e^{-2\pi i(\tau_1+\tau_2)}\right)^{|Y|} Z_Y^{\text{vector}}(a, \tau_1, \tau_2)Z_Y^{fund}(a, \mu_1, \tau_1, \tau_2), \quad \text{(D.14)}$$

$$E_3 \; : \; Z_{E_3}^{\text{inst}}(a, \mu_1, \mu_2, \tau_1, \tau_2) = \sum_Y \left(\mathfrak{q}e^{-2\pi i(\tau_1+\tau_2)}\right)^{|Y|} Z_Y^{\text{vector}}(a, \tau_1, \tau_2)\times$$

$$\times Z_Y^{fund}(a, \mu_1, \tau_1, \tau_2)Z_Y^{a\text{-}fund}(a, \mu_2, \tau_1, \tau_2).$$

In these conventions, the instanton counting parameter is:

$$E_n \; : \; \mathfrak{q} = \lambda, \qquad \widetilde{E}_1 \; : \; \mathfrak{q} = -\lambda. \tag{D.15}$$

Let us also note that the series expansion in the instanton counting parameter is an expression for the partition function valid in the gauge-theory limit of the $E_n$ theories. In this regard, we will identify:

$$Q = e^{4\pi i a} \approx \frac{1}{U^2}, \tag{D.16}$$

when comparing to the results obtained from the SW curves.

**4d partition function.** The 4d partition functions can be obtained in the small-circle limit, by introducing the appropriate factors of $\beta$ in the 5d expressions, namely:

$$\tau_i \to \beta \epsilon_i, \qquad a \to \beta a, \qquad \mu_i \to \beta m_i. \tag{D.17}$$

The $SU(2)$ vector multiplet contribution is then given by:

$$Z_Y^{\text{vector}}(a, \epsilon_1, \epsilon_2) = \prod_{\alpha,\beta=1}^{2} \frac{1}{N_{\alpha\beta}(a, t, q)}, \tag{D.18}$$

with the 4d version of the product:

$$N_{\alpha\beta}(a, \epsilon_1, \epsilon_2) = \prod_{s\in Y_\alpha} \left(a_\beta - a_\alpha - L_{Y_\alpha}(s)\,\epsilon_1 + (A_{Y_\beta}(s)+1)\,\epsilon_2\right)$$

$$\prod_{\widetilde{s}\in Y_\beta} \left(a_\beta - a_\alpha + (L_{Y_\beta}(\widetilde{s})+1)\,\epsilon_1 - A_{Y_\alpha}(\widetilde{s})\,\epsilon_2\right), \tag{D.19}$$

again with $a_1 = -a_2 = -a$. The fundamental matter contribution is given by:

$$Z_Y^{fund}(a, m, \epsilon_1, \epsilon_2) = \prod_{\alpha=1}^{2} \prod_{s\in Y_\alpha} \left(a_\alpha + m + \left(i - \frac{1}{2}\right)\epsilon_1 + \left(j - \frac{1}{2}\right)\epsilon_2\right). \tag{D.20}$$

We are interested in computing the partition function for the 4d $SU(2)$ theories with $N_f \le 3$. This is given by:

$$Z_{N_f}^{\text{inst}}(a, m, \epsilon_1, \epsilon_2) = \sum_{(Y_1, Y_2)} \mathfrak{q}^{\,|Y_1|+|Y_2|} Z_Y^{\text{vector}}(a, \epsilon_1, \epsilon_2) \prod_{i=1}^{N_f} Z_Y^{fund}(a, m_i, \epsilon_1, \epsilon_2). \tag{D.21}$$

For $N_f = 4$, there is an additional contribution needed to fully decouple the $U(1)$ factor from the above $U(2)$ prescription of the partition function – see *e.g.* [36] The $U(N)$ prescription can be used for $SU(N)$ gauge theories, by identifying and factoring out the additional $U(1)$ contribution. In four dimensions, this factor was first identified in the context of the AGT correspondence [215]. We further note that the 4d limit obtained from the $E_3$ partition function discussed in the main text slightly differs from the above expression. In particular, the difference appears at the level of the prepotential, but it only involves an $a$-independent term [36]. This is related to our choice of quantisation for the 5d fermions.

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
