# Peer review of "The $U$-plane of rank-one 4d $\mathcal{N}=2$ KK theories"

_SciPost Physics, doi:SciPost Phys. 12, 065 (2022)_

## Round 2 · Referee Report · Anonymous (Referee 1) · 2021-10-8

Strengths

1- Very well written, and clear. Excellent review and overview of the vast literature on the subject, including also the needed results from mathematics. The paper aims and succeeds in being relatively self-contained. The referencing is also very complete. 2-Contains interesting new results about the rank 1 5d SCFTs compactified on a circle to 4d, with clarifications and extensions of previous results from the literature. 3- Many interesting, useful interconnections with string theory and geometry are further explored, extending previous results from the literature. 4-Extensive, helpful discussion of the details in many special cases, including RG flow relations between theories.

Weaknesses

1- The strengths listed above unavoidably lead to a long paper, with the new results interspersed with the nice review. A rushed reader can find the highlights of the new results summarized in the excellent introduction.

Report

This work is a complete study of the Coulomb branch of all rank 1 5d SCFTs, including effects from putting the theories on a circle. This connects the 5d rank 1 theories to the 4d rank 1 theories with additional KK effect contributions. As discussed, this is natural from the perspective of geometric engineering of the 4d rank 1 theories from IIA, since IIA D-branes give the KK tower of M-theory on a circle. The paper gives thorough analysis of the interconnections between the 5d theories, the 4d theories and KK tower, and the connections with string theory, M-theory, and geometry. The various symmetries, including enhanced symmetry points and higher form discrete symmetry are connected to the geometry and results and classifications in mathematics. The detailed modular properties are thoroughly discussed. Aspects of the gravitational couplings are also discussed. The deformations by mass parameters (background global symmetries), and associated RG flows between theories, is discussed. The paper is very clearly written and full of new results, along with a nice, rather complete review of the extensive literature of past work on rank 1 theories in 4d and 5d. Many important special cases and interconnections are discussed in detail, including both mathematical and physical aspects. The reader will learn (or recall) the background, and also learn many interesting new things from reading this paper (I certainly did). This paper is likely to become a primary, go-to reference for researchers who want to find or reference a result about the Coulomb branches of 4d or 5d rank 1 theories and their interconnections. It was written with a lot of care, and I did not find any suggested corrections or improvements. I recommend that it be published in its current form.

Requested changes

None.

---

## Round 2 · Referee Report · Anonymous (Referee 2) · 2021-12-15

Report

This is a sound and original work of some interest to the field theory and string theory community. It investigates in a unified way the Coulomb branch geometries of rank-1 5d N=1 supersymmetric field theories on R^4 x S^1. It organizes these geometries in terms of the known classification of rational elliptic surfaces, extending a discussion of the purely 4d rank-1 N=2 superconformal field theories given by Caorsi & Cecotti (ref. [56] of this paper) to these 5d theories.
This paper is exceptionally clearly-written. Also, it is nicely self-contained, as it includes very clear short reviews of the necessary background field theory, string theory constructions, and math.

---

## Editorial Decision

published